# FEEL: Quantifying Heterogeneity in Physiological Signals for Generalizable Emotion Recognition

**Pragya Singh**[*]
IIIT-Delhi
India
pragyas@iiitd.ac.in

**Ankush Gupta**[†]
IIIT-Delhi
India
ankush21232@iiitd.ac.in

**Somay Jalan**[†]
IIIT-Delhi
India
somay22505@iiitd.ac.in

**Mohan Kumar**
RIT
NY, USA
mjkvcs@rit.edu

**Pushpendra Singh**
IIIT-Delhi
India
psingh@iiitd.ac.in

## Abstract

Emotion recognition from physiological signals has substantial potential for applications in mental health and emotion-aware systems. However, the lack of standardized, large-scale evaluations across heterogeneous datasets limits progress and model generalization. We introduce FEEL (Framework for Emotion Evaluation), the first large-scale benchmarking study of emotion recognition using electrodermal activity (EDA) and photoplethysmography (PPG) signals across 19 publicly available datasets. We evaluate 16 architectures spanning traditional machine learning, deep learning, and self-supervised pretraining approaches, structured into four representative modeling paradigms. Our study includes both within-dataset and cross-dataset evaluations, analyzing generalization across variations in experimental settings, device types, and labeling strategies. Our results showed that fine-tuned contrastive signal-language pretraining (CLSP) models (71/114) achieve the highest F1 across arousal and valence classification tasks, while simpler models like Random Forests, LDA, and MLP remain competitive (36/114). Models leveraging handcrafted features (107/114) consistently outperform those trained on raw signal segments, underscoring the value of domain knowledge in low-resource, noisy settings. Further cross-dataset analyses reveal that models trained on real-life setting data generalize well to lab (F1 = 0.79) and constraint-based settings (F1 = 0.78). Similarly, models trained on expert-annotated data transfer effectively to stimulus-labeled (F1 = 0.72) and self-reported datasets (F1 = 0.76). Moreover, models trained on lab-based devices also demonstrated high transferability to both custom wearable devices (F1 = 0.81) and the Empatica E4 (F1 = 0.73), underscoring the influence of heterogeneity. Overall, FEEL provides a unified framework for benchmarking physiological emotion recognition, delivering insights to guide the development of generalizable emotion-aware technologies. Code implementation is available here. More information about FEEL can be found on our website.

## 1 Introduction

Emotion recognition is an increasingly important area of research with broad applications in mental health, behavior understanding, and human-computer interaction (HCI) (Thieme et al. [2020]). Recent advances in artificial intelligence have significantly improved our ability to infer emotional

---

[*]Corresponding Author, Website: https://alchemy18.github.io/pragyasingh/
[†]Both authors contributed equally.

39th Conference on Neural Information Processing Systems (NeurIPS 2025) Track on Datasets and Benchmarks.

states from multimodal data sources, including speech, facial expressions, and physiological signals (Abbaspourazad et al. [2024], Heimerl et al. [2023], Ma et al. [2023]). Among these, physiological signals, such as EDA and PPG, offer a unique advantage as they provide a direct, involuntary window into emotional states, less prone to conscious control or social masking than behavioral cues like facial expressions (Yang et al. [2018]) or vocal features (Ma et al. [2023]). Combined with the growing ubiquity of wearable devices, this makes physiological signals a promising foundation for real-world emotion-aware applications (Saganowski et al. [2023], Wang et al. [2014]).

However, the field remains limited in its ability to deliver large-scale, generalizable machine learning models, unlike fields such as computer vision or natural language processing, where major breakthroughs have been driven by massive, unified datasets. In physiological emotion recognition, data collection is fundamentally more constrained due to reliance on human participants, specialized sensors, and ethical considerations (Singh et al. [2025a,b], Saganowski et al. [2023]). As a result, most publicly available datasets are small and diverse in their structure, collected under different protocols, sensor settings, and labeling schemes, making it difficult to combine or use them effectively. While a large number of small-scale datasets have been released, their potential remains largely untapped due to the lack of standardization across datasets, which introduces *heterogeneity* that leads to domain shifts, preventing models trained on one dataset from generalizing to others (Zhang et al. [2024], Mishra et al. [2020], Han et al. [2024]). This fragmentation impedes progress, as it discourages data harmonization, hinders reproducibility, and limits the ability to train robust, cross-domain models.

To move toward scalable, high-impact physiological signal-based emotion recognition systems, the field must begin to treat data as a **"shared resource"** and should work towards harmonizing datasets across key dimensions such as signal representations and labeling strategies. This is vital not only for enabling large-scale training and effective domain adaptation but also for establishing fair and reproducible benchmarks. In the absence of such coordination, research remains fragmented, and findings from one dataset may fail to generalize to others. Furthermore, the field currently lacks a systematic, dataset-level benchmark for evaluating emotion recognition models across widely used physiological signals, particularly EDA and PPG signals, which are increasingly common in wearable sensing platforms (Singh et al. [2024]). Such a benchmark would enable standardized model evaluation, facilitate signal-specific insights, and support assessment of generalization across datasets. By providing consistent pre-processing and labeling protocols, it promotes fairness, reproducibility, and meaningful comparison. This foundation is essential for accelerating progress toward deployable emotion recognition systems in real-world, wearable contexts.

To address the lack of standardized evaluation for heterogeneity and its impact on model performance, in this work, we curated a diverse collection of **19 publicly available datasets** covering a wide range of experimental conditions and labeling strategies (as detailed in appendix A.1). We performed a **meta-analysis** of data quality and benchmarked this dataset suite using **four representative modeling approaches** commonly employed in prior studies (Han et al. [2024], Zhang et al. [2024], Singh et al. [2024]): (i) traditional machine learning using handcrafted features, (ii) deep learning applied to handcrafted features, (iii) deep learning directly on segments of raw physiological signals, and (iv) pre-trained representation learning methods that leverage signal embeddings learned from external tasks or domains, and presented **comprehensive performance comparisons** to highlight the challenges and opportunities posed by heterogeneous data. In addition to performance evaluation, we also present a comprehensive **cross-dataset analysis** to examine key dimensions of dataset heterogeneity that impact model generalization, with the goal of addressing fundamental questions about which modeling paradigms are effective under varying conditions. Specifically, we examined three harmonization dimensions: Experimental Setting, Device Type, and Labeling Method. In addition, we conducted transferability experiments focusing on participants' demographic characteristics. By systematically analyzing these dimensions, we uncovered how design choices across datasets contribute to performance variability in cross-data models.

We present *FEEL*, the first unified cross-dataset evaluation framework for emotion recognition from physiological signals, enabling a systematic analysis of model generalizability and transferability across diverse data collection scenarios. By moving beyond isolated dataset evaluations, FEEL facilitates a holistic assessment of model performance under varying experimental conditions. Our contributions include: (1) a comprehensive benchmark of 19 publicly available emotion recognition datasets based on physiological signals; (2) a unified binning strategy for data harmonization; (3) a novel fine-tuning strategy for contrastive language-signal pretraining (CLSP) applied to datasets lacking textual modalities; and (4) extensive cross-dataset analyses to evaluate model transferability

across variations in labeling strategies, devices, and settings, as well as transferability across demographic groups. Together, FEEL lays the groundwork for developing scalable, robust emotion recognition models for real-world affective computing applications.

## 2 Related Work

The core concept of physiological emotion recognition is to investigate how emotional states can be inferred from measurable bodily signals. Its development intertwines with advances in psychology, neuroscience, biomedical engineering, and computer science. Early theoretical foundations were laid by the James-Lange theory of emotion, which posited a direct link between physiological states and emotional experience (James [1948]). Subsequent research in psychophysiology provided empirical evidence that specific emotional states correlate with autonomic nervous system activity, such as heart rate variability (HRV), electrodermal activity (EDA), and respiration patterns (Kreibig [2010]). With the emergence of affective computing, researchers began leveraging machine learning to model emotion from physiological signals. Picard's foundational work introduced the concept of machines capable of recognizing and responding to human emotions (Picard [2000]), catalyzing interest in using computational proxies for emotion recognition. In the following years, various supervised learning approaches were proposed to classify emotions using multimodal physiological data, including EEG, ECG, PPG, EDA, and skin temperature (Chanel et al. [2006], Kim and André [2008], Shukla et al. [2021]). Recently, the focus has shifted to utilizing deep learning models to capture complex, non-linear relationships in physiological data. Recurrent neural networks (RNNs) and convolutional neural networks (CNNs) have demonstrated promising performance in end-to-end emotion classification tasks (Zitouni et al. [2022]). Additionally, attention-based models and contrastive learning have been explored to address challenges in subject variability and generalization (Singh et al. [2025a], Tripathi et al. [2017], Matton et al. [2023]).

While early emotion recognition systems were primarily developed and evaluated in controlled laboratory environments (Koelstra et al. [2012], Subramanian et al. [2018], Schmidt et al. [2018]), recent research has shifted toward recognizing emotions in real-world settings. This transition is motivated by the growing demand for emotion-aware systems in domains such as mobile health monitoring, human-robot interaction, and affect-aware virtual agents. Wearable and mobile devices have enabled continuous, unobtrusive monitoring of physiological signals, making it potentially feasible to infer emotional states during daily activities (Gjoreski et al. [2016]). Despite technical advances, emotion recognition from physiological signals faces ongoing challenges, including increased signal noise, environmental variability, high inter-subject variability, and a lack of large-scale, standardized datasets (Saganowski et al. [2023]). Recent works have begun to explore self-supervised and contrastive learning paradigms to reduce reliance on labeled data and develop pre-trained models (Pillai et al. [2024], Abbaspourazad et al. [2024], Narayanswamy et al. [2024]). However, data availability remains a major bottleneck, as many emotion recognition datasets are not publicly released due to ethical and privacy concerns. Even when datasets are accessible, they often require direct communication with the authors, creating logistical barriers that slow research progress and contribute to fragmentation, hindering consistent benchmarking and cross-study comparability (Mishra et al. [2020], Han et al. [2024]). Despite their limitations, these small-scale datasets are an indispensable resource. When appropriately aggregated and evaluated, they offer a unique opportunity to benchmark models across heterogeneous settings, fostering more transparent, comparative research. To address this gap, in this work, we propose the first multi-dataset evaluation framework designed to maximize the utility of existing public datasets (with PPG and EDA signals) and advance the development of more generalizable and deployable emotion recognition models.

## 3 Methods

### 3.1 Data Curation

To enable a comprehensive evaluation of heterogeneity in emotion recognition, we curated a collection of 19 datasets comprising PPG and EDA. All datasets are either publicly accessible through research repositories or available upon request from the authors. These datasets collectively represent diverse demographic populations, recording environments, and experiment protocols, providing a basis for evaluating heterogeneity and its influences. PPG and EDA signals were specifically chosen due to their non-invasive nature, widespread implementation in commercial wearable devices, and demonstrated

| Dataset | #Participants | Devices | Settings | Task Descriptions | Labeling |
|---|---|---|---|---|---|
| WESAD | 15 | E4 | Lab | Neutral Reading, Funny Video Clips, Trier Social Stress Test (TSST), Meditation | Stimulus-Label |
| NURSE | 15 | E4 | Real | Stress in a Work Environment (Hospital) | Self-report |
| EMOGNITION | 43 | E4 | Lab | Short Film Clips | Stimulus-Label |
| UBFC_PHYS | 56 | E4 | Lab | Speech Task - Interview/Holiday description, Arithmetic Task - Countdown | Stimulus-Label |
| VERBIO | 49 | E4 | Lab | Public speaking anxiety in real and virtual environments | Self-report |
| PhyMER | 30 | E4 | Lab | Video Stimuli | Self-report |
| EmoWear | 48 | E4 | Lab | Video Stimuli | Self-report |
| MAUS | 22 | Procomp Infinit | Lab | N-Back Task | Stimulus-Label |
| CLAS | 62 | Shimmer3 GSR+ | Lab | Video Stimuli, Math Problems, Logic Problems, and Stroop Test | Stimulus-Label |
| CASE | 30 | ThoughtTech SA9309M, SA9308M | Lab | Video Clips | Self-report |
| Unobtrusive | 24 | E4 | Lab+Real | Lab: Mental Arithmetic, Sudoku, N-back, Stroop, Eye-Closing, Relaxation; Real Life: Work from Home | Stimulus-Label |
| CEAP-360VR | 32 | E4 | Lab | VR Video Clips | Self-report |
| ScientISST MOVE | 15 | E4 | Constraint | Lift a Chair, Greetings, Gesticulate, Jumps, Walk, Run | Stimulus-Label |
| LAUREATE | 44 | E4 | Real | 13-Week Study in University Settings | Self-report |
| ForDigitStress | 38 | IOMbiofeedback | Constraint | Digital Job Interviews | Expert-Annotation |
| Dapper | 88 | Custom Wristband | Real | Emotional Experiences in Daily Life Over Five Days | Self-reports |
| ADARP | 11 | E4 | Real | Daily Diary Study (4 Times/14 Days) – Individuals with Alcohol Use Disorders | Self-report |
| MOCAS | 21 | E4 | Lab | CCTV Monitoring Task Scenario | Self-report |
| Exercise | 36, 31, 30 | E4 | Constraint | Stroop, Trier Mental Challenge, Debate, Counting, Anaerobic/Aerobic Exercise, Rest | Stimulus-Label |

Table 1: Overview of our 19 Emotion Datasets: Participant Count, Devices Used, Experimental Settings, Task Descriptions, and Labeling Methods. More information added in the appendix A.1.

utility for detecting emotional states in ecological settings relevant to real-world applications. The detailed list of our selected datasets, participant counts, devices used, experimental settings, task descriptions, and labeling methods is provided in Table 1 and Appendix A.1.

## 3.2 Data Preprocessing and Standardization

To ensure consistency across the heterogeneous formats of the 19 datasets, we developed a unified preprocessing pipeline. For each dataset, we generated standardized CSV files containing (i) extracted features along with participant ID (PID), arousal, and valence labels, and (ii) raw physiological signal data with corresponding metadata. Separate files were created for EDA, PPG, and their combined modalities. We first segmented the data on a per-participant, task-wise basis for the dataset collected in lab or constraint settings with fixed stimuli or tasks. In datasets with no task-specific segments (real-world datasets), we subdivided the signals into hourly segments (Han et al. [2024]) as per self-reports. Each segment was labeled using a **unified binary scheme** where all data was mapped to arousal and valence dimensions (Tian et al. [2022]). For datasets where self-reported arousal-valence labels were available, we used them directly; otherwise, we inferred arousal and valence levels based on stimulus type, task metadata, or other self-report data available. This approach harmonized the labeling schemes across datasets and enabled consistent categorization of our data into binary categories. Additional dataset-specific information and binning procedures are documented in the appendix A.1. Following binning, the signal segments were preprocessed to remove artifacts. We then extracted features (see appendix A.4.1) separately for EDA and PPG, followed by their concatenation to form a combined feature set. To account for inter-individual variability and prevent dominance of any single feature due to scale differences, participant-wise min-max normalization was applied to the extracted features (Singh et al. [2024]). Feature selection was guided by prior work (Han et al. [2024], Singh et al. [2024]). Additionally, on raw signal segments, we applied z-score normalization on a

per-participant basis to standardize signal distributions, ensuring comparability across datasets while preserving temporal structure and within-participant variability (Mishra et al. [2020]). To visualize the data-wise distribution of our features after unification, see figure 2. Additionally, to enable more nuanced emotion classification experiments, we performed a four-class binning based on the widely accepted circumplex model of affect (Russell [1980]). Each segment was further labeled into the following four classes: High Arousal Positive Valence (HAPV), High Arousal Negative Valence (HANV), Low Arousal Positive Valence (LAPV), and Low Arousal Negative Valence (LANV), based on the available arousal and valence labels. This four-class approach was chosen for the following reasons: **i) Theoretical grounding and granularity** – it is widely used in the emotion recognition and affective computing community and captures more fine-grained emotional distinctions than binary labels. **ii) Practical feasibility** - arousal and valence annotations are the most commonly available labels across our benchmark datasets, enabling consistent application of this scheme. **iii) Alignment with cross-dataset harmonization** – it maintains our experimental setting and label unification strategy while better reflecting the complexity of human emotions.

### 3.3 Datasets Benchmarking

We benchmarked our 19 physiological emotion datasets (Table 1) across four representative modeling paradigms to evaluate the performance of different signal modalities.

**1) Traditional Machine Learning (ML)**: In this paradigm, we used our extracted handcrafted statistical features $f_{HC}$ directly to train classical ML classifiers - RF and LDA. This paradigm was chosen because it serves as a strong baseline and remains prevalent in prior literature (Schmidt et al. [2018], Subramanian et al. [2018], Zhou et al. [2025]), especially for small-sized datasets. Implementation details are provided in the appendix A.4.2.

**2) Deep Learning with Handcrafted Features**: In this paradigm we used our extracted handcrafted features $f_{HC}$ as input to deep learning architectures - MLP ($f_{HC}+MLP$), ResNet ($f_{HC}+ResNet$), LSTM ($f_{HC}+LSTM+MLP$), and Attention-based model ($f_{HC}+Attention+MLP$). This paradigm was chosen because it combines domain knowledge embedded in engineered features with non-linear learning capabilities, representing practical scenarios where interpretability and complex pattern recognition are both required (Ali et al. [2018], Lee et al. [2020], Zhao et al. [2023], Ehiabhi and Wang [2023]). Implementation details are provided in appendix A.4.3.

**3) Deep Learning on Raw Signals**: In this paradigm we used raw time-series signals x(t) directly as input to deep learning architectures - *ResNet, LSTM+MLP*, and *CNN + Transformer Encoder Block*. This paradigm was chosen because it enables end-to-end learning without manual feature extraction, allowing models to autonomously discover novel representations and potentially capture subtle signal characteristics overlooked in traditional feature engineering (Dzieżyc et al. [2020]). Implementation details are provided in Appendix A.4.4.

**4) Pretrained Representation Learning**: In this paradigm, we evaluated the zero-shot and fine-tuned performance of each of our datasets using models based on *Contrastive Language-Signal Pretraining* (CLSP) (Singh et al. [2024]). This paradigm was chosen to evaluate how each of our datasets performs when utilizing pre-trained models with or without dataset-specific training. We selected CLSP models because, to the best of our knowledge, they are the only available pre-trained models specifically developed for physiological emotion recognition, trained on the EEVR dataset, and incorporating both PPG and EDA modalities. Moreover, CLSP models have been shown to exhibit strong cross-dataset generalization, which motivated our selection. We first performed Zero-shot inference without any dataset-specific adaptation to assess the direct transferability of pretrained representations. Then we performed Dataset-specific fine-tuning to examine how well these representations can be adapted to each dataset's characteristics and influence their classification performance. For fine-tuning, we split the dataset into participant-wise 50-50 train and test sets, and then we employed three progressively increasing data efficiency regimes: **few-shot (5%), low-resource (25%)**, and **partial-participant (50%)** of samples per class from the training set. This systematic approach enables us to comprehensively benchmark not only the baseline zero-shot performance but also the adaptation potential of pre-trained physiological representations across our diverse dataset collection.

For fine-tuning, we adopted *Conditional Context Optimization* (CoCoOp) (Zhou et al. [2022]) (figure 1) with two CoCoOp MetaNet-inspired modulation networks to condition textual prompts based on physiological input signals. This approach was chosen because it enables adaptation of pretrained

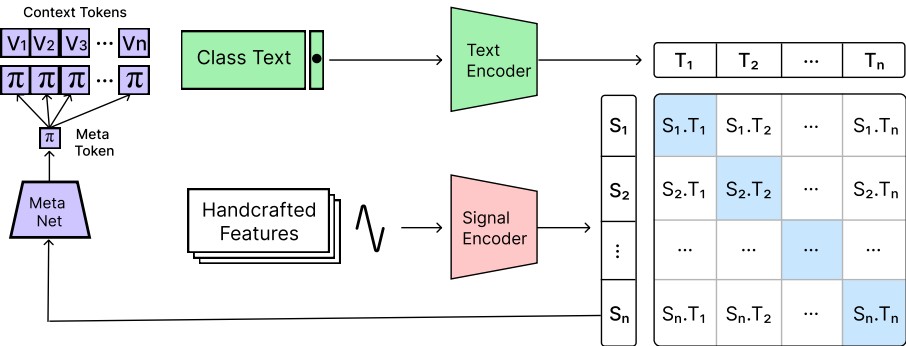

Figure 1: Our CLSP Fine-Tuning Approach, consists of lightweight neural network (Meta-Net) that generates for each signal segment an input-conditional context token.

representations without requiring ground-truth text annotations in the target datasets. The modulation networks were designed in two architectural variants: one using **two linear layers**, and another employing **two stacked 1D convolutional layers** applied to the signal embedding sequence. In our implementation, we expanded beyond basic *class text* labels ("high arousal," "low arousal," "positive valence," or "negative valence") to incorporate more nuanced emotion-specific textual descriptions (detailed in appendix A.4.5) processed through the CLSP text-encoder. These enriched descriptions, along with context tokens, function as context-adaptive textual prompts tailored to each physiological input, facilitating more precise mapping between signal patterns and emotional states. Complete implementation specifications are in Appendix A.4.5.

## 3.4 Cross-Dataset Generalization Analysis

We conducted a systematic cross-dataset analysis to quantify the impacts of contextual heterogeneity. We grouped our datasets (see A.6) based on three dimensions chosen for their potential to introduce systematic variability in data and modeling outcomes. The dimensions include: **1) Experimental Setting:** to capture the influence of differences in ecological validity on model performance. Our dataset collection included three settings: the lab, a semi-realistic constraint setting, and a real-life setting. **2) Device Type:** to account for variation in sensor hardware. Our collection included wrist-worn research-grade wearables (Empatica E4), custom-designed wearables, and lab-based devices. **3) Labeling Method:** to reflect the influence of emotion labeling technique. Our collection includes three annotation strategies: stimulus-based labels (assigned based on predefined stimulus-response assumptions), self-reported labels (reflecting participants' subjective emotional states), and expert annotations (provided by trained observers based on behavioral cues and self-reports). Together, these harmonization dimensions offer a structured approach to assessing the contextual factors. To examine the influence of demographic factors on model transferability, we conducted supplementary experiments across two demographic attributes. For gender-based analysis, we utilized binary male/female labels available across 8 datasets. For age-based analysis, we partitioned subjects into two groups: Young (18–25 years) and Old (25+ years) across 7 datasets, based on data availability. More details are provided in Section A.6.

## 4 Experiments

We conducted separate experiments for arousal, valence, and four-class classification using three input configurations: EDA only, PPG only, and EDA+PPG. For four-class classification, CLSP fine-tuning benchmarking could not be performed on several datasets: NURSE, UBFC_PHYS, MAUS, Unobtrusive, VERBIO, and ADARP lacked samples from one or more classes, while MOCAS and ScientISST MOVE had insufficient samples for 5% fine-tuning. Additionally, we did not benchmark DL-based models for four-class classification (except HC+MLP) due to their poor performance in binary classification. Overall, this set of experiments enabled a systematic comparison of the standalone utility of each modality and its combined contribution across various modeling paradigms.

**ML and DL Paradigm:** We evaluated the performance of our ML and DL models using Leave-One-Subject-Out Cross-Validation (LOSO-CV), to ensure subject-independent evaluation and reflect

real-world deployment conditions. Performance was measured using average accuracy and F1 scores, standard metrics in physiological emotion recognition (Schmidt et al. [2018], Singh et al. [2024]) computed across all LOSO folds. For raw signal-based DL models, we used a sliding window approach with a window size of 60 samples and 50% overlap between consecutive windows. To mitigate class imbalance in both arousal and valence classification, we applied random oversampling to all datasets and SMOTE (Synthetic Minority Over-sampling Technique) to datasets with significant imbalance (defined as a class size difference exceeding one-third). We did not apply oversampling for CLSP fine-tuning except in case of significant imbalance. Additional architectural and training details are provided in Appendix A.4.

**Pretraining Paradigm:** We evaluated pretrained models under subject-independent conditions using three data efficiency regimes: 5%, 25%, and 50% of training data per class. Each dataset was split participant-wise into 50% training and 50% testing sets. To ensure robustness, we repeated the experiment with the train and test splits swapped and computed the average accuracy and F1 score across both folds. For additional details, see Appendix A.4.5.

**Benchmarking Analysis:** To evaluate performance variations across datasets, we conducted a comprehensive meta-analysis, examining the impact of data quality factors (see A.2), sensor modality (EDA, PPG, EDA+PPG), and task (Arousal and Valence) on model effectiveness. We further ranked overall dataset performances and performed a qualitative analysis to interpret the observed trends. This involved assessing model behavior across different modalities and tasks, and attributing performance differences to underlying factors in the data collection pipeline, specifically, the recording environment, labeling methodology, elicitation task, and sensing devices used.

**Cross-Dataset Analysis:** To evaluate generalization performance, we first identified the top three performing models for each dataset. Through majority voting across these top models, we observed that classical machine learning approaches (LDA, RF), the hybrid handcrafted feature-based HC+MLP architecture, and CLSP models with MLP and CNN meta-learners consistently outperformed more complex signal-based deep learning models. Based on this analysis, we selected LDA, RF, HC+MLP, and all CLSP variants for cross-dataset evaluation. To specifically assess cross-domain transferability, each selected model was retrained on its corresponding training partition and evaluated on non-overlapping dataset groups. These results were then compared against two key baselines: **1)** the leave-one-dataset-out (LODO) performance, representing in-domain transferability within the training cohort, and **2)** the CLSP zero-shot performance, which served as a pretrained, no-adaptation baseline for out-of-domain generalization. Where the LODO evaluation was conducted using RF and HC+MLP models to examine within-cohort generalization, i.e., how well models trained on all but one dataset performed when applied to the held-out dataset from the same domain. This comparison enabled a comprehensive analysis of the generalization capabilities of both traditional and pre-trained models when applied to unseen datasets (see more details in Appendix A.5).

**Computation:** For our benchmarking experiment, the computational cost varied by dataset complexity and size. The five largest datasets, including CLAS, CASE, Unobtrusive, DAPPER, and LAUREATE, required approximately 600–720 GPU hours (5–6 days) each. While the remaining 14 datasets required 24–30 hours each, totaling approximately 336–420 GPU hours. Raw signal-based deep learning models were the most computationally intensive, whereas traditional ML models were significantly faster. Fine-tuning CLSP models was highly efficient, requiring only 1–30 minutes of GPU training time, depending on dataset size. All experiments were conducted on 4 NVIDIA A100, 2 H100, and 2 H200 GPUs; more information about the compute resources is provided in Appendix A.3.

# 5 Results

We summarize the main results in this section, with full details provided in Appendix A.8. We begin by outlining the performance trends observed across the individual datasets, followed by a detailed explanation of benchmarking performance and cross-dataset evaluation.

## 5.1 Benchmarking Performance across Modeling Paradigms

**Overall Comparision:** We summarize the overall benchmarking results in Tables 2, 3, and 10. On average, pretrained models, particularly various CLSP variants, consistently outperformed other

| DataSet | EDA | | PPG | | EDA+PPG | |
|---------|-----|---|-----|---|---------|---|
| | Best Model | F1 | Best Model | F1 | Best Model | F1 |
| WESAD | Signal + Resnet | 0.83 | HC+MLP | 0.8 | HC+MLP | **0.91** |
| NURSE | CLSP+CNN (5%) | **0.62** | CLSP+MLP (50%) | 0.52 | CLSP+CNN (5%) | **0.62** |
| EMOGNITION | CLSP+MLP (5%) | **0.68** | CLSP+MLP (50%) | 0.62 | CLSP+CNN (5%) | 0.57 |
| UBFC_PHYS | CLSP+MLP (5%) | **0.45** | CLSP+CNN (25%) | 0.34 | CLSP+MLP (5%) | 0.41 |
| PhyMER | CLSP - Zero Shot | **0.51** | LDA | 0.42 | LDA | 0.42 |
| EmoWear | RF | 0.64 | RF | 0.64 | CLSP+MLP (50%) | **0.67** |
| MAUS | Signal + Resnet | **0.83** | RF | 0.82 | RF | 0.82 |
| CLAS | RF | 0.69 | RF | 0.66 | RF | **0.70** |
| CASE | Signal+CNN+Transformer | **0.47** | HC+MLP | 0.30 | CLSP+MLP (5%)* | 0.40 |
| Unobtrusive | RF | **0.88** | RF | 0.87 | CLSP+MLP (50%) | 0.86 |
| CEAP-360VR | CLSP+CNN (5%) | **0.56** | CLSP+CNN (5%) | 0.43 | CLSP+MLP (5%) | 0.45 |
| ScientISST MOVE | HC+Attention+MLP | 0.77 | RF [†] | 0.81 | HC+MLP | **0.88** |
| Dapper | CLSP+CNN (50%) | 0.77 | CLSP+CNN (5%) | 0.70 | CLSP+MLP (5%) | **0.81** |
| ForDigitStress | CLSP+MLP (25%) | 0.94 | RF [†] | 0.99 | RF [†] | 0.99 |
| ADARP | CLSP+MLP (25%) | **0.83** | CLSP+CNN (50%) | 0.80 | CLSP+MLP (25%) | 0.62 |
| Exercise | CLSP - Zero Shot | **0.63** | CLSP+CNN (25%) | 0.57 | CLSP+MLP (5%) | 0.54 |
| MOCAS | CLSP+CNN (5%) | **0.65** | CLSP+MLP (5%) | 0.62 | CLSP+MLP (25%) | 0.63 |
| LAUREATE | RF[†] | 0.69 | CLSP+MLP (50%) | 0.77 | CLSP Zero-Shot | **0.82** |
| VERBIO | CLSP+CNN (50%) | **0.83** | CLSP+CNN (50%) | 0.77 | CLSP+CNN (50%) | 0.72 |

Table 2: Best-performing model and corresponding F1 score for **arousal classification** across all datasets and modalities (EDA, PPG, EDA+PPG). The table lists, for each dataset and modality, the model that achieved the highest F1 score. **\*** : reflects results that are achieved after applying random sampling before CLSP fine-tuning. †: reflects results achieved after applying SMOTE.

approaches across datasets for binary classification, contributing to 71 of the 114 top-performing model instances for binary classification. Among classical machine learning techniques, RF and LDA followed, with 17 and 8 top-performing entries, respectively. Within the deep learning category, the handcrafted feature-based MLP achieved 11 top results, while signal-based deep models accounted for 3, and handcrafted features combined with attention mechanisms contributed 2 best-performing instances. Within the CLSP model family, we observed that fine-tuning played a crucial role in achieving strong cross-dataset performance. In 29 out of 73 top-performing instances, models required fine-tuning on up to 50% of the target dataset, indicating that while CLSP models offer transferability, moderate domain adaptation is often necessary. Notably, 21 instances achieved competitive results with only 5% of the data used for fine-tuning, suggesting that CLSP models can exhibit effective few-shot generalization. A smaller subset (9 instances) performed best with 25% fine-tuning, reinforcing the spectrum of adaptation needs across datasets. Among CLSP variants, CLSP+CNN demonstrated the highest overall performance, contributing to 37 top-performing cases, followed by CLSP+MLP with 22. Zero-shot variants of CLSP, which require no fine-tuning, were top performers in 14 cases, highlighting the generalizability of the CLSP baseline. Collectively, these findings suggest that while zero-shot CLSP offers a useful starting point, performance can be significantly improved through lightweight dataset-specific fine-tuning, particularly with a CNN metanet that better captures transferable patterns in physiological signals. Detailed comparison of all modeling paradigms and their performances across our datasuite for both recognition tasks and all three modality variations are shown in Figures 5, 6, 7, 8, 9, and 10. For four-class classification, machine learning models (RF and LDA) dominate with 54% of best-performing cases. CLSP-based models appear in 35% of instances, while deep learning models (HC+MLP) account for only 11% of the best-performing outcomes (see Table 10).

**Dataset Specific Performance Variations:** Model performance for binary classification exhibited significant variability across datasets, ranging from a minimum F1 score of 0.30 (e.g., in CASE and ADARP) to a maximum of 0.98 (WESAD), as detailed in Table 11. Our qualitative analysis of dataset collection methodologies (summarized in Table 12 and visualized in Figures 11, 12, and 13 revealed that high-performing datasets generally shared key characteristics: well-balanced experimental setups, ecologically valid elicitation protocols, and robust labeling techniques that accounted for participants' perspectives (Singh et al. [2025a]). In contrast, suboptimal performance was often linked to weak elicitation strategies, misaligned labels, the absence of stimuli covering the full emotional spectrum, and the presence of signal artifacts. Overall, models utilizing EDA consistently outperformed those based on other modalities, as shown in Figure 3. EDA-based models achieved top performance in 12 out of 19 datasets for arousal classification and in 13 out of 19 for valence classification, highlighting the robustness of EDA signals in emotion recognition tasks. Notably, EDA+PPG also showed strong

performance, particularly in real-life and constrained task settings, where multimodal input helped mitigate signal noise. While models based solely on PPG performed comparably in some cases, they generally yielded lower performance than EDA-based approaches, suggesting that PPG may be less sensitive to subtle emotional variations. Overall, our results highlight the critical importance of thoughtfully designing data collection protocols to effectively capture meaningful emotional variations and of aligning labeling strategies that accurately reflect these underlying physiological changes. The four-class classification results varied considerably across datasets, with F1 scores ranging from 0.987 (WESAD) to 0.269 (ADARP). Among the lowest-performing datasets were EmoWear and CLAS, which, despite being lab-based, relied solely on video stimuli, suggesting that limited or low-arousal stimuli can constrain physiological differentiation. ADARP, collected in real-life daily settings, exhibited low performance primarily due to the imbalance in samples across four classes (as shown in Table 4). Meanwhile, CEAP-360VR, CASE, and MOCAS, which combined lab and semi-realistic stimuli, also led to overall poor performance compared to other datasets. These patterns suggest that dataset quality, stimulus richness, and ecological validity collectively influence classification outcomes, beyond simple distinctions between laboratory and real-world environments. Consistent with the binary classification results, models using EDA+PPG or EDA alone achieved relatively high performance across 15 datasets (see figure 4), highlighting the robustness and central role of EDA signals in physiological emotion recognition.

| DataSet | EDA | | PPG | | EDA+PPG | |
|---------|-----------|------|-----------|------|-----------|------|
| | **Best Model** | **F1** | **Best Model** | **F1** | **Best Model** | **F1** |
| WESAD | CLSP+CNN (50%) | 0.83 | CLSP+CNN (50%) | 0.83 | HC+MLP | **0.98** |
| NURSE | CLSP - Zero Shot | **0.62** | CLSP+CNN (5%) $^\dagger$ | 0.39 | CLSP Zero-Shot | 0.38 |
| EMOGNITION | CLSP - Zero Shot | **0.53** | CLSP+MLP (5%) | 0.50 | CLSP+CNN (5%) | 0.39 |
| UBFC_PHYS | RF | **0.76** | LDA | 0.68 | RF | 0.72 |
| PhyMER | CLSP - Zero Shot | **0.72** | CLSP+CNN (50%) | 0.69 | CLSP+MLP (50%) | 0.70 |
| EmoWear | CLSP+CNN (50%) | **0.78** | CLSP+CNN (50%) | 0.77 | RF | 0.77 |
| MAUS | HC+MLP | 0.58 | LDA | 0.56 | LDA | **0.59** |
| CLAS | CLSP - Zero Shot | **0.64** | CLSP+CNN (25%) | 0.61 | HC+Attention+MLP | 0.63 |
| CASE | CLSP+MLP (5%) | **0.54** | LDA | 0.48 | LDA | 0.49 |
| Unobtrusive | CLSP - Zero Shot | **0.71** | RF | **0.71** | CLSP+CNN (25%) | 0.70 |
| CEAP-360VR | CLSP+CNN (5%) | **0.62** | CLSP+CNN (5%) | 0.61 | LDA | 0.50 |
| ScientISST MOVE | CLSP+MLP (50%) | **0.82** | CLSP+CNN (50%) | 0.80 | CLSP+CNN (50%) | **0.82** |
| Dapper | CLSP+CNN (50%) | 0.87 | CLSP+CNN (50%) | 0.85 | CLSP+CNN (50%) | **0.94** |
| ForDigitStress | CLSP+CNN (5%) | 0.87 | RF$^\dagger$ | 0.92 | RF$^\dagger$ | **0.92** |
| ADARP | CLSP - Zero Shot | 0.30 | CLSP Zero-Shot | 0.40 | HC+MLP | **0.47** |
| Exercise | CLSP - Zero Shot | **0.75** | CLSP+CNN (50%) | 0.72 | CLSP+MLP (50%) | 0.71 |
| MOCAS | CLSP - Zero Shot | **0.89** | CLSP+CNN (50%) | 0.87 | CLSP+CNN (25%) | 0.82 |
| LAUREATE | HC+MLP | 0.36 | HC+MLP | **0.41** | CLSP+MLP (50%)$^*$ | 0.40 |
| VERBIO | HC+MLP | **0.40** | HC+MLP | 0.38 | CLSP+MLP (5%) | 0.34 |

Table 3: Best-performing model and corresponding F1 score for **valence classification** across all datasets and modalities (EDA, PPG, EDA+PPG). The table lists, for each dataset and modality, the model that achieved the highest F1 score. * : reflects results that are achieved after applying random sampling before CLSP fine-tuning. $\dagger$: reflects results achieved after applying SMOTE.

## 5.2 Performances Across Harmonizing Dimensions

**Experiment Setting:** Our experiments reveal how training environments impact cross-domain generalization. Models trained on real-world data exhibited good transferability to both lab (max F1 = 0.79 with CLSP MLP at 5%) and constraint-based settings (max F1 = 0.78 with RF). However, their intra-domain performance was notably poor (max F1 = 0.49), reflecting high variability within real-world data. In contrast, models trained in constraint-based settings showed exceptional transferability to real-life data (best F1 = 0.88 with RF), yet only moderate success in lab settings and within their own domain, especially struggling with arousal prediction compared to valence. Further lab-trained models demonstrated a similar pattern, performing well when tested on real-life and constraint-based data (max F1 = 0.76), but underperformed within their own dataset ($F1 \approx 0.5$), possibly due to cohort-specific biases. Detailed results are added in Table 13, 14, and figure 16.

**Device:** Our analysis further revealed significant variability in cross-cohort transferability. Models trained on the wearable (Empatica E4) based data exhibited good generalization to the custom wearable (best F1 = 0.82 with CLSP MLP 50%), but transferred poorly to lab-based datasets (min F1 = 0.45), suggesting limited modality alignment. Conversely, models trained on lab-based devices

showed strong generalization to both the custom wearable (best F1 = 0.81 with LDA) and E4 (best F1 = 0.73 with CLSP CNN 50%). Moreover, custom wearable models transferred poorly overall, with slightly better performance for arousal detection, particularly when tested on lab-based data (best F1 = 0.64). Zero-shot models (CLSP) using EDA demonstrated promising results, achieving up to 0.83 F1 on the custom wearable, and showing moderate generalizability to other cohorts. Detailed results are added in Table 15, 16, and Figure 15.

**Labeling:** Labeling strategy emerged as a major factor influencing model generalization. Expert-annotated datasets produced models that transferred well to both stimulus-labeled (best F1 = 0.72) and self-reported data (best F1 = 0.76), underscoring the robustness of expert-generated labels. Interestingly, stimulus-labeled models also performed strongly on expert-annotated data (best F1 = 0.87 with LDA). However, stimulus-labeled models showed weak generalization to self-reported data (best F1 = 0.63), suggesting limitations in capturing personal emotional nuance. Similarly, self-report trained models were inconsistent, performing well when tested on expert data (best F1 = 0.87 with CLSP CNN 5%) but poorly on stimulus-label. Once again, zero-shot CLSP models using EDA demonstrated outstanding results, achieving the highest F1 overall (0.91) on expert-labeled data. Detailed results are added in Table 17, 18, and figure 14.

**Age and Gender:** For arousal classification, results indicate moderate generalization both within and across gender groups (F1 $\approx$ 0.50–0.56), whereas in zero-shot CLSP transfer, EDA consistently outperformed PPG and EDA+PPG, often by a substantial margin. In contrast, valence classification exhibits an opposite trend: cross-gender transfer achieves notably higher performance (F1 $\approx$ 0.69–0.71 using CLSP fine-tuned models) than within-gender evaluation, where F1 drops to 0.47–0.55. A similar pattern emerges in age-wise transfer experiments, with cross-age-group generalization for valence (F1 $\approx$ 0.67–0.73 using CLSP fine-tuned models) outperforming within-age-group performance. These findings suggest that valence representations are more transferable across demographic groups, while arousal models are more sensitive to participant-specific physiological variability.

Overall, CLSP-based architectures consistently outperformed others, with MLP and RF emerging as weaker baselines, underscoring the importance of model choice and cohort harmonization for robust cross-domain emotion recognition. Collectively, our findings underscore the importance of carefully selecting device and labeling techniques, the potential of constraint-based training for real-world deployment, and the clear advantages of zero-shot and CLSP-based models for cross-domain emotion recognition. Detailed discussion added in A.7.

# 6   Conclusion

**FEEL** presents a comprehensive framework for benchmarking physiological signal-based emotion data, serving as a foundation for developing robust emotion recognition systems. Beyond identifying high-performing models, our findings underscore the nuanced trade-offs between model complexity, performance, and generalizability. For instance, the strong performance of handcrafted feature-based models suggests that signal-specific priors remain essential for high-performing physiological emotion recognition. Meanwhile, the success of fine-tuning CLSP methods indicates the untapped potential of leveraging cross-modal supervision, even in the absence of explicit textual inputs. Importantly, our cross-dataset analyses showed the potential for harmonizing small-scale models for large-scale pretraining. As emotion-aware systems move toward deployment in health, education, and HCI, FEEL serves as a vital step toward reproducible, scalable, and ethically grounded models. In the future, we aim to expand FEEL by incorporating a broader set of publicly available datasets and encouraging community participation in benchmarking and dataset standardization.

**Limitations and Social Impact** This work focuses on benchmarking a representative set of modeling paradigms to establish a broad baseline, primarily using traditional and general-purpose architectures. However, more advanced or domain-specific models tailored to physiological signals were not included and remain an important direction for future benchmarking. Additionally, our current harmonization strategy is limited due to a lack of a common labeling technique. Moreover, our analysis does not account for key sources of heterogeneity, including cultural context and health status; we plan to incorporate them in the future. By systematically studying these limitations, we hope to contribute to the development of responsible, generalizable, and impactful emotion recognition technologies for applications in HCI, affective computing, and mental health support.

# 7 Acknowledgment

We would like to acknowledge the valuable contributions of all the researchers and institutions who made these nineteen datasets available for further research. We emphasize that our work focuses solely on benchmarking; therefore, anyone wishing to use these datasets must follow the access procedures established by the respective dataset owners to ensure privacy and data protection. Additionally, this research work described in this paper made use of the LAUREATE Database, which is owned by the Università della Svizzera italiana (USI), Switzerland. Other datasets are sourced from publicly available repositories, as detailed on our project website. We claim no ownership or rights over any of these datasets, and we will not be able to share any derived features or data. All dataset access requests must be directed to the respective owners. Further, we want to acknowledge the support from the Center of Excellence Human Centered Computing, Infosys Center for Artificial Intelligence, iHub-Anubhuti-IIITD Foundation, established under the NM-ICPS scheme of the DST, and the Center of Excellence in Healthcare (CoEHe) at IIIT-Delhi

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

# A  Technical Appendices

## A.1  Individual Dataset Binning Details

This section outlines the summary of the dataset and information about the binning scheme applied to harmonize our 19 physiological signal-based emotion datasets, facilitating both benchmarking and cross-dataset evaluation. To ensure consistency in labeling across datasets, emotion elicitation tasks or available self-reports were mapped to binary arousal and valence categories when direct arousal and valence self-reports were not provided. Detailed descriptions for each dataset are presented below.

### A.1.1  WESAD

The WESAD dataset (Schmidt et al. [2018]) is a multimodal dataset containing physiological and motion data from 15 participants recorded via wrist- and chest-worn sensors during a lab study. We utilize the wrist data (gathered using Empatica E4 wristbands) for our experiments, as it includes our target modalities: electrodermal activity (EDA) and photoplethysmogram (PPG). The dataset consists of four affective conditions: baseline (neutral reading task), amusement (viewing humorous videos), stress (Trier Social Stress Test), and meditation (guided breathing). For binary arousal labeling, baseline and meditation are grouped as low arousal, while stress and amusement are labeled as high arousal. For valence, baseline and stress are treated as negative valence due to potential anticipatory anxiety or stress induction, and amusement and meditation are labeled as positive valence.

### A.1.2  NURSE

The Nurse dataset (Hosseini et al. [2022], Xiang et al. [2025]) is a multimodal collection of physiological data obtained from 15 nurses working in real-world hospital settings during the COVID-19 pandemic. The data were recorded using Empatica E4 wristbands, capturing signals such as electrodermal activity (EDA), heart rate (HR), skin temperature (TEMP), blood volume pulse (BVP), inter-beat intervals (IBI), and three-axis acceleration (ACC). In addition, periodic smartphone-administered surveys were used to gather context, documenting self-reported stress events and their contributing factors. For our analysis, we focused on the EDA and PPG signals. The labeling mechanism was based on self-reported stress events, where periods labeled as 'stress events' were assigned high arousal and negative valence, while all other periods were categorized as low arousal and positive valence. This dataset was highly imbalanced, as most of the data consisted of periods of high stress, given the challenging hospital environment during the COVID-19 pandemic. This resulted in an overrepresentation of high arousal and negative valence labels, with fewer periods of low stress.

### A.1.3  EMOGNITION

The Emognition dataset (Saganowski et al. [2022]) comprises multimodal physiological and facial expression data collected from 43 participants. Participants viewed short film clips designed to elicit nine discrete emotions: amusement, awe, enthusiasm, liking, surprise, anger, disgust, fear, and sadness. Physiological signals were recorded using three wearable devices: Muse 2 (EEG, ACC, GYRO), Empatica E4 (BVP, EDA, SKT, ACC), and Samsung Galaxy Watch (BVP, HR, ACC, GYRO). Upper-body videos were simultaneously captured to analyze facial expressions. For our analysis, we focus on EDA and PPG signals collected from E4. Emotional states were annotated using elicitation task labels that were subsequently mapped to arousal and valence dimensions: high arousal includes amusement, surprise, anger, enthusiasm, fear, and awe, while low arousal includes sadness, disgust, baseline, neutral, and liking. Positive valence includes amusement, liking, enthusiasm, and awe, while negative valence includes sadness, disgust, baseline, surprise, anger, neutral, and fear.

### A.1.4  UBFC_PHYS

The UBFC-Phys dataset (Sabour et al. [2023]) is a multimodal dataset comprising physiological and video data collected from 56 participants during a three-phase protocol inspired by the Trier Social Stress Test (TSST). Participants underwent a rest phase (T1), a speech task (T2), and an arithmetic task (T3), with each phase designed to elicit varying levels of stress. Physiological signals, including blood volume pulse (BVP) and electrodermal activity (EDA), were recorded using Empatica E4 wristbands. For our analysis, we focus on the PPG and EDA signals. Labeling was performed by

categorizing the rest phase (T1) as low arousal and positive valence, while the speech (T2) and arithmetic (T3) tasks were labeled as high arousal and negative valence, reflecting the increased stress levels associated with these tasks.

### A.1.5   VERBIO

The VerBIO dataset (Nirjhar and Chaspari [2024]) is a multimodal bio-behavioral dataset comprising physiological and audio data collected from 49 participants (originally 55 participants, but data was only available for 49) during 344 public speaking sessions in both real-life and virtual environments. Participants delivered short speeches on assigned topics, with physiological signals recorded using Empatica E4 and Actiwave devices. The original version of the dataset included audio recordings, physiological signals, and self-reported anxiety measures. An updated version incorporates time-continuous stress annotations provided by four annotators, enabling the analysis of moment-to-moment stress levels. For our study, we focus on wrist-derived physiological signals, specifically EDA and PPG. Labeling was performed by categorizing the self-reported measures into periods with high stress annotations as high arousal and negative valence, while periods with low stress annotations were labeled as low arousal and positive valence.

### A.1.6   PhyMER

The PhyMER dataset (Pant et al. [2023]) is a multimodal physiological dataset designed for emotion recognition, incorporating personality traits as contextual information. It comprises physiological signals and personality assessments collected from 30 participants (15 male and 15 female). The physiological data were recorded using two wearable devices: the Emotiv Epoc X headset for EEG signals and the Empatica E4 wristband for EDA, BVP, and peripheral skin temperature. Participants self-reported their emotional responses using a web-based annotation tool, providing ratings on arousal and valence on the SAM scale and categorizing their emotions into seven basic categories: anger, disgust, fear, happiness, neutral, sadness, and surprise on a Likert scale of 1-9. For our study, we focus on wrist-derived physiological signals, specifically EDA and BVP. Labeling was performed by binarizing the self-reported SAM ratings.

### A.1.7   EmoWear

The EmoWear dataset (Rahmani et al. [2024]) is a multimodal physiological and motion dataset designed for emotion recognition and context awareness. It comprises data from 48 participants (21 females, 27 males) who engaged in a series of tasks, including watching 38 emotionally eliciting video clips, walking a predefined route, reading sentences aloud, and drinking water. Physiological signals were recorded using the Empatica E4 wristband (capturing BVP, EDA, SKT, and accelerometry) and the Zephyr BioHarness (recording ECG, respiration, SKT, and accelerometry). Additionally, three ST SensorTile.box devices were employed to collect accelerometer and gyroscope data. Participants self-assessed their emotional states using the circumplex model of affect, providing ratings on valence, arousal, and dominance scales. For our study, we focus on wrist-derived physiological signals, specifically EDA and BVP. Labeling was performed by binarizing the self-reported arousal valence ratings.

### A.1.8   MAUS

The MAUS dataset (Beh et al. [2021]) is a multimodal physiological dataset designed for mental workload assessment using wearable sensors. It comprises physiological signals collected from 22 participants (2 females) during N-back tasks of varying difficulty levels. The PixArt Watch was used to download the photoplethysmogram (PPG) data. Additionally, a clinical procomp Infinit device was used to record electrocardiography (ECG), galvanic skin response (GSR), and fingertip PPG signals. Participants completed the Pittsburgh Sleep Quality Index (PSQI) questionnaire at the beginning of the experiment and the NASA Task Load Index (NASA-TLX) questionnaire after each N-back task to provide subjective assessments of their sleep quality and perceived workload. For our study, we focus on Procomp data, specifically EDA and PPG. Labeling was performed by categorizing n-back tasks into periods with high workload as high arousal and negative valence, while periods with low workload were labeled as low arousal and positive valence. Specifically, tasks labeled as "0_back" were considered low in cognitive demand and thus assigned low arousal and positive

valence. Conversely, tasks labeled as "2_back" or "3_back" were deemed higher in cognitive load, leading to their classification as high arousal and negative valence.

### A.1.9 CLAS

The CLAS (Cognitive Load, Affect, and Stress) dataset (Markova et al. [2019]) is a multimodal physiological dataset designed to support research on the automated assessment of mental states, including cognitive load, affect, and stress. It comprises synchronized recordings of physiological signals—electrocardiography (ECG), photoplethysmography (PPG), electrodermal activity (EDA), and accelerometer data—from 62 healthy volunteers engaged in five tasks: three interactive tasks (math problems, logic problems, and the Stroop test) aimed at eliciting different types of cognitive effort, and two perceptive tasks involving images and videos selected to evoke emotions. For our study, we focus specifically EDA and PPG collected using Shimmer3 GSR+ unit. We applied a binary labeling scheme for valence and arousal based on task types and stimuli. Tasks such as math tests, Stroop tests, and IQ tests were labeled as high arousal and negative valence, reflecting high cognitive load. Emotion-eliciting videos and images were categorized accordingly: stimuli like videos 2.mp4, 5.mp4, and image set "pics1" were labeled as high arousal and positive valence, while others like videos 13.mp4, 14.mp4, and image set "pics2" were labeled as low arousal and negative valence.

### A.1.10 CASE

The CASE (Continuously Annotated Signals of Emotion) (Sharma et al. [2019]) dataset is a multimodal physiological dataset designed for emotion analysis. It comprises physiological signals and continuous affect annotations collected from 30 participants (15 male and 15 female) while watching various video stimuli. Physiological data were recorded using ThoughtTech sensors measuring ECG, BVP, EMG, EDA, respiration, and skin temperature. Participants provided real-time continuous annotations of their emotional experiences using a joystick-based interface, simultaneously reporting valence and arousal levels. Labeling was performed by calculating the mean of participants' continuous self-reported annotations for each video segment. Segments with mean valence above the overall average were labeled as positive valence, while those below were labeled as negative valence. Similarly, segments with mean arousal above the average were labeled as high arousal, and those below as low arousal.

### A.1.11 Unobtrusive

The Unobtrusive dataset (Anders et al. [2024]) is a multimodal physiological dataset designed for cognitive load assessment in both controlled (lab) and uncontrolled (real-life) environments. It comprises approximately 315 hours of data collected from 24 participants during a four-hour cognitive load elicitation with self-chosen tasks in the real-life setting and a four-hour mental workload elicitation in a lab setting. Physiological signals were recorded using consumer-grade wearable devices, including the Muse S headband and the Empatica E4 wristband, capturing EEG, EDA, PPG, and accelerometer data. Participants performed office-like tasks such as mental arithmetic, Stroop, N-Back, and Sudoku with two defined difficulty levels in the lab, and tasks like researching, programming, and writing emails in the uncontrolled environments. Each task was labeled by participants using two 5-point Likert scales of mental workload and stress, as well as the pairwise NASA-TLX questionnaire. For our study, we focus on wrist-derived physiological signals, specifically EDA and PPG. Labeling was performed by categorizing tasks containing keywords such as 'hig', 'nor', 'stroop', 'n_back', 'arithmetix', or 'sudoku' as high arousal and rest tasks as low arousal. Similarly, tasks with keywords like 'hig_mw', 'vhg_mw', or 'hard' were labelled as negative valence, while the rest were labeled as positive valence.

### A.1.12 CEAP-360VR

The CEAP-360VR (Xue et al. [2023]) is a multimodal dataset designed to study emotional responses within immersive virtual reality (VR) environments. It comprises data from 32 participants who each viewed eight one-minute 360° video clips using an HTC Vive Pro Eye head-mounted display. During the viewing sessions, participants provided continuous valence and arousal annotations via a joystick interface. Physiological signals, including electrodermal activity (EDA), blood volume pulse (BVP), heart rate (HR), inter-beat interval (IBI), and skin temperature (SKT), were recorded using the Empatica E4 wristband. Additionally, behavioral data such as head and eye movements and

pupil diameter were collected. The dataset also includes responses to questionnaires assessing motion sickness (SSQ), presence (IPQ), and workload (NASA-TLX). For our study, we focus on wrist-derived physiological signals, specifically EDA and BVP. Labeling was performed by calculating the mean of participants' continuous self-reported annotations for each video segment. Segments with mean valence above the overall average were labeled as positive valence, while those below were labeled as negative valence. Similarly, segments with mean arousal above the average were labeled as high arousal, and those below as low arousal.

### A.1.13 ScientISST MOVE

The ScientISST MOVE dataset (Saraiva et al. [2023], Goldberger et al. [2000]) is a multimodal physiological dataset designed to study the effects of natural everyday activities on biosignal acquisition. It comprises synchronized recordings from 15 healthy participants (originally 17, but the data only included 15 participants) performing activities such as lifting a chair, greeting, gesticulating, walking, and running. Data were collected using three wearable devices: a chestband, an armband, and the Empatica E4 wristband, capturing signals including Electrodermal Activity (EDA), Photo-plethysmography (PPG), Electrocardiography (ECG), Electromyography (EMG), skin temperature, and actigraphy. For our study, we focus on wrist-derived physiological signals, specifically EDA and PPG. Labeling was performed by categorizing activities based on their physical intensity and associated emotional valence. High-energy activities such as 'jumps' and 'run' were labeled as high arousal and negative valence, while low-energy activities like 'walk_before_downstairs' and 'baseline' were labeled as low arousal and positive valence. Similarly, activities perceived as positive, such as 'greetings' and 'gesticulate', were labeled as high arousal and positive valence, whereas more strenuous or repetitive tasks like 'lift' were labeled as low arousal and negative valence.

### A.1.14 LAUREATE

The LAUREATE dataset (Laporte et al. [2023]) is a comprehensive multimodal dataset designed to facilitate research into the relationship between physiological responses, affective states, and academic performance in real-world educational settings. It comprises physiological data collected from 42 students and 2 lecturers over a 13-week university semester, encompassing 52 sessions that include classes, quizzes, and exams. Participants wore Empatica E4 wristband devices to record physiological signals such as electrodermal activity (EDA), photoplethysmography (PPG), skin temperature, and acceleration signal data. Additionally, daily post-lecture self-reports were gathered using the PANAVA-KS scale and additional custom-designed questions to capture information on lifestyle habits (e.g., study hours, physical activity, sleep quality), perceived engagement, attention, and emotional states of the students. Similarly, the lecturers' post-class survey included similar questions. For our study, we focus on EDA and PPG data of students and lecturers collected during classes/lecture sessions. The survey items for both students and lecturers included assessments of lecture engagement, such as enthusiasm, motivation, stress, tiredness, peacefulness, happiness, and calmness. These self-reported measures were used to compute composite scores for arousal and valence during lectures. The composite scores were then utilized to determine arousal and valence classes for each physiological data segment collected during lectures.

### A.1.15 ForDigitStress

The ForDigitStress dataset (Heimerl et al. [2023], Becker et al. [2023]) is a multimodal dataset designed to facilitate automatic stress recognition. It comprises data from 38 participants (originally 40 participants, but we could not find self-report files for 2 participants) who engaged in simulated digital job interviews, a scenario chosen to elicit psychosocial stress in a controlled yet realistic environment. Each session included a preparatory phase, an interview session conducted via video call, and a post-interview assessment. During the interviews, participants were subjected to challenging questions regarding their strengths and weaknesses, salary expectations, and hypothetical job-related situations, aimed at inducing stress. The dataset encompasses multiple modalities, including audio recordings, video data capturing facial expressions and body movements, eye-tracking information, and physiological signals, including PPG and EDA signals. To annotate the data, participants provided self-reports on their stress levels and emotions experienced during the interviews. Furthermore, two trained psychologists conducted frame-by-frame annotations of stress and emotions such as shame, anger, anxiety, and surprise, with high inter-rater reliability (Cohen's k > 0.7). For our study, we

focus on EDA and PPG data and binary arousal and valence annotations provided by experts after analyzing self-reports and participants' behavior.

### A.1.16 Dapper

The DAPPER dataset (Shui et al. [2021]) is a comprehensive multimodal dataset designed to study emotional experiences in real-world settings. It includes data from 142 participants, with 88 of them providing physiological recordings over five consecutive days. Participants wore custom-designed wrist-worn devices to collect physiological signals such as heart rate (via photoplethysmography), galvanic skin response (GSR), and three-axis acceleration during daytime hours. To capture psychological states, the study employed both the Experience Sampling Method (ESM) and the Day Reconstruction Method (DRM). The ESM involved prompting participants six times daily to report their momentary emotional states, while the DRM required them to recall and describe at least six significant events each day, providing associated emotional ratings. This dual-method approach offered a nuanced view of participants' emotional state in their natural environments. In our experiments, we focused on participants' self-reported arousal and valence levels obtained through ESM. To align physiological data with these self-reports, we extracted GSR and PPG signals recorded within a specific time window, two hours preceding and fifteen minutes following each ESM prompt. This approach enabled us to examine the temporal relationship between physiological responses and reported emotional states in real-world settings.

### A.1.17 ADARP

The ADARP (Alcohol and Drug Abuse Research Program) dataset (Sah et al. [2022], Alinia et al. [2021], Sah et al. [2022]) is a comprehensive multimodal dataset developed to facilitate research on stress detection and alcohol relapse quantification in real-world settings. It encompasses data from 11 individuals (10 females) diagnosed with alcohol use disorder (AUD), collected through a combination of physiological monitoring, self-reported assessments, and structured interviews. Participants in the study wore Empatica E4 wristbands, which continuously recorded physiological signals. In parallel, participants completed ecological momentary assessments (EMA) four times daily over a period of up to 14 days. These EMA surveys captured self-reported data on emotions, including stress, feeling overwhelmed, and anxiety, using the Positive and Negative Affect Schedule (PANAS) scale. For our study, we focused on the EDA and PPG data and performed binning based on the self-reports provided in the dataset. Segments where participants reported no experiences of stress, feeling overwhelmed, or anxiety were treated as positive valence, while all other segments were treated as negative valence. Regarding arousal, the presence of anxiety was classified as high arousal, whereas reports of stress and feeling overwhelmed were classified as low arousal to balance the dataset. We labeled the physiological data segments within a specific time window: two hours preceding and fifteen minutes following each EMA prompt. This dataset exhibited a significant class imbalance due to all participants being diagnosed with Alcohol Use Disorder (AUD), which led to a predominance of negative affective states such as stress, anxiety, and feeling overwhelmed in the self-reports.

### A.1.18 MOCAS

The MOCAS (Jo et al. [2024]) dataset is a comprehensive resource to facilitate research on human cognitive workload (CWL) assessment in real-world settings. Unlike existing datasets that rely on virtual game stimuli, MOCAS data were collected from realistic closed-circuit television (CCTV) monitoring tasks, enhancing its applicability to practical scenarios. The dataset comprises data from 21 human subjects who performed simultaneous tasks while monitoring CCTV footage. An Empatica E4 wearable watch, Emotive Insight, and a webcam were used to collect data. Physiological signals, including electroencephalography (EEG), BVP, EDA, Skin Temperature, and Accelerometer data, alongside behavioral features such as facial expressions, eye movements, and mouse activity. After each task, participants reported their CWL by completing the NASA-Task Load Index (NASA-TLX) and Instantaneous Self-Assessment (ISA). Additionally, arousal and valence were self-reported from the Self-Assessment Manikin (SAM) scale. We directly used the SAM scale rating for our labeling by categorizing them into two classes, alongside and EDA and PPG signal data segments.

### A.1.19 Exercise

The Exercise dataset (Hongn et al. [2025]) provides a comprehensive collection of non-invasive physiological data aimed at advancing research in stress detection and physical activity classification. Data were recorded using the Empatica E4 wearable device, which captures electrodermal activity (EDA), skin temperature, three-axis accelerometry, and blood volume pulse (BVP). The dataset encompasses records from 36 healthy individuals during a structured stress induction protocol, 30 during aerobic exercise, and 31 during anaerobic exercise. The stress induction protocol involved Stroop Test, Trier Mental Challenge Test which included mathematical tasks (with annoying background audio), vocalize their opinion about for and against a controversial topics, and counting backward from 1022 in decrements of 13, each designed to elicit negative physiological responses, while a stationary cycling routine was developed to distinguish between aerobic and anaerobic activities where anaerobic activity has cool-down periods in between cycling sprints and aerobic has increasing resistance cycling with cool-down at the end. For this study, we used EDA and PPG data, and we labeled the physiological data based on the specific tasks performed and the associated stress levels. For the stress induction protocol, we have used stress self-reports where high stress scores were mapped to high arousal negative valence. In contrast, the low stress scores were labeled as low arousal and positive valence. Regarding the exercise sessions, for both aerobic and anaerobic activities, an initial baseline, warm-up, and later cool-down, and rest are labeled as low arousal, while the cycling period was treated as high arousal. For the valence label in aerobic exercise data, speeds up to 85 rpm were taken as positive valence, while the rest of the data was taken as negative valence. For anaerobic exercise, the data of the initial two sprints is taken as positive valence, while later sprints are taken as negative valence.

### A.1.20 EEVR

The EEVR dataset (Singh et al. [2024]) is a multimodal dataset designed to advance emotion recognition research by integrating physiological signals with textual descriptions of emotional experiences. It includes data from 37 participants who were exposed to various emotional stimuli presented through 360° virtual reality (VR) videos. The physiological signals, including electrodermal activity (EDA) and photoplethysmography (PPG), were recorded using a 4-channel Biopac MP36. The dataset encompasses a range of emotional experiences, covering all four quadrants of Russell's circumplex model of emotion. To facilitate emotion classification tasks, the dataset includes annotations for valence and arousal, as well as individual emotions, collected using the SAM and PANAS surveys, along with self-reported textual descriptions of emotions gathered through qualitative interviews. For our experiments, we have directly used the pre-trained models (available here) provided by the authors, which were trained on the EEVR datasets.

### A.2 Meta Analysis

In this section, we present our meta-analysis that included computing the number of high and low arousal and positive and negative valence samples in each dataset to assess class balance and artifact percentages in our datasets. This was done as part of our qualitative analysis of benchmarking performances to understand if these factors could have impacted the overall performance. We begin by explaining the definition of artifacts as per prior literature, their impact on data quality and performance, and then we explain our artifact detection methods. Finally, we presented our comprehensive dataset-wise analysis in Table 4.

**Artifact Detection**

**Definitions: Artifacts in biosignals** (EDA, PPG) are "non-avoidable distortions which get superimposed on the signals representing emotional changes that originate from external (e.g., movement, ambient light) or internal (e.g., electrode-skin impedance changes) sources" ( Islam et al. [2020]). **Artifacts in electrodermal activity (EDA)** are "transient, non-sweat-gland–related perturbations of the skin conductance signal that do not reflect sympathetic nervous system activity. Common sources of these perturbations are abrupt motions, unstable electrode contact, or environmental factors like humidity or temperature fluctuations Gashi et al. [2020]." Similarly, **photoplethysmography (PPG) artifacts** are "distortions in the optical pulse waveform not caused by pulsatile blood-volume changes, degrading the actual cardiac-related component. Limb motion and sensor contact pressure are significant sources of these distortions (Islam et al. [2020])."

**Impact of Artifacts:** Artifacts in EDA signals impact the quality of emotion-representation in the data by inflating noise relative to true sympathetic responses. In-the-wild studies show that including artifact-contaminated segments can reduce arousal–valence classification accuracy by up to 20 percentage points compared to manually cleaned data, directly undermining the reliability of affective state. Moreover, unfiltered motion and contact artifacts bias tonic–phasic decomposition methods, misestimating key features like mean SCR amplitude, thereby impacting machine-learning emotion classifiers that rely on these features ( Venkatachalam et al. [2011]). In PPG signals, motion artifacts interrupt the detection of valid inter-beat intervals, leading to heart-rate variability (HRV) metrics with mean absolute errors exceeding 30 ms and impacting arousal prediction models ( Zheng et al. [2023]).

### Artifact Detection Methods

**EDA Artifacts:** For our artifact detection, we have used the EDA artifact detection pipeline proposed in EDArtifact ( Gashi et al. [2020]), which involves a structured sequence of preprocessing, feature extraction, and classification. The raw EDA signals are initially sampled at 4 Hz and segmented into non-overlapping windows of 60 samples (equivalent to 15 seconds). Each segment undergoes Haar wavelet decomposition up to level 3 to capture multi-resolution signal characteristics. A comprehensive set of 36 features is extracted from each segment, encompassing statistical properties (e.g., mean, variance), first- and second-order derivative statistics, wavelet coefficients, and characteristics of skin conductance response (SCR) peaks. The SCR peaks are identified using a minimum amplitude threshold of 0.01 µS, with an onset validation offset of 1 sample, a pre-apex search window of 3 seconds, and a post-apex half-amplitude decay window of 10 seconds. The extracted features are then normalised and input into a pre-trained XGBoost classifier, which has been trained to distinguish between clean and artifact-contaminated segments.

**PPG Artifacts:** We used the Tiny-PPG motion artifact detection pipeline for PPG signals, leveraging a lightweight pre-trained 1d convolutional neural network designed for real-time deployment on edge devices ( Zheng et al. [2023]). Raw PPG time series data is loaded and, for each subject (PID), segmented into non-overlapping windows of 60 samples. Each window is reshaped to the model's expected input tensor shape [batch, channel, length] = [1, 1, 60]. Inference is performed window-wise, where each segment is passed through the model to obtain a segmentation mask representing the likelihood of motion artifact presence. A sigmoid activation is applied to convert logits into probabilities, followed by binarisation using a threshold of 0.5. The prediction for the entire signal is aggregated using a mean-based criterion. If the average probability of artifact presence exceeds 0.5, the segment is labelled as containing a motion artifact.

| Dataset Name | High Arousal | Low Arousal | Positive Valence | Negative Valence | EDA Artifacts (%) | PPG Artifacts (%) |
|---|---|---|---|---|---|---|
| WESAD | **120** | **120** | **120** | **120** | 11.34 ± 21.22 | **91.84 ± 7.56** |
| NURSE | 130 | 162 | 248 | 44 | 18.76 ± 14.99 | **100 ± 0** |
| EMOGNITION | 252 | 212 | 295 | 169 | 33.75 ± 36.92 | **94.09 ± 5.08** |
| UBFC_PHYS | 208 | 464 | **328** | **344** | 8.3 ± 19.18 | 0 ± 0 |
| VERBIO | 254 | 115 | 288 | 81 | 24.7 ± 33.37 | 53.45 ± 10.35 |
| PhyMER | 1036 | 1684 | 1128 | 1592 | 2.89 ± 10.09 | 0 ± 0 |
| EmoWear | 1950 | 1602 | 1216 | 2336 | 7.70 ± 14.03 | 0.03 ± 1.68 |
| MAUS | 352 | 176 | 352 | 176 | 0.64 ± 1.52 | **100 ± 0** |
| CLAS | **1560** | **1440** | **1560** | **1440** | 70.82 ± 27.5 | 0 ± 0 |
| CASE | 332 | 988 | 816 | 504 | **98.67 ± 1.9** | **100 ± 0** |
| Unobtrusive | 707 | 213 | 359 | 561 | 12.6 ± 4.5 | 3.80 ± 19.13 |
| CEAP-360VR | 98 | 158 | **135** | **121** | 13.41 ± 9.67 | 6.67 ± 3.5 |
| ScientISST MOVE | 30 | 60 | 31 | 59 | 59.81 ± 10.9 | **96.67 ± 17.95** |
| LAUREATE | 718 | 254 | 703 | 269 | 18.92 ± 18.42 | **92.01 ± 4.67** |
| ForDigitStress | 505 | 82 | 154 | 433 | 0 ± 0 | 11.83 ± 10.02 |
| Dapper | 2244 | 757 | 307 | 2694 | 7.28 ± 2.94 | 37.6 ± 2.48 |
| ADARP | 13 | 4 | 14 | 3 | 68.7 ± 14.47 | **100 ± 0** |
| MOCAS | **98** | **98** | 36 | 160 | 33.67 ± 10.3 | 31.9 ± 14.03 |
| Exercise | **427** | **455** | 347 | 535 | 50.55 ± 7.47 | **95.08 ± 1.17** |

Table 4: **Data quality statistics** across datasets. Bold values in the Arousal and Valence columns indicate **near-balanced class distributions** (min/max ≥ 0.9). Bold values in the artifact columns denote cases where over **90% of the data** across participants is affected by artifacts.

### Meta Analysis Observations

| Model | Data Type | FLOPS | Parameters | Latency (ms) |
|---|---|---|---|---|
| RF | EDA | - | - | 15.01 |
| | PPG | - | - | 14.78 |
| | EDA+PPG | - | - | 15.59 |
| LDA | EDA | 59 | 16 | 0.34 |
| | PPG | 143 | 37 | 0.38 |
| | EDA+PPG | 203 | 52 | 0.42 |
| HC+MLP | EDA | 3200 | 1701 | 0.33 |
| | PPG | 7400 | 3801 | 0.38 |
| | EDA+PPG | 10400 | 5301 | 0.43 |
| HC+RESNET | EDA | 6225029 | 414082 | 0.61 |
| | PPG | 14939525 | 414082 | 0.62 |
| | EDA+PPG | 21164165 | 414082 | 0.61 |
| HC+LSTM+NN | EDA | 3084037 | 265346 | 0.25 |
| | PPG | 7309573 | 265346 | 0.32 |
| | EDA+PPG | 10327813 | 265346 | 0.37 |
| HC+Attention+NN | EDA | 67717 | 68226 | 0.20 |
| | PPG | 70405 | 265346 | 0.21 |
| | EDA+PPG | 72325 | 72834 | 0.21 |
| Signal+CNN+Transformer | EDA | 5589888 | 2203086 | 1.45 |
| | PPG | 5589888 | 2203086 | 1.43 |
| | EDA+PPG | - | - | - |
| Signal+LSTM+NN | EDA | 12138757 | 265346 | 0.41 |
| | PPG | 12138757 | 265346 | 0.40 |
| | EDA+PPG | - | - | - |
| Signal+RESNET | EDA | 24898949 | 414082 | 0.60 |
| | PPG | 24898949 | 414082 | 0.61 |
| | EDA+PPG | - | - | - |
| CLSP | EDA | 850.51 | 66.0932 M | 3.8492 |
| | PPG | 850.51 | 66.0943 M | 3.8977 |
| | EDA+PPG | 850.51 | 66.095 M | 3.8676 |
| CLSP-Finetune | EDA | 3230.91 | 66.1247 M | 13.0432 |
| | PPG | 3230.91 | 66.1257 M | 12.9931 |
| | EDA+PPG | 3230.91 | 66.1265 M | 12.9105 |

Table 5: Comparison of model performance across data types (EDA, PPG, and EDA+PPG). Here, "M" denotes millions. Missing entries indicate that no experiment was conducted for those cases. For the Random Forest (RF) model, FLOPS and parameter counts are not directly applicable.

Artifact analysis was conducted prior to the pre-processing or standardization step. The results are presented in Table 4, where artifact presence is reported as the mean artifact percentage ± standard deviation across participants. Our artifact detection results showed that the datasets collected in real-life settings (such as Nurse, ADARP, and Laurate) or using physical stressors as an emotion elicitation task (like Exercise and Scientisst_MOVE) tend to exhibit more EDA artifacts and PPG motion artifacts. We further observed that datasets including WESAD and CLAS have a balanced class distribution across arousal and valence, whereas datasets like ForDigitStress, Nurse, and VERBIO show a high imbalance, skewing toward negative samples. Moreover, we identified that data collected "in the wild" settings generally featured more negative-valence instances, while video-based elicitation datasets have a higher proportion of high-arousal samples. Further analysis is added in Table 12.

## A.3 Computation Cost

In this section, we present our computation cost across all modeling paradigms; see Table 5 for details.

## A.4    Benchmarking Models

In this section, we present a detailed discussion of the feature extraction, model architecture, and hyperparameters of the models employed in our benchmarking experiments. All models were initialized with a seed value of 42. We selected all the model's hyperparameters based on hyperparameter tuning to identify the optimal configuration.

### A.4.1    Feature Extraction

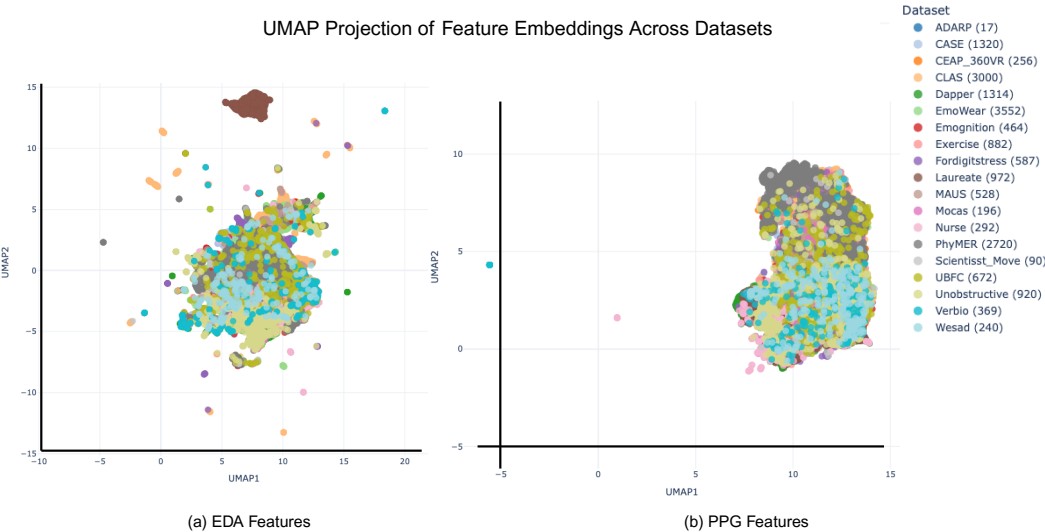

Figure 2: UMAP projection of Electrodermal Activity (EDA) and Photoplethysmography (PPG) signals across 19 physiological datasets. Each point represents EDA and PPG features from the dataset, color-coded by dataset source.

**EDA:** To extract meaningful statistical and physiological features from the EDA signal, we first decompose the raw signal using NeuroKit2's eda_phasic function, which separates it into tonic (slow-changing baseline) and phasic (fast, event-related) components. The phasic signal captures skin conductance responses (SCRs), while the tonic signal represents the baseline skin conductance level (SCL). Following this, the basic statistical features are computed from the raw EDA signal.

**PPG:** To extract physiological features from the PPG signal, we first iterate over each windowed segment in the raw_ppg_window dataset. Each segment signal is then cleaned using NeuroKit2's ppg_clean function to remove noise and artifacts. The cleaned PPG signal is processed using ppg_process, which extracts key features such as heart rate and waveform characteristics. Then, ppg_analyze is applied to compute PPG-specific features. Table 6 outlines the handcrafted statistical features chosen for training our models.

### A.4.2    ML Models

**Random Forest:** We used a Random Forest Classifier from the scikit-learn library with the primary hyperparameters set as n_estimators=100, and n_jobs=5, while keeping all other parameters at their default values.

**LDA:** We used a Linear Discriminant Analysis (LDA) model from the scikit-learn library with the hyperparameter n_components=1, solver = svd, while keeping all other parameters at their default values. The n_components=1 setting indicates that the model projects the input data onto a single linear discriminant axis, which is used because it is generally useful for binary classification problems and for reducing the feature space to a single dimension while preserving class separability.

| Signal | Features Selected |
|--------|-------------------|
| EDA | ku_eda, sk_eda, dynrange, slope, variance, entropy, insc, first_derivative_mean, max_scr, min_scr, nSCR, meanAmpSCR, meanRespSCR, sumAmpSCR, sumRespSCR |
| PPG | BPM, PPG_Rate_Mean, HRV_MedianNN, HRV_Prc20NN, HRV_MinNN, HRV_HTI, HRV_TINN, HRV_LF, HRV_VHF, HRV_LFn, HRV_HFn, HRV_LnHF, HRV_SD1SD2, HRV_CVI, HRV_PSS, HRV_PAS, HRV_PI, HRV_C1d, HRV_C1a, HRV_DFA_alpha1, HRV_MFDFA_alpha1_Width, HRV_MFDFA_alpha1_Peak, HRV_MFDFA_alpha1_Mean, HRV_MFDFA_alpha1_Max, HRV_MFDFA_alpha1_Delta, HRV_MFDFA_alpha1_Asymmetry, HRV_ApEn, HRV_ShanEn, HRV_FuzzyEn, HRV_MSEn, HRV_CMSEn, HRV_RCMSEn, HRV_CD, HRV_HFD, HRV_KFD, HRV_LZC |
| EDA+PPG | ku_eda, sk_eda, dynrange, slope, variance, entropy, insc, first_derivative_mean, max_scr, min_scr, nSCR, meanAmpSCR, meanRespSCR, sumAmpSCR, sumRespSCR, BPM, PPG_Rate_Mean, HRV_MedianNN, HRV_Prc20NN, HRV_MinNN, HRV_HTI, HRV_TINN, HRV_LF, HRV_VHF, HRV_LFn, HRV_HFn, HRV_LnHF, HRV_SD1SD2, HRV_CVI, HRV_PSS, HRV_PAS, HRV_PI, HRV_C1d, HRV_C1a, HRV_DFA_alpha1, HRV_MFDFA_alpha1_Width, HRV_MFDFA_alpha1_Peak, HRV_MFDFA_alpha1_Mean, HRV_MFDFA_alpha1_Max, HRV_MFDFA_alpha1_Delta, HRV_MFDFA_alpha1_Asymmetry, HRV_ApEn, HRV_ShanEn, HRV_FuzzyEn, HRV_MSEn, HRV_CMSEn, HRV_RCMSEn, HRV_CD, HRV_HFD, HRV_KFD, HRV_LZC |

Table 6: Handcrafted Features Selected for EDA, PPG, and Combined (EDA+PPG) Signals

### A.4.3 Handcrafted Features + DL Models

**MLP:** We trained a Multi-Layer Perceptron (MLP) classifier using scikit-learn's MLPClassifier with the hyperparameters hidden_layer_sizes=(100,) while leaving all other parameters at their default values such as the activation function (relu), solver (adam), learning rate strategy (constant), and maximum iterations (200).

**RESNET:** We used a 1D ResNet-based architecture tailored for temporal classification tasks. The core building block is a residual module comprising three sequential Conv1D layers with kernel sizes [8, 5, 3], each followed by BatchNorm and ReLU activations except final layer. A shortcut connection using a 1×1 Conv1D aligns input-output dimensions, enabling residual learning. The network stacks two such blocks, followed by an adaptive average pooling layer that compresses the temporal dimension. The pooled features are passed through a fully connected layer and softmax activation for classification. The model is optimized using Adam (lr=0.001) with cross-entropy loss and trained end-to-end with mini-batch gradient descent. We ran for epoch 100 with a batch size of 16.

**LSTM+MLP:** We implement a sequence classification model based on Long Short-Term Memory (LSTM) networks, followed by a multi-layer perceptron (MLP). The model begins with a two-layer LSTM configured with a hidden size of 128 and a dropout rate of 0.3 to mitigate overfitting. The LSTM operates on univariate time series data and captures temporal dependencies in the input. The output from the final time step of the LSTM is passed through an MLP consisting of two fully connected layers. The first layer maps the hidden representation to 256 dimensions, applies a ReLU activation, and includes a dropout of 0.4. The second layer reduces the dimensionality to 128, again followed by ReLU and a dropout of 0.3. A final linear layer maps the features to the number of output classes, and a softmax activation is applied to obtain class probabilities. The model is trained using the Adam optimizer with a learning rate of 0.001 and cross-entropy loss. Training is conducted over 100 epochs with a batch size of 16, using shuffled data and GPU acceleration when available.

**Attention Layer + MLP:** We implement an attention-based neural network classifier designed to operate directly on handcrafted statistical features extracted from physiological signals. These features, computed as summary statistics from time-series data, are treated as a single flat input vector without any temporal or sequential structure. The input vector is first projected into a 128-dimensional representation using a linear layer. A 4-head self-attention mechanism is then applied to model inter-feature dependencies, allowing the model to dynamically weight the contribution of different features during learning. Unlike a full Transformer, our approach does not involve positional encoding or stacked attention layers; rather, it uses a single self-attention block to enhance feature interactions before passing the output through a two-layer MLP with dimensions of 256 and 128, incorporating dropout rates of 0.4 and 0.3, respectively. The final output is produced via a softmax layer, and the model is trained using the Adam optimizer with a learning rate of 0.001 and cross-entropy loss, processing data in batches of 16 samples over 100 epochs.

### A.4.4 Signal Segments + DL Models

**Resnet:** We implemented a deep residual convolutional neural network tailored for classifying physiological time-series signals. First, the raw signals were segmented into fixed-length overlapping windows using a sliding window approach. To ensure all windows had consistent length, we applied zero-padding to the right when segments were shorter than the desired size. These segments were then converted into padded tensors suitable for batched input to the model. Our network comprises two stacked ResNetBlock modules, each engineered to extract hierarchical temporal features. Within each block, the input passes through a sequence of three 1D convolution layers with kernel sizes of 8, 5, and 3, respectively. Each convolution uses 'same' padding to preserve the temporal dimension, followed by batch normalization and ReLU activation to improve training stability and non-linearity. Each block includes a shortcut connection implemented via a 1×1 convolution to facilitate gradient flow. The number of filters is fixed at 128 throughout the network, allowing for rich intermediate representations. After the convolution layers, an adaptive average pooling layer was included to reduce the output to a fixed-length vector, regardless of the input window size. This vector is passed through a fully connected linear layer for classification, followed by a softmax activation to produce probability distributions over the target classes. The model is trained using the Adam optimizer with a learning rate of 0.001, and cross-entropy loss is used to guide optimization. We ran for epoch 100 with a batch size of 16.

**LSTM+MLP:** We implemented a hybrid LSTM-MLP classifier to model temporal dependencies in univariate physiological time-series data. Prior to modeling, we segmented each signal into overlapping fixed-size windows using a sliding window mechanism. This ensured that even signals of varying lengths could be represented as uniform tensors, with shorter segments padded using zeros. Each windowed sequence was treated as a 1D time series and passed to a multi-layer LSTM module consisting of two stacked layers, each with 128 hidden units and dropout regularization to reduce overfitting. The final hidden state from the LSTM was used as a condensed summary of the temporal dynamics within each window. This representation was then passed through a multi-layer perceptron (MLP) with two hidden layers (256 and 128 units), ReLU activations, and dropout layers. The final classification was performed using a fully connected layer followed by a softmax activation to produce class probabilities. The model was optimized using the Adam optimizer with a learning rate of 0.001 and trained using the cross-entropy loss. We ran for epoch 100 with a batch size of 8.

**CNN+ Transformer Encoder Block:** We implemented a hybrid neural architecture combining a convolutional and a Transformer encoder block for 1D physiological signals. The Feature Extractor module uses three parallel 1D convolutional blocks, each with kernel sizes of 5, 9, and 13, respectively, to capture temporal patterns at multiple receptive fields. Each block consists of two convolutional layers: the first maps from 1 input channel to 32 filters, and the second expands from 32 to 64 filters, both with padding="same". Each convolutional layer is followed by batch normalization, ReLU activation, and dropout (rate = 0.2). Outputs from all three branches are concatenated, resulting in a feature map with 192 channels (3 blocks × 64 filters). To adaptively reweight these feature channels, we incorporated a Squeeze-and-Excitation (SE) block with a reduction ratio of 16. This mechanism reduces the channel dimensionality from 192 to 12 via a fully connected layer, then projects it back to 192 using a second linear layer followed by a sigmoid activation, generating channel-wise attention weights. The aggregated feature representation is passed through a global average pooling layer and projected into a 128-dimensional embedding space using a linear layer followed by Layer Normalization and dropout (rate = 0.1). We employed a Transformer encoder with 4 layers, each using 8 attention heads, a model dimension (d_model) of 128, and feedforward sublayers with hidden dimensions of 512. The encoder is preceded by sinusoidal positional encodings added to the input sequence to provide temporal order information. Each encoder layer includes multi-head self-attention, residual connections, layer normalization, and a dropout rate of 0.1. The Transformer output is globally pooled (mean over sequence dimension) and passed to a two-layer MLP classifier: the first layer maps from 128 to 64 units with ReLU and dropout (0.1), and the second maps from 64 to the two emotion classes. The entire architecture is trained using the Adam optimizer with a learning rate of 0.001 and cross-entropy loss, over 50 epochs with a batch size of 16.

### A.4.5 Fine-tuned CLSP Models

**MLP-based MetaNet:** For the linear variant of the MetaNet modulation network, we employ a simple yet effective two-layer multilayer perceptron (MLP) to generate instance-conditioned prompts. The network first projects the input signal features into a hidden representation of dimension 32

using a fully connected layer, followed by a ReLU nonlinearity. This intermediate representation is then passed through a second linear layer to produce output vectors in the same dimensionality as the CLSP text embedding space. These outputs are reshaped into instance-specific bias vectors and added to a set of **16 learnable context tokens**, resulting in dynamically modulated prompts tailored to each input sample. The overall system is optimized using Adam optimizer with a batch size of 4 and a learning rate of 5e-5 for 15 epoch, enabling efficient fine-tuning. This design facilitates task adaptation by conditioning the text encoder on input signal characteristics, without requiring access to class labels or ground-truth text data during training.

**1D-CNN-based MetaNet:** For training our model based on 1D-CNN metanet adoption of CoCoOp, we used a batch size of 4 and optimized with the Adam optimizer using a learning rate of 5e-5 for 15 epochs. For prompt learning, we set the number of learnable context tokens to 24, each embedded in a 768-dimensional space aligned with the text encoder. The Meta network comprises two stacked 1D convolutional layers. The first convolution uses an input channel of 1 and outputs 24 hidden channels (with kernel size 3, stride 1, and padding 1), followed by a ReLU activation, and a second convolution compresses the representation back to a single output (with kernel size of 3 and padding 1) channel, maintaining the temporal dimension. This network transforms the signal features into instance-specific context bias vectors, which are added to a set of **24 learnable context tokens**, forming dynamic prompts. These prompts are prepended to tokenized class descriptions and passed to a frozen DistilBERT encoder, enabling adaptive conditioning of text embeddings based solely on input signals. This 1D-CNN modulation strategy provides a computationally efficient and effective means of aligning physiological data with textual semantics in the absence of ground-truth annotations.

For our fine-tuned CLSP experiments, we employed a set of carefully constructed textual prompts corresponding to each emotional category. These prompts were designed to provide richer semantic grounding for the text encoder, enabling the model to better capture the conceptual meaning of each class even in the absence of explicit textual supervision. Each prompt describes general physiological and affective cues, such as variations in energy or bodily reactions, that are broadly recognized across cultures, thereby reflecting universal aspects of emotional experience rather than culture-specific expressions. Importantly, our approach also extends beyond static textual class definitions, since in our fine-tuning approach, each prompt is augmented with context tokens that are learned for every input segment. These adaptive tokens enable the model to dynamically refine the prompt representation based on the input, thereby mitigating potential rigidity and bias that may arise from the nature of textual prompts. The complete set of textual prompts used for our fine-tuning experiments is presented below:

**1) Textual Prompts for Arousal Classification**:

- **High Arousal**: *"The participant felt a strong physical reaction, like a racing heart or tense body, and experienced high-energy emotions such as excitement, enthusiasm, surprise, anger, and nervousness."*
- **Low Arousal**: *"The participant felt low energy and relaxed, with calm emotions like peacefulness, relaxation, neutral, boredom, and lack of interest."*

**2) Textual Prompts for Valence Classification**:

- **Negative Valence**: *"The participant felt bad and was in a negative mood, with emotions like sadness, fear, anger, worry, hopelessness, and frustration."*
- **Positive Valence**: *"The participant experienced a positive mood characterized by emotions such as happiness, joy, gratitude, serenity, interest, hope, pride, amusement, inspiration, awe, and love."*

**3) Textual Prompts for Four Class Classification**:

- **High Arousal Negative Valence**: *"Strong physical reaction with intense negative emotions like anger, fear, frustration, anxiety, or panic."*
- **High Arousal Positive Valence**: *"Strong physical activation with energizing positive emotions like joy, enthusiasm, exhilaration, or amusement."*
- **Low Arousal Negative Valence**: *"Low energy with subdued negative emotions like sadness, boredom, tiredness, disappointment, or hopelessness."*

- **Low Arousal Positive Valence**: *"Calm and relaxed with subtle positive emotions like contentment, peace, satisfaction, and mild happiness."*

Table 7: Dataset Categorization by Experimental Setting

| Setting Group | Datasets |
| --- | --- |
| Lab | WESAD, EMOGNITION, UBFC_PHYS, VERBIO, PhyMER, EmoWear, CEAP-360VR, CASE, MOCAS, MAUS, CLAS |
| Constraint | ForDigitStress, ScientISST MOVE, Exercise |
| Lab+Real | Unobtrusive |
| Real | ADARP, Dapper, NURSE, LAUREATE |

Table 8: Dataset Categorization by Device Type

| Device Group | Datasets |
| --- | --- |
| Wearable (Empatica E4) | WESAD, NURSE, EMOGNITION, UBFC_PHYS, VERBIO, PhyMER, EmoWear, Unobtrusive, CEAP-360VR, ScientISST MOVE, LAUREATE, MOCAS, Exercise, ADARP |
| Lab Based Device | CASE (ThoughtTech SA9309M, ThoughtTech SA9308M) CLAS (Shimmer3 GSR+ Unit) MAUS (Procomp Infinit) ForDigitStress (IOMbiofeedback sensor) |
| Custom Wearable | Dapper (Custom Designed Wristband) |

## A.5 Cross-Dataset Analysis

The cross-data models were run using the same set of parameters as in the benchmarking stage, with the seed of 42 and the same number of epochs as defined model-wise in A.4.

## A.6 Dataset Grouping

The dataset grouping across different harmonizing dimensions - Experimental Setting, Device Type, and Labeling Method is added in Table 7, 8, and 9. Furthermore, to evaluate gender-based transferability, we selected nine datasets containing gender metadata: WESAD, ScientISST MOVE, UBFC_PHYS, Exercise, PhyMER, EmoWear, CASE, CEAP-360VR, and NURSE (female subjects only). For age-based transferability analysis, we employed seven datasets: WESAD, ScientISST MOVE, Exercise, PhyMER, EmoWear, CASE, and CEAP-360VR.

Table 9: Dataset Categorization by Labeling Method

| Label Group | Datasets |
| --- | --- |
| Stimulus-Label | WESAD, EMOGNITION, UBFC_PHYS, MAUS, CLAS, ScientISST MOVE, Exercise |
| Self-report | VERBIO, PhyMER, EmoWear, CASE, Unobtrusive, NURSE, CEAP-360VR, LAUREATE, Dapper, ADARP, MOCAS |
| Expert-Annotated | ForDigitStress |

## A.7 Results Discussion

### A.7.1 Benchmarking Results

Overall, our benchmarking results reveal several key insights for the physiological signal–based emotion recognition community. Below, we summarize our reflections:

- **Evaluating Meaningful Performance**: In our benchmarking experiments, we observed that datasets collected under well-designed laboratory conditions, such as ForDigitStress, WESAD, and MOCAS, tend to yield high classification performance. This is likely due to the high degree of experimental control during data collection, which ensures consistent labeling and minimizes noise. However, it is important to note that model performance on such datasets may not accurately reflect real-world behavior. We encourage the emotion

recognition and affective computing community to complement lab-based evaluations with studies using more naturalistic, diverse, and ecologically valid datasets. Doing so will enable the development of models that generalize better to everyday contexts and support fairer and more meaningful comparisons across methods.

- **Variability in Valence and Arousal Performance**: Across datasets, we observed that arousal classification often outperformed valence and vice versa in some cases, while four-class emotion classification remained relatively low. This highlights that physiological responses may reflect certain dimensions of emotional states (e.g., arousal) more reliably than others, and this might vary as per the experiment settings and labeling techniques used, but it also reflects that our inferences are heavily dependent on how we interpret these signals. It raises fundamental questions about whether current approaches to labeling and benchmarking truly capture the nuances of emotion (Singh et al. [2025b]). For benchmarking, we mapped diverse emotion labels into arousal and valence dimensions, but this simplification may obscure important distinctions and assumptions about what physiological changes represent. These observations suggest a need for more precise labeling paradigms, better theoretical grounding, and careful consideration of whether our models and evaluation frameworks are aligned with the actual phenomena we aim to study.

- **Data Quality and Its Impact on Inference:** Our benchmarking results highlight that the quality of labels, task design, class balance, and stimulus strength all critically influence model performance and the reliability of inferences from physiological data. Datasets with expert annotations or experience sampling (e.g., ForDigitStress, Dapper) consistently outperform those relying on post-hoc or abstract self-reports, demonstrating the value of accurate and temporally aligned labels. Realistic, ecologically valid tasks (e.g., Unobtrusive, LAUREATE) elicit more meaningful physiological responses, supporting better generalization, while datasets with class imbalance or limited emotional diversity (e.g., NURSE, ADARP) hinder learning and reduce robustness. Similarly, weak or immersive-limited stimuli, such as VR or mild emotion elicitation (e.g., CEAP-360VR, EMOGNITION), lead to lower performance, highlighting the importance of strong and contextually relevant emotional triggers. Together, these findings suggest that our inferences from physiological signals depend not only on model architecture but fundamentally on the quality and design of the underlying data, emphasizing the need for balanced, realistic, and carefully labeled datasets to improve generalization and the interpretability of emotion recognition systems.

In summary, future research should integrate ecologically valid elicitation methods, temporally precise annotation, and balanced multimodal datasets to strengthen model robustness and real-world transferability. As illustrated in Table 12, our qualitative data-wise evaluation emphasizes that rigorous study design, balancing sample representation, participant diversity, ecological realism, and alignment of labels with physiological responses, is fundamental for building reliable emotion recognition systems.

### A.7.2   Cross-Dataset Generalization Analysis

We evaluate generalization across settings, annotation types, devices, and demographics to understand the robustness and transferability of our best-performing models. Below, we share our findings:

**1. Cross-Setting Generalization is Asymmetric but Promising**

- **Real → Lab/Constraint:** Models trained on real-world data generalized well to lab and constraint-based settings (F1 up to 0.79 for valence), particularly with CLSP-based models and EDA input.

- **Constraint → Real:** Constraint-trained models showed strong transferability to real-world data, achieving the highest overall F1 score (0.88) for valence with RF, indicating that well-structured elicitation with moderate variability helps bridge domain gaps.

- **Lab → Real/Constraint:** Lab-trained models yielded moderate transfer (F1 up to 0.76), but lacked consistency, possibly due to overfitting to controlled conditions that limit generalizability.

Detailed results for cross-setting are added in Table 14, 13, and visualized in figure 16.

**2. Cross-Label Generalization Benefits from High-Quality Annotations**

- **Expert-Annotated → Self/Stimulus:** Expert-annotated training led to strong generalization across label types (F1 up to 0.76 for valence), showing that high-quality temporal labels help bridge subjective and task-derived annotations.

- **Self-report → Expert/Stimulus:** Surprisingly strong performance was observed when self-report-trained models were tested on expert labels (F1 up to 0.87), suggesting alignment between subjective awareness and physiological signals when well-labeled.

- **Stimulus → Others:** Stimulus-labeled training underperformed when generalized to self-report or expert data (F1 as low as 0.52), likely due to weak alignment between stimuli and experienced emotions.

Detailed results for cross-label are added in Table 17, 18, and visualized figure 14.

### 3. Cross-Device Generalization is Strongest with High-Fidelity Sensors

- **Wearable → Custom:** Training on commercial wearable devices transferred well to custom wearables (F1 up to 0.82 for valence), especially using CLSP models, suggesting sensor fidelity supports robust feature learning.

- **Wearable → Lab-Based Device**: Transfer from Wearable (E4) to lab-based devices showed poor to moderate results (F1 mostly below 0.62), suggesting limited compatibility due to differing device characteristics and signal quality.

- **Lab-Based Device → Custom Wearable/Wearable:** Models trained on lab-based devices generalized well to both custom wearables and Wearable E4 devices, achieving strong valence performance (F1 up to 0.81 with LDA on EDA+PPG for custom wearables, and around 0.73 with CLSP CNN for Wearable E4), demonstrating that high-quality lab data can effectively support cross-device generalization using advanced models.

- **Custom → Lab/Wearable:** Models trained on custom devices generalized poorly to other hardware (F1 often < 0.66), likely due to noise, motion artifacts, and inconsistent signal quality.

Detailed results for cross-device are added in Table 15, 16, and visualized in figure 15.

**4. Influence of Demographics on Generalization:** Cross-demographic evaluations indicate strong transferability across gender and age for valence classification (F1 = 0.71 and 0.73, respectively). In contrast, arousal transfer remains weak, nearing random performance, implying that arousal-related physiological patterns are more susceptible to individual variability than those associated with valence. Detailed results for gender-wise transferability experiments are added in Tables 19, 20, and for age-wise transferability are added in Tables 21, 22.

In summary, cross-setting evaluations reveal that models trained on real-world data generalize well to lab and constraint environments, especially for valence detection. This suggests that diverse, in-the-wild datasets enhance model robustness. Similarly, training on constraint-based data transfers effectively to real-world scenarios, whereas models trained solely on lab data tend to overfit and show limited generalization due to the controlled nature of those settings. For Cross-Label evaluations, transferability between different annotation types is generally promising. Models trained on expert annotations perform well when tested on stimulus-based and self-reported labels, indicating common underlying emotional features across labeling methods. In contrast, models trained on self-reports transfer well to expert annotations but struggle to generalize within self-report targets themselves, reflecting the inherent subjectivity and complexity of self-reported emotions. Regarding Cross-Device evaluations, models trained on high-quality lab-based devices serve as strong foundations, demonstrating effective transfer to both custom and commercial wearable devices, particularly when leveraging advanced CLSP architectures and combined EDA+PPG inputs. However, models trained on custom wearable devices tend to generalize poorly due to sensor variability and noise. Encouragingly, zero-shot CLSP models exhibit robust, device-agnostic performance, underscoring their potential for flexible deployment across diverse hardware platforms. Overall, models trained across different settings, labels, and devices demonstrate varying degrees of generalization, with real-world data, expert annotations, and high-quality lab devices offering strong transfer potential. Notably, CLSP-based models consistently show robust performance across all evaluation scenarios, highlighting their adaptability and effectiveness for cross-domain affective computing applications.

## A.8 All Results

In this section, we present our detailed summary of our results for data benchmarking and cross-dataset experiments. Check out the supplementary material for more detailed results.

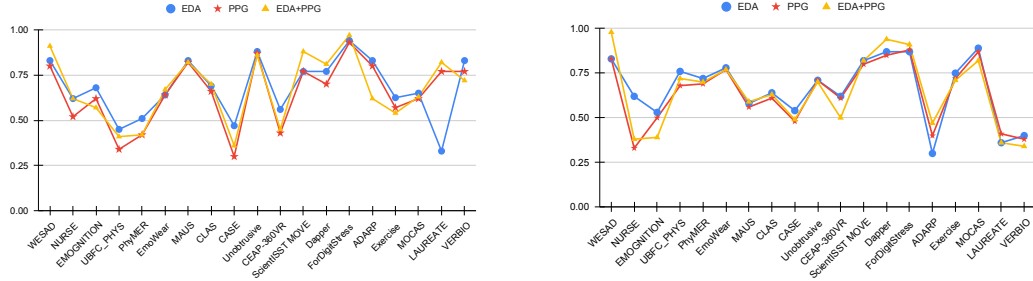

Figure 3: Comparative performance (F1 score) of the best-performing models per dataset across three physiological modalities (EDA, PPG, EDA+PPG) for emotion recognition. Each line represents a modality, showing how its top-performing model varies in effectiveness across the 19 datasets. Left: For arousal classification. Right: For valence classification.

### A.8.1 Benchmarking Results

In this section, we present the benchmarking results for valence, arousal, and four-class classification. Table 10 presents the best-performing models and their F1 scores for four-class classification across all datasets. We then show dataset-wise results across all 16 modeling paradigms in Figures 5, 6, 7, 8, 9, and 10 using radar plots for both arousal and valence classification across three modalities: EDA, PPG and EDA+PPG. In Figures 11, 12, and 13, we present dataset-wise performance visualizations to compare datasets and their relative positioning with respect to each other.

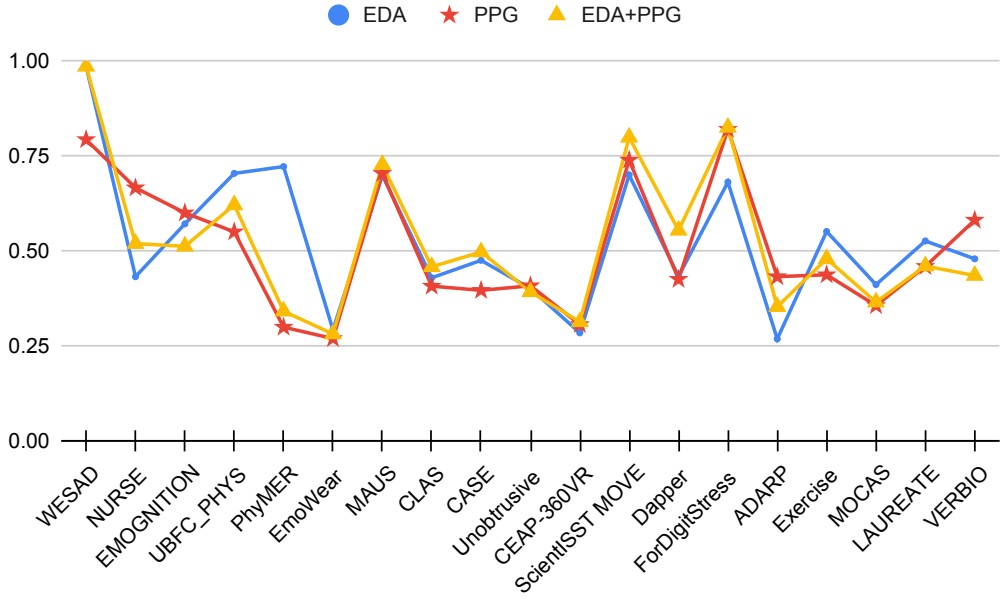

Figure 4: Comparative performance (F1 score) of the best-performing models for four-class classification per dataset across three physiological modalities (EDA, PPG, EDA+PPG) for emotion recognition. Each line represents a modality, showing how its top-performing model varies in effectiveness across the 19 datasets.

| Dataset | EDA | | PPG | | EDA+PPG | |
|---|---|---|---|---|---|---|
| | **Best Model** | **F1** | **Best Model** | **F1** | **Best Model** | **F1** |
| WESAD | RF | **0.987** | RF | 0.794 | LDA | **0.987** |
| NURSE | CLSP Zero Shot | 0.433 | CLSP Zero Shot | **0.667** | CLSP Zero Shot | 0.52 |
| EMOGNITION | RF | 0.572 | CLSP+CNN (50%) | **0.601** | RF | 0.513 |
| UBFC_PHYS | CLSP Zero Shot | **0.705** | LDA | 0.551 | LDA | 0.622 |
| PhyMER | CLSP+CNN (50%) | **0.723** | RF | 0.3 | RF | 0.342 |
| EmoWear | CLSP+CNN (50%) | **0.293** | HC+MLP | 0.27 | HC+MLP | 0.282 |
| MAUS | HC+MLP | 0.7 | RF | 0.705 | RF | **0.728** |
| CLAS | RF | 0.43 | HC+MLP | 0.408 | RF | **0.459** |
| CASE | RF | 0.476 | RF | 0.397 | RF | **0.498** |
| Unobtrusive | RF | 0.402 | CLSP Zero Shot | **0.409** | HC+MLP | 0.393 |
| CEAP-360VR | CLSP+MLP (25%) | 0.285 | RF | 0.307 | RF | **0.314** |
| ScientISST MOVE | CLSP+MLP (25%) | 0.701 | CLSP+CNN (50%) | 0.74 | CLSP+CNN (50%) | **0.8** |
| Dapper | RF | 0.434 | RF | 0.426 | RF | **0.555** |
| ForDigitStress | LDA | 0.682 | RF | 0.821 | RF | **0.826** |
| ADARP | CLSP Zero Shot | 0.269 | CLSP Zero Shot | **0.433** | CLSP Zero Shot | 0.354 |
| Exercise | CLSP+CNN (25%) | **0.552** | HC+MLP | 0.438 | RF | 0.48 |
| MOCAS | RF | **0.412** | RF | 0.357 | RF | 0.366 |
| LAUREATE | CLSP+MLP (5%) | **0.527** | RF | 0.46 | RF | 0.461 |
| VERBIO | CLSP Zero Shot | 0.48 | CLSP Zero Shot | **0.582** | CLSP Zero Shot | 0.436 |

Table 10: Best-performing model and corresponding F1 score for **four class classification** across all datasets and modalities (EDA, PPG, EDA+PPG). The table lists, for each dataset and modality, the model that achieved the highest F1 score.

| Statistic | Arousal | | | Valence | | |
|---|---|---|---|---|---|---|
| | **EDA** | **PPG** | **EDA+PPG** | **EDA** | **PPG** | **EDA+PPG** |
| MIN | 0.33 | 0.30 | 0.36 | 0.30 | 0.33 | 0.34 |
| MAX | 0.94 | 0.93 | 0.97 | 0.89 | 0.88 | 0.98 |
| AVG | 0.68 | 0.65 | 0.67 | 0.66 | 0.67 | 0.64 |
| STD | 0.16 | 0.18 | 0.18 | 0.18 | 0.18 | 0.20 |

| Rank | Arousal | | | Valence | | |
|---|---|---|---|---|---|---|
| | **EDA** | **PPG** | **EDA+PPG** | **EDA** | **PPG** | **EDA+PPG** |
| 1 | ForDigitStress | ForDigitStress | ForDigitStress | MOCAS | ForDigitStress | WESAD |
| 2 | Unobtrusive | Unobtrusive | WESAD | Dapper | MOCAS | Dapper |
| 3 | ADARP | MAUS | ScientISST MOVE | ForDigitStress | Dapper | ForDigitStress |
| 4 | MAUS | ADARP | Unobtrusive | WESAD | WESAD | MOCAS |
| 5 | VERBIO | WESAD | LAUREATE | ScientISST MOVE | ScientISST MOVE | ScientISST MOVE |
| 6 | WESAD | LAUREATE | MAUS | EmoWear | EmoWear | EmoWear |
| 7 | Dapper | ScientISST MOVE | Dapper | UBFC_PHYS | Exercise | UBFC_PHYS |
| 8 | ScientISST MOVE | VERBIO | VERBIO | Exercise | Unobtrusive | Exercise |
| 9 | CLAS | Dapper | CLAS | PhyMER | PhyMER | Unobtrusive |
| 10 | EMOGNITION | CLAS | EmoWear | Unobtrusive | UBFC_PHYS | PhyMER |
| 11 | MOCAS | EmoWear | MOCAS | CLAS | CEAP-360VR | CLAS |
| 12 | EmoWear | MOCAS | ADARP | CEAP-360VR | CLAS | MAUS |
| 13 | Exercise | EMOGNITION | NURSE | NURSE | MAUS | CEAP-360VR |
| 14 | NURSE | Exercise | EMOGNITION | MAUS | EMOGNITION | CASE |
| 15 | CEAP-360VR | NURSE | Exercise | CASE | CASE | ADARP |
| 16 | PhyMER | CEAP-360VR | CEAP-360VR | EMOGNITION | LAUREATE | EMOGNITION |
| 17 | CASE | PhyMER | PhyMER | VERBIO | ADARP | NURSE |
| 18 | UBFC_PHYS | UBFC_PHYS | UBFC_PHYS | LAUREATE | VERBIO | LAUREATE |
| 19 | LAUREATE | CASE | CASE | ADARP | NURSE | VERBIO |

Table 11: F1 score statistics (MIN, MAX, AVG, STD) and rankings of our 19 datasets according to their performance for arousal and valence prediction using EDA, PPG, and their combination (EDA+PPG).

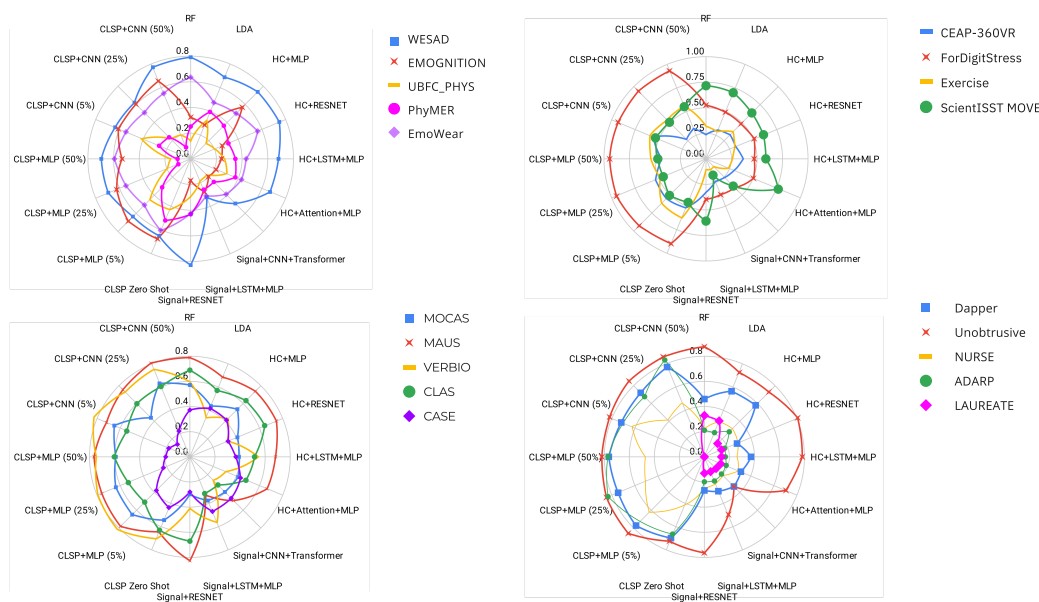

Figure 5: Benchmarking results for Arousal Classification on EDA signal Data across 19 datasets for 4 modeling paradigms and 16 model variants.

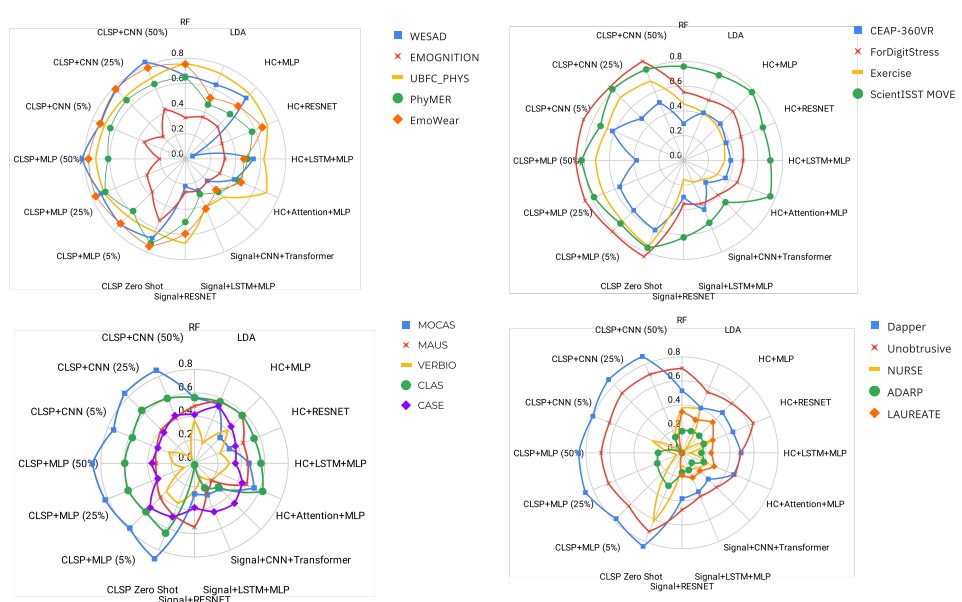

Figure 6: Benchmarking results for Valence Classification on EDA signal Data across 19 datasets for 4 modeling paradigms and 16 model variants.

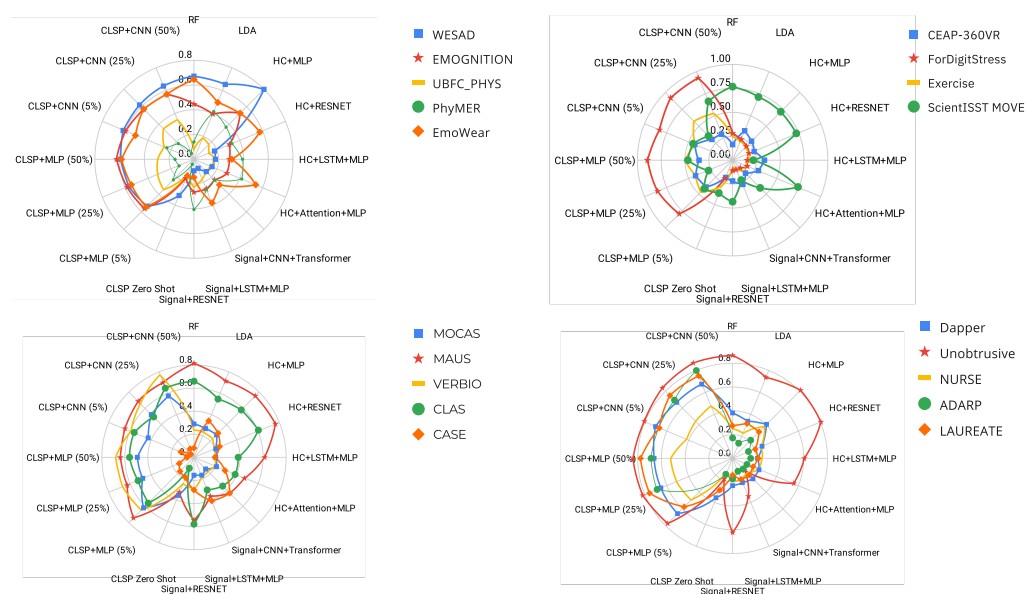

Figure 7: Benchmarking results for Arousal Classification on PPG signal Data across 19 datasets for 4 modeling paradigms and 16 model variants.

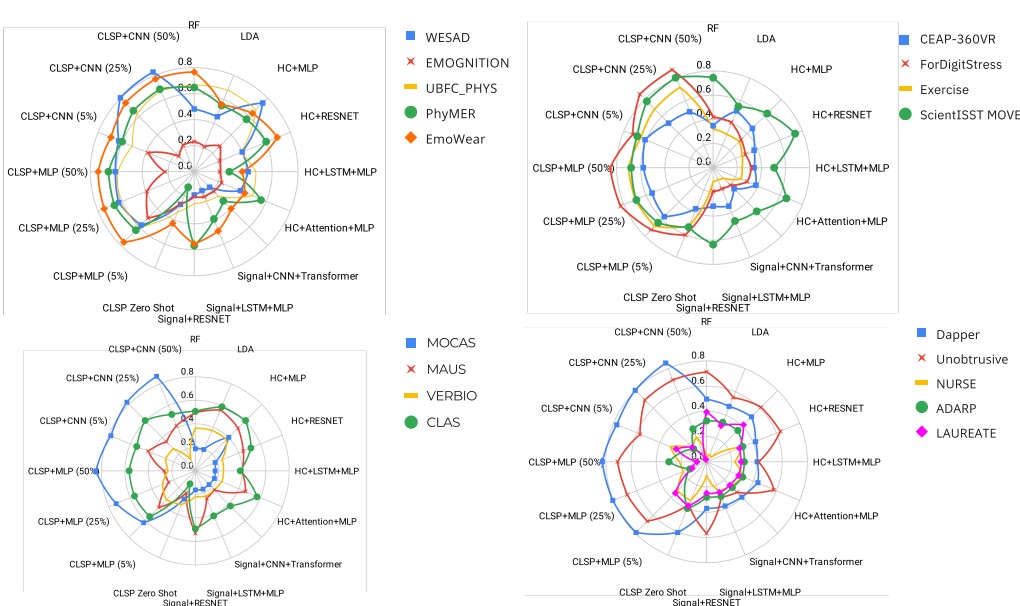

Figure 8: Benchmarking results for Valence Classification on PPG signal Data across 19 datasets for 4 modeling paradigms and 16 model variants.

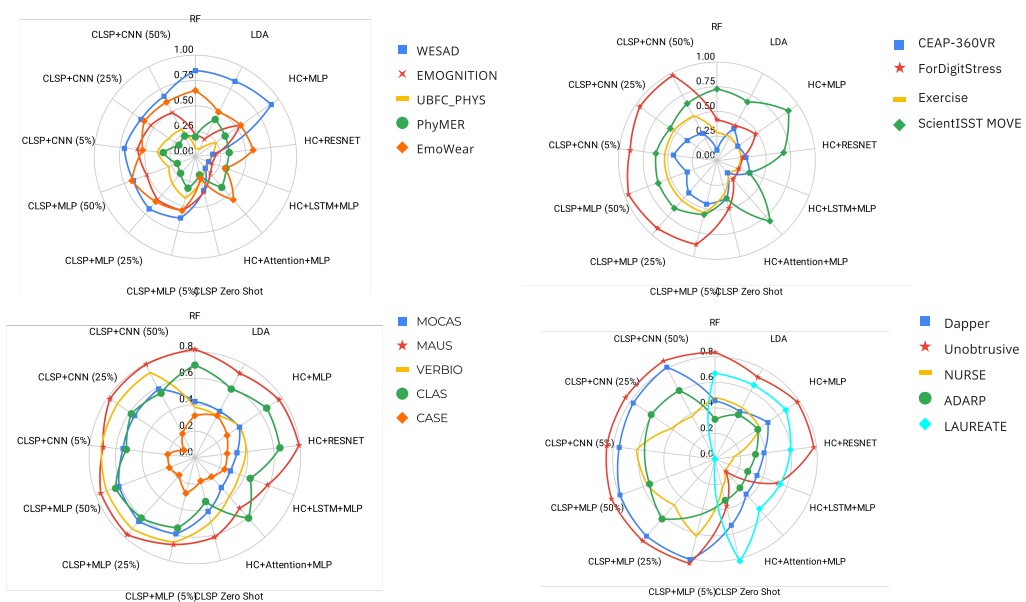

Figure 9: Benchmarking results for Arousal Classification on EDA+PPG Data across 19 datasets for 4 modeling paradigms and 16 model variants.

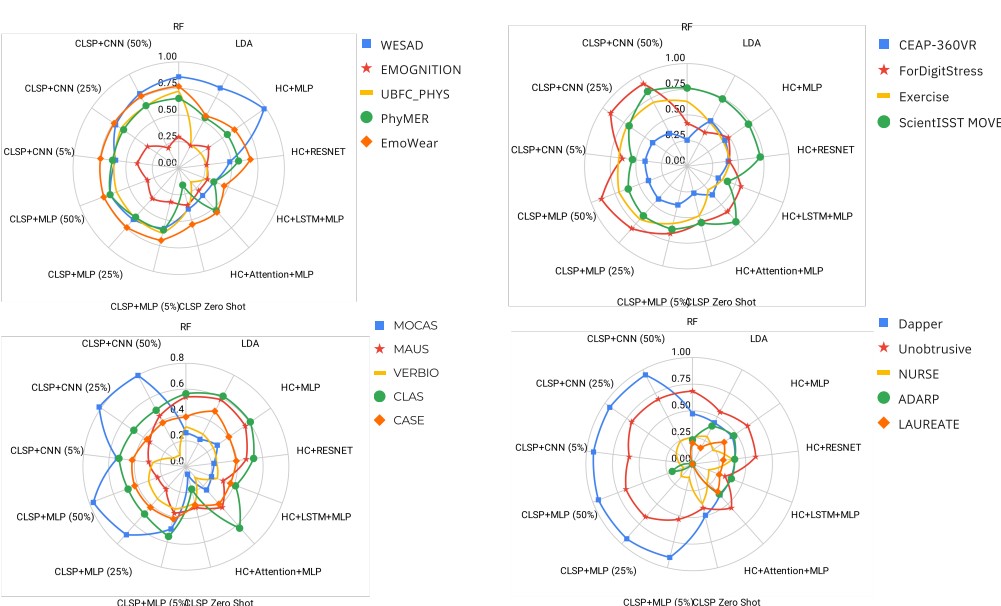

Figure 10: Benchmarking results for Valence Classification on EDA+PPG Data across 19 datasets for 4 modeling paradigms and 16 model variants.

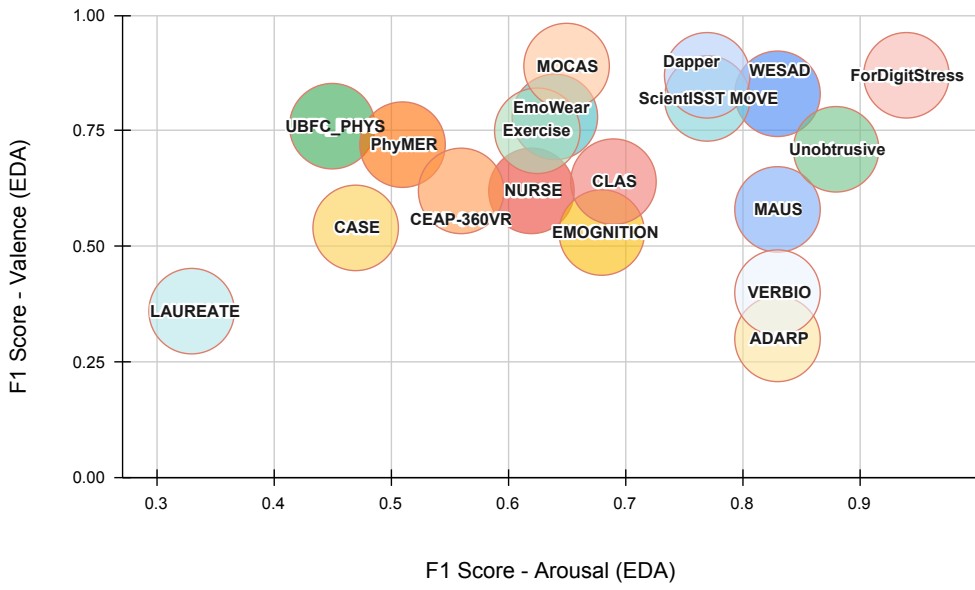

Figure 11: **Benchmarking results: EDA only.** This bubble plot illustrates the impact of EDA signals on F1 performance (best model) for arousal and valence classification across 19 datasets.

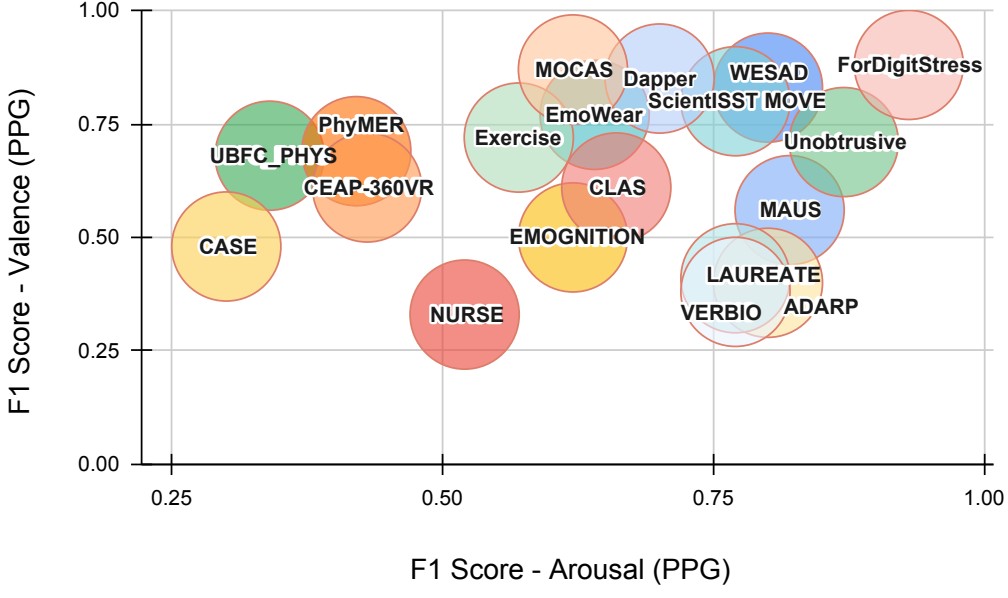

Figure 12: **Benchmarking results: PPG only.** This bubble plot illustrates the impact of PPG signals on F1 performance (best model) for arousal and valence classification across 19 datasets.

Table 12: Performance summary highlights the strengths and limitations of each dataset in supporting effective arousal and valence classification. We have divided the table based on the overall high-performing dataset and datasets with poor or contrasting performances across arousal and valence tasks.

| Dataset | Context and Elicitation Details | Performance Summary (with Interpretation) |
|---|---|---|
| **ForDigitStress** | Lab-based, semi-controlled digital interview tasks with expert annotation. | Top-performing for arousal and strong for valence, due to ecologically valid stressors and high-quality expert labels that align well with physiological responses. |
| **WESAD** | Lab setting with diverse elicitation (videos, stress tasks, meditation). | High performance for both stimulus labeled arousal and valence, particularly with EDA+PPG, attributed to the well-designed and emotionally rich elicitation protocol. |
| **MOCAS** | CCTV-based monitoring task with SAM self-reports. | Strong valence classification as participants found it easier to self-report valence; arousal performance was weaker due to less intense stimuli. |
| **ScientISST MOVE** | Physical tasks (e.g., handshake, jumping) with self-report. | Best arousal classification with EDA+PPG due to physical exertion eliciting strong signals; valence performance is also high despite motion artifacts. |
| **Unobtrusive** | Office-like cognitive tasks in lab and real-life with Likert-scale labels. | High performance in arousal classification, driven by realistic work scenarios; valence classification was moderate due to complexity of labeling subtle emotions. |
| **Dapper** | Real-life setting with ESM and custom wearable devices. | Strong valence classification, as ESM allowed timely and accurate self-reporting of naturally occurring emotions. |
| **MAUS** | N-Back cognitive task targeting mental workload. | Strong arousal classification, as task reliably induced measurable physiological changes linked to cognitive load. |
| **NURSE** | Real-world data from nurses, with reports focused on stressful events. | Poor performance for both arousal and valence, largely due to class imbalance and lack of positive emotion coverage. |
| **Emowear** | Lab-based audiovisual stimuli with self-report. | Weak overall performance; elicitation was too mild and may have led to participant bias due to the artificial lab setting. |
| **UBFC_PHYS** | Constrained speech and arithmetic tasks with stimulus labels. | Slightly better performance for valence, but overall limited by labeling that may not reflect true emotional states. |
| **VERBIO** | Public speaking in VR and real-world environments. | Arousal classification slightly better, as anxiety-inducing tasks aligned with arousal; valence performance limited by mild emotional variability. |
| **EMOGNITION** | Short film clips in lab setting with self-report. | Weak performance overall; limited emotional range in stimuli and potential response bias due to lab-based setting. |
| **CEAP-360VR** | VR video clips with self-report. | Valence classification marginally better, though weak stimuli and VR setting may not have triggered strong emotional variation. |
| **Exercise** | Lab-based cognitive and physical stress tasks with stimulus labels. | Valence was captured much better than arousal, possibly due to the mixed-task setup diluting arousal signals. |
| **PhyMER** | Lab-based video clips with self-reports. | Slightly better valence classification, though performance overall is limited by the lack of emotionally intense stimuli. |
| **CLAS** | Lab-based logic, math, video tasks with self-reports. | Near-random performance due to weak elicitation tasks and low-quality labels that failed to capture real emotion. |
| **CASE** | Lab-based video stimuli with joystick-based self-reporting. | Poor performance across both tasks; continuous labeling may have distracted participants and reduced emotional immersion. |
| **LAUREATE** | Real-life classroom setting with self-reports on engagement and attention. | Strong arousal classification with PPG and EDA+PPG, reflecting real-world physiological patterns; poor valence performance due to abstract labeling not directly tied to emotion. |
| **ADARP** | Real-life setting with AUD participants; self-report labels. | Strong arousal performance for EDA and PPG separately; the combined signal underperformed. Valence results were poor due to label imbalance (stress-heavy dataset). |

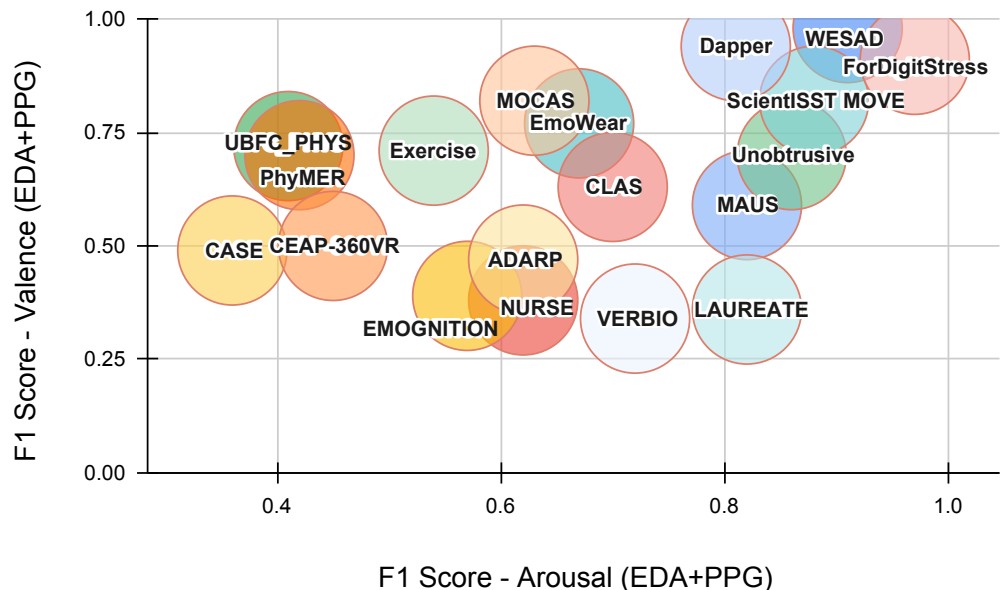

Figure 13: **Benchmarking results: EDA + PPG combined.** This bubble plot illustrates the impact of combining EDA and PPG signals on F1 performance (best model) for arousal and valence classification across 19 datasets.

### A.8.2 Cross-Data Analysis Results

In this section, we present our results for cross-dataset analysis as detailed in Tables 15, 16 for device dimension, Tables 13, 14 for setting dimension, and Tables 17, 18 for labeling dimension. We further visualized the impact of these dimensions on the EDA and PPG features using UMAP as shown in figures 14, 15, 16. Moreover, the summarized results for gender-wise transferability are added in Tables 19 and 20, and age-wise transferability is added in Tables 21 and 22.

| Testing Cohort | Training Cohort | EDA | | PPG | | EDA+PPG | |
|---|---|---|---|---|---|---|---|
| | | **Best Model** | **F1** | **Best Model** | **F1** | **Best Model** | **F1** |
| Lab | Real | **CLSP CNN 5%** | **0.72** | CLSP MLP 5% | 0.57 | **CLSP MLP 5%** | **0.71** |
| Lab | Constraint | CLSP MLP 50% | 0.56 | **RF** | **0.61** | RF | 0.60 |
| Lab | Lab | RF | 0.50 | RF | 0.50 | RF | 0.52 |
| Lab | CLSP ZeroShot | - | 0.58 | - | 0.15 | - | 0.29 |
| Constraint | Real | RF | 0.68 | RF | 0.51 | **CLSP MLP 5%** | **0.64** |
| Constraint | Lab | HC+MLP | 0.44 | **LDA** | **0.67** | **LDA** | **0.64** |
| Constraint | Constraint | HC+MLP | 0.48 | RF | 0.48 | RF | 0.48 |
| Constraint | CLSP ZeroShot | - | **0.74** | - | 0.27 | - | 0.40 |
| Real | Constraint | **CLSP MLP 5%** | **0.65** | RF | 0.59 | **CLSP MLP 5%** | **0.73** |
| Real | Lab | HC+MLP | 0.59 | **LDA** | **0.69** | CLSP MLP 25% | 0.72 |
| Real | Real | HC+MLP | 0.49 | RF | 0.48 | RF | 0.46 |
| Real | CLSP ZeroShot | - | **0.65** | - | 0.31 | - | 0.52 |

Table 13: For each data-collection setting category (Lab, Constraint, and Real) and modality (EDA, PPG, and EDA+PPG), the table identifies the top-performing model and its F1 score for **arousal classification**.

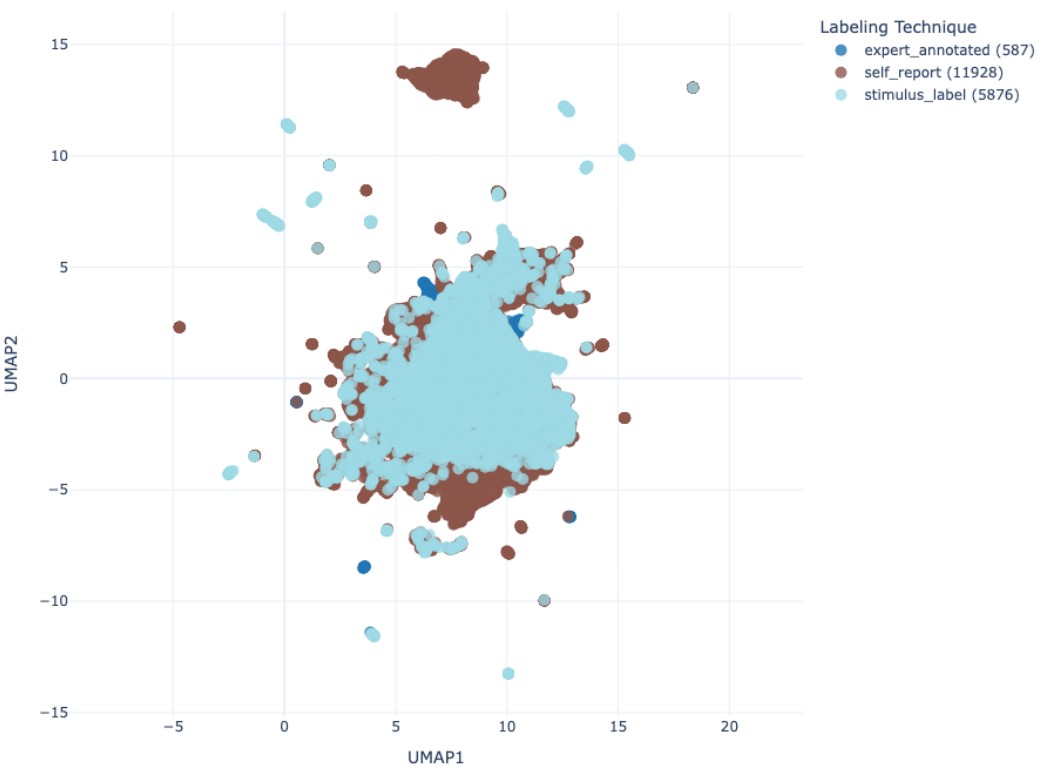

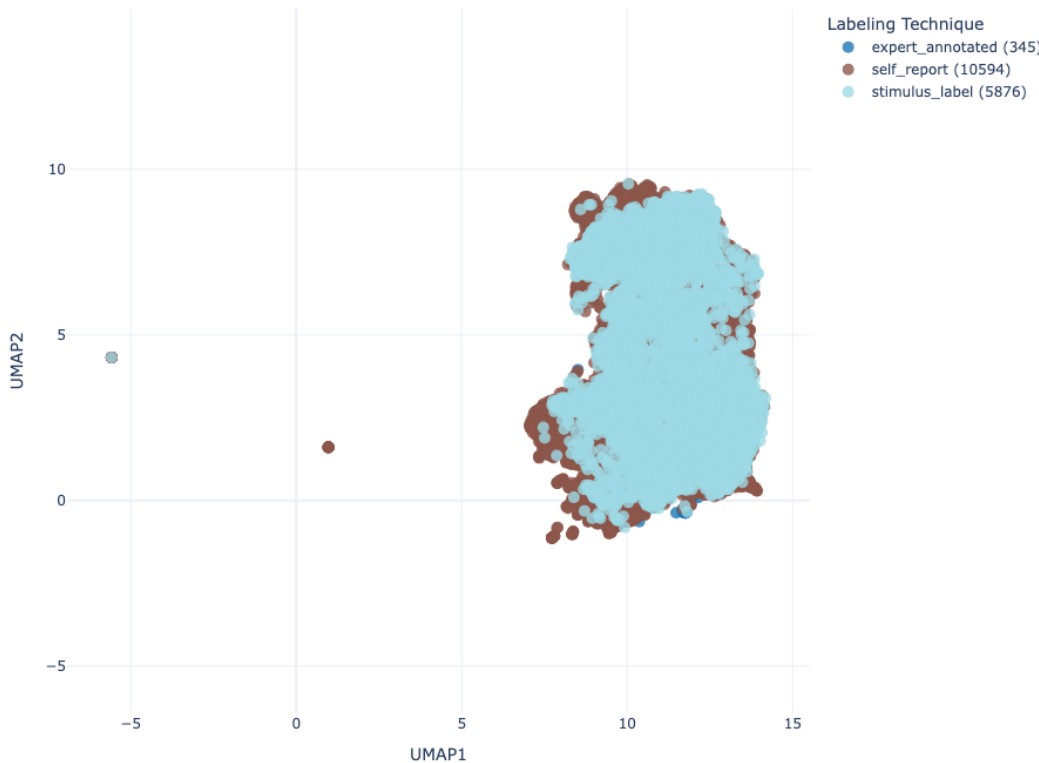

Figure 14: UMPA Visualization of EDA and PPG Features color coded by Labeling Techniques

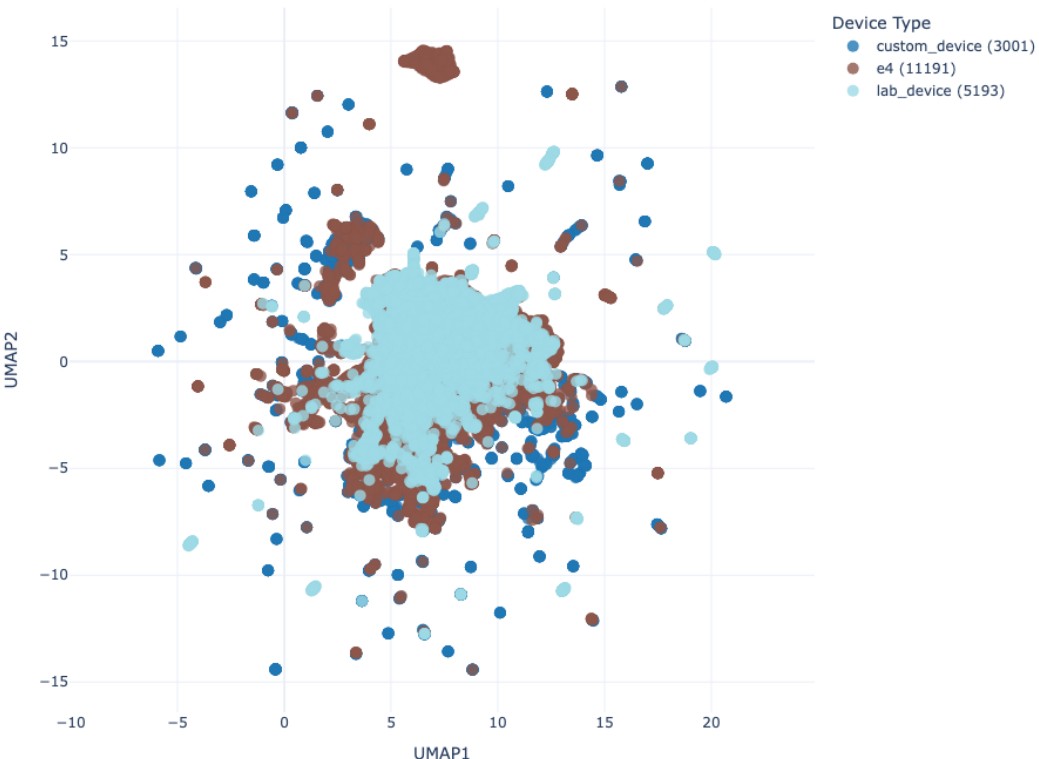

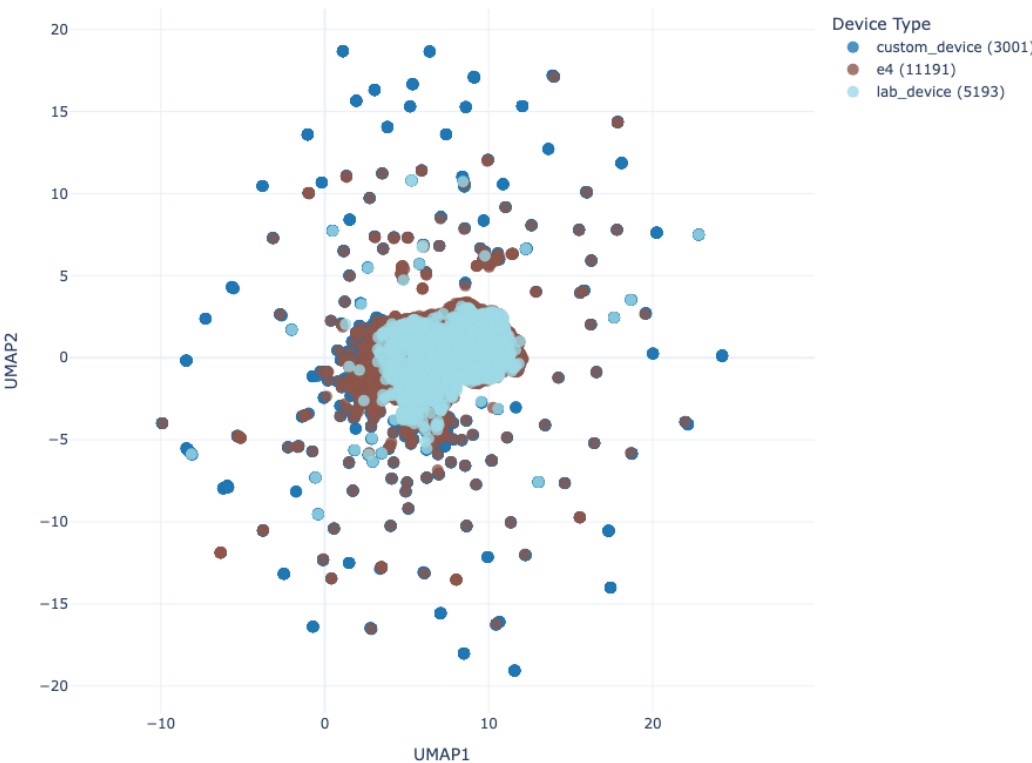

Figure 15: UMPA Visualization of EDA and PPG Features color coded by Device Type. Note: e4 here is wearable cohort, since all wristworn wearable devices were empatice e4.

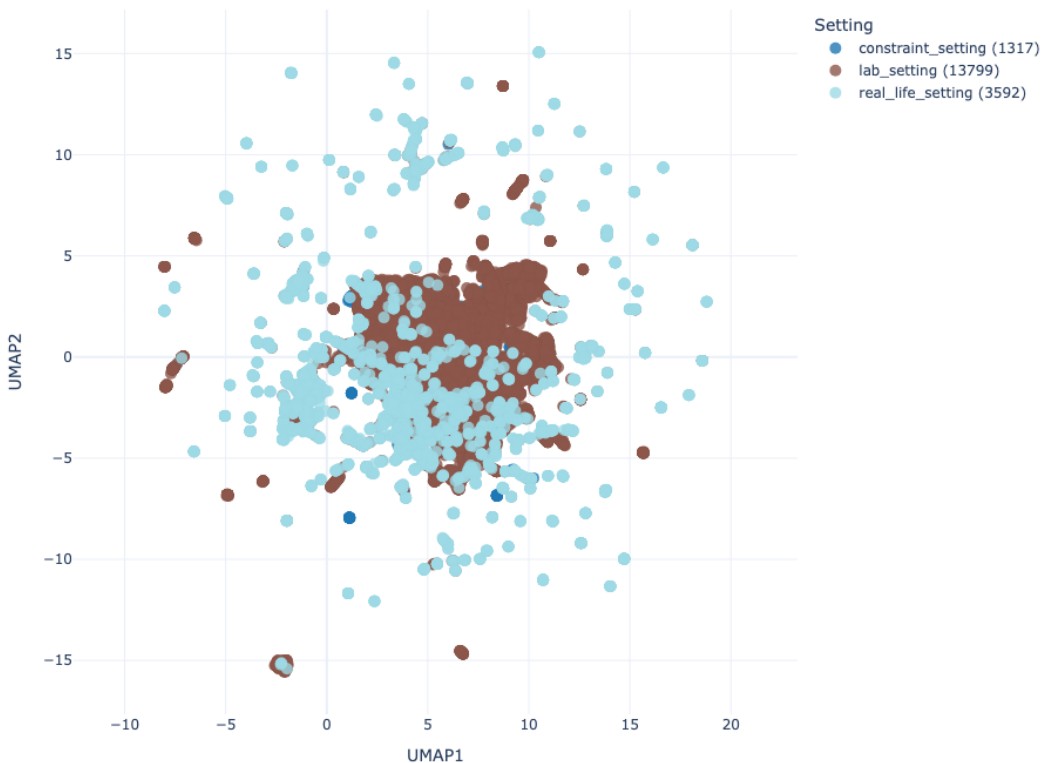

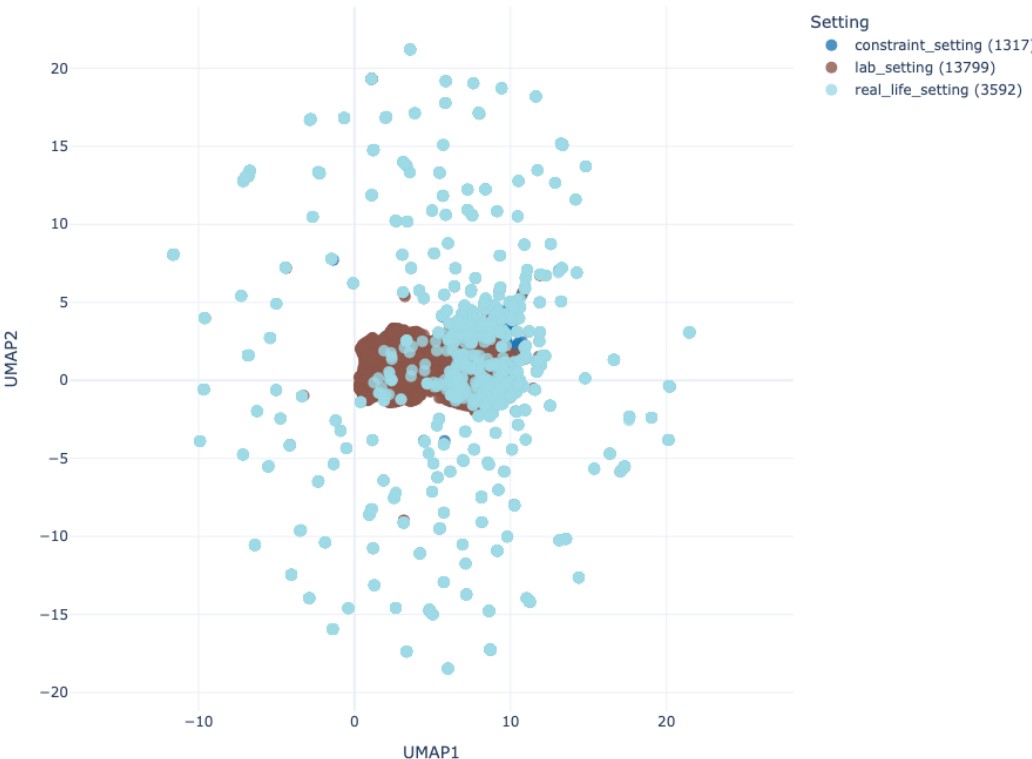

Figure 16: UMPA Visualization of EDA and PPG Features color coded by Experiment Collection Setting

| Testing Cohort | Training Cohort | EDA | | PPG | | EDA+PPG | |
|---|---|---|---|---|---|---|---|
| | | **Best Model** | **F1** | **Best Model** | **F1** | **Best Model** | **F1** |
| Lab | Real | **CLSP MLP 5%** | **0.79** | **RF** | **0.69** | **CLSP MLP 25%** | **0.79** |
| Lab | Constraint | CLSP MLP 25% | 0.66 | CLSP CNN 5% | 0.67 | CLSP MLP 25% | 0.68 |
| Lab | Lab | RF | 0.54 | HC+MLP | 0.50 | HC+MLP | 0.51 |
| Lab | CLSP ZeroShot | - | 0.67 | - | 0.28 | - | 0.36 |
| Constraint | Real | RF | 0.76 | **RF** | **0.78** | **RF** | **0.77** |
| Constraint | Lab | RF | 0.76 | RF | 0.72 | RF | 0.74 |
| Constraint | Constraint | RF | 0.63 | RF | 0.64 | RF | 0.65 |
| Constraint | CLSP ZeroShot | - | **0.79** | - | 0.55 | - | 0.52 |
| Real | Constraint | RF | 0.76 | **RF** | **0.70** | **RF** | **0.88** |
| Real | Lab | RF | 0.72 | CLSP MLP 25% | 0.64 | RF | 0.76 |
| Real | Real | HC+MLP | 0.41 | HC+MLP | 0.41 | HC+MLP | 0.42 |
| Real | CLSP ZeroShot | - | **0.77** | - | 0.50 | - | 0.49 |

Table 14: For each data-collection setting category (Lab, Constraint, and Real) and modality (EDA, PPG, and EDA+PPG), the table identifies the top-performing model and its F1 score for **valence classification**.

| Testing Cohort | Training Cohort | EDA | | PPG | | EDA+PPG | |
|---|---|---|---|---|---|---|---|
| | | **Best Model** | **F1** | **Best Model** | **F1** | **Best Model** | **F1** |
| Custom Wearable | Wearable | LDA | 0.69 | **LDA** | **0.69** | CLSP MLP 50% | 0.73 |
| Custom Wearable | Lab Based Device | **RF** | **0.72** | HC+MLP | 0.65 | **CLSP CNN 25%** | **0.78** |
| Custom Wearable | Custom Wearable | HC+MLP | 0.57 | HC+MLP | 0.40 | HC+MLP | 0.50 |
| Custom Wearable | CLSP ZeroShot | - | **0.72** | - | 0.44 | - | 0.58 |
| Lab Based Device | Wearable | LDA | 0.58 | HC+MLP | 0.45 | CLSP CNN 50% | 0.50 |
| Lab Based Device | Custom Wearable | **CLSP CNN 5%** | **0.67** | **RF** | **0.62** | **CLSP CNN 5%** | **0.65** |
| Lab Based Device | Lab Based Device | RF | 0.54 | RF | 0.54 | RF | 0.56 |
| Lab Based Device | CLSP ZeroShot | - | 0.63 | - | 0.35 | - | 0.50 |
| Wearable | Custom Wearable | CLSP MLP 5% | 0.65 | **CLSP CNN 5%** | **0.60** | **CLSP CNN 25%** | **0.66** |
| Wearable | Lab Based Device | **CLSP MLP 25%** | **0.68** | **HC+MLP** | **0.60** | CLSP CNN 25% | 0.59 |
| Wearable | Wearable | HC+MLP | 0.49 | HC+MLP | 0.48 | HC+MLP | 0.52 |
| Wearable | CLSP ZeroShot | - | 0.59 | - | 0.43 | - | 0.52 |

Table 15: For each device category (Wearables, Custom Wearable, and Lab-Based Device) and modality (EDA, PPG, and EDA+PPG), the table identifies the top-performing model and its F1 score for **arousal classification**. Here wearable is Empatica E4.

| Testing Cohort | Training Cohort | EDA | | PPG | | EDA+PPG | |
| --- | --- | --- | --- | --- | --- | --- | --- |
| | | **Best Model** | **F1** | **Best Model** | **F1** | **Best Model** | **F1** |
| Custom Wearable | Wearable | CLSP MLP 50% | 0.82 | **CLSP CNN 5%** | **0.72** | CLSP MLP 5% | 0.78 |
| Custom Wearable | Lab Based Device | LDA | 0.67 | CLSP MLP 50% | 0.67 | **LDA** | **0.81** |
| Custom Wearable | Custom Wearable | RF | 0.52 | HC+MLP | 0.50 | RF | 0.47 |
| Custom Wearable | CLSP ZeroShot | - | **0.83** | - | 0.60 | - | 0.54 |
| Lab Based Device | Wearable | CLSP CNN 50% | 0.62 | CLSP MLP 25% | 0.61 | CLSP CNN 5% | 0.62 |
| Lab Based Device | Custom Wearable | **CLSP CNN 50%** | **0.64** | **CLSP CNN 50%** | **0.63** | **CLSP CNN 5%** | **0.63** |
| Lab Based Device | Lab Based Device | RF | 0.53 | RF | 0.51 | RF | 0.52 |
| Lab Based Device | CLSP ZeroShot | - | 0.60 | - | 0.59 | - | 0.45 |
| Wearable | Lab Based Device | **CLSP CNN 50%** | **0.72** | **CLSP CNN 50%** | **0.73** | **CLSP CNN 50%** | **0.73** |
| Wearable | Custom Wearable | CLSP CNN 25% | 0.51 | CLSP MLP 5% | 0.58 | LDA | 0.53 |
| Wearable | Wearable | HC+MLP | 0.50 | HC+MLP | 0.55 | HC+MLP | 0.54 |
| Wearable | CLSP ZeroShot | - | 0.70 | - | 0.62 | - | 0.53 |

Table 16: For each device category (Wearables, Custom Wearable, and Lab-Based Device) and modality (EDA, PPG, and EDA+PPG), the table identifies the top-performing model and its F1 score for **valence classification**.

| Testing Cohort | Training Cohort | EDA | | PPG | | EDA+PPG | |
| --- | --- | --- | --- | --- | --- | --- | --- |
| | | **Best Model** | **F1** | **Best Model** | **F1** | **Best Model** | **F1** |
| Stimulus-Label | Expert-Annotated | **CLSP MLP 5%** | **0.64** | **RF** | **0.72** | **CLSP MLP 50%** | **0.65** |
| Stimulus-Label | Self-report | RF | 0.62 | CLSP CNN 5% | 0.44 | CLSP CNN 5% | 0.57 |
| Stimulus-Label | Stimulus-Label | RF | 0.54 | RF | 0.51 | HC+MLP | 0.55 |
| Stimulus-Label | CLSP ZeroShot | - | 0.60 | - | 0.37 | - | 0.50 |
| Self-report | Expert-Annotated | **CLSP MLP 5%** | **0.65** | **CLSP CNN 50%** | **0.64** | **CLSP MLP 5%** | **0.69** |
| Self-report | Stimulus-Label | HC+MLP | 0.57 | CLSP CNN 50% | 0.51 | RF | 0.63 |
| Self-report | Self-report | HC+MLP | 0.53 | HC+MLP | 0.52 | HC+MLP | 0.52 |
| Self-report | CLSP ZeroShot | - | 0.60 | - | 0.43 | - | 0.54 |
| Expert-Annotated | Self-report | RF | 0.87 | **LDA** | **0.69** | **RF** | **0.84** |
| Expert-Annotated | Stimulus-Label | CLSP CNN 50% | 0.79 | CLSP MLP 50% | 0.70 | RF | 0.82 |
| Expert-Annotated | Expert-Annotated | RF | 0.52 | RF | 0.28 | HC+MLP | 0.48 |
| Expert-Annotated | CLSP ZeroShot | - | **0.91** | - | 0.39 | - | 0.68 |

Table 17: For each labeling method category (Stimulus-Labels, Self-report, and Expert-Annotated) and modality (EDA, PPG, and EDA+PPG), the table identifies the top-performing model and its F1 score for **arousal classification**.

| Testing Cohort | Training Cohort | EDA | | PPG | | EDA+PPG | |
|---|---|---|---|---|---|---|---|
| | | Best Model | F1 | Best Model | F1 | Best Model | F1 |
| Stimulus-Label | Expert-Annotated | **CLSP MLP 5%** | **0.65** | **CLSP CNN 50%** | **0.65** | **CLSP CNN 25%** | **0.65** |
| Stimulus-Label | Self-report | CLSP CNN 25% | 0.63 | CLSP CNN 5% | 0.61 | CLSP CNN 5% | 0.61 |
| Stimulus-Label | Stimulus-Label | RF | 0.61 | RF | 0.53 | RF | 0.52 |
| Stimulus-Label | CLSP ZeroShot | - | 0.63 | - | 0.59 | - | 0.48 |
| Self-report | Expert-Annotated | CLSP MLP 50% | 0.69 | **RF** | **0.72** | **CLSP CNN 50%** | **0.76** |
| Self-report | Stimulus-Label | LDA | 0.57 | CLSP MLP 5% | 0.59 | LDA | 0.56 |
| Self-report | Self-report | RF | 0.53 | HC+MLP | 0.48 | HC+MLP | 0.52 |
| Self-report | CLSP ZeroShot | - | **0.70** | - | 0.62 | - | 0.53 |
| Expert-Annotated | Self-report | CLSP CNN 25% | 0.83 | **CLSP CNN 50%** | **0.85** | **CLSP CNN 5%** | **0.87** |
| Expert-Annotated | Stimulus-Label | **LDA** | **0.87** | **RF** | **0.85** | CLSP CNN 50% | 0.74 |
| Expert-Annotated | Expert-Annotated | HC+MLP | 0.56 | RF | 0.42 | HC+MLP | 0.49 |
| Expert-Annotated | CLSP ZeroShot | - | 0.83 | - | 0.60 | - | 0.43 |

Table 18: For each labeling method category (Stimulus-Labels, Self-report, and Expert-Annotated) and modality (EDA, PPG, and EDA+PPG), the table identifies the top-performing model and its F1 score for **valence classification**.

| Testing Cohort | Training Cohort | EDA | | PPG | | EDA+PPG | |
|---|---|---|---|---|---|---|---|
| | | Best Model | F1 | Best Model | F1 | Best Model | F1 |
| Male | Female | HC+MLP | **0.56** | LDA | **0.51** | LDA | 0.54 |
| Male | Male | RF | **0.56** | HC+MLP | **0.51** | RF | **0.56** |
| Male | CLSP ZeroShot | - | 0.56 | - | 0.16 | - | 0.24 |
| Female | Male | LDA | 0.50 | LDA | 0.51 | LDA | 0.53 |
| Female | Female | RF | 0.52 | HC+MLP | **0.55** | HC+MLP | **0.56** |
| Female | CLSP ZeroShot | - | **0.54** | - | 0.15 | - | 0.25 |

Table 19: Best-performing models for **arousal classification** across gender groups and modalities. For each dataset, gender group (Male, Female), and modality combination (EDA, PPG, EDA+PPG), we report the model achieving the highest F1 score.

| Testing Cohort | Training Cohort | EDA | | PPG | | EDA+PPG | |
|---|---|---|---|---|---|---|---|
| | | Best Model | F1 | Best Model | F1 | Best Model | F1 |
| Male | Female | CLSP MLP 25% | **0.69** | RF | **0.71** | CLSP CNN 50% | **0.70** |
| Male | Male | HC+MLP | 0.53 | HC+MLP | 0.52 | RF | 0.47 |
| Male | CLSP ZeroShot | - | **0.69** | - | 0.35 | - | 0.42 |
| Female | Male | CLSP MLP 50% | **0.71** | CLSP CNN 50% | **0.70** | CLSP MLP 25% | **0.70** |
| Female | Female | HC+MLP | 0.55 | RF | 0.49 | HC+MLP | 0.54 |
| Female | CLSP ZeroShot | - | 0.69 | - | 0.34 | - | 0.42 |

Table 20: Best-performing models for **valence classification** across gender groups and modalities. For each dataset, gender group (Male, Female), and modality combination (EDA, PPG, EDA+PPG), we report the model achieving the highest F1 score.

| Testing Cohort | Training Cohort | EDA | | PPG | | EDA+PPG | |
|---|---|---|---|---|---|---|---|
| | | **Best Model** | **F1** | **Best Model** | **F1** | **Best Model** | **F1** |
| Old | Young | LDA | 0.51 | HC+MLP | 0.56 | HC+MLP | **0.56** |
| Old | Old | RF | 0.55 | RF | 0.55 | RF | 0.53 |
| Old | CLSP ZeroShot | - | **0.56** | - | **0.7** | - | 0.19 |
| Young | Old | LDA | 0.5 | LDA | 0.43 | CLSP MLP 50% | 0.47 |
| Young | Young | HC+MLP | 0.55 | RF | 0.53 | HC+MLP | **0.58** |
| Young | CLSP ZeroShot | - | **0.56** | - | **0.72** | - | 0.28 |

Table 21: Best-performing models for **arousal classification** across age groups and modalities. For each dataset, age group (Young: 18–25 years, Old: 25+ years), and modality combination (EDA, PPG, EDA+PPG), we report the model achieving the highest F1 score.

| Testing Cohort | Training Cohort | EDA | | PPG | | EDA+PPG | |
|---|---|---|---|---|---|---|---|
| | | **Best Model** | **F1** | **Best Model** | **F1** | **Best Model** | **F1** |
| Old | Young | CLSP MLP 50% | **0.73** | CLSP CNN 50% | **0.72** | RF | **0.73** |
| Old | Old | RF | 0.53 | RF | 0.57 | RF | 0.53 |
| Old | CLSP ZeroShot | - | 0.14 | - | 0.37 | - | 0.47 |
| Young | Old | CLSP MLP 5% | **0.72** | RF | **0.67** | RF | **0.69** |
| Young | Young | RF | 0.54 | RF | 0.51 | RF | 0.48 |
| Young | CLSP ZeroShot | - | 0.16 | - | 0.31 | - | 0.35 |

Table 22: Best-performing models for **valence classification** across age groups and modalities. For each dataset, age group (Young: 18–25 years, Old: 25+ years), and modality combination (EDA, PPG, EDA+PPG), we report the model achieving the highest F1 score.

