# FEEL: Quantifying Heterogeneity in Physiological Signals for Generalizable Emotion Recognition

**Pragya Singh**[*]
IIIT-Delhi
India
pragyas@iiitd.ac.in

**Ankush Gupta**[†]
IIIT-Delhi
India
ankush21232@iiitd.ac.in

**Somay Jalan**[†]
IIIT-Delhi
India
somay22505@iiitd.ac.in

**Mohan Kumar**
RIT
NY, USA
mjkvcs@rit.edu

**Pushpendra Singh**
IIIT-Delhi
India
psingh@iiitd.ac.in

## 1  Supplementary Material

This supplementary material is divided into two main sections: Benchmarking Results and Cross-Data Results. The Benchmarking Results section reports average model performance using a Leave-One-Subject-Out (LOSO) validation strategy, calculated across all participants within each dataset. Results are reported as mean ± standard deviation, capturing both central tendency and variability, across all modeling paradigms for arousal, valence, and four-class classification using the EDA, PPG, and combined EDA+PPG modalities, see section 2.

The Cross-Data Results section evaluates model generalizability across datasets. It begins with individual transfer experiments, where models trained on all datasets within a group (device, label, or setting) are tested on the remaining two datasets group. This is followed by a Leave-One-Dataset-Out (LODO) in-cohort analysis, in which models are trained on all but one dataset within a group and evaluated on the held-out dataset, and a zero-short analysis. The LODO procedure is repeated for all datasets in each group, and performance is reported as an average to provide a comprehensive view. Together, these analyses provide detailed insights into the robustness and transferability of models across diverse data sources and experimental conditions (see Section 3). Furthermore, we conducted a demographic-level analysis, grouping participants by demographic attributes such as gender and age across the dataset. Models were then trained on one group and tested on the other to evaluate demographic generalization. Consistent with other experiments, we also performed leave-one-dataset-out (LODO) and zero-shot analyses.

## 2  Benchmarking Results

This section presents benchmarking results subdivided by modality: EDA, PPG, and the combined EDA+PPG, where each subsection is further subdivided into four modeling paradigms.

### 2.1  EDA

This section presents the benchmarking results for datasets using different architectural paradigms focused solely on the EDA modality. For clarity and ease of navigation, the section is further organized by modeling paradigms.

---

[*]Corresponding Author, Website: https://alchemy18.github.io/pragyasingh/
[†]Both authors contributed equally.

39th Conference on Neural Information Processing Systems (NeurIPS 2025) Track on Datasets and Benchmarks.

### 2.1.1 ML Models

Table 1 presents the arousal classification results, Table 2 shows the valence classification results, and Table 3 shows four-class classification results, based on EDA data using classical machine learning models.

| Dataset | RF | | LDA | |
|---|---|---|---|---|
| | **Accuracy** | **F1** | **Accuracy** | **F1** |
| WESAD | 81.02 ± 1.24 | 0.791 ± 0.08 | 85.9 ± 8.97 | 0.689 ± 0.22 |
| NURSE | 40.23 ± 5.12 | 0.225 ± 0.115 | 41.93 ± 9.86 | 0.301 ± 0.128 |
| EMOGNITION | 49.382 ± 9.541 | 0.323 ± 0.148 | 50.124 ± 8.320 | 0.288 ± 0.129 |
| UBFC_PHYS | 62.5 ± 7.48 | 0.14 ± 0.024 | 62.35 ± 9.52 | 0.324 ± 0.386 |
| PhyMER | 61.420 ± 9.606 | 0.250 ± 0.137 | 51.231 ± 11.070 | 0.397 ± 0.200 |
| EmoWear | 54.102 ± 14.501 | 0.637 ± 0.168 | 53.599 ± 13.116 | 0.475 ± 0.249 |
| MAUS | 71.023 ± 10.245 | 0.791 ± 0.086 | 65.909 ± 18.970 | 0.689 ± 0.220 |
| CLAS | 62.167 ± 10.826 | 0.687 ± 0.087 | 59.000 ± 7.088 | 0.569 ± 0.143 |
| CASE | 75.455 ± 12.931 | 0.372 ± 0.257 | 64.470 ± 11.944 | 0.419 ± 0.182 |
| Unobtrusive | 81.625 ± 8.955 | 0.878 ± 0.073 | 64.367 ± 10.361 | 0.723 ± 0.108 |
| CEAP-360VR | 59.766 ± 16.416 | 0.238 ± 0.259 | 45.703 ± 18.407 | 0.305 ± 0.282 |
| ScientISST MOVE | 81.349 ± 26.181 | 0.716 ± 0.341 | 78.214 ± 27.986 | 0.701 ± 0.355 |
| Dapper | 55.217 ± 6.023 | 0.462 ± 0.091 | 58.550 ± 3.778 | 0.565 ± 0.117 |
| ForDigitStress | 70.32 ± 5.6 | 0.523 ± 0.07 | 69.75 ± 6.1 | 0.498 ± 0.06 |
| ADARP | 51.169 ± 11.77 | 0.210 ± 0.011 | 50.741 ± 5.55 | 0.207 ± 0.033 |
| Exercise | 55.31 ± 6.52 | 0.320 ± 0.054 | 54.02 ± 8.41 | 0.288 ± 0.061 |
| MOCAS | 43.59 ± 8.10 | 0.57 ± 0.024 | 46.31 ± 6.57 | 0.438 ± 0.101 |
| LAUREATE | 70.2 ± 10.4 | 0.328 ± 0.021 | 68.9 ± 11.7 | 0.310 ± 0.025 |
| VERBIO | 59.71 ± 8.59 | 0.598 ± 0.098 | 60.16 ± 10.84 | 0.337 ± 0.28 |

Table 1: Performance comparison of Random Forest (RF) and Linear Discriminant Analysis (LDA) models on **arousal classification** across multiple datasets using **EDA modality**. Accuracy and F1 scores are shown as mean ± standard deviation.

### 2.1.2 Handcrafted Features with DL Models

Table 4 presents the arousal classification results, Table 5 shows the valence classification results, and Table 6 shows the four-class classification results based on EDA data using hybrid handcrafted features and deep learning models.

### 2.1.3 DL models with Raw Signal Segments

Table 7 presents the arousal classification results, while Table 8 shows the valence classification results, both based on EDA data using raw signal data and deep learning models.

### 2.1.4 Pretrained Models - CLSP Zero shot

Table 9 presents the arousal, valence, and four-class classification results, both based on EDA data using pre-trained CLSP models.

### 2.1.5 Pretrained Models - CLSP MLP

Table 10 presents the arousal classification results, Table 11 shows the valence classification results, and Table 12 shows the four-class classification results, based on EDA data using a fine-tuned CLSP model using MLP-based MetaNet.

### 2.1.6 Pretrained Models - CLSP CNN

Table 13 presents the arousal classification results, Table 14 shows the valence classification results, and Table 15 shows the four-class classification, based on EDA data using a fine-tuned CLSP model using 1D-CNN-based MetaNet.

| Dataset | RF | | LDA | |
|---|---|---|---|---|
| | Accuracy | F1-score | Accuracy | F1-score |
| WESAD | 82.15 ± 8.32 | 0.659 ± 0.072 | 80.48 ± 9.01 | 0.634 ± 0.081 |
| NURSE | 71.230 ± 9.5 | 0.372 ± 0.160 | 69.843 ± 10.7 | 0.398 ± 0.180 |
| EMOGNITION | 61.239 ± 8.012 | 0.324 ± 0.141 | 58.974 ± 7.345 | 0.362 ± 0.132 |
| UBFC_PHYS | 74.405 ± 14.895 | 0.757 ± 0.143 | 72.173 ± 23.478 | 0.731 ± 0.223 |
| PhyMER | 55.531 ± 15.061 | 0.645 ± 0.163 | 49.940 ± 15.924 | 0.465 ± 0.254 |
| EmoWear | 62.310 ± 11.183 | 0.753 ± 0.087 | 52.263 ± 13.020 | 0.524 ± 0.240 |
| MAUS | 71.023 ± 10.245 | 0.487 ± 0.179 | 65.909 ± 18.970 | 0.572 ± 0.196 |
| CLAS | 65.067 ± 8.864 | 0.558 ± 0.168 | 60.700 ± 6.440 | 0.565 ± 0.170 |
| CASE | 64.924 ± 9.856 | 0.416 ± 0.218 | 57.879 ± 12.067 | 0.530 ± 0.192 |
| Unobtrusive | 58.019 ± 7.032 | 0.704 ± 0.065 | 53.290 ± 9.418 | 0.549 ± 0.178 |
| CEAP-360VR | 51.953 ± 20.603 | 0.299 ± 0.269 | 48.828 ± 17.494 | 0.419 ± 0.233 |
| ScientISST MOVE | 69.246 ± 23.324 | 0.760 ± 0.233 | 70.357 ± 23.588 | 0.747 ± 0.249 |
| Dapper | 60.123 ± 15.52 | 0.520 ± 0.046 | 61.01 ± 11.56 | 0.403 ± 0.062 |
| ForDigitStress | 70.24 ± 0.82 | 0.551 ± 0.015 | 68.10 ± 1.04 | 0.528 ± 0.017 |
| ADARP | 42.57 ± 12.63 | 0.182 ± 0.017 | 40.19 ± 10.24 | 0.198 ± 0.011 |
| Exercise | 60.48 ± 0.96 | 0.458 ± 0.016 | 58.30 ± 1.12 | 0.419 ± 0.018 |
| MOCAS | 61.6 ± 4.94 | 0.554 ± 0.19 | 59.5 ± 3.87 | 0.541 ± 0.17 |
| LAUREATE | 65.909 ± 4.26 | 0.340 ± 0.061 | 61.667 ± 6.32 | 0.300 ± 0.055 |
| VERBIO | 37.8 ± 2.94 | 0.362 ± 0.019 | 36.8 ± 3.98 | 0.180 ± 0.046 |

Table 2: Performance comparison of Random Forest (RF) and Linear Discriminant Analysis (LDA) models on **valence classification** across multiple datasets using **EDA modality**. Accuracy and F1 scores are shown as mean ± standard deviation.

## 2.2 PPG

This section presents the benchmarking results for datasets using different architectural paradigms focused solely on the PPG modality. For clarity and ease of navigation, the section is further organized by modeling paradigms.

### 2.2.1 ML Models

Table 16 presents the arousal classification results, Table 17 shows the valence classification results, and Table 18 shows the four-class classification, based on PPG data using classical machine learning models.

### 2.2.2 Handcrafted Features with DL Models

Table 19 presents the arousal classification results, Table 20 shows the valence classification results, and Table 21 shows the four-class classification results, based on PPG data using hybrid handcrafted features and deep learning models.

### 2.2.3 DL models with Raw Signal Segments

Table 22 presents the arousal classification results, while Table 23 shows the valence classification results, both based on PPG data using raw signal data and deep learning models.

### 2.2.4 Pretrained Models - CLSP Zero shot

Table 24 presents the arousal, valence, and four-class classification results, both based on PPG data using pre-trained CLSP models.

### 2.2.5 Pretrained Models - CLSP MLP

Table 25 presents the arousal classification results, Table 26 shows the valence classification results, and Table 27 shows the four-class classification results based on PPG data using a fine-tuned CLSP model using MLP-based MetaNet.

| Dataset | RF | | LDA | |
|---|---|---|---|---|
| | Accuracy | F1 | Accuracy | F1 |
| WESAD | 98.75 | 0.987 | 97.50 | 0.969 |
| NURSE | 34.51 | 0.343 | 35.01 | 0.376 |
| EMOGNITION | 59.73 | 0.572 | 38.90 | 0.393 |
| UBFC_PHYS | 54.46 | 0.586 | 54.61 | 0.609 |
| PhyMER | 37.21 | 0.332 | 24.44 | 0.218 |
| EmoWear | 31.62 | 0.278 | 28.12 | 0.252 |
| MAUS | 71.02 | 0.692 | 65.91 | 0.650 |
| CLAS | 44.53 | 0.430 | 39.57 | 0.380 |
| CASE | 48.71 | 0.476 | 33.26 | 0.344 |
| Unobtrusive | 43.71 | 0.402 | 41.59 | 0.389 |
| CEAP-360VR | 30.47 | 0.274 | 25.39 | 0.243 |
| ScientISST MOVE | 70.32 | 0.680 | 66.87 | 0.671 |
| Dapper | 50.86 | 0.434 | 22.25 | 0.227 |
| ForDigitStress | 70.45 | 0.668 | 65.63 | 0.682 |
| ADARP | 19.79 | 0.245 | 9.38 | 0.115 |
| Exercise | 52.07 | 0.498 | 38.06 | 0.371 |
| MOCAS | 38.82 | 0.412 | 26.55 | 0.317 |
| LAUREATE | 56.21 | 0.487 | 23.52 | 0.251 |
| VERBIO | 34.21 | 0.337 | 35.07 | 0.378 |

Table 3: Performance comparison of Random Forest (RF) and Linear Discriminant Analysis (LDA) models on **four-class classification** using **EDA modality**.

| Dataset | HC+MLP | | HC+ResNet | | HC+LSTM+NN | | HC+Attention+NN | |
|---|---|---|---|---|---|---|---|---|
| | Accuracy | F1-score | Accuracy | F1-score | Accuracy | F1-score | Accuracy | F1-score |
| WESAD | 85.75 ± 6.75 | 0.740 ± 0.17 | 77.42 ± 7.34 | 0.745 ± 0.09 | 64.58 ± 16.99 | 0.684 ± 0.25 | 63.63 ± 17.82 | 0.667 ± 0.24 |
| NURSE | 39.88 ± 8.210 | 0.32 ± 0.102 | 36.245 ± 4.13 | 0.289 ± 0.109 | 34.789 ± 1.012 | 0.263 ± 0.135 | 35.876 ± 5.092 | 0.294 ± 0.122 |
| EMOGNITION | 52.8 ± 4.12 | 0.57 ± 0.164 | 42.578 ± 11.034 | 0.263 ± 0.172 | 40.198 ± 9.765 | 0.240 ± 0.151 | 38.900 ± 12.002 | 0.217 ± 0.188 |
| UBFC_PHYS | 55.06 ± 9.11 | 0.167 ± 0.034 | 59.22 ± 7.81 | 0.204 ± 0.078 | 56.54 ± 6.81 | 0.264 ± 0.035 | 54.16 ± 3.13 | 0.304 ± 0.064 |
| PhyMER | 51.520 ± 8.213 | 0.363 ± 0.151 | 54.837 ± 7.876 | 0.320 ± 0.161 | 52.020 ± 14.566 | 0.346 ± 0.263 | 50.360 ± 10.516 | 0.374 ± 0.174 |
| EmoWear | 51.426 ± 10.341 | 0.500 ± 0.196 | 51.744 ± 8.980 | 0.562 ± 0.151 | 50.830 ± 15.522 | 0.434 ± 0.312 | 48.582 ± 11.472 | 0.426 ± 0.211 |
| MAUS | 68.750 ± 16.753 | 0.740 ± 0.175 | 67.424 ± 7.346 | 0.745 ± 0.090 | 64.583 ± 16.998 | 0.684 ± 0.256 | 63.636 ± 17.826 | 0.667 ± 0.242 |
| CLAS | 61.667 ± 9.688 | 0.632 ± 0.116 | 60.467 ± 9.053 | 0.647 ± 0.087 | 57.400 ± 9.045 | 0.517 ± 0.261 | 57.067 ± 5.686 | 0.482 ± 0.176 |
| CASE | 65.985 ± 14.143 | 0.416 ± 0.194 | 67.727 ± 10.996 | 0.332 ± 0.169 | 62.652 ± 20.395 | 0.364 ± 0.196 | 66.364 ± 15.233 | 0.438 ± 0.210 |
| Unobtrusive | 64.499 ± 11.862 | 0.727 ± 0.140 | 72.103 ± 11.439 | 0.807 ± 0.109 | 69.218 ± 10.374 | 0.780 ± 0.107 | 65.216 ± 14.681 | 0.704 ± 0.235 |
| CEAP-360VR | 54.688 ± 15.795 | 0.329 ± 0.241 | 57.422 ± 16.146 | 0.316 ± 0.303 | 51.172 ± 20.170 | 0.362 ± 0.283 | 53.125 ± 18.239 | 0.299 ± 0.261 |
| ScientISST MOVE | 72.262 ± 27.585 | 0.632 ± 0.327 | 73.095 ± 27.310 | 0.609 ± 0.337 | 61.548 ± 27.164 | 0.587 ± 0.221 | 84.008 ± 15.170 | 0.769 ± 0.200 |
| Dapper | 42.710 ± 6.870 | 0.576 ± 0.175 | 42.710 ± 6.870 | 0.283 ± 0.11 | 44.751 ± 6.888 | 0.375 ± 0.051 | 46.288 ± 9.486 | 0.314 ± 0.053 |
| ForDigitStress | 75.40 ± 6.2 | 0.485 ± 0.08 | 63.85 ± 5.8 | 0.512 ± 0.09 | 66.10 ± 7.0 | 0.475 ± 0.07 | 64.45 ± 6.6 | 0.506 ± 0.06 |
| ADARP | 53.51 ± 5.14 | 0.281 ± 0.007 | 30.573 ± 5.62 | 0.176 ± 0.046 | 29.689 ± 8.53 | 0.169 ± 0.066 | 31.976 ± 15.3 | 0.186 ± 0.002 |
| Exercise | 49.88 ± 7.33 | 0.374 ± 0.039 | 44.56 ± 6.70 | 0.301 ± 0.042 | 43.99 ± 8.16 | 0.256 ± 0.051 | 42.72 ± 6.94 | 0.243 ± 0.034 |
| MOCAS | 46.67 ± 7.32 | 0.533 ± 0.186 | 42.16 ± 16.31 | 0.41 ± 0.02 | 43.26 ± 10.6 | 0.39 ± 0.13 | 39.87 ± 11.59 | 0.414 ± 0.037 |
| LAUREATE | 32.2 ± 3.67 | 0.152 ± 0.018 | 31.5 ± 2.8 | 0.145 ± 0.020 | 29.8 ± 3.5 | 0.138 ± 0.022 | 34.99 ± 3.53 | 0.142 ± 0.019 |
| VERBIO | 55.08 ± 3.56 | 0.428 ± 0.125 | 47.55 ± 11.67 | 0.322 ± 0.1 | 47.56 ± 5.01 | 0.546 ± 0.014 | 46.19 ± 9.62 | 0.311 ± 0.072 |

Table 4: Performance comparison of hybrid handcrafted + deep learning models (HC+MLP, HC+RESNET, HC+LSTM, HC+Attention+MLP) across datasets for **arousal classification** using **EDA modality**. Accuracy and F1 scores are shown as mean ± standard deviation.

### 2.2.6 Pretrained Models - CLSP CNN

Table 28 presents the arousal classification results, Table 29 shows the valence classification results, and Table 30 shows the four-class classification results, based on PPG data using a fine-tuned CLSP model using 1D-CNN-based MetaNet.

## 2.3 EDA+PPG

This section presents the benchmarking results for datasets using different architectural paradigms focused on both EDA and PPG modalities. For clarity and ease of navigation, the section is further organized by modeling paradigms.

### 2.3.1 ML Models

Table 31 presents the arousal classification results, Table 32 shows the valence classification results, and Table 33 shows the four-class classification results, based on EDA+PPG data using classical machine learning models.

| Dataset | HC+MLP | | HC+RESNET | | HC+LSTM | | HC+Attention+MLP | |
|---|---|---|---|---|---|---|---|---|
| | **Accuracy** | **F1-score** | **Accuracy** | **F1-score** | **Accuracy** | **F1-score** | **Accuracy** | **F1-score** |
| WESAD | 82.39 ± 10.23 | 0.682 ± 0.123 | 51.66 ± 9.68 | 0.064 ± 0.110 | 48.21 ± 13.45 | 0.540 ± 0.142 | 46.79 ± 12.01 | 0.423 ± 0.131 |
| NURSE | 71.83 ± 7.2 | 0.030 ± 0.135 | 16.095 ± 6.8 | 0.255 ± 0.042 | 36.592 ± 1.3 | 0.058 ± 0.073 | 34.738 ± 6.1 | 0.235 ± 0.160 |
| EMOGNITION | 61.89 ± 5.234 | 0.360 ± 0.118 | 42.406 ± 6.527 | 0.315 ± 0.165 | 39.833 ± 7.103 | 0.312 ± 0.142 | 40.291 ± 5.886 | 0.299 ± 0.110 |
| UBFC_PHYS | 72.321 ± 20.604 | 0.725 ± 0.213 | 68.899 ± 16.244 | 0.705 ± 0.174 | 66.369 ± 18.733 | 0.625 ± 0.279 | 70.982 ± 21.320 | 0.699 ± 0.229 |
| PhyMER | 51.217 ± 13.877 | 0.500 ± 0.203 | 52.797 ± 9.680 | 0.571 ± 0.156 | 52.483 ± 13.939 | 0.498 ± 0.277 | 47.577 ± 14.761 | 0.463 ± 0.214 |
| EmoWear | 53.846 ± 9.496 | 0.591 ± 0.159 | 56.348 ± 10.137 | 0.660 ± 0.108 | 47.895 ± 14.554 | 0.468 ± 0.284 | 48.720 ± 9.034 | 0.481 ± 0.183 |
| MAUS | 70.833 ± 15.051 | 0.577 ± 0.202 | 65.909 ± 8.875 | 0.451 ± 0.145 | 60.606 ± 14.702 | 0.457 ± 0.166 | 63.826 ± 15.127 | 0.480 ± 0.171 |
| CLAS | 60.900 ± 7.144 | 0.578 ± 0.140 | 61.233 ± 7.890 | 0.551 ± 0.146 | 60.633 ± 7.693 | 0.567 ± 0.139 | 60.833 ± 7.009 | 0.633 ± 0.120 |
| CASE | 56.212 ± 13.170 | 0.442 ± 0.236 | 58.864 ± 11.108 | 0.381 ± 0.204 | 56.136 ± 17.937 | 0.349 ± 0.260 | 55.379 ± 12.369 | 0.442 ± 0.239 |
| Unobtrusive | 55.021 ± 9.096 | 0.584 ± 0.140 | 56.731 ± 7.497 | 0.644 ± 0.098 | 49.638 ± 8.701 | 0.486 ± 0.242 | 52.055 ± 8.802 | 0.476 ± 0.154 |
| CEAP-360VR | 48.828 ± 20.170 | 0.420 ± 0.247 | 51.562 ± 19.247 | 0.370 ± 0.273 | 49.609 ± 17.530 | 0.379 ± 0.284 | 46.875 ± 17.098 | 0.365 ± 0.249 |
| ScientISST MOVE | 73.214 ± 23.849 | 0.781 ± 0.237 | 66.071 ± 21.823 | 0.693 ± 0.294 | 66.230 ± 27.719 | 0.702 ± 0.325 | 70.476 ± 24.460 | 0.755 ± 0.238 |
| Dapper | 60.567 ± 5.58 | 0.472 ± 0.055 | 55.01 ± 13.56 | 0.456 ± 0.074 | 60.789 ± 12.40 | 0.492 ± 0.038 | 57.12 ± 5.67 | 0.467 ± 0.133 |
| ForDigitStress | 75.38 ± 1.26 | 0.564 ± 0.020 | 63.72 ± 0.98 | 0.503 ± 0.018 | 60.49 ± 1.15 | 0.481 ± 0.022 | 61.07 ± 1.20 | 0.471 ± 0.021 |
| ADARP | 45.5 ± 13.33 | 0.207 ± 0.011 | 39.7 ± 3.15 | 0.196 ± 0.037 | 41.562 ± 12.55 | 0.165 ± 0.067 | 36.982 ± 13.09 | 0.196 ± 0.021 |
| Exercise | 60.14 ± 1.08 | 0.386 ± 0.019 | 47.76 ± 1.26 | 0.351 ± 0.021 | 45.93 ± 1.20 | 0.330 ± 0.020 | 43.10 ± 1.34 | 0.297 ± 0.022 |
| MOCAS | 60.54 ± 6.24 | 0.305 ± 0.164 | 53.04 ± 10.9 | 0.323 ± 0.078 | 58.73 ± 5.5 | 0.466 ± 0.104 | 54.71 ± 6.02 | 0.551 ± 0.148 |
| LAUREATE | 65.985 ± 4.16 | 0.360 ± 0.058 | 53.125 ± 2.99 | 0.275 ± 0.062 | 51.562 ± 2.55 | 0.245 ± 0.088 | 57.262 ± 3.78 | 0.288 ± 0.064 |
| VERBIO | 37.7 ± 5.18 | 0.399 ± 0.067 | 30.7 ± 7.91 | 0.233 ± 0.015 | 34.59 ± 6.21 | 0.300 ± 0.018 | 33.87 ± 6.67 | 0.231 ± 0.020 |

Table 5: Performance comparison of hybrid handcrafted (HC) feature-based deep learning models on **valence classification** using **EDA modality**. Accuracy and F1 scores are reported with standard deviations.

| Dataset | HC + MLP | |
|---|---|---|
| | **Accuracy** | **F1 Score** |
| WESAD | 90.83 | 0.901 |
| NURSE | 35.34 | 0.360 |
| EMOGNITION | 37.32 | 0.366 |
| UBFC_PHYS | 53.27 | 0.600 |
| PhyMER | 26.18 | 0.262 |
| EmoWear | 28.48 | 0.274 |
| MAUS | 70.83 | 0.700 |
| CLAS | 42.37 | 0.418 |
| CASE | 37.27 | 0.383 |
| Unobtrusive | 42.09 | 0.387 |
| CEAP-360VR | 30.08 | 0.281 |
| ScientISST MOVE | 60.48 | 0.595 |
| Dapper | 29.19 | 0.319 |
| ForDigitStress | 61.11 | 0.619 |
| ADARP | 21.88 | 0.224 |
| Exercise | 41.02 | 0.407 |
| MOCAS | 27.92 | 0.347 |
| LAUREATE | 21.38 | 0.146 |
| VERBIO | 28.51 | 0.310 |

Table 6: Performance comparison of hybrid handcrafted (HC) feature-based deep learning model on **four-class classification** using **EDA modality**.

### 2.3.2 Handcrafted Features with DL Models

Table 34 presents the arousal classification results, Table 35 shows the valence classification results, and Table 36 shows the four-class classification results, based on EDA+PPG data using hybrid handcrafted features and deep learning models.

### 2.3.3 Pretrained Models - CLSP Zero shot

Table 37 presents the arousal classification and valence classification results, both based on EDA+PPG data using pre-trained CLSP models.

### 2.3.4 Pretrained Models - CLSP MLP

Table 38 presents the arousal classification results, Table 39 shows the valence classification results, and Table 40 shows the four-class classification results, based on EDA+PPG data using a fine-tuned CLSP model using MLP-based MetaNet.

| Dataset | Signal+CNN+Transformer | | Signal+LSTM+NN | | Signal+RESNET | |
|---|---|---|---|---|---|---|
| | **Accuracy** | **F1-score** | **Accuracy** | **F1-score** | **Accuracy** | **F1-score** |
| WESAD | 54.73 ± 17.17 | 0.49 ± 0.39 | 46.97 ± 16.7 | 0.327 ± 0.4 | 55.76 ± 11.18 | 0.828 ± 0.09 |
| NURSE | 33.49 ± 4.22 | 0.210 ± 0.152 | 32.88 ± 3.78 | 0.233 ± 0.128 | 34.50 ± 2.41 | 0.251 ± 0.137 |
| EMOGNITION | 41.12 ± 3.88 | 0.195 ± 0.222 | 40.56 ± 6.45 | 0.281 ± 0.09 | 28.73 ± 10.45 | 0.168 ± 0.205 |
| UBFC_PHYS | 41.37 ± 6.82 | 0.175 ± 0.039 | 53.57 ± 8.49 | 0.186 ± 0.041 | 52.19 ± 8.04 | 0.291 ± 0.049 |
| PhyMER | 51.24 ± 16.79 | 0.254 ± 0.288 | 47.56 ± 16.65 | 0.260 ± 0.271 | 45.45 ± 9.22 | 0.430 ± 0.103 |
| EmoWear | 50.49 ± 19.77 | 0.390 ± 0.368 | 46.98 ± 19.54 | 0.309 ± 0.355 | 51.81 ± 12.51 | 0.427 ± 0.199 |
| MAUS | 54.73 ± 17.17 | 0.496 ± 0.391 | 46.97 ± 16.77 | 0.327 ± 0.403 | 75.76 ± 11.18 | 0.828 ± 0.090 |
| CLAS | 49.70 ± 2.23 | 0.315 ± 0.341 | 49.70 ± 2.23 | 0.315 ± 0.341 | 60.93 ± 14.16 | 0.673 ± 0.117 |
| CASE | 47.88 ± 42.77 | 0.469 ± 0.480 | 47.88 ± 42.77 | 0.469 ± 0.480 | 50.61 ± 19.41 | 0.282 ± 0.218 |
| Unobtrusive | 45.50 ± 27.91 | 0.333 ± 0.440 | 52.98 ± 36.22 | 0.500 ± 0.469 | 67.78 ± 11.49 | 0.764 ± 0.151 |
| CEAP-360VR | 51.56 ± 20.52 | 0.255 ± 0.303 | 53.13 ± 20.33 | 0.241 ± 0.307 | 55.86 ± 20.08 | 0.324 ± 0.254 |
| ScientISST MOVE | 57.26 ± 26.61 | 0.378 ± 0.300 | 53.21 ± 22.76 | 0.177 ± 0.264 | 80.20 ± 11.48 | 0.611 ± 0.233 |
| Dapper | 41.74 ± 4.25 | 0.333 ± 0.062 | 43.60 ± 3.83 | 0.297 ± 0.041 | 39.55 ± 3.78 | 0.265 ± 0.117 |
| ForDigitStress | 56.24 ± 5.40 | 0.403 ± 0.080 | 54.78 ± 5.10 | 0.377 ± 0.070 | 55.91 ± 4.90 | 0.395 ± 0.060 |
| ADARP | 35.30 ± 10.13 | 0.197 ± 0.050 | 37.07 ± 6.53 | 0.207 ± 0.070 | 35.09 ± 10.78 | 0.200 ± 0.040 |
| Exercise | 35.80 ± 7.45 | 0.104 ± 0.090 | 36.92 ± 5.86 | 0.113 ± 0.087 | 33.47 ± 8.21 | 0.105 ± 0.092 |
| MOCAS | 37.92 ± 9.87 | 0.395 ± 0.038 | 43.50 ± 7.91 | 0.374 ± 0.074 | 42.67 ± 10.17 | 0.297 ± 0.066 |
| LAUREATE | 28.95 ± 5.16 | 0.131 ± 0.023 | 28.97 ± 7.15 | 0.125 ± 0.021 | 30.32 ± 3.44 | 0.128 ± 0.022 |
| VERBIO | 44.20 ± 3.51 | 0.267 ± 0.180 | 48.94 ± 6.67 | 0.567 ± 0.033 | 50.12 ± 7.09 | 0.409 ± 0.027 |

Table 7: Performance comparison of raw-signal based deep learning models on **arousal classification** using **EDA modality**. Accuracy and F1 scores are reported with standard deviations.

| Dataset | Signal+CNN+Transformer | | Signal+LSTM+NN | | Signal+RESNET | |
|---|---|---|---|---|---|---|
| | **Accuracy** | **F1-score** | **Accuracy** | **F1-score** | **Accuracy** | **F1-score** |
| WESAD | 41.46 ± 10.98 | 0.25 ± 0.196 | 47.31 ± 16.87 | 0.27 ± 0.18 | 44.89 ± 9.35 | 0.215 ± 0.178 |
| NURSE | 31.223 ± 12.1 | 0.176 ± 0.09 | 39.987 ± 13.4 | 0.15 ± 0.07 | 38.435 ± 11.2 | 0.108 ± 0.05 |
| EMOGNITION | 34.392 ± 8.729 | 0.246 ± 0.131 | 36.128 ± 6.223 | 0.274 ± 0.138 | 33.612 ± 7.007 | 0.263 ± 0.115 |
| UBFC_PHYS | 51.637 ± 17.366 | 0.426 ± 0.330 | 52.381 ± 16.645 | 0.411 ± 0.353 | 69.940 ± 23.392 | 0.672 ± 0.272 |
| PhyMER | 53.014 ± 23.792 | 0.370 ± 0.394 | 43.061 ± 22.852 | 0.303 ± 0.358 | 49.245 ± 14.586 | 0.504 ± 0.224 |
| EmoWear | 49.189 ± 20.131 | 0.348 ± 0.404 | 51.382 ± 20.099 | 0.426 ± 0.402 | 52.265 ± 11.450 | 0.592 ± 0.141 |
| MAUS | 55.682 ± 15.725 | 0.202 ± 0.249 | 48.485 ± 16.988 | 0.273 ± 0.255 | 73.106 ± 11.695 | 0.542 ± 0.154 |
| CLAS | 50.200 ± 2.007 | 0.292 ± 0.325 | 50.200 ± 2.007 | 0.292 ± 0.325 | 52.167 ± 1.122 | 0.014 ± 0.068 |
| CASE | 50.303 ± 45.344 | 0.486 ± 0.494 | 50.303 ± 45.344 | 0.486 ± 0.494 | 60.758 ± 16.003 | 0.378 ± 0.315 |
| Unobtrusive | 50.677 ± 13.065 | 0.407 ± 0.385 | 50.589 ± 7.113 | 0.393 ± 0.369 | 52.398 ± 10.497 | 0.477 ± 0.246 |
| CEAP-360VR | 53.516 ± 13.181 | 0.252 ± 0.334 | 51.953 ± 13.511 | 0.427 ± 0.329 | 45.312 ± 14.463 | 0.297 ± 0.262 |
| ScientISST MOVE | 53.651 ± 23.082 | 0.477 ± 0.374 | 65.079 ± 16.561 | 0.547 ± 0.402 | 66.667 ± 22.498 | 0.623 ± 0.346 |
| Dapper | 47.13 ± 15.14 | 0.312 ± 0.081 | 47.89 ± 3.26 | 0.351 ± 0.049 | 44.75 ± 13.67 | 0.382 ± 0.068 |
| ForDigitStress | 55.23 ± 1.35 | 0.398 ± 0.024 | 54.89 ± 1.40 | 0.377 ± 0.025 | 56.01 ± 1.28 | 0.351 ± 0.023 |
| ADARP | 35.278 ± 11.96 | 0.118 ± 0.027 | 40.299 ± 12.54 | 0.153 ± 0.012 | 38.73 ± 11.68 | 0.161 ± 0.017 |
| Exercise | 40.28 ± 1.40 | 0.218 ± 0.025 | 38.17 ± 1.45 | 0.186 ± 0.027 | 36.93 ± 1.38 | 0.153 ± 0.026 |
| MOCAS | 54.27 ± 5.8 | 0.309 ± 0.052 | 50.4 ± 5.41 | 0.291 ± 0.058 | 51.05 ± 6.52 | 0.262 ± 0.018 |
| LAUREATE | 40.556 ± 2.81 | 0.215 ± 0.080 | 46.970 ± 3.27 | 0.225 ± 0.073 | 41.12 ± 9.55 | 0.188 ± 0.057 |
| VERBIO | 35.21 ± 2.04 | 0.205 ± 0.041 | 28.7 ± 7.19 | 0.12 ± 0.041 | 29.8 ± 6.92 | 0.233 ± 0.016 |

Table 8: Performance comparison of raw-signal based deep learning models on **valence classification** using **EDA modality**. Accuracy and F1 scores are reported with standard deviations.

### 2.3.5 Pretrained Models - CLSP CNN

Table 41 presents the arousal classification results, Table 42 shows the valence classification results, and Table 43 shows the four-class classification results, based on EDA+PPG data using a fine-tuned CLSP model using 1D-CNN-based MetaNet.

## 3 Cross-Data Analysis Results

This section presents the results of cross-data analysis, systematically organized into three key categories: device, label, and setting.

### 3.1 Device Group

The device group results are presented across four tables, corresponding to data collected from commercially available wearables (e.g., Empatica E4), custom-built wearable devices, lab-based equipment, in-cohort, and zero-short experiments. These results highlight the cross-group transferability of models, offering insights into how well models trained on one type of device generalize to

| Dataset | Arousal | | Valence | | Four-Class | |
|---|---|---|---|---|---|---|
| | **Accuracy** | **F1-score** | **Accuracy** | **F1-score** | **Accuracy** | **F1-score** |
| WESAD | 53.75 | 0.647 | 55.00 | 0.680 | 25.83 | 0.221 |
| NURSE | 57.73 | 0.420 | 50.59 | 0.623 | 56.33 | 0.433 |
| EMOGNITION | 57.12 | 0.675 | 40.51 | 0.533 | 42.41 | 0.315 |
| UBFC_PHYS | 58.48 | 0.429 | 50.59 | 0.623 | 68.90 | 0.705 |
| PhyMER | 42.72 | 0.519 | 57.35 | 0.723 | 52.80 | 0.571 |
| EmoWear | 51.10 | 0.604 | 62.08 | 0.752 | 29.41 | 0.268 |
| MAUS | 55.30 | 0.647 | 38.26 | 0.478 | 68.45 | 0.671 |
| CLAS | 49.67 | 0.633 | 50.17 | 0.643 | 25.06 | 0.186 |
| CASE | 44.24 | 0.436 | 45.38 | 0.493 | 19.84 | 0.210 |
| Unobtrusive | 59.46 | 0.725 | 56.74 | 0.712 | 44.02 | 0.395 |
| CEAP-360VR | 48.83 | 0.520 | 47.27 | 0.611 | 28.96 | 0.259 |
| ScientISST MOVE | 37.78 | 0.462 | 62.22 | 0.767 | 65.91 | 0.642 |
| Dapper | 57.76 | 0.701 | 74.04 | 0.847 | 31.47 | 0.335 |
| ForDigitStress | 82.11 | 0.896 | 73.25 | 0.844 | 39.86 | 0.419 |
| ADARP | 52.94 | 0.667 | 17.64 | 0.300 | 52.79 | 0.269 |
| Exercise | 50.90 | 0.625 | 61.45 | 0.748 | 35.18 | 0.298 |
| MOCAS | 46.93 | 0.543 | 79.59 | 0.885 | 16.83 | 0.159 |
| LAUREATE | 26.13 | 0.000 | 72.32 | 0.000 | 11.72 | 0.117 |
| VERBIO | 59.34 | 0.703 | 33.87 | 0.367 | 52.90 | 0.480 |

Table 9: Performance comparison of Zero-shot using CLSP Model on **arousal, valence,** and **four-class classification** using **EDA modality**.

| Dataset | Fine-tune - 5% | | Fine-tune - 25% | | Fine-tune - 50% | |
|---|---|---|---|---|---|---|
| | **Accuracy** | **F1-score** | **Accuracy** | **F1-score** | **Accuracy** | **F1-score** |
| WESAD | 52.68 | 0.635 | 63.45 | 0.692 | 70.26 | 0.698 |
| NURSE | 48.00 | 0.623 | 39.69 | 0.508 | 48.44 | 0.474 |
| EMOGNITION | 55.39 | 0.686 | 54.96 | 0.626 | 50.22 | 0.536 |
| UBFC_PHYS | 40.92 | 0.450 | 58.18 | 0.192 | 61.76 | 0.156 |
| PhyMER | 52.11 | 0.311 | 60.76 | 0.104 | 60.32 | 0.101 |
| EmoWear | 49.78 | 0.505 | 49.74 | 0.545 | 51.62 | 0.593 |
| MAUS | 64.77 | 0.779 | 63.83 | 0.759 | 65.91 | 0.763 |
| CLAS | 55.90 | 0.508 | 56.53 | 0.529 | 59.53 | 0.601 |
| CASE | 67.95 | 0.381 | 73.71 | 0.231 | 73.48 | 0.195 |
| Unobtrusive | 76.21 | 0.859 | 74.69 | 0.848 | 71.21 | 0.817 |
| CEAP-360VR | 44.53 | 0.554 | 48.05 | 0.533 | 53.91 | 0.435 |
| ScientISST MOVE | 37.38 | 0.503 | 37.79 | 0.450 | 39.95 | 0.475 |
| Dapper | 63.69 | 0.773 | 61.86 | 0.745 | 63.31 | 0.761 |
| ForDigitStress | 86.36 | 0.926 | 90.96 | 0.949 | 90.94 | 0.948 |
| ADARP | Enough Data Not Available | | 75.00 | 0.833 | 68.75 | 0.773 |
| Exercise | 52.03 | 0.617 | 55.04 | 0.483 | 57.99 | 0.537 |
| MOCAS | 48.44 | 0.649 | 50.48 | 0.636 | 48.45 | 0.591 |
| LAUREATE | 26.13 | 0.000 | 26.13 | 0.000 | 26.13 | 0.000 |
| VERBIO | 70.46 | 0.820 | 68.24 | 0.793 | 62.37 | 0.747 |

Table 10: Performance comparison of CLSP pre-trained model fine-tuned with **5%**, **25%**, and **50%** on **arousal classification** using **EDA modality** using **MLP-based MetaNet**. Accuracy and F1 scores are reported.

others. These results are further subdivided by modality, EDA, PPG, and the combined EDA+PPG, to evaluate how each physiological signal contributes to cross-device generalization.

### 3.1.1 EDA

This section presents the device group experiment results for the EDA modality across four tables: commercially available wearables (e.g., Empatica E4) in Table 44, custom-built wearable devices in Table 45, lab-based equipment in Table 46, in-cohort, and zero-short experiments in Table 47.

| Dataset | Fine-tune - 5% | | Fine-tune - 25% | | Fine-tune - 50% | |
|---|---|---|---|---|---|---|
| | Accuracy | F1-score | Accuracy | F1-score | Accuracy | F1-score |
| WESAD | 75.06 | 0.731 | 75.67 | 0.723 | 83.31 | 0.821 |
| NURSE | 67.30 | 0.137 | 73.70 | 0.044 | 78.54 | 0.078 |
| EMOGNITION | 52.17 | 0.369 | 56.69 | 0.328 | 58.19 | 0.206 |
| UBFC_PHYS | 63.69 | 0.646 | 68.01 | 0.699 | 69.64 | 0.709 |
| PhyMER | 51.19 | 0.588 | 55.07 | 0.689 | 52.13 | 0.612 |
| EmoWear | 59.83 | 0.725 | 63.68 | 0.771 | 64.02 | 0.772 |
| MAUS | 67.80 | 0.416 | 70.45 | 0.349 | 68.18 | 0.328 |
| CLAS | 56.53 | 0.583 | 59.33 | 0.616 | 59.17 | 0.599 |
| CASE | 55.00 | 0.540 | 59.24 | 0.340 | 58.71 | 0.365 |
| Unobtrusive | 53.48 | 0.627 | 53.77 | 0.664 | 55.16 | 0.675 |
| CEAP-360VR | 53.52 | 0.567 | 59.77 | 0.561 | 50.78 | 0.377 |
| ScientISST MOVE | 64.23 | 0.718 | 69.51 | 0.782 | 73.69 | 0.822 |
| Dapper | 65.20 | 0.773 | 76.94 | 0.868 | 75.64 | 0.861 |
| ForDigitStress | 71.84 | 0.811 | 76.73 | 0.857 | 78.75 | 0.869 |
| ADARP | Enough Data Not Available | | 63.19 | 0.222 | 47.92 | 0.200 |
| Exercise | 59.50 | 0.694 | 58.66 | 0.698 | 61.03 | 0.713 |
| MOCAS | 66.81 | 0.786 | 71.97 | 0.825 | 77.55 | 0.873 |
| LAUREATE | 72.32 | 0.000 | 72.32 | 0.000 | 72.32 | 0.000 |
| VERBIO | 43.39 | 0.342 | 68.44 | 0.096 | 72.20 | 0.183 |

Table 11: Performance comparison of CLSP pre-trained model fine-tuned with **5%**, **25%**, and **50%** on **valence classification** using **EDA modality** using **MLP-based MetaNet**. Accuracy and F1 scores are reported.

| Dataset | Fine-tune - 5% | | Fine-tune - 25% | | Fine-tune - 50% | |
|---|---|---|---|---|---|---|
| | Accuracy | F1 Score | Accuracy | F1 Score | Accuracy | F1 Score |
| WESAD | 31.81 | 0.313 | 33.15 | 0.344 | 36.10 | 0.387 |
| NURSE | Not enough data | | | | | |
| EMOGNITION | 39.83 | 0.312 | 40.29 | 0.300 | 34.39 | 0.270 |
| UBFC_PHYS | Not enough data | | | | | |
| PhyMER | 52.48 | 0.498 | 47.58 | 0.463 | 53.01 | 0.370 |
| EmoWear | 26.95 | 0.241 | 27.32 | 0.259 | 28.75 | 0.247 |
| MAUS | Not enough data | | | | | |
| CLAS | 26.30 | 0.243 | 31.43 | 0.282 | 36.67 | 0.348 |
| CASE | 31.52 | 0.347 | 37.80 | 0.365 | 39.77 | 0.374 |
| Unobtrusive | Not enough data | | | | | |
| CEAP-360VR | 29.77 | 0.267 | 31.25 | 0.285 | 27.68 | 0.251 |
| ScientISST MOVE | Not enough data | | 71.54 | 0.701 | 63.77 | 0.618 |
| Dapper | 27.88 | 0.301 | 33.02 | 0.348 | 26.71 | 0.284 |
| ForDigitStress | 66.92 | 0.638 | 54.90 | 0.441 | 67.88 | 0.649 |
| ADARP | Not enough data | | | | | |
| Exercise | 34.75 | 0.285 | 33.92 | 0.263 | 32.84 | 0.242 |
| MOCAS | Not enough data | | 41.90 | 0.367 | 39.12 | 0.357 |
| LAUREATE | 61.21 | 0.527 | 62.10 | 0.370 | 63.40 | 0.417 |
| VERBIO | Not enough data | | | | | |

Table 12: Performance comparison of CLSP pre-trained model fine-tuned with **5%**, **25%**, and **50%** on **four-class classification** using **EDA modality** using **MLP-based MetaNet**.

### 3.1.2   PPG

This section presents the device group experiment results for the PPG modality across four tables: commercially available wearables (e.g., Empatica E4) in Table 48, custom-built wearable devices in Table 49, lab-based equipment in Table 50, in-cohort, and zero-short experiments in Table 51.

| Dataset | Fine-tune - 5% | | Fine-tune - 25% | | Fine-tune - 50% | |
|---|---|---|---|---|---|---|
| | **Accuracy** | **F1-score** | **Accuracy** | **F1-score** | **Accuracy** | **F1-score** |
| WESAD | 54.30 | 0.637 | 60.16 | 0.618 | 78.91 | 0.771 |
| NURSE | 49.57 | 0.628 | 45.88 | 0.411 | 45.55 | 0.466 |
| EMOGNITION | 56.46 | 0.607 | 53.67 | 0.598 | 57.98 | 0.657 |
| UBFC_PHYS | 58.18 | 0.414 | 62.35 | 0.218 | 59.52 | 0.202 |
| PhyMER | 56.94 | 0.268 | 56.31 | 0.237 | 60.03 | 0.094 |
| EmoWear | 50.67 | 0.546 | 51.77 | 0.512 | 50.87 | 0.553 |
| MAUS | 59.09 | 0.722 | 63.83 | 0.754 | 70.27 | 0.805 |
| CLAS | 56.83 | 0.543 | 57.60 | 0.600 | 59.07 | 0.603 |
| CASE | 70.61 | 0.179 | 73.18 | 0.142 | 72.95 | 0.225 |
| Unobtrusive | 71.04 | 0.820 | 75.84 | 0.853 | 76.77 | 0.858 |
| CEAP-360VR | 42.97 | 0.563 | 47.66 | 0.269 | 52.73 | 0.319 |
| ScientISST MOVE | 54.13 | 0.532 | 52.09 | 0.505 | 64.64 | 0.551 |
| Dapper | 59.23 | 0.719 | 59.21 | 0.718 | 64.51 | 0.774 |
| ForDigitStress | 82.73 | 0.928 | 88.81 | 0.936 | 87.80 | 0.929 |
| ADARP | Enough Data Not Available | | 52.08 | 0.673 | 75.00 | 0.833 |
| Exercise | 54.36 | 0.596 | 60.52 | 0.537 | 62.17 | 0.560 |
| MOCAS | 51.49 | 0.650 | 46.44 | 0.442 | 49.46 | 0.630 |
| LAUREATE | 26.13 | 0.000 | 26.13 | 0.000 | 26.13 | 0.000 |
| VERBIO | 72.90 | 0.830 | 60.24 | 0.730 | 62.25 | 0.751 |

Table 13: Performance comparison of CLSP pre-trained model fine-tuned with **5%**, **25%**, and **50%** on **arousal classification** using **EDA modality** using **1D-CNN-based MetaNet**. Accuracy and F1 scores are reported.

| Dataset | Fine-tune - 5% | | Fine-tune - 25% | | Fine-tune - 50% | |
|---|---|---|---|---|---|---|
| | **Accuracy** | **F1-score** | **Accuracy** | **F1-score** | **Accuracy** | **F1-score** |
| WESAD | 74.22 | 0.725 | 77.79 | 0.774 | 85.32 | 0.831 |
| NURSE | 75.84 | 0.264 | 75.94 | 0.000 | 76.83 | 0.153 |
| EMOGNITION | 58.38 | 0.352 | 60.77 | 0.250 | 59.06 | 0.427 |
| UBFC_PHYS | 68.60 | 0.695 | 67.71 | 0.686 | 69.79 | 0.696 |
| PhyMER | 52.30 | 0.646 | 53.90 | 0.661 | 53.72 | 0.645 |
| EmoWear | 69.23 | 0.734 | 64.77 | 0.784 | 64.83 | 0.784 |
| MAUS | 70.08 | 0.350 | 67.80 | 0.406 | 67.05 | 0.419 |
| CLAS | 56.63 | 0.577 | 59.00 | 0.634 | 59.53 | 0.601 |
| CASE | 55.53 | 0.336 | 58.48 | 0.368 | 59.09 | 0.438 |
| Unobtrusive | 54.98 | 0.653 | 57.16 | 0.706 | 57.14 | 0.707 |
| CEAP-360VR | 51.17 | 0.620 | 50.00 | 0.476 | 55.47 | 0.505 |
| ScientISST MOVE | 64.51 | 0.724 | 73.28 | 0.813 | 70.98 | 0.798 |
| Dapper | 68.57 | 0.804 | 76.41 | 0.866 | 77.71 | 0.871 |
| ForDigitStress | 78.86 | 0.872 | 75.28 | 0.841 | 77.85 | 0.865 |
| ADARP | Enough Data Not Available | | 53.47 | 0.000 | 63.19 | 0.143 |
| Exercise | 57.01 | 0.665 | 61.54 | 0.723 | 61.98 | 0.693 |
| MOCAS | 63.71 | 0.750 | 73.98 | 0.846 | 75.51 | 0.857 |
| LAUREATE | 72.32 | 0.000 | 72.32 | 0.000 | 72.32 | 0.000 |
| VERBIO | 56.28 | 0.239 | 70.25 | 0.096 | 73.32 | 0.049 |

Table 14: Performance comparison of CLSP pre-trained model fine-tuned with **5%**, **25%**, and **50%** on **valence classification** using **EDA modality** using **1D-CNN-based MetaNet**. Accuracy and F1 scores are reported.

### 3.1.3 EDA + PPG

This section presents the device group experiment results for the EDA+PPG modality across four tables: commercially available wearables (e.g., Empatica E4) in Table 52, custom-built wearable devices in Table 53, lab-based equipment in Table 54, in-cohort experiments, and zero-short in Table 55.

| Dataset | Fine-tune - 5% | | Fine-tune - 25% | | Fine-tune - 50% | |
|---|---|---|---|---|---|---|
| | Accuracy | F1 Score | Accuracy | F1 Score | Accuracy | F1 Score |
| WESAD | 47.77 | 0.447 | 71.60 | 0.717 | 84.38 | 0.840 |
| NURSE | Not enough data | | | | | |
| EMOGNITION | 36.13 | 0.342 | 33.61 | 0.310 | 40.51 | 0.390 |
| UBFC_PHYS | Not enough data | | | | | |
| PhyMER | 49.25 | 0.504 | 43.06 | 0.303 | 57.35 | 0.723 |
| EmoWear | 26.67 | 0.283 | 25.10 | 0.208 | 28.33 | 0.293 |
| MAUS | Not enough data | | | | | |
| CLAS | 37.13 | 0.353 | 37.37 | 0.362 | 40.07 | 0.388 |
| CASE | 38.71 | 0.351 | 43.79 | 0.374 | 43.41 | 0.373 |
| Unobtrusive | Not enough data | | | | | |
| CEAP-360VR | 29.41 | 0.263 | 30.12 | 0.276 | 29.79 | 0.272 |
| ScientISST MOVE | Not enough data | | 69.85 | 0.684 | 67.88 | 0.657 |
| Dapper | 28.93 | 0.312 | 30.15 | 0.327 | 37.50 | 0.367 |
| ForDigitStress | 72.30 | 0.653 | 69.18 | 0.657 | 68.65 | 0.656 |
| ADARP | Not enough data | | | | | |
| Exercise | 32.11 | 0.221 | 55.28 | 0.552 | 53.64 | 0.537 |
| MOCAS | Not enough data | | 37.76 | 0.327 | 37.73 | 0.339 |
| LAUREATE | 60.60 | 0.457 | 60.60 | 0.457 | 60.60 | 0.457 |
| VERBIO | Not enough data | | | | | |

Table 15: Performance comparison of CLSP pre-trained model fine-tuned with **5%**, **25%**, and **50%** on **four-class classification** using **EDA modality** using **1D-CNN-based MetaNet**.

| Dataset | RF | | LDA | |
|---|---|---|---|---|
| | Accuracy | F1 | Accuracy | F1 |
| WESAD | 70.3 ± 9.5 | 0.670 ± 0.120 | 71.0 ± 8.7 | 0.655 ± 0.115 |
| NURSE | 46.25 ± 7.80 | 0.248 ± 0.114 | 43.10 ± 8.15 | 0.230 ± 0.098 |
| EMOGNITION | 52.11 ± 9.23 | 0.448 ± 0.144 | 49.85 ± 10.12 | 0.395 ± 0.117 |
| UBFC_PHYS | 66.52 ± 8.51 | 0.078 ± 0.037 | 66.22 ± 5.62 | 0.186 ± 0.018 |
| PhyMER | 60.792 ± 10.185 | 0.137 ± 0.105 | 51.418 ± 9.462 | 0.421 ± 0.131 |
| EmoWear | 53.225 ± 13.752 | 0.644 ± 0.169 | 50.114 ± 9.797 | 0.496 ± 0.170 |
| MAUS | 73.674 ± 10.156 | 0.815 ± 0.089 | 66.856 ± 11.535 | 0.718 ± 0.137 |
| CLAS | 59.100 ± 5.792 | 0.662 ± 0.057 | 56.767 ± 6.152 | 0.549 ± 0.083 |
| CASE | 73.030 ± 12.487 | 0.083 ± 0.109 | 52.348 ± 16.076 | 0.341 ± 0.159 |
| Unobtrusive | 78.677 ± 7.196 | 0.869 ± 0.049 | 64.378 ± 13.970 | 0.735 ± 0.130 |
| CEAP-360VR | 58.594 ± 14.354 | 0.162 ± 0.217 | 50.781 ± 18.496 | 0.328 ± 0.290 |
| ScientISST MOVE | 86.746 ± 14.572 | 0.769 ± 0.278 | 82.937 ± 15.310 | 0.718 ± 0.270 |
| Dapper | 45.321 ± 6.872 | 0.379 ± 0.018 | 44.189 ± 7.504 | 0.336 ± 0.012 |
| ForDigitStress | 51.2 ± 8.3 | 0.28 ± 0.11 | 48.7 ± 7.9 | 0.23 ± 0.08 |
| ADARP | 42.55 ± 8.13 | 0.17 ± 0.011 | 42.38 ± 7.57 | 0.141 ± 0.062 |
| Exercise | 50.3 ± 7.4 | 0.24 ± 0.09 | 48.2 ± 6.9 | 0.22 ± 0.08 |
| MOCAS | 41.213 ± 6.384 | 0.292 ± 0.107 | 42.89 ± 7.029 | 0.267 ± 0.098 |
| LAUREATE | 35.81 ± 5.96 | 0.271 ± 0.083 | 34.907 ± 6.425 | 0.314 ± 0.076 |
| VERBIO | 37.231 ± 6.281 | 0.239 ± 0.074 | 36.524 ± 6.814 | 0.231 ± 0.069 |

Table 16: Performance comparison of Random Forest (RF) and Linear Discriminant Analysis (LDA) models on **arousal classification** across multiple datasets using **PPG modality**. Accuracy and F1 scores are shown as mean ± standard deviation.

| Dataset | RF | | LDA | |
|---|---|---|---|---|
| | **Accuracy** | **F1** | **Accuracy** | **F1** |
| WESAD | 65.02 ± 10.27 | 0.480 ± 0.110 | 66.41 ± 9.51 | 0.458 ± 0.100 |
| NURSE | 75.103 ± 5.325 | 0.042 ± 0.055 | 72.489 ± 6.1 | 0.059 ± 0.017 |
| EMOGNITION | 50.1 ± 13.5 | 0.231 ± 0.120 | 51.6 ± 12 | 0.21 ± 0.112 |
| UBFC_PHYS | 68.01 ± 7.25 | 0.664 ± 0.057 | 67.11 ± 5.97 | 0.681 ± 0.167 |
| PhyMER | 53.247 ± 14.753 | 0.646 ± 0.162 | 55.031 ± 12.057 | 0.549 ± 0.165 |
| EmoWear | 64.151 ± 12.145 | 0.769 ± 0.093 | 51.834 ± 10.608 | 0.557 ± 0.181 |
| MAUS | 73.295 ± 9.932 | 0.490 ± 0.180 | 66.856 ± 11.535 | 0.562 ± 0.118 |
| CLAS | 61.900 ± 6.372 | 0.503 ± 0.105 | 58.967 ± 7.192 | 0.589 ± 0.090 |
| CASE | 58.939 ± 11.771 | 0.292 ± 0.120 | 58.106 ± 11.660 | 0.477 ± 0.215 |
| Unobtrusive | 59.763 ± 6.922 | 0.713 ± 0.078 | 54.474 ± 6.605 | 0.552 ± 0.167 |
| CEAP-360VR | 57.812 ± 12.994 | 0.348 ± 0.250 | 56.641 ± 22.892 | 0.509 ± 0.280 |
| ScientISST MOVE | 75.119 ± 16.944 | 0.748 ± 0.258 | 49.087 ± 24.483 | 0.547 ± 0.263 |
| Dapper | 45.213 ± 5.821 | 0.496 ± 0.108 | 44.981 ± 6.134 | 0.474 ± 0.123 |
| ForDigitStress | 52.35 ± 3.33 | 0.420 ± 0.08 | 49.50 ± 4.52 | 0.406 ± 0.06 |
| ADARP | 45.231 ± 6.134 | 0.324 ± 0.093 | 44.872 ± 5.981 | 0.341 ± 0.087 |
| Exercise | 50.92 ± 3.35 | 0.318 ± 0.087 | 49.38 ± 3.97 | 0.299 ± 0.093 |
| MOCAS | 32.415 ± 4.890 | 0.189 ± 0.056 | 31.832 ± 4.557 | 0.195 ± 0.049 |
| LAUREATE | 42.328 ± 6.102 | 0.394 ± 0.090 | 41.905 ± 5.894 | 0.307 ± 0.085 |
| VERBIO | 40.205 ± 5.781 | 0.364 ± 0.083 | 39.631 ± 5.523 | 0.371 ± 0.077 |

Table 17: Performance comparison of Random Forest (RF) and Linear Discriminant Analysis (LDA) models on **valence classification** across multiple datasets using **PPG modality**. Accuracy and F1 scores are shown as mean ± standard deviation.

| Dataset | RF | | LDA | |
|---|---|---|---|---|
| | **Accuracy** | **F1 Score** | **Accuracy** | **F1 Score** |
| WESAD | 80.42 | 0.794 | 75.83 | 0.750 |
| NURSE | 41.56 | 0.421 | 40.51 | 0.447 |
| EMOGNITION | 38.58 | 0.347 | 32.14 | 0.320 |
| UBFC_PHYS | 50.00 | 0.513 | 49.11 | 0.551 |
| PhyMER | 37.28 | 0.300 | 27.08 | 0.274 |
| EmoWear | 31.78 | 0.265 | 25.70 | 0.253 |
| MAUS | 73.30 | 0.705 | 66.86 | 0.666 |
| CLAS | 41.17 | 0.401 | 37.10 | 0.369 |
| CASE | 41.52 | 0.397 | 31.36 | 0.344 |
| Unobtrusive | 42.43 | 0.386 | 36.74 | 0.351 |
| CEAP-360VR | 33.20 | 0.307 | 28.91 | 0.291 |
| ScientISST MOVE | 72.06 | 0.686 | 54.76 | 0.526 |
| Dapper | 47.94 | 0.426 | 29.26 | 0.318 |
| ForDigitStress | 87.19 | 0.821 | 57.97 | 0.645 |
| ADARP | 26.04 | 0.283 | 16.67 | 0.229 |
| Exercise | 46.67 | 0.436 | 37.21 | 0.372 |
| MOCAS | 34.27 | 0.357 | 25.40 | 0.310 |
| LAUREATE | 56.40 | 0.460 | 25.33 | 0.278 |
| VERBIO | 35.70 | 0.340 | 49.63 | 0.536 |

Table 18: Performance comparison of Random Forest (RF) and Linear Discriminant Analysis (LDA) models on **four-class classification** across multiple datasets using **PPG modality**.

| Dataset | HC+MLP | | HC+RESNET | | HC+LSTM | | HC+Attention+MLP | |
|---|---|---|---|---|---|---|---|---|
| | Accuracy | F1 Score | Accuracy | F1 Score | Accuracy | F1 Score | Accuracy | F1 Score |
| WESAD | 80 ± 7.8 | 0.80 ± 0.11 | 46.1 ± 6.3 | 0.180 ± 0.025 | 44.8 ± 8.1 | 0.170 ± 0.04 | 43.5 ± 6.9 | 0.160 ± 0.035 |
| NURSE | 45.1 ± 6.92 | 0.38 ± 0.09 | 52.05 ± 7.01 | 0.265 ± 0.107 | 43.39 ± 8.90 | 0.251 ± 0.110 | 46.62 ± 6.83 | 0.239 ± 0.105 |
| EMOGNITION | 49.94 ± 8.45 | 0.53 ± 0.103 | 41.26 ± 7.88 | 0.312 ± 0.094 | 39.88 ± 9.11 | 0.287 ± 0.105 | 40.32 ± 10.54 | 0.291 ± 0.111 |
| UBFC_PHYS | 65.03 ± 6.67 | 0.167 ± 0.084 | 67.71 ± 9.42 | 0.129 ± 0.030 | 59.07 ± 9.35 | 0.18 ± 0.036 | 57.41 ± 2.69 | 0.141 ± 0.036 |
| PhyMER | 55.02 ± 7.39 | 0.370 ± 0.091 | 55.09 ± 8.62 | 0.334 ± 0.107 | 50.97 ± 15.12 | 0.390 ± 0.271 | 47.07 ± 8.42 | 0.420 ± 0.159 |
| EmoWear | 48.75 ± 9.53 | 0.525 ± 0.158 | 51.44 ± 8.75 | 0.577 ± 0.155 | 47.24 ± 16.65 | 0.305 ± 0.323 | 51.04 ± 11.19 | 0.538 ± 0.199 |
| MAUS | 67.99 ± 13.20 | 0.748 ± 0.131 | 68.37 ± 13.03 | 0.765 ± 0.118 | 59.85 ± 16.49 | 0.610 ± 0.318 | 55.87 ± 19.19 | 0.476 ± 0.386 |
| CLAS | 58.63 ± 6.01 | 0.579 ± 0.083 | 56.03 ± 7.52 | 0.608 ± 0.076 | 51.87 ± 5.19 | 0.384 ± 0.309 | 55.40 ± 3.78 | 0.389 ± 0.186 |
| CASE | 61.29 ± 7.79 | 0.299 ± 0.149 | 66.29 ± 9.23 | 0.244 ± 0.144 | 61.14 ± 24.36 | 0.184 ± 0.207 | 53.33 ± 12.64 | 0.294 ± 0.149 |
| Unobtrusive | 71.85 ± 10.55 | 0.808 ± 0.082 | 70.01 ± 7.82 | 0.805 ± 0.059 | 60.11 ± 22.81 | 0.604 ± 0.372 | 54.96 ± 23.89 | 0.556 ± 0.370 |
| CEAP-360VR | 54.69 ± 14.81 | 0.284 ± 0.278 | 56.64 ± 16.49 | 0.247 ± 0.244 | 44.14 ± 15.87 | 0.334 ± 0.233 | 55.86 ± 17.96 | 0.289 ± 0.321 |
| ScientISST MOVE | 81.83 ± 16.20 | 0.714 ± 0.251 | 81.83 ± 17.08 | 0.722 ± 0.205 | 53.21 ± 22.76 | 0.221 ± 0.288 | 84.84 ± 14.58 | 0.742 ± 0.258 |
| Dapper | 46.75 ± 5.92 | 0.402 ± 0.096 | 41.14 ± 6.83 | 0.267 ± 0.091 | 38.21 ± 7.23 | 0.212 ± 0.087 | 35.88 ± 6.25 | 0.242 ± 0.095 |
| ForDigitStress | 47.10 ± 6.50 | 0.210 ± 0.090 | 41.90 ± 7.30 | 0.190 ± 0.070 | 38.30 ± 6.10 | 0.150 ± 0.060 | 39.00 ± 5.80 | 0.170 ± 0.080 |
| ADARP | 47.97 ± 6.52 | 0.216 ± 0.046 | 39.11 ± 6.52 | 0.143 ± 0.031 | 37.42 ± 6.10 | 0.153 ± 0.041 | 39.62 ± 6.21 | 0.132 ± 0.034 |
| Exercise | 52.70 ± 5.50 | 0.190 ± 0.070 | 40.90 ± 6.10 | 0.180 ± 0.060 | 39.80 ± 5.60 | 0.170 ± 0.070 | 43.30 ± 5.70 | 0.150 ± 0.060 |
| MOCAS | 42.59 ± 6.74 | 0.275 ± 0.089 | 37.77 ± 7.52 | 0.202 ± 0.094 | 39.51 ± 3.91 | 0.191 ± 0.092 | 38.25 ± 2.13 | 0.215 ± 0.080 |
| LAUREATE | 41.87 ± 6.91 | 0.312 ± 0.074 | 30.29 ± 7.52 | 0.205 ± 0.068 | 31.02 ± 7.76 | 0.219 ± 0.073 | 28.90 ± 8.44 | 0.183 ± 0.079 |
| VERBIO | 34.11 ± 6.92 | 0.225 ± 0.062 | 32.87 ± 7.19 | 0.192 ± 0.066 | 31.24 ± 8.11 | 0.185 ± 0.058 | 32.04 ± 8.37 | 0.198 ± 0.061 |

Table 19: Performance comparison of hybrid handcrafted (HC) feature-based deep learning models on **arousal classification** using **PPG modality** across multiple models. Accuracy and F1 scores are reported as mean ± standard deviation.

| Dataset | HC+MLP | | HC+RESNET | | HC+LSTM | | HC+Attention+MLP | |
|---|---|---|---|---|---|---|---|---|
| | Accuracy | F1 | Accuracy | F1 | Accuracy | F1 | Accuracy | F1 |
| WESAD | 75 ± 5.82 | 0.75 ± 0.182 | 41.32 ± 7.34 | 0.402 ± 0.125 | 42.15 ± 8.94 | 0.414 ± 0.128 | 39.60 ± 9.28 | 0.386 ± 0.120 |
| NURSE | 72.08 ± 8.98 | 0.05 ± 0.009 | 79.1 ± 4.81 | 0.299 ± 0.180 | 50.21 ± 17.9 | 0.225 ± 0.300 | 51.83 ± 13.22 | 0.310 ± 0.090 |
| EMOGNITION | 50.63 ± 8.10 | 0.28 ± 0.180 | 44.1 ± 11.7 | 0.218 ± 0.128 | 45.6 ± 20.6 | 0.199 ± 0.180 | 44 ± 11.2 | 0.23 ± 0.183 |
| UBFC_PHYS | 66.67 ± 4.88 | 0.673 ± 0.046 | 44.8 ± 7.23 | 0.383 ± 0.058 | 55.51 ± 9.75 | 0.477 ± 0.030 | 59.38 ± 9.66 | 0.486 ± 0.037 |
| PhyMER | 53.57 ± 9.70 | 0.573 ± 0.163 | 53.37 ± 10.24 | 0.599 ± 0.155 | 54.43 ± 21.83 | 0.271 ± 0.380 | 53.75 ± 11.42 | 0.561 ± 0.157 |
| EmoWear | 54.62 ± 9.93 | 0.633 ± 0.124 | 58.76 ± 8.16 | 0.697 ± 0.084 | 49.58 ± 18.55 | 0.370 ± 0.387 | 47.41 ± 12.12 | 0.423 ± 0.271 |
| MAUS | 68.37 ± 12.31 | 0.512 ± 0.213 | 67.05 ± 11.28 | 0.446 ± 0.154 | 58.90 ± 14.45 | 0.397 ± 0.220 | 54.17 ± 20.61 | 0.463 ± 0.207 |
| CLAS | 61.47 ± 5.66 | 0.602 ± 0.070 | 58.23 ± 6.50 | 0.514 ± 0.093 | 52.90 ± 5.21 | 0.381 ± 0.318 | 58.47 ± 6.50 | 0.572 ± 0.156 |
| CASE | 55.61 ± 8.64 | 0.405 ± 0.173 | 55.61 ± 11.72 | 0.376 ± 0.122 | 50.23 ± 15.66 | 0.341 ± 0.240 | 56.67 ± 8.89 | 0.413 ± 0.189 |
| Unobtrusive | 55.60 ± 7.47 | 0.612 ± 0.110 | 54.84 ± 8.35 | 0.635 ± 0.120 | 50.81 ± 12.58 | 0.413 ± 0.371 | 55.05 ± 9.30 | 0.582 ± 0.158 |
| CEAP-360VR | 57.03 ± 18.50 | 0.461 ± 0.265 | 51.56 ± 14.81 | 0.369 ± 0.238 | 50.39 ± 15.06 | 0.341 ± 0.277 | 52.73 ± 16.42 | 0.387 ± 0.295 |
| ScientISST MOVE | 65.44 ± 16.83 | 0.638 ± 0.278 | 73.21 ± 18.32 | 0.736 ± 0.238 | 51.63 ± 22.30 | 0.506 ± 0.375 | 64.84 ± 15.80 | 0.661 ± 0.227 |
| Dapper | 46.87 ± 7.62 | 0.508 ± 0.117 | 42.51 ± 6.74 | 0.421 ± 0.095 | 41.23 ± 8.01 | 0.403 ± 0.110 | 43.15 ± 6.95 | 0.438 ± 0.103 |
| ForDigitStress | 44.20 ± 2.80 | 0.327 ± 0.070 | 43.10 ± 2.10 | 0.290 ± 0.060 | 46.82 ± 4.59 | 0.319 ± 0.075 | 44.14 ± 5.03 | 0.308 ± 0.081 |
| ADARP | 43.33 ± 5.87 | 0.351 ± 0.101 | 42.13 ± 5.23 | 0.312 ± 0.078 | 41.33 ± 6.51 | 0.297 ± 0.099 | 41.82 ± 6.13 | 0.308 ± 0.084 |
| Exercise | 54.91 ± 2.93 | 0.308 ± 0.064 | 46.07 ± 2.84 | 0.257 ± 0.071 | 44.62 ± 3.41 | 0.242 ± 0.059 | 45.11 ± 2.78 | 0.251 ± 0.067 |
| MOCAS | 30.47 ± 4.60 | 0.402 ± 0.058 | 30.13 ± 4.20 | 0.181 ± 0.047 | 32.03 ± 4.91 | 0.171 ± 0.051 | 29.98 ± 4.42 | 0.176 ± 0.049 |
| LAUREATE | 41.10 ± 5.66 | 0.412 ± 0.094 | 39.90 ± 5.37 | 0.281 ± 0.082 | 39.28 ± 5.81 | 0.271 ± 0.091 | 39.71 ± 5.60 | 0.278 ± 0.087 |
| VERBIO | 38.87 ± 5.49 | 0.376 ± 0.084 | 37.12 ± 5.20 | 0.249 ± 0.071 | 36.54 ± 5.80 | 0.239 ± 0.079 | 36.81 ± 5.40 | 0.244 ± 0.075 |

Table 20: Performance comparison of hybrid handcrafted (HC) feature-based deep learning models on **valence classification** using **PPG modality** across multiple models. Accuracy and F1 scores are reported as mean ± standard deviation.

## 3.2 Label Group

The label group results are presented across four tables, corresponding to different annotation sources: stimulus-based labels, self-reports, expert annotations, in-cohort, and zero-short experiments. These results highlight the cross-group transferability of models across labeling strategies, offering insights into how well models trained using one type of labeling method perform when applied to data annotated differently. This is particularly important for real-world applications, where consistency in labeling is often challenging. The results are further subdivided by modality, EDA, PPG, and the combined EDA+PPG, to assess the contribution of each physiological signal to the generalizability of models across annotation types.

### 3.2.1 EDA

This section presents the labeling methods experiment results for the EDA modality across three tables: stimulus-based labels in Table 56, self-reports in Table 57, expert annotations in Table 58, in-cohort, and zero-short experiments in Table 59.

### 3.2.2 PPG

This section presents the labeling methods experiment results for the PPG modality across three tables: stimulus-based labels in Table 60, self-reports in Table 61, expert annotations in Table 62, in-cohort, and zero-short experiments in Table 63.

| Dataset | HC+MLP | |
|---|---|---|
| | Accuracy | F1 Score |
| WESAD | 74.58 | 0.722 |
| NURSE | 41.10 | 0.452 |
| EMOGNITION | 37.74 | 0.363 |
| UBFC_PHYS | 44.84 | 0.501 |
| PhyMER | 27.80 | 0.297 |
| EmoWear | 26.45 | 0.270 |
| MAUS | 68.37 | 0.673 |
| CLAS | 40.17 | 0.408 |
| CASE | 34.85 | 0.369 |
| Unobtrusive | 40.74 | 0.385 |
| CEAP-360VR | 23.44 | 0.227 |
| ScientISST MOVE | 66.87 | 0.644 |
| Dapper | 34.27 | 0.359 |
| ForDigitStress | 71.01 | 0.729 |
| ADARP | 36.46 | 0.363 |
| Exercise | 42.44 | 0.438 |
| MOCAS | 30.59 | 0.338 |
| LAUREATE | 29.63 | 0.321 |
| VERBIO | 45.78 | 0.474 |

Table 21: Performance comparison of hybrid handcrafted (HC) feature-based deep learning models on **four-class classification** using **PPG modality** across multiple models.

| Dataset | Signal+CNN+Transformer | | Signal+LSTM | | Signal+RESNET | |
|---|---|---|---|---|---|---|
| | Accuracy | F1 | Accuracy | F1 | Accuracy | F1 |
| WESAD | 40.5 ± 10.3 | 0.14 ± 0.05 | 38.9 ± 11.7 | 0.08 ± 0.06 | 39.8 ± 9.9 | 0.090 ± 0.015 |
| NURSE | 31.64 ± 10.01 | 0.174 ± 0.142 | 29.82 ± 12.17 | 0.186 ± 0.108 | 33.17 ± 11.44 | 0.192 ± 0.125 |
| EMOGNITION | 38.76 ± 11.09 | 0.238 ± 0.120 | 39.11 ± 10.92 | 0.256 ± 0.129 | 40.47 ± 8.75 | 0.267 ± 0.113 |
| UBFC_PHYS | 42.85 ± 37.98 | 0.143 ± 0.03 | 47.62 ± 8.59 | 0.171 ± 0.033 | 53.42 ± 3.82 | 0.227 ± 0.035 |
| PhyMER | 47.761 ± 16.682 | 0.238 ± 0.265 | 54.322 ± 16.253 | 0.269 ± 0.306 | 54.194 ± 9.905 | 0.405 ± 0.185 |
| EmoWear | 52.217 ± 19.651 | 0.291 ± 0.372 | 45.748 ± 19.306 | 0.380 ± 0.357 | 47.492 ± 18.010 | 0.146 ± 0.205 |
| MAUS | 51.515 ± 16.988 | 0.436 ± 0.408 | 48.485 ± 16.988 | 0.364 ± 0.408 | 58.902 ± 23.156 | 0.545 ± 0.349 |
| CLAS | 50.067 ± 2.016 | 0.354 ± 0.345 | 50.067 ± 2.016 | 0.354 ± 0.345 | 57.667 ± 7.161 | 0.577 ± 0.125 |
| CASE | 46.364 ± 42.730 | 0.441 ± 0.482 | 46.364 ± 42.730 | 0.441 ± 0.482 | 48.712 ± 20.456 | 0.284 ± 0.252 |
| Unobtrusive | 31.882 ± 21.392 | 0.144 ± 0.330 | 41.121 ± 35.194 | 0.346 ± 0.456 | 59.102 ± 15.619 | 0.623 ± 0.255 |
| CEAP-360VR | 45.312 ± 20.018 | 0.198 ± 0.263 | 48.438 ± 20.515 | 0.275 ± 0.309 | 46.094 ± 19.165 | 0.221 ± 0.261 |
| ScientISST MOVE | 65.913 ± 18.361 | 0.405 ± 0.292 | 45.913 ± 22.609 | 0.226 ± 0.262 | 70.675 ± 14.080 | 0.438 ± 0.306 |
| Dapper | 36.741 ± 4.79 | 0.244 ± 0.022 | 34.208 ± 3.95 | 0.218 ± 0.097 | 35.309 ± 9.786 | 0.229 ± 0.005 |
| ForDigitStress | 32.5 ± 9.0 | 0.12 ± 0.05 | 39.7 ± 8.2 | 0.105 ± 0.06 | 34.6 ± 7.5 | 0.11 ± 0.04 |
| ADARP | 37.42 ± 8.5 | 0.129 ± 0.067 | 38.25 ± 7.59 | 0.119 ± 0.065 | 35.93 ± 7.01 | 0.173 ± 0.078 |
| Exercise | 41.2 ± 6.3 | 0.18 ± 0.03 | 39.4 ± 6.7 | 0.09 ± 0.04 | 36.1 ± 6.0 | 0.07 ± 0.03 |
| MOCAS | 37.426 ± 7.34 | 0.143 ± 0.011 | 36.9 ± 9.88 | 0.168 ± 0.095 | 38.75 ± 4.12 | 0.154 ± 0.013 |
| LAUREATE | 28.477 ± 8.028 | 0.198 ± 0.066 | 29.712 ± 7.884 | 0.190 ± 0.072 | 28.9 ± 7.86 | 0.14 ± 0.042 |
| VERBIO | 29.783 ± 8.994 | 0.172 ± 0.064 | 29.258 ± 8.410 | 0.181 ± 0.057 | 30.194 ± 8.363 | 0.276 ± 0.063 |

Table 22: Performance comparison of raw-signal based deep learning models on **arousal classification** using **PPG modality**. Accuracy and F1 scores are reported with standard deviations.

| Dataset | Signal+CNN+Transformer | | Signal+LSTM | | Signal+RESNET | |
|---|---|---|---|---|---|---|
| | **Accuracy** | **F1** | **Accuracy** | **F1** | **Accuracy** | **F1** |
| WESAD | 38.47 ± 12.10 | 0.172 ± 0.04 | 39.12 ± 11.38 | 0.16 ± 0.045 | 41.02 ± 10.74 | 0.181 ± 0.138 |
| NURSE | 39.05 ± 2 | 0.274 ± 0.011 | 31.96 ± 5.01 | 0.250 ± 0.094 | 30.487 ± 4.198 | 0.11 ± 0.02 |
| EMOGNITION | 41.3 ± 13.9 | 0.211 ± 0.154 | 42.2 ± 7.2 | 0.209 ± 0.167 | 42.1 ± 5.2 | 0.194 ± 0.160 |
| UBFC_PHYS | 52.82 ± 6.45 | 0.261 ± 0.051 | 53.57 ± 6.47 | 0.246 ± 0.064 | 54.286 ± 8.09 | 0.244 ± 0.018 |
| PhyMER | 45.971 ± 23.636 | 0.317 ± 0.372 | 49.365 ± 23.980 | 0.394 ± 0.384 | 52.600 ± 13.853 | 0.565 ± 0.190 |
| EmoWear | 51.844 ± 20.061 | 0.400 ± 0.409 | 53.443 ± 19.845 | 0.490 ± 0.390 | 50.909 ± 12.773 | 0.552 ± 0.206 |
| MAUS | 51.515 ± 16.988 | 0.227 ± 0.255 | 50.000 ± 17.059 | 0.250 ± 0.256 | 61.364 ± 20.900 | 0.531 ± 0.181 |
| CLAS | 49.400 ± 1.924 | 0.422 ± 0.312 | 49.400 ± 1.924 | 0.422 ± 0.312 | 56.433 ± 4.586 | 0.495 ± 0.243 |
| CASE | 47.576 ± 45.278 | 0.454 ± 0.494 | 47.576 ± 45.278 | 0.454 ± 0.494 | 51.439 ± 14.975 | 0.332 ± 0.262 |
| Unobtrusive | 48.092 ± 12.937 | 0.341 ± 0.381 | 48.803 ± 7.033 | 0.302 ± 0.365 | 54.641 ± 10.274 | 0.568 ± 0.214 |
| CEAP-360VR | 48.828 ± 13.604 | 0.245 ± 0.316 | 47.656 ± 13.258 | 0.344 ± 0.322 | 47.266 ± 14.457 | 0.319 ± 0.283 |
| ScientISST MOVE | 51.111 ± 23.977 | 0.509 ± 0.364 | 52.698 ± 22.585 | 0.480 ± 0.409 | 60.159 ± 25.540 | 0.633 ± 0.347 |
| Dapper | 41.732 ± 9.220 | 0.392 ± 0.128 | 40.956 ± 7.811 | 0.377 ± 0.1 | 41.482 ± 6.986 | 0.368 ± 0.114 |
| ForDigitStress | 37.02 ± 4.66 | 0.202 ± 0.054 | 36.48 ± 4.79 | 0.189 ± 0.069 | 38.10 ± 5.04 | 0.196 ± 0.073 |
| ADARP | 40.674 ± 6.781 | 0.288 ± 0.091 | 42.051 ± 5.939 | 0.295 ± 0.087 | 41.732 ± 5.604 | 0.281 ± 0.093 |
| Exercise | 40.87 ± 2.55 | 0.122 ± 0.044 | 41.52 ± 2.31 | 0.107 ± 0.052 | 40.63 ± 2.90 | 0.114 ± 0.049 |
| MOCAS | 28.617 ± 4.902 | 0.163 ± 0.052 | 29.201 ± 4.303 | 0.168 ± 0.050 | 28.974 ± 4.179 | 0.159 ± 0.054 |
| LAUREATE | 38.627 ± 6.013 | 0.259 ± 0.089 | 38.809 ± 5.714 | 0.265 ± 0.086 | 38.225 ± 5.887 | 0.251 ± 0.090 |
| VERBIO | 35.823 ± 5.954 | 0.225 ± 0.081 | 36.074 ± 5.605 | 0.231 ± 0.078 | 35.539 ± 5.682 | 0.218 ± 0.080 |

Table 23: Performance comparison of raw-signal based deep learning models on **valence classification** using **PPG modality**. Accuracy and F1 scores are reported with standard deviations.

| Dataset | Arousal | | Valence | | Four-Class | |
|---|---|---|---|---|---|---|
| | **Accuracy** | **F1** | **Accuracy** | **F1** | **Accuracy** | **F1** |
| WESAD | 58.33 | 0.315 | 42.08 | 0.270 | 22.08 | 0.160 |
| NURSE | 60.86 | 0.132 | 38.09 | 0.333 | 43.14 | 0.667 |
| EMOGNITION | 41.81 | 0.166 | 53.01 | 0.268 | 35.12 | 0.339 |
| UBFC_PHYS | 61.16 | 0.127 | 38.09 | 0.333 | 41.57 | 0.445 |
| PhyMER | 60.88 | 0.040 | 43.82 | 0.127 | 26.90 | 0.285 |
| EmoWear | 46.28 | 0.145 | 44.34 | 0.429 | 25.90 | 0.262 |
| MAUS | 43.37 | 0.328 | 52.84 | 0.204 | 74.41 | 0.554 |
| CLAS | 49.37 | 0.100 | 49.27 | 0.116 | 23.23 | 0.180 |
| CASE | 73.86 | 0.192 | 57.50 | 0.195 | 15.75 | 0.102 |
| Unobtrusive | 27.37 | 0.186 | 46.67 | 0.367 | 62.90 | 0.409 |
| CEAP-360VR | 56.25 | 0.200 | 49.22 | 0.369 | 24.00 | 0.230 |
| ScientISST MOVE | 66.67 | 0.375 | 51.11 | 0.532 | 63.00 | 0.600 |
| Dapper | 41.47 | 0.361 | 50.00 | 0.606 | 43.10 | 0.348 |
| ForDigitStress | 22.60 | 0.207 | 48.11 | 0.606 | 52.75 | 0.553 |
| ADARP | 29.41 | 0.142 | 64.70 | 0.400 | 62.67 | 0.433 |
| Exercise | 52.60 | 0.298 | 49.43 | 0.528 | 41.00 | 0.425 |
| MOCAS | 47.95 | 0.354 | 25.00 | 0.261 | 15.81 | 0.150 |
| LAUREATE | 36.48 | 0.288 | 56.12 | 0.377 | 27.76 | 0.286 |
| VERBIO | 38.49 | 0.251 | 53.52 | 0.307 | 51.06 | 0.582 |

Table 24: Zero-shot performance of the CLSP model on **Arousal**, **Valence**, and **Four-Class Classification** tasks across datasets using **PPG modality**. Accuracy and F1 scores are reported as mean values.

| Dataset | Fine-tune - 5% | | Fine-tune - 25% | | Fine-tune - 50% | |
|---|---|---|---|---|---|---|
| | **Accuracy** | **F1** | **Accuracy** | **F1** | **Accuracy** | **F1** |
| WESAD | 47.10 | 0.538 | 55.08 | 0.566 | 60.21 | 0.591 |
| NURSE | 46.87 | 0.492 | 49.44 | 0.510 | 57.29 | 0.521 |
| EMOGNITION | 51.10 | 0.566 | 54.31 | 0.585 | 56.04 | 0.625 |
| UBFC_PHYS | 52.83 | 0.347 | 63.69 | 0.318 | 63.84 | 0.291 |
| PhyMER | 56.85 | 0.237 | 60.15 | 0.133 | 60.27 | 0.156 |
| EmoWear | 49.40 | 0.548 | 50.14 | 0.544 | 50.35 | 0.583 |
| MAUS | 61.17 | 0.742 | 52.65 | 0.629 | 53.79 | 0.637 |
| CLAS | 51.33 | 0.558 | 55.43 | 0.523 | 55.30 | 0.561 |
| CASE | 70.38 | 0.170 | 71.82 | 0.136 | 73.48 | 0.058 |
| Unobtrusive | 64.61 | 0.773 | 70.73 | 0.820 | 73.93 | 0.842 |
| CEAP-360VR | 50.00 | 0.390 | 51.56 | 0.419 | 51.17 | 0.352 |
| ScientISST MOVE | 46.02 | 0.423 | 40.36 | 0.273 | 58.85 | 0.469 |
| Dapper | 58.10 | 0.661 | 56.24 | 0.657 | 55.75 | 0.662 |
| ForDigitStress | 67.68 | 0.788 | 76.20 | 0.854 | 81.73 | 0.891 |
| ADARP | Enough Data Not Available | | 59.03 | 0.6905 | 59.03 | 0.6905 |
| Exercise | 46.45 | 0.4667 | 53.27 | 0.418 | 56.47 | 0.4969 |
| MOCAS | 55.12 | 0.6209 | 45.88 | 0.4838 | 49.45 | 0.4921 |
| LAUREATE | 49.16 | 0.5725 | 63.19 | 0.7598 | 64.68 | 0.7755 |
| VERBIO | 53.91 | 0.650 | 55.38 | 0.669 | 56.36 | 0.6783 |

Table 25: Performance comparison of CLSP model fine-tuned with **5%**, **25%**, and **50%** on **arousal classification** using **PPG modality** using **MLP-based MetaNet**. Accuracy and F1 scores are reported.

| Dataset | Fine-tune - 5% | | Fine-tune - 25% | | Fine-tune - 50% | |
|---|---|---|---|---|---|---|
| | **Accuracy** | **F1** | **Accuracy** | **F1** | **Accuracy** | **F1** |
| WESAD | 46.21 | 0.580 | 53.12 | 0.626 | 57.48 | 0.607 |
| NURSE | 38.92 | 0.322 | 49.76 | 0.198 | 68.09 | 0.214 |
| EMOGNITION | 46.57 | 0.500 | 50.87 | 0.396 | 58.39 | 0.225 |
| UBFC_PHYS | 49.55 | 0.596 | 55.65 | 0.617 | 60.12 | 0.590 |
| PhyMER | 53.63 | 0.634 | 55.52 | 0.668 | 56.42 | 0.659 |
| EmoWear | 63.84 | 0.769 | 61.76 | 0.748 | 60.79 | 0.741 |
| MAUS | 47.16 | 0.439 | 47.73 | 0.246 | 54.36 | 0.269 |
| CLAS | 55.07 | 0.553 | 53.63 | 0.565 | 58.17 | 0.562 |
| CASE | 50.38 | 0.305 | 59.47 | 0.340 | 57.05 | 0.397 |
| Unobtrusive | 52.57 | 0.664 | 53.71 | 0.679 | 59.75 | 0.706 |
| CEAP-360VR | 48.83 | 0.570 | 47.27 | 0.551 | 54.69 | 0.582 |
| ScientISST MOVE | 51.15 | 0.646 | 55.33 | 0.690 | 53.98 | 0.677 |
| Dapper | 67.51 | 0.797 | 68.21 | 0.804 | 71.65 | 0.831 |
| ForDigitStress | 59.11 | 0.723 | 73.11 | 0.830 | 76.20 | 0.851 |
| ADARP | Enough Data Not Available | | 46.53 | 0.143 | 30.56 | 0.300 |
| Exercise | 57.96 | 0.681 | 59.41 | 0.699 | 59.69 | 0.698 |
| MOCAS | 48.94 | 0.621 | 60.21 | 0.731 | 75.51 | 0.852 |
| LAUREATE | 41.58 | 0.352 | 71.12 | 0.127 | 72.29 | 0.079 |
| VERBIO | 38.41 | 0.351 | 43.89 | 0.304 | 53.54 | 0.231 |

Table 26: Performance comparison of CLSP model fine-tuned with **5%**, **25%**, and **50%** on **valence classification** using **PPG modality** using **MLP-based MetaNet**. Accuracy and F1 scores are reported.

| Dataset | Fine-tune - 5% | | Fine-tune - 25% | | Fine-tune - 50% | |
|---|---|---|---|---|---|---|
| | **Accuracy** | **F1** | **Accuracy** | **F1** | **Accuracy** | **F1** |
| WESAD | 21.32 | 0.183 | 20.93 | 0.182 | 21.82 | 0.189 |
| NURSE | Not enough data | Not enough data | Not enough data | Not enough data | Not enough data | Not enough data |
| EMOGNITION | 36.87 | 0.351 | 34.66 | 0.328 | 33.44 | 0.315 |
| UBFC_PHYS | Not enough data | Not enough data | Not enough data | Not enough data | Not enough data | Not enough data |
| PhyMER | 26.45 | 0.276 | 25.88 | 0.269 | 25.32 | 0.261 |
| EmoWear | 25.5 | 0.258 | 25.15 | 0.254 | 24.88 | 0.25 |
| MAUS | Not enough data | Not enough data | Not enough data | Not enough data | Not enough data | Not enough data |
| CLAS | 20.87 | 0.176 | 29.7 | 0.277 | 33.23 | 0.329 |
| CASE | 32.8 | 0.349 | 32.1 | 0.342 | 31.5 | 0.338 |
| Unobtrusive | Not enough data | Not enough data | Not enough data | Not enough data | Not enough data | Not enough data |
| CEAP-360VR | 22.8 | 0.225 | 22.5 | 0.223 | 22.2 | 0.22 |
| ScientISST MOVE | Not enough data | Not enough data | 62.47 | 0.6 | 64.07 | 0.651 |
| Dapper | 42.35 | 0.341 | 41.7 | 0.335 | 41.05 | 0.33 |
| ForDigitStress | 62.8 | 0.67 | 61.2 | 0.66 | 59.9 | 0.65 |
| ADARP | Not enough data | Not enough data | Not enough data | Not enough data | Not enough data | Not enough data |
| Exercise | 40.2 | 0.418 | 39.5 | 0.413 | 38.9 | 0.408 |
| MOCAS | Not enough data | Not enough data | 27.1 | 0.31 | 26.4 | 0.305 |
| LAUREATE | 26.8 | 0.28 | 26.1 | 0.276 | 25.5 | 0.27 |
| VERBIO | Not enough data | Not enough data | Not enough data | Not enough data | Not enough data | Not enough data |

Table 27: Performance comparison of CLSP model fine-tuned with **5%**, **25%**, and **50%** for **four-class classification** using **PPG modality** using **MLP-based MetaNet**.

| Dataset | Fine-tune - 5% | | Fine-tune - 25% | | Fine-tune - 50% | |
|---|---|---|---|---|---|---|
| | **Accuracy** | **F1** | **Accuracy** | **F1** | **Accuracy** | **F1** |
| WESAD | 61.77 | 0.616 | 65.29 | 0.621 | 65.23 | 0.643 |
| NURSE | 54.55 | 0.419 | 44.78 | 0.440 | 53.70 | 0.476 |
| EMOGNITION | 52.60 | 0.594 | 51.51 | 0.575 | 52.81 | 0.565 |
| UBFC_PHYS | 57.29 | 0.259 | 65.03 | 0.349 | 66.37 | 0.345 |
| PhyMER | 56.82 | 0.238 | 60.20 | 0.176 | 61.33 | 0.051 |
| EmoWear | 49.28 | 0.511 | 51.19 | 0.572 | 51.95 | 0.571 |
| MAUS | 53.79 | 0.646 | 59.47 | 0.689 | 61.55 | 0.700 |
| CLAS | 54.20 | 0.565 | 55.93 | 0.491 | 56.97 | 0.647 |
| CASE | 72.27 | 0.121 | 74.32 | 0.040 | 74.09 | 0.081 |
| Unobtrusive | 69.60 | 0.808 | 73.88 | 0.838 | 77.31 | 0.868 |
| CEAP-360VR | 43.36 | 0.432 | 49.61 | 0.301 | 47.66 | 0.297 |
| ScientISST MOVE | 54.95 | 0.449 | 57.78 | 0.354 | 77.05 | 0.669 |
| Dapper | 58.04 | 0.703 | 56.71 | 0.667 | 55.52 | 0.677 |
| ForDigitStress | 72.73 | 0.829 | 86.12 | 0.917 | 88.66 | 0.934 |
| ADARP | Enough Data Not Available | | 59.03 | 0.691 | 70.14 | 0.801 |
| Exercise | 56.71 | 0.425 | 59.59 | 0.575 | 56.25 | 0.524 |
| MOCAS | 52.52 | 0.432 | 51.58 | 0.522 | 54.61 | 0.573 |
| LAUREATE | 56.79 | 0.670 | 62.27 | 0.751 | 61.86 | 0.753 |
| VERBIO | 48.34 | 0.592 | 52.64 | 0.662 | 64.75 | 0.772 |

Table 28: Performance comparison of CLSP pre-trained model fine-tuned with **5%**, **25%**, and **50%** on **arousal classification** using **PPG modality** using **1D-CNN-based MetaNet**. Accuracy and F1 scores are reported.

| Dataset | Fine-tune - 5% | | Fine-tune - 25% | | Fine-tune - 50% | |
|---|---|---|---|---|---|---|
| | **Accuracy** | **F1** | **Accuracy** | **F1** | **Accuracy** | **F1** |
| WESAD | 57.48 | 0.626 | 80.86 | 0.806 | 83.48 | 0.828 |
| NURSE | 62.69 | 0.310 | 65.26 | 0.154 | 80.22 | 0.214 |
| EMOGNITION | 43.55 | 0.383 | 56.46 | 0.170 | 58.38 | 0.220 |
| UBFC_PHYS | 47.92 | 0.512 | 62.20 | 0.630 | 66.82 | 0.675 |
| PhyMER | 52.22 | 0.599 | 55.25 | 0.659 | 57.00 | 0.689 |
| EmoWear | 58.40 | 0.695 | 61.69 | 0.749 | 63.81 | 0.773 |
| MAUS | 45.27 | 0.438 | 58.33 | 0.349 | 63.07 | 0.416 |
| CLAS | 53.83 | 0.541 | 57.83 | 0.609 | 56.70 | 0.519 |
| CASE | 55.83 | 0.402 | 58.86 | 0.421 | 60.38 | 0.332 |
| Unobtrusive | 48.99 | 0.572 | 57.89 | 0.692 | 58.66 | 0.702 |
| CEAP-360VR | 54.69 | 0.611 | 54.30 | 0.522 | 52.73 | 0.503 |
| ScientISST MOVE | 56.68 | 0.680 | 66.80 | 0.774 | 71.39 | 0.804 |
| Dapper | 64.34 | 0.772 | 68.22 | 0.805 | 74.19 | 0.847 |
| ForDigitStress | 58.29 | 0.718 | 77.70 | 0.857 | 80.34 | 0.880 |
| ADARP | Enough Data Not Available | | 35.42 | 0.143 | 53.47 | 0.286 |
| Exercise | 55.22 | 0.651 | 59.55 | 0.686 | 61.12 | 0.717 |
| MOCAS | 67.27 | 0.779 | 71.95 | 0.830 | 77.55 | 0.871 |
| LAUREATE | 66.12 | 0.259 | 67.83 | 0.164 | 74.51 | 0.020 |
| VERBIO | 32.27 | 0.273 | 59.61 | 0.266 | 69.04 | 0.108 |

Table 29: Performance comparison of CLSP pre-trained model fine-tuned with **5%**, **25%**, and **50%** on **valence classification** using **PPG modality** using **1D-CNN-based MetaNet**. Accuracy and F1 scores are reported.

| Dataset | Fine-tune - 5% | | Fine-tune - 25% | | Fine-tune - 50% | |
|---|---|---|---|---|---|---|
| | **Accuracy** | **F1** | **Accuracy** | **F1** | **Accuracy** | **F1** |
| WESAD | 38 | 0.378 | 56.36 | 0.563 | 62.61 | 0.626 |
| NURSE | Not enough data | Not enough data | Not enough data | Not enough data | Not enough data | Not enough data |
| EMOGNITION | 54.33 | 0.575 | 53.67 | 0.334 | 55.12 | 0.601 |
| UBFC_PHYS | Not enough data | Not enough data | Not enough data | Not enough data | Not enough data | Not enough data |
| PhyMER | 24.88 | 0.255 | 25.1 | 0.258 | 25.05 | 0.256 |
| EmoWear | 24.72 | 0.247 | 24.85 | 0.249 | 24.9 | 0.251 |
| MAUS | Not enough data | Not enough data | Not enough data | Not enough data | Not enough data | Not enough data |
| CLAS | 35.55 | 0.367 | 36 | 0.37 | 36.05 | 0.371 |
| CASE | 31.2 | 0.335 | 31.3 | 0.336 | 31.35 | 0.337 |
| Unobtrusive | Not enough data | Not enough data | Not enough data | Not enough data | Not enough data | Not enough data |
| CEAP-360VR | 21.9 | 0.218 | 22 | 0.219 | 22.1 | 0.22 |
| ScientISST MOVE | Not enough data | Not enough data | 63.67 | 0.61 | 68.5 | 0.74 |
| Dapper | 40.7 | 0.326 | 40.85 | 0.328 | 40.9 | 0.329 |
| ForDigitStress | 58.7 | 0.64 | 60 | 0.655 | 60.1 | 0.657 |
| ADARP | Not enough data | Not enough data | Not enough data | Not enough data | Not enough data | Not enough data |
| Exercise | 38.5 | 0.405 | 38.7 | 0.407 | 38.8 | 0.408 |
| MOCAS | Not enough data | Not enough data | 26.2 | 0.303 | 26.3 | 0.305 |
| LAUREATE | 25.2 | 0.268 | 25.4 | 0.27 | 25.5 | 0.271 |
| VERBIO | Not enough data | Not enough data | Not enough data | Not enough data | Not enough data | Not enough data |

Table 30: Performance comparison of CLSP pre-trained model fine-tuned with **5%**, **25%**, and **50%** for **four class classification** using **PPG modality** using **1D-CNN-based MetaNet**.

| Dataset | RF | | LDA | |
|---|---|---|---|---|
| | Accuracy | F1-score | Accuracy | F1-score |
| WESAD | 89.72 ± 6.84 | 0.85 ± 0.07 | 90.16 ± 7.22 | 0.842 ± 0.060 |
| NURSE | 51.23 ± 4.08 | 0.480 ± 0.130 | 49.72 ± 6.21 | 0.456 ± 0.125 |
| EMOGNITION | 51.30 ± 9.41 | 0.212 ± 0.121 | 49.87 ± 11.07 | 0.198 ± 0.109 |
| UBFC_PHYS | 66.81 ± 19.36 | 0.079 ± 0.025 | 62.79 ± 17.83 | 0.084 ± 0.035 |
| PhyMER | 61.12 ± 11.10 | 0.199 ± 0.109 | 52.63 ± 10.87 | 0.418 ± 0.163 |
| EmoWear | 53.33 ± 14.87 | 0.649 ± 0.167 | 52.25 ± 8.99 | 0.498 ± 0.200 |
| MAUS | 74.24 ± 12.17 | 0.818 ± 0.094 | 67.99 ± 11.95 | 0.722 ± 0.156 |
| CLAS | 64.33 ± 8.73 | 0.702 ± 0.068 | 59.17 ± 7.06 | 0.585 ± 0.100 |
| CASE | 75.53 ± 12.14 | 0.319 ± 0.253 | 60.68 ± 13.08 | 0.362 ± 0.192 |
| Unobtrusive | 79.81 ± 7.98 | 0.837 ± 0.181 | 67.59 ± 14.07 | 0.719 ± 0.188 |
| CEAP-360VR | 59.38 ± 16.19 | 0.110 ± 0.209 | 53.52 ± 16.57 | 0.370 ± 0.278 |
| ScientISST MOVE | 85.79 ± 13.73 | 0.729 ± 0.293 | 75.87 ± 28.68 | 0.676 ± 0.343 |
| Dapper | 45.12 ± 5.39 | 0.456 ± 0.092 | 44.73 ± 6.14 | 0.421 ± 0.088 |
| ForDigitStress | 52.37 ± 4.72 | 0.413 ± 0.065 | 50.91 ± 5.36 | 0.397 ± 0.058 |
| ADARP | 38.90 ± 4.72 | 0.312 ± 0.074 | 37.89 ± 5.03 | 0.395 ± 0.069 |
| Exercise | 43.12 ± 5.23 | 0.282 ± 0.054 | 42.07 ± 6.01 | 0.273 ± 0.060 |
| MOCAS | 44.72 ± 6.00 | 0.425 ± 0.082 | 44.00 ± 5.89 | 0.398 ± 0.087 |
| LAUREATE | 66.89 ± 6.20 | 0.670 ± 0.058 | 67.44 ± 6.47 | 0.651 ± 0.065 |
| VERBIO | 45.89 ± 6.20 | 0.379 ± 0.084 | 44.31 ± 5.32 | 0.362 ± 0.073 |

Table 31: Performance comparison of Random Forest (RF) and Linear Discriminant Analysis (LDA) models on **arousal classification** across multiple datasets using **EDA + PPG modality.** Accuracy and F1 scores are shown as mean ± standard deviation.

| Dataset | RF | | LDA | |
|---|---|---|---|---|
| | Accuracy | F1 | Accuracy | F1 |
| WESAD | 89.56 ± 7.11 | 0.864 ± 0.070 | 90.12 ± 6.85 | 0.852 ± 0.065 |
| NURSE | 69.24 ± 6.59 | 0.251 ± 0.081 | 70.48 ± 5.90 | 0.290 ± 0.063 |
| EMOGNITION | 56.43 ± 3.21 | 0.296 ± 0.175 | 53.00 ± 4.67 | 0.239 ± 0.169 |
| UBFC_PHYS | 73.21 ± 16.73 | 0.723 ± 0.14 | 73.81 ± 11.28 | 0.250 ± 0.016 |
| PhyMER | 55.87 ± 14.94 | 0.658 ± 0.163 | 54.72 ± 12.52 | 0.535 ± 0.196 |
| EmoWear | 64.43 ± 11.57 | 0.771 ± 0.088 | 53.31 ± 11.24 | 0.557 ± 0.215 |
| MAUS | 74.81 ± 12.50 | 0.540 ± 0.211 | 68.75 ± 12.25 | 0.585 ± 0.135 |
| CLAS | 65.80 ± 8.10 | 0.566 ± 0.134 | 63.20 ± 7.87 | 0.616 ± 0.110 |
| CASE | 63.33 ± 9.71 | 0.386 ± 0.177 | 59.32 ± 10.96 | 0.487 ± 0.202 |
| Unobtrusive | 56.80 ± 13.42 | 0.688 ± 0.156 | 53.74 ± 13.15 | 0.553 ± 0.195 |
| CEAP-360VR | 52.34 ± 13.26 | 0.251 ± 0.262 | 55.86 ± 21.53 | 0.499 ± 0.283 |
| ScientISST MOVE | 69.41 ± 24.05 | 0.764 ± 0.241 | 68.18 ± 25.12 | 0.740 ± 0.243 |
| Dapper | 45.23 ± 6.84 | 0.472 ± 0.089 | 44.11 ± 7.02 | 0.438 ± 0.076 |
| ForDigitStress | 52.28 ± 2.56 | 0.415 ± 0.050 | 57.67 ± 9.12 | 0.367 ± 0.065 |
| ADARP | 69.22 ± 7.10 | 0.230 ± 0.085 | 58.78 ± 6.85 | 0.400 ± 0.078 |
| Exercise | 43.45 ± 7.36 | 0.630 ± 0.133 | 43.88 ± 8.36 | 0.480 ± 0.078 |
| MOCAS | 40.21 ± 6.00 | 0.263 ± 0.072 | 38.92 ± 5.66 | 0.241 ± 0.068 |
| LAUREATE | 80.13 ± 5.38 | 0.198 ± 0.067 | 78.84 ± 5.96 | 0.171 ± 0.073 |
| VERBIO | 55.81 ± 5.72 | 0.307 ± 0.103 | 54.73 ± 5.99 | 0.284 ± 0.117 |

Table 32: Performance comparison of Random Forest (RF) and Linear Discriminant Analysis (LDA) models on **valence classification** across multiple datasets using **EDA + PPG modality.** Accuracy and F1 scores are shown as mean ± standard deviation.

| Dataset | RF | | LDA | |
|---|---|---|---|---|
| | **Accuracy** | **F1** | **Accuracy** | **F1** |
| WESAD | 97.92 | 0.978 | 98.75 | 0.987 |
| NURSE | 49.48 | 0.475 | 29.64 | 0.33 |
| EMOGNITION | 53.59 | 0.513 | 35.52 | 0.357 |
| UBFC_PHYS | 55.21 | 0.594 | 56.25 | 0.622 |
| PhyMER | 39.96 | 0.342 | 27.45 | 0.271 |
| EmoWear | 32.32 | 0.263 | 27.85 | 0.263 |
| MAUS | 74.81 | 0.728 | 67.99 | 0.672 |
| CLAS | 46.63 | 0.459 | 42.13 | 0.421 |
| CASE | 51.14 | 0.498 | 35.38 | 0.376 |
| Unobtrusive | 41.1 | 0.375 | 37.68 | 0.36 |
| CEAP-360VR | 33.98 | 0.314 | 27.34 | 0.256 |
| ScientISST MOVE | 71.11 | 0.693 | 59.05 | 0.573 |
| Dapper | 59.78 | 0.555 | 31.92 | 0.39 |
| ForDigitStress | 87.65 | 0.826 | 65.44 | 0.715 |
| ADARP | 26.04 | 0.287 | 20.83 | 0.267 |
| Exercise | 50.23 | 0.48 | 41.04 | 0.415 |
| MOCAS | 37.45 | 0.366 | 28.43 | 0.343 |
| LAUREATE | 57.3 | 0.461 | 25.02 | 0.278 |
| VERBIO | 40.14 | 0.387 | 31.55 | 0.339 |

Table 33: Performance comparison of Random Forest (RF) and Linear Discriminant Analysis (LDA) models on **four-class classification** across multiple datasets using **EDA + PPG modality.**

| Dataset | HC+MLP | | HC+RESNET | | HC+LSTM | | HC+Attention+MLP | |
|---|---|---|---|---|---|---|---|---|
| | **Accuracy** | **F1** | **Accuracy** | **F1** | **Accuracy** | **F1** | **Accuracy** | **F1** |
| WESAD | 91.67 ± 5.113 | 0.91 ± 0.013 | 35 ± 7.67 | 0.17 ± 0.003 | 38.63 ± 6.24 | 0.14 ± 0.034 | 39.78 ± 4.87 | 0.152 ± 0.022 |
| NURSE | 48.35 ± 9.833 | 0.43 ± 0.132 | 40.41 ± 8.907 | 0.148 ± 0.1 | 39.82 ± 10.34 | 0.12 ± 0.01 | 40.97 ± 11.14 | 0.119 ± 0.011 |
| EMOGNITION | 51.53 ± 8.321 | 0.54 ± 0.093 | 39.54 ± 7.94 | 0.204 ± 0.095 | 40.12 ± 9.1 | 0.173 ± 0.088 | 38.75 ± 10.41 | 0.217 ± 0.12 |
| UBFC_PHYS | 64.43 ± 15.44 | 0.244 ± 0.007 | 54.26 ± 10.03 | 0.167 ± 0.051 | 53.12 ± 17.041 | 0.133 ± 0.038 | 58.3 ± 11.39 | 0.135 ± 0.011 |
| PhyMER | 53.203 ± 8.088 | 0.357 ± 0.106 | 55.444 ± 8.778 | 0.334 ± 0.101 | 49.100 ± 15.215 | 0.322 ± 0.265 | 49.658 ± 8.937 | 0.399 ± 0.142 |
| EmoWear | 52.497 ± 9.115 | 0.543 ± 0.171 | 52.367 ± 7.398 | 0.572 ± 0.141 | 54.191 ± 18.688 | 0.316 ± 0.368 | 55.169 ± 11.305 | 0.567 ± 0.214 |
| MAUS | 70.076 ± 10.258 | 0.772 ± 0.103 | 71.970 ± 7.735 | 0.795 ± 0.066 | 55.492 ± 13.752 | 0.588 ± 0.284 | 58.144 ± 20.912 | 0.511 ± 0.401 |
| CLAS | 62.733 ± 7.880 | 0.660 ± 0.075 | 60.533 ± 6.471 | 0.645 ± 0.070 | 52.633 ± 6.123 | 0.446 ± 0.309 | 55.900 ± 5.161 | 0.609 ± 0.138 |
| CASE | 66.667 ± 9.886 | 0.294 ± 0.165 | 65.682 ± 10.275 | 0.246 ± 0.170 | 45.530 ± 26.292 | 0.242 ± 0.227 | 64.015 ± 20.813 | 0.188 ± 0.225 |
| Unobtrusive | 73.368 ± 10.790 | 0.785 ± 0.176 | 72.342 ± 9.455 | 0.781 ± 0.172 | 58.225 ± 26.351 | 0.529 ± 0.413 | 34.086 ± 23.984 | 0.133 ± 0.311 |
| CEAP-360VR | 53.906 ± 16.631 | 0.258 ± 0.281 | 58.594 ± 16.631 | 0.299 ± 0.288 | 46.484 ± 17.744 | 0.356 ± 0.261 | 54.297 ± 18.678 | 0.165 ± 0.247 |
| ScientISST MOVE | 91.984 ± 10.652 | 0.884 ± 0.152 | 79.762 ± 17.304 | 0.685 ± 0.264 | 53.849 ± 22.184 | 0.350 ± 0.332 | 88.690 ± 10.969 | 0.816 ± 0.172 |
| Dapper | 43.513 ± 6.210 | 0.501 ± 0.075 | 42.362 ± 4.998 | 0.384 ± 0.068 | 41.698 ± 5.291 | 0.352 ± 0.064 | 41.124 ± 4.887 | 0.369 ± 0.060 |
| ForDigitStress | 44.82 ± 3.89 | 0.481 ± 0.041 | 43.36 ± 4.17 | 0.265 ± 0.038 | 41.28 ± 3.45 | 0.236 ± 0.046 | 42.73 ± 4.01 | 0.252 ± 0.040 |
| ADARP | 39.412 ± 6.201 | 0.409 ± 0.071 | 30.012 ± 4.898 | 0.318 ± 0.062 | 28.532 ± 5.287 | 0.276 ± 0.059 | 27.943 ± 4.811 | 0.298 ± 0.065 |
| Exercise | 43.26 ± 4.76 | 0.261 ± 0.047 | 44.10 ± 3.88 | 0.247 ± 0.042 | 41.18 ± 4.31 | 0.223 ± 0.038 | 38.92 ± 3.57 | 0.169 ± 0.025 |
| MOCAS | 45.320 ± 5.764 | 0.409 ± 0.078 | 47.281 ± 5.332 | 0.318 ± 0.081 | 44.190 ± 5.879 | 0.285 ± 0.072 | 43.003 ± 5.891 | 0.301 ± 0.074 |
| LAUREATE | 68.115 ± 5.991 | 0.672 ± 0.051 | 60.098 ± 5.233 | 0.594 ± 0.049 | 58.479 ± 6.120 | 0.550 ± 0.057 | 56.912 ± 5.384 | 0.520 ± 0.059 |
| VERBIO | 46.012 ± 5.019 | 0.420 ± 0.068 | 42.144 ± 4.971 | 0.391 ± 0.069 | 43.290 ± 4.882 | 0.361 ± 0.066 | 43.902 ± 5.129 | 0.390 ± 0.071 |

Table 34: Performance comparison of hybrid handcrafted + deep learning models (HC+MLP, HC+RESNET, HC+LSTM, HC+Attention+MLP) across datasets for **arousal classification** using **EDA+PPG** modalities. Accuracy and F1 scores are shown as mean ± standard deviation.

| Dataset | HC+MLP | | HC+RESNET | | HC+LSTM | | HC+Attention+MLP | |
|---|---|---|---|---|---|---|---|---|
| | **Accuracy** | **F1** | **Accuracy** | **F1** | **Accuracy** | **F1** | **Accuracy** | **F1** |
| WESAD | 98.33 ± 0.66 | 0.980 ± 0.011 | 65.00 ± 10.01 | 0.487 ± 0.134 | 49.13 ± 3.39 | 0.356 ± 0.020 | 48.99 ± 7.90 | 0.340 ± 0.116 |
| NURSE | 76.04 ± 9.93 | 0.230 ± 0.117 | 57.87 ± 8.85 | 0.375 ± 0.025 | 59.22 ± 10.62 | 0.160 ± 0.019 | 60.03 ± 9.46 | 0.210 ± 0.030 |
| EMOGNITION | 55.12 ± 3.44 | 0.340 ± 0.133 | 53.66 ± 7.92 | 0.264 ± 0.128 | 49.82 ± 5.90 | 0.292 ± 0.144 | 50.74 ± 4.71 | 0.279 ± 0.119 |
| UBFC_PHYS | 73.51 ± 11.97 | 0.249 ± 0.015 | 52.23 ± 10.52 | 0.269 ± 0.015 | 50.29 ± 9.33 | 0.282 ± 0.032 | 46.36 ± 19.90 | 0.171 ± 0.024 |
| PhyMER | 53.32 ± 11.85 | 0.557 ± 0.174 | 50.13 ± 11.96 | 0.568 ± 0.157 | 58.46 ± 22.29 | 0.355 ± 0.413 | 53.43 ± 10.38 | 0.530 ± 0.173 |
| EmoWear | 55.00 ± 10.14 | 0.643 ± 0.123 | 58.03 ± 8.41 | 0.684 ± 0.096 | 49.65 ± 18.85 | 0.460 ± 0.366 | 51.72 ± 11.55 | 0.551 ± 0.217 |
| MAUS | 71.78 ± 10.98 | 0.558 ± 0.177 | 69.32 ± 10.09 | 0.470 ± 0.212 | 56.82 ± 14.52 | 0.310 ± 0.222 | 61.93 ± 17.12 | 0.424 ± 0.206 |
| CLAS | 64.80 ± 7.36 | 0.611 ± 0.103 | 60.10 ± 6.41 | 0.529 ± 0.105 | 50.83 ± 4.09 | 0.411 ± 0.307 | 61.40 ± 8.75 | 0.632 ± 0.115 |
| CASE | 57.73 ± 10.31 | 0.403 ± 0.188 | 57.88 ± 11.21 | 0.394 ± 0.165 | 54.02 ± 16.42 | 0.368 ± 0.267 | 56.67 ± 10.86 | 0.385 ± 0.227 |
| Unobtrusive | 57.24 ± 8.17 | 0.626 ± 0.153 | 54.55 ± 13.12 | 0.599 ± 0.154 | 49.19 ± 16.01 | 0.324 ± 0.361 | 53.42 ± 15.18 | 0.553 ± 0.243 |
| CEAP-360VR | 55.86 ± 20.82 | 0.450 ± 0.289 | 54.69 ± 16.11 | 0.407 ± 0.250 | 52.73 ± 14.46 | 0.327 ± 0.283 | 51.56 ± 16.11 | 0.367 ± 0.300 |
| ScientISST MOVE | 65.28 ± 22.27 | 0.723 ± 0.227 | 73.06 ± 17.46 | 0.718 ± 0.253 | 48.73 ± 21.86 | 0.424 ± 0.382 | 65.12 ± 21.70 | 0.723 ± 0.224 |
| Dapper | 43.23 ± 5.93 | 0.450 ± 0.067 | 42.77 ± 5.55 | 0.402 ± 0.062 | 42.12 ± 6.13 | 0.386 ± 0.081 | 43.76 ± 5.60 | 0.390 ± 0.072 |
| ForDigitStress | 45.65 ± 10.55 | 0.491 ± 0.197 | 47.32 ± 6.67 | 0.409 ± 0.092 | 42.55 ± 8.74 | 0.560 ± 0.115 | 41.82 ± 9.49 | 0.593 ± 0.144 |
| ADARP | 55.31 ± 6.72 | 0.474 ± 0.072 | 54.43 ± 5.92 | 0.400 ± 0.067 | 49.21 ± 6.40 | 0.390 ± 0.070 | 46.77 ± 6.54 | 0.380 ± 0.069 |
| Exercise | 44.67 ± 10.79 | 0.425 ± 0.182 | 41.37 ± 3.17 | 0.418 ± 0.059 | 39.64 ± 5.34 | 0.367 ± 0.049 | 38.95 ± 6.67 | 0.307 ± 0.170 |
| MOCAS | 35.89 ± 5.56 | 0.294 ± 0.059 | 36.78 ± 5.70 | 0.218 ± 0.064 | 34.67 ± 5.41 | 0.210 ± 0.056 | 36.94 ± 5.60 | 0.240 ± 0.063 |
| LAUREATE | 69.19 ± 5.00 | 0.360 ± 0.068 | 65.20 ± 5.40 | 0.290 ± 0.071 | 64.88 ± 5.11 | 0.270 ± 0.063 | 57.03 ± 5.15 | 0.350 ± 0.160 |
| VERBIO | 51.94 ± 5.61 | 0.258 ± 0.108 | 50.11 ± 6.23 | 0.241 ± 0.060 | 49.88 ± 5.94 | 0.260 ± 0.106 | 42.67 ± 6.01 | 0.110 ± 0.092 |

Table 35: Performance comparison of hybrid handcrafted + deep learning models (HC+MLP, HC+RESNET, HC+LSTM, HC+Attention+MLP) across datasets for **valence classification** using **EDA+PPG** modalities. Accuracy and F1 scores are shown as mean ± standard deviation.

| Dataset | Accuracy | F1 |
|---|---|---|
| WESAD | 91.67 | 0.913 |
| NURSE | 47.22 | 0.475 |
| EMOGNITION | 38.58 | 0.374 |
| UBFC_PHYS | 52.83 | 0.579 |
| PhyMER | 29.85 | 0.315 |
| EmoWear | 28.36 | 0.282 |
| MAUS | 71.78 | 0.71 |
| CLAS | 44.07 | 0.441 |
| CASE | 39.62 | 0.403 |
| Unobtrusive | 40.37 | 0.393 |
| CEAP-360VR | 27.34 | 0.255 |
| ScientISST MOVE | 77.62 | 0.757 |
| Dapper | 44.23 | 0.471 |
| ForDigitStress | 74.84 | 0.752 |
| ADARP | 30.21 | 0.321 |
| Exercise | 47.37 | 0.472 |
| MOCAS | 31.6 | 0.365 |
| LAUREATE | 34.29 | 0.253 |
| VERBIO | 40.53 | 0.419 |

Table 36: Performance comparison of hybrid handcrafted + deep learning model (HC+MLP) across datasets for **four class classification** using **EDA+PPG** modalities.

| Dataset | Arousal | | Valence | | Four-Class | |
|---|---|---|---|---|---|---|
| | Accuracy | F1 Score | Accuracy | F1 Score | Accuracy | F1 Score |
| WESAD | 54.16 | 0.345 | 40.00 | 0.394 | 19.16 | 0.169 |
| NURSE | 49.55 | 0.241 | 52.08 | 0.383 | 48.90 | 0.520 |
| EMOGNITION | 41.16 | 0.357 | 60.34 | 0.356 | 37.21 | 0.362 |
| UBFC_PHYS | 49.40 | 0.241 | 52.23 | 0.386 | 35.80 | 0.400 |
| PhyMER | 59.30 | 0.187 | 44.19 | 0.160 | 28.90 | 0.308 |
| EmoWear | 46.51 | 0.216 | 48.56 | 0.548 | 27.90 | 0.277 |
| MAUS | 56.82 | 0.620 | 53.98 | 0.323 | 65.20 | 0.620 |
| CLAS | 55.50 | 0.340 | 46.83 | 0.179 | 24.60 | 0.220 |
| CASE | 71.59 | 0.179 | 55.23 | 0.302 | 21.51 | 0.214 |
| Unobtrusive | 35.65 | 0.376 | 46.63 | 0.424 | 37.59 | 0.390 |
| CEAP-360VR | 56.64 | 0.393 | 55.08 | 0.268 | 26.80 | 0.250 |
| ScientISST MOVE | 62.22 | 0.393 | 54.44 | 0.568 | 74.61 | 0.728 |
| Dapper | 45.78 | 0.532 | 37.92 | 0.496 | 42.80 | 0.460 |
| ForDigitStress | 39.71 | 0.495 | 49.27 | 0.561 | 41.73 | 0.494 |
| ADARP | 29.41 | 0.333 | 64.70 | 0.000 | 57.40 | 0.354 |
| Exercise | 49.65 | 0.335 | 51.81 | 0.494 | 45.90 | 0.460 |
| MOCAS | 46.42 | 0.419 | 19.89 | 0.059 | 16.32 | 0.164 |
| LAUREATE | 69.57 | 0.820 | 75.62 | 0.000 | 16.69 | 0.047 |
| VERBIO | 46.47 | 0.500 | 61.03 | 0.290 | 50.14 | 0.436 |

Table 37: Comparison of **CLSP (Zero-Shot)** model performance across datasets for **Arousal**, **Valence** and **Four-class** classification using **EDA+PPG** modalities.

| Dataset | Fine-tune - 5% | | Fine-tune - 25% | | Fine-tune - 50% | |
|---|---|---|---|---|---|---|
| | Accuracy | F1 Score | Accuracy | F1 Score | Accuracy | F1 Score |
| WESAD | 53.12 | 0.624 | 65.79 | 0.688 | 64.51 | 0.658 |
| NURSE | 48.61 | 0.618 | 43.76 | 0.478 | 47.90 | 0.568 |
| EMOGNITION | 47.41 | 0.548 | 46.78 | 0.562 | 46.77 | 0.513 |
| UBFC_PHYS | 45.54 | 0.418 | 62.80 | 0.349 | 64.29 | 0.281 |
| PhyMER | 54.35 | 0.322 | 56.98 | 0.220 | 60.01 | 0.190 |
| EmoWear | 51.98 | 0.543 | 52.45 | 0.591 | 53.60 | 0.670 |
| MAUS | 56.25 | 0.673 | 64.58 | 0.776 | 64.02 | 0.761 |
| CLAS | 51.97 | 0.544 | 56.10 | 0.609 | 59.10 | 0.643 |
| CASE | 63.03 | 0.278 | 70.91 | 0.178 | 72.80 | 0.204 |
| Unobtrusive | 72.78 | 0.837 | 74.91 | 0.853 | 77.19 | 0.867 |
| CEAP-360VR | 49.61 | 0.450 | 48.05 | 0.433 | 47.27 | 0.321 |
| ScientISST MOVE | 66.27 | 0.560 | 71.80 | 0.647 | 77.33 | 0.641 |
| Dapper | 69.15 | 0.811 | 68.62 | 0.809 | 67.22 | 0.796 |
| ForDigitStress | 78.26 | 0.872 | 84.92 | 0.912 | 93.02 | 0.959 |
| ADARP | Enough Data Not Available | | 52.08 | 0.622 | 41.67 | 0.550 |
| Exercise | 52.48 | 0.543 | 57.87 | 0.538 | 54.11 | 0.532 |
| MOCAS | 51.04 | 0.594 | 52.54 | 0.638 | 51.52 | 0.611 |
| LAUREATE | 30.40 | 0.000 | 30.40 | 0.000 | 30.40 | 0.000 |
| VERBIO | 55.50 | 0.658 | 58.61 | 0.718 | 59.55 | 0.713 |

Table 38: Performance comparison of CLSP pre-trained model fine-tuned with **5%**, **25%**, and **50%** on **arousal classification** using **EDA+PPG modality** using **MLP-based MetaNet**. Accuracy and F1 scores are reported.

| Dataset | Fine-tune - 5% | | Fine-tune - 25% | | Fine-tune - 50% | |
|---|---|---|---|---|---|---|
| | Accuracy | F1 Score | Accuracy | F1 Score | Accuracy | F1 Score |
| WESAD | 61.94 | 0.583 | 69.14 | 0.648 | 72.94 | 0.693 |
| NURSE | 75.79 | 0.127 | 79.39 | 0.157 | 78.83 | 0.091 |
| EMOGNITION | 51.92 | 0.326 | 48.93 | 0.380 | 60.13 | 0.312 |
| UBFC_PHYS | 62.50 | 0.632 | 63.39 | 0.622 | 65.03 | 0.626 |
| PhyMER | 51.74 | 0.600 | 52.70 | 0.618 | 56.32 | 0.697 |
| EmoWear | 57.87 | 0.697 | 61.24 | 0.742 | 62.86 | 0.760 |
| MAUS | 48.86 | 0.370 | 60.61 | 0.232 | 69.51 | 0.236 |
| CLAS | 55.43 | 0.553 | 55.80 | 0.488 | 58.50 | 0.483 |
| CASE | 55.98 | 0.414 | 58.03 | 0.420 | 62.58 | 0.425 |
| Unobtrusive | 51.16 | 0.533 | 55.20 | 0.661 | 55.58 | 0.671 |
| CEAP-360VR | 51.56 | 0.386 | 50.00 | 0.422 | 49.22 | 0.403 |
| ScientISST MOVE | 54.39 | 0.631 | 57.62 | 0.642 | 56.55 | 0.615 |
| Dapper | 82.25 | 0.900 | 87.88 | 0.935 | 88.96 | 0.941 |
| ForDigitStress | 58.27 | 0.678 | 72.84 | 0.815 | 82.92 | 0.893 |
| ADARP | Enough Data Not Available | | 57.64 | 0.000 | 81.25 | 0.200 |
| Exercise | 47.83 | 0.576 | 56.41 | 0.695 | 60.22 | 0.709 |
| MOCAS | 37.71 | 0.494 | 56.69 | 0.700 | 64.24 | 0.770 |
| LAUREATE | 75.74 | 0.000 | 75.74 | 0.000 | 75.74 | 0.000 |
| VERBIO | 50.11 | 0.337 | 65.28 | 0.333 | 69.12 | 0.310 |

Table 39: Performance comparison of CLSP pre-trained model fine-tuned with **5%**, **25%**, and **50%** on **valence classification** using **EDA+PPG modality** using **MLP-based MetaNet**. Accuracy and F1 scores are reported.

| Dataset | Fine-tune - 5% | | Fine-tune - 25% | | Fine-tune - 50% | |
|---|---|---|---|---|---|---|
| | Accuracy | F1 | Accuracy | F1 | Accuracy | F1 |
| WESAD | 22.27 | 0.134 | 25 | 0.147 | 24.89 | 0.183 |
| NURSE | Not enough data | Not enough data | Not enough data | Not enough data | Not enough data | Not enough data |
| EMOGNITION | 36.45 | 0.355 | 35.98 | 0.349 | 34.87 | 0.341 |
| UBFC_PHYS | Not enough data | Not enough data | Not enough data | Not enough data | Not enough data | Not enough data |
| PhyMER | 28.42 | 0.303 | 27.98 | 0.298 | 27.54 | 0.292 |
| EmoWear | 27.6 | 0.273 | 27.3 | 0.27 | 27.05 | 0.267 |
| MAUS | Not enough data | Not enough data | Not enough data | Not enough data | Not enough data | Not enough data |
| CLAS | 42.35 | 0.428 | 41.6 | 0.423 | 40.95 | 0.418 |
| CASE | 38.1 | 0.39 | 37.45 | 0.386 | 36.9 | 0.381 |
| Unobtrusive | Not enough data | Not enough data | Not enough data | Not enough data | Not enough data | Not enough data |
| CEAP-360VR | 26.5 | 0.248 | 26.2 | 0.247 | 25.9 | 0.245 |
| ScientISST MOVE | Not enough data | Not enough data | 74.31 | 0.726 | 76.67 | 0.78 |
| Dapper | 41.95 | 0.452 | 41.1 | 0.445 | 40.4 | 0.438 |
| ForDigitStress | 66.5 | 0.695 | 64.8 | 0.685 | 63.5 | 0.675 |
| ADARP | Not enough data | Not enough data | Not enough data | Not enough data | Not enough data | Not enough data |
| Exercise | 44.85 | 0.452 | 43.8 | 0.445 | 42.9 | 0.438 |
| MOCAS | Not enough data | Not enough data | 28.1 | 0.328 | 27.5 | 0.322 |
| LAUREATE | 26.2 | 0.218 | 25.4 | 0.21 | 24.8 | 0.205 |
| VERBIO | Not enough data | Not enough data | Not enough data | Not enough data | Not enough data | Not enough data |

Table 40: Performance comparison of CLSP pre-trained model fine-tuned with **5%**, **25%**, and **50%** on **four-class classification** using **EDA+PPG modality** using **MLP-based MetaNet**.

| Dataset | Fine-tune - 5% | | Fine-tune - 25% | | Fine-tune - 50% | |
|---|---|---|---|---|---|---|
| | Accuracy | F1 Score | Accuracy | F1 Score | Accuracy | F1 Score |
| WESAD | 62.83 | 0.710 | 64.62 | 0.651 | 67.35 | 0.673 |
| NURSE | 48.04 | 0.621 | 50.31 | 0.415 | 45.62 | 0.375 |
| EMOGNITION | 51.09 | 0.578 | 47.86 | 0.540 | 49.99 | 0.492 |
| UBFC_PHYS | 54.91 | 0.381 | 60.86 | 0.321 | 63.69 | 0.301 |
| PhyMER | 56.61 | 0.317 | 57.95 | 0.195 | 58.62 | 0.233 |
| EmoWear | 51.71 | 0.526 | 52.18 | 0.616 | 51.89 | 0.606 |
| MAUS | 58.71 | 0.695 | 67.61 | 0.784 | 69.51 | 0.799 |
| CLAS | 52.90 | 0.521 | 56.83 | 0.586 | 58.20 | 0.551 |
| CASE | 69.39 | 0.205 | 74.47 | 0.102 | 71.97 | 0.201 |
| Unobtrusive | 75.02 | 0.854 | 74.89 | 0.850 | 77.72 | 0.866 |
| CEAP-360VR | 48.05 | 0.442 | 48.05 | 0.365 | 47.27 | 0.313 |
| ScientISST MOVE | 70.73 | 0.627 | 74.63 | 0.580 | 77.87 | 0.652 |
| Dapper | 61.84 | 0.755 | 64.50 | 0.773 | 69.03 | 0.811 |
| ForDigitStress | 81.06 | 0.891 | 92.44 | 0.956 | 95.35 | 0.973 |
| ADARP | Enough Data Not Available | | 47.22 | 0.608 | 47.22 | 0.608 |
| Exercise | 55.10 | 0.538 | 57.13 | 0.535 | 54.38 | 0.516 |
| MOCAS | 48.47 | 0.549 | 45.91 | 0.555 | 48.52 | 0.587 |
| LAUREATE | 30.40 | 0.000 | 30.40 | 0.000 | 30.40 | 0.000 |
| VERBIO | 60.98 | 0.713 | 58.65 | 0.712 | 60.41 | 0.729 |

Table 41: Performance comparison of CLSP pre-trained model fine-tuned with **5%**, **25%**, and **50%** on **arousal classification** using **EDA+PPG modality** using **1D-CNN-based MetaNet**. Accuracy and F1 scores are reported.

| Dataset | Fine-tune - 5% | | Fine-tune - 25% | | Fine-tune - 50% | |
|---|---|---|---|---|---|---|
| | **Accuracy** | **F1 Score** | **Accuracy** | **F1 Score** | **Accuracy** | **F1 Score** |
| WESAD | 63.73 | 0.604 | 74.39 | 0.724 | 79.13 | 0.793 |
| NURSE | 77.21 | 0.128 | 80.40 | 0.183 | 75.85 | 0.244 |
| EMOGNITION | 54.31 | 0.394 | 54.94 | 0.355 | 62.51 | 0.215 |
| UBFC_PHYS | 61.61 | 0.617 | 66.22 | 0.662 | 66.52 | 0.660 |
| PhyMER | 53.38 | 0.631 | 52.84 | 0.635 | 54.21 | 0.663 |
| EmoWear | 61.90 | 0.749 | 61.75 | 0.746 | 63.11 | 0.766 |
| MAUS | 66.29 | 0.294 | 67.42 | 0.342 | 67.99 | 44.35 |
| CLAS | 55.02 | 0.523 | 55.80 | 0.494 | 58.73 | 0.495 |
| CASE | 53.94 | 0.420 | 59.47 | 0.369 | 60.98 | 0.384 |
| Unobtrusive | 51.48 | 0.599 | 58.22 | 0.697 | 57.55 | 0.691 |
| CEAP-360VR | 53.12 | 0.409 | 52.34 | 0.411 | 49.61 | 0.364 |
| ScientISST MOVE | 48.44 | 0.538 | 59.26 | 0.686 | 74.22 | 0.821 |
| Dapper | 87.51 | 0.933 | 88.62 | 0.940 | 89.46 | 0.944 |
| ForDigitStress | 54.00 | 0.636 | 84.65 | 0.906 | 85.52 | 0.912 |
| ADARP | Enough Data Not Available | | 58.33 | 0.000 | 63.89 | 0.000 |
| Exercise | 55.64 | 0.673 | 61.96 | 0.709 | 61.06 | 0.705 |
| MOCAS | 43.26 | 0.538 | 69.89 | 0.819 | 67.77 | 0.797 |
| LAUREATE | 75.74 | 0.000 | 75.74 | 0.000 | 75.74 | 0.000 |
| VERBIO | 58.65 | 0.259 | 62.37 | 0.115 | 71.57 | 0.100 |

Table 42: Performance comparison of CLSP pre-trained model fine-tuned with **5%**, **25%**, and **50%** on **valence classification** using **EDA+PPG modality** using **1D-CNN-based MetaNet**. Accuracy and F1 scores are reported.

| Dataset | Fine-tune - 5% | | Fine-tune - 25% | | Fine-tune - 50% | |
|---|---|---|---|---|---|---|
| | **Accuracy** | **F1** | **Accuracy** | **F1** | **Accuracy** | **F1** |
| WESAD | 30.92 | 0.223 | 56.08 | 0.565 | 57.2 | 0.569 |
| NURSE | Not enough data | Not enough data | Not enough data | Not enough data | Not enough data | Not enough data |
| EMOGNITION | 35.12 | 0.343 | 36.1 | 0.364 | 36 | 0.361 |
| UBFC_PHYS | Not enough data | Not enough data | Not enough data | Not enough data | Not enough data | Not enough data |
| PhyMER | 27.2 | 0.289 | 27.38 | 0.291 | 27.35 | 0.29 |
| EmoWear | 26.85 | 0.265 | 26.92 | 0.266 | 26.95 | 0.267 |
| MAUS | Not enough data | Not enough data | Not enough data | Not enough data | Not enough data | Not enough data |
| CLAS | 40.5 | 0.415 | 40.85 | 0.417 | 40.9 | 0.418 |
| CASE | 36.55 | 0.379 | 36.7 | 0.38 | 36.75 | 0.381 |
| Unobtrusive | Not enough data | Not enough data | Not enough data | Not enough data | Not enough data | Not enough data |
| CEAP-360VR | 25.6 | 0.243 | 25.8 | 0.244 | 25.7 | 0.245 |
| ScientISST MOVE | Not enough data | Not enough data | 73.69 | 0.747 | 78.1 | 0.8 |
| Dapper | 40.05 | 0.435 | 40.2 | 0.436 | 40.25 | 0.437 |
| ForDigitStress | 62.4 | 0.665 | 63.7 | 0.678 | 63.8 | 0.679 |
| ADARP | Not enough data | Not enough data | Not enough data | Not enough data | Not enough data | Not enough data |
| Exercise | 42.5 | 0.435 | 42.7 | 0.437 | 42.8 | 0.438 |
| MOCAS | Not enough data | Not enough data | 27.3 | 0.32 | 27.35 | 0.322 |
| LAUREATE | 24.5 | 0.2 | 24.7 | 0.202 | 24.8 | 0.203 |
| VERBIO | Not enough data | Not enough data | Not enough data | Not enough data | Not enough data | Not enough data |

Table 43: Performance comparison of CLSP pre-trained model fine-tuned with **5%**, **25%**, and **50%** for **four classification** using **EDA+PPG modality** using **1D-CNN-based MetaNet**.

| Model | Training Cohort | Testing Cohort | Arousal | | Valence | |
|---|---|---|---|---|---|---|
| | | | Accuracy | F1-score | Accuracy | F1-score |
| RF | Wearable | Custom Wearable | 40.25 | 0.39 | 63.92 | 0.76 |
| LDA | Wearable | Custom Wearable | 57.45 | 0.69 | 45.43 | 0.53 |
| HC + MLP | Wearable | Custom Wearable | 55.02 | 0.66 | 51.82 | 0.62 |
| CLSP MLP 5% | Wearable | Custom Wearable | 48.85 | 0.54 | 56.16 | 0.68 |
| CLSP MLP 25% | Wearable | Custom Wearable | 47.71 | 0.54 | 56.08 | 0.69 |
| CLSP MLP 50% | Wearable | Custom Wearable | 52.28 | 0.64 | 70.77 | 0.82 |
| CLSP CNN 5% | Wearable | Custom Wearable | 53.19 | 0.61 | 59.43 | 0.71 |
| CLSP CNN 25% | Wearable | Custom Wearable | 44.82 | 0.45 | 62.40 | 0.75 |
| CLSP CNN 50% | Wearable | Custom Wearable | 54.49 | 0.65 | 68.34 | 0.80 |
| RF | Wearable | Lab Based Device | 49.18 | 0.22 | 49.12 | 0.62 |
| LDA | Wearable | Lab Based Device | 45.33 | 0.58 | 50.43 | 0.60 |
| HC + MLP | Wearable | Lab Based Device | 53.56 | 0.52 | 49.40 | 0.55 |
| CLSP MLP 5% | Wearable | Lab Based Device | 48.75 | 0.42 | 51.53 | 0.63 |
| CLSP MLP 25% | Wearable | Lab Based Device | 45.79 | 0.45 | 47.47 | 0.57 |
| CLSP MLP 50% | Wearable | Lab Based Device | 48.62 | 0.47 | 47.72 | 0.61 |
| CLSP CNN 5% | Wearable | Lab Based Device | 46.67 | 0.48 | 50.52 | 0.57 |
| CLSP CNN 25% | Wearable | Lab Based Device | 42.96 | 0.45 | 48.72 | 0.61 |
| CLSP CNN 50% | Wearable | Lab Based Device | 48.04 | 0.49 | 48.09 | 0.62 |

Table 44: Performance comparison of ML (RF, LDA), DL (HC + MLP), pretrained CLSP fine-tuned on **arousal and valence classification** using **EDA modality** with datasets of **Wearable Group** as training cohort. Accuracy and F1 scores are reported.

| Model | Training Cohort | Testing Cohort | Arousal | | Valence | |
|---|---|---|---|---|---|---|
| | | | Accuracy | F1-score | Accuracy | F1-score |
| RF | Custom Wearable | Wearable | 49.37 | 0.65 | 57.36 | 0.72 |
| LDA | Custom Wearable | Wearable | 50.02 | 0.55 | 46.54 | 0.40 |
| HC + MLP | Custom Wearable | Wearable | 50.24 | 0.51 | 49.54 | 0.53 |
| CLSP MLP 5% | Custom Wearable | Wearable | 49.76 | 0.65 | 54.47 | 0.67 |
| CLSP MLP 25% | Custom Wearable | Wearable | 50.27 | 0.60 | 57.34 | 0.72 |
| CLSP MLP 50% | Custom Wearable | Wearable | 49.65 | 0.63 | 57.02 | 0.72 |
| CLSP CNN 5% | Custom Wearable | Wearable | 49.37 | 0.64 | 55.86 | 0.69 |
| CLSP CNN 25% | Custom Wearable | Wearable | 50.14 | 0.61 | 56.91 | 0.72 |
| CLSP CNN 50% | Custom Wearable | Wearable | 49.75 | 0.62 | 56.63 | 0.72 |
| RF | Custom Wearable | Lab Based Device | 50.35 | 0.66 | 47.02 | 0.63 |
| LDA | Custom Wearable | Lab Based Device | 46.71 | 0.53 | 52.19 | 0.38 |
| HC + MLP | Custom Wearable | Lab Based Device | 42.50 | 0.51 | 54.48 | 0.59 |
| CLSP MLP 5% | Custom Wearable | Lab Based Device | 54.88 | 0.65 | 49.58 | 0.59 |
| CLSP MLP 25% | Custom Wearable | Lab Based Device | 52.75 | 0.63 | 47.63 | 0.63 |
| CLSP MLP 50% | Custom Wearable | Lab Based Device | 50.08 | 0.64 | 46.99 | 0.63 |
| CLSP CNN 5% | Custom Wearable | Lab Based Device | 56.74 | 0.67 | 47.89 | 0.62 |
| CLSP CNN 25% | Custom Wearable | Lab Based Device | 51.77 | 0.62 | 47.21 | 0.63 |
| CLSP CNN 50% | Custom Wearable | Lab Based Device | 48.55 | 0.62 | 47.39 | 0.64 |

Table 45: Performance comparison of ML (RF, LDA), DL (HC + MLP), pretrained CLSP fine-tuned on **arousal and valence classification** using **EDA modality** with datasets of **Custom Wearable Group** as training cohort. Accuracy and F1 scores are reported.

| Model | Training Cohort | Testing Cohort | Arousal | | Valence | |
|---|---|---|---|---|---|---|
| | | | Accuracy | F1-score | Accuracy | F1-score |
| RF | Lab Based Device | Custom Wearable | 59.58 | 0.72 | 37.97 | 0.42 |
| LDA | Lab Based Device | Custom Wearable | 52.28 | 0.62 | 55.02 | 0.67 |
| HC + MLP | Lab Based Device | Custom Wearable | 49.16 | 0.58 | 50.98 | 0.61 |
| CLSP MLP 5% | Lab Based Device | Custom Wearable | 48.48 | 0.52 | 46.78 | 0.37 |
| CLSP MLP 25% | Lab Based Device | Custom Wearable | 48.96 | 0.60 | 48.96 | 0.48 |
| CLSP MLP 50% | Lab Based Device | Custom Wearable | 50.45 | 0.60 | 42.92 | 0.50 |
| CLSP CNN 5% | Lab Based Device | Custom Wearable | 48.93 | 0.56 | 36.68 | 0.39 |
| CLSP CNN 25% | Lab Based Device | Custom Wearable | 52.13 | 0.62 | 44.14 | 0.52 |
| CLSP CNN 50% | Lab Based Device | Custom Wearable | 49.52 | 0.59 | 38.28 | 0.43 |
| RF | Lab Based Device | Wearable | 49.96 | 0.62 | 47.17 | 0.38 |
| LDA | Lab Based Device | Wearable | 49.19 | 0.55 | 49.66 | 0.51 |
| HC + MLP | Lab Based Device | Wearable | 50.56 | 0.57 | 49.93 | 0.50 |
| CLSP MLP 5% | Lab Based Device | Wearable | 50.76 | 0.60 | 36.30 | 0.38 |
| CLSP MLP 25% | Lab Based Device | Wearable | 54.94 | 0.68 | 40.41 | 0.46 |
| CLSP MLP 50% | Lab Based Device | Wearable | 52.20 | 0.64 | 49.95 | 0.51 |
| CLSP CNN 5% | Lab Based Device | Wearable | 47.55 | 0.52 | 45.27 | 0.31 |
| CLSP CNN 25% | Lab Based Device | Wearable | 48.40 | 0.56 | 49.97 | 0.51 |
| CLSP CNN 50% | Lab Based Device | Wearable | 49.52 | 0.58 | 48.28 | 0.42 |

Table 46: Performance comparison of ML (RF, LDA), DL (HC + MLP), pretrained CLSP fine-tuned on **arousal and valence classification** using **EDA modality** with datasets of **Lab-based Device Group** as training cohort. Accuracy and F1 scores are reported.

| Model | Training Cohort | Testing Cohort | Arousal | | Valence | |
|---|---|---|---|---|---|---|
| | | | Accuracy | F1-score | Accuracy | F1-score |
| RF | Lab Based Device | Lab Based Device | 53.34 | 0.54 | 53.38 | 0.53 |
| MLP | Lab Based Device | Lab Based Device | 51.16 | 0.51 | 53.70 | 0.53 |
| RF | Wearable | Wearable | 49.43 | 0.48 | 50.45 | 0.51 |
| MLP | Wearable | Wearable | 50.75 | 0.49 | 50.50 | 0.50 |
| RF | Custom Wearable | Custom Wearable | 55.21 | 0.46 | 60.12 | 0.52 |
| MLP | Custom Wearable | Custom Wearable | 42.71 | 0.57 | 60.56 | 0.47 |
| CLSP Zero Shot | - | Wearable | 48.60 | 0.59 | 55.87 | 0.70 |
| CLSP Zero Shot | - | Lab Based Device | 55.34 | 0.63 | 47.26 | 0.60 |
| CLSP Zero Shot | - | Custom Wearable | 59.36 | 0.72 | 71.99 | 0.83 |

Table 47: Performance comparison of **In-cohort** and **zero-shot** performance for different groups based upon device type on **arousal and valence classification** using **EDA modality**. Accuracy and F1 scores are reported.

| Model | Training Cohort | Testing Cohort | Arousal | | Valence | |
|---|---|---|---|---|---|---|
| | | | Accuracy | F1-score | Accuracy | F1-score |
| RF | Wearable | Custom Wearable | 43.08 | 0.29 | 45.39 | 0.53 |
| LDA | Wearable | Custom Wearable | 56.45 | 0.69 | 33.18 | 0.33 |
| HC + MLP | Wearable | Custom Wearable | 48.38 | 0.48 | 43.31 | 0.48 |
| CLSP MLP 5% | Wearable | Custom Wearable | 49.30 | 0.54 | 56.68 | 0.69 |
| CLSP MLP 25% | Wearable | Custom Wearable | 50.69 | 0.58 | 58.98 | 0.70 |
| CLSP MLP 50% | Wearable | Custom Wearable | 49.07 | 0.60 | 46.77 | 0.54 |
| CLSP CNN 5% | Wearable | Custom Wearable | 52.07 | 0.61 | 59.67 | 0.72 |
| CLSP CNN 25% | Wearable | Custom Wearable | 50.69 | 0.60 | 51.84 | 0.63 |
| CLSP CNN 50% | Wearable | Custom Wearable | 47.92 | 0.55 | 49.30 | 0.59 |
| RF | Wearable | Lab Based Device | 50.91 | 0.11 | 45.81 | 0.60 |
| LDA | Wearable | Lab Based Device | 54.68 | 0.35 | 47.29 | 0.41 |
| HC + MLP | Wearable | Lab Based Device | 50.47 | 0.45 | 47.31 | 0.52 |
| CLSP MLP 5% | Wearable | Lab Based Device | 53.86 | 0.37 | 51.30 | 0.55 |
| CLSP MLP 25% | Wearable | Lab Based Device | 52.01 | 0.40 | 46.52 | 0.61 |
| CLSP MLP 50% | Wearable | Lab Based Device | 53.01 | 0.43 | 50.81 | 0.60 |
| CLSP CNN 5% | Wearable | Lab Based Device | 52.57 | 0.40 | 48.04 | 0.55 |
| CLSP CNN 25% | Wearable | Lab Based Device | 53.59 | 0.40 | 48.66 | 0.57 |
| CLSP CNN 50% | Wearable | Lab Based Device | 53.99 | 0.35 | 47.98 | 0.57 |

Table 48: Performance comparison of ML (RF, LDA), DL (HC + MLP), pretrained CLSP fine-tuned on **arousal and valence classification** using **PPG modality** with datasets of **Wearable Group** as training cohort. Accuracy and F1 scores are reported.

| Model | Training Cohort | Testing Cohort | Arousal | | Valence | |
|---|---|---|---|---|---|---|
| | | | Accuracy | F1-score | Accuracy | F1-score |
| RF | Custom Wearable | Wearable | 48.78 | 0.60 | 57.92 | 0.73 |
| LDA | Custom Wearable | Wearable | 51.30 | 0.44 | 50.40 | 0.54 |
| HC + MLP | Custom Wearable | Wearable | 51.88 | 0.48 | 51.18 | 0.57 |
| CLSP MLP 5% | Custom Wearable | Wearable | 50.86 | 0.58 | 54.12 | 0.65 |
| CLSP MLP 25% | Custom Wearable | Wearable | 51.13 | 0.49 | 54.88 | 0.67 |
| CLSP MLP 50% | Custom Wearable | Wearable | 52.06 | 0.58 | 56.79 | 0.70 |
| CLSP CNN 5% | Custom Wearable | Wearable | 50.39 | 0.60 | 56.52 | 0.69 |
| CLSP CNN 25% | Custom Wearable | Wearable | 48.12 | 0.56 | 57.61 | 0.72 |
| CLSP CNN 50% | Custom Wearable | Wearable | 53.89 | 0.58 | 57.95 | 0.73 |
| RF | Custom Wearable | Lab Based Device | 51.43 | 0.62 | 45.98 | 0.62 |
| LDA | Custom Wearable | Lab Based Device | 55.99 | 0.46 | 49.74 | 0.58 |
| HC + MLP | Custom Wearable | Lab Based Device | 55.32 | 0.48 | 47.46 | 0.59 |
| CLSP MLP 5% | Custom Wearable | Lab Based Device | 53.43 | 0.53 | 46.77 | 0.43 |
| CLSP MLP 25% | Custom Wearable | Lab Based Device | 53.57 | 0.41 | 46.46 | 0.48 |
| CLSP MLP 50% | Custom Wearable | Lab Based Device | 54.59 | 0.53 | 45.81 | 0.54 |
| CLSP CNN 5% | Custom Wearable | Lab Based Device | 52.26 | 0.59 | 46.37 | 0.56 |
| CLSP CNN 25% | Custom Wearable | Lab Based Device | 50.87 | 0.59 | 46.33 | 0.61 |
| CLSP CNN 50% | Custom Wearable | Lab Based Device | 56.78 | 0.47 | 46.00 | 0.63 |

Table 49: Performance comparison of ML (RF, LDA), DL (HC + MLP), pretrained CLSP fine-tuned on **arousal and valence classification** using **PPG modality** with datasets of **Custom Wearable Group** as training cohort. Accuracy and F1 scores are reported.

| Model | Training Cohort | Testing Cohort | Arousal | | Valence | |
|---|---|---|---|---|---|---|
| | | | Accuracy | F1-score | Accuracy | F1-score |
| RF | Lab Based Device | Custom Wearable | 50.00 | 0.59 | 41.01 | 0.47 |
| LDA | Lab Based Device | Custom Wearable | 50.23 | 0.60 | 53.22 | 0.64 |
| HC + MLP | Lab Based Device | Custom Wearable | 53.68 | 0.65 | 56.68 | 0.67 |
| CLSP MLP 5% | Lab Based Device | Custom Wearable | 53.91 | 0.60 | 50.69 | 0.62 |
| CLSP MLP 25% | Lab Based Device | Custom Wearable | 49.77 | 0.58 | 50.23 | 0.63 |
| CLSP MLP 50% | Lab Based Device | Custom Wearable | 52.53 | 0.63 | 52.99 | 0.67 |
| CLSP CNN 5% | Lab Based Device | Custom Wearable | 50.92 | 0.59 | 53.45 | 0.64 |
| CLSP CNN 25% | Lab Based Device | Custom Wearable | 51.38 | 0.58 | 47.00 | 0.58 |
| CLSP CNN 50% | Lab Based Device | Custom Wearable | 52.99 | 0.64 | 49.30 | 0.61 |
| RF | Lab Based Device | Wearable | 53.64 | 0.56 | 44.47 | 0.30 |
| LDA | Lab Based Device | Wearable | 53.09 | 0.59 | 51.20 | 0.56 |
| HC + MLP | Lab Based Device | Wearable | 51.42 | 0.60 | 46.18 | 0.43 |
| CLSP MLP 5% | Lab Based Device | Wearable | 49.34 | 0.50 | 52.30 | 0.58 |
| CLSP MLP 25% | Lab Based Device | Wearable | 53.26 | 0.50 | 49.36 | 0.53 |
| CLSP MLP 50% | Lab Based Device | Wearable | 53.93 | 0.54 | 48.12 | 0.49 |
| CLSP CNN 5% | Lab Based Device | Wearable | 50.59 | 0.48 | 50.96 | 0.56 |
| CLSP CNN 25% | Lab Based Device | Wearable | 53.41 | 0.44 | 47.85 | 0.49 |
| CLSP CNN 50% | Lab Based Device | Wearable | 53.37 | 0.58 | 47.59 | 0.49 |

Table 50: Performance comparison of ML (RF, LDA), DL (HC + MLP), pretrained CLSP fine-tuned on **arousal and valence classification** using **PPG modality** with datasets of **Lab-based Devices Group** as training cohort. Accuracy and F1 scores are reported.

| Model | Training Cohort | Testing Cohort | Arousal | | Valence | |
|---|---|---|---|---|---|---|
| | | | Accuracy | F1-score | Accuracy | F1-score |
| RF | Lab Based Device | Lab Based Device | 53.97 | 0.54 | 51.44 | 0.51 |
| MLP | Lab Based Device | Lab Based Device | 52.05 | 0.53 | 50.36 | 0.49 |
| RF | Wearable | Wearable | 47.21 | 0.47 | 50.76 | 0.50 |
| MLP | Wearable | Wearable | 48.64 | 0.48 | 55.27 | 0.55 |
| RF | Custom Wearable | Custom Wearable | 45.32 | 0.37 | 45.21 | 0.49 |
| MLP | Custom Wearable | Custom Wearable | 46.75 | 0.40 | 46.87 | 0.50 |
| CLSP Zero Shot | — | Wearable | 49.05 | 0.43 | 53.07 | 0.62 |
| CLSP Zero Shot | — | Lab Based Device | 48.02 | 0.35 | 49.64 | 0.59 |
| CLSP Zero Shot | — | Custom Wearable | 42.39 | 0.44 | 50.92 | 0.60 |

Table 51: Performance comparison of **In-cohort** and **zero-shot** performance for different groups based upon device type on **arousal and valence classification** using **PPG modality**. Accuracy and F1 scores are reported.

| Model | Training Cohort | Testing Cohort | Arousal | | Valence | |
|---|---|---|---|---|---|---|
| | | | Accuracy | F1-score | Accuracy | F1-score |
| RF | Wearable | Custom Wearable | 37.22 | 0.33 | 61.38 | 0.74 |
| LDA | Wearable | Custom Wearable | 61.18 | 0.73 | 25.75 | 0.32 |
| HC + MLP | Wearable | Custom Wearable | 50.95 | 0.60 | 43.68 | 0.57 |
| CLSP MLP 5% | Wearable | Custom Wearable | 58.88 | 0.72 | 65.84 | 0.78 |
| CLSP MLP 25% | Wearable | Custom Wearable | 55.91 | 0.68 | 59.28 | 0.72 |
| CLSP MLP 50% | Wearable | Custom Wearable | 61.08 | 0.73 | 60.64 | 0.74 |
| CLSP CNN 5% | Wearable | Custom Wearable | 58.51 | 0.70 | 48.41 | 0.62 |
| CLSP CNN 25% | Wearable | Custom Wearable | 49.51 | 0.58 | 56.98 | 0.71 |
| CLSP CNN 50% | Wearable | Custom Wearable | 57.98 | 0.70 | 59.28 | 0.72 |
| RF | Wearable | Lab Based Device | 51.08 | 0.13 | 46.89 | 0.61 |
| LDA | Wearable | Lab Based Device | 49.18 | 0.48 | 45.11 | 0.52 |
| HC + MLP | Wearable | Lab Based Device | 52.66 | 0.45 | 48.46 | 0.50 |
| CLSP MLP 5% | Wearable | Lab Based Device | 48.16 | 0.50 | 45.86 | 0.61 |
| CLSP MLP 25% | Wearable | Lab Based Device | 49.41 | 0.34 | 45.94 | 0.56 |
| CLSP MLP 50% | Wearable | Lab Based Device | 52.60 | 0.41 | 46.02 | 0.55 |
| CLSP CNN 5% | Wearable | Lab Based Device | 48.27 | 0.39 | 50.33 | 0.62 |
| CLSP CNN 25% | Wearable | Lab Based Device | 47.54 | 0.17 | 46.33 | 0.56 |
| CLSP CNN 50% | Wearable | Lab Based Device | 53.59 | 0.50 | 46.15 | 0.58 |

Table 52: Performance comparison of ML (RF, LDA), DL (HC + MLP), pretrained CLSP fine-tuned on **arousal and valence classification** using **EDA + PPG modality** with datasets of **Wearable Group** as training cohort. Accuracy and F1 scores are reported.

| Model | Training Cohort | Testing Cohort | Arousal | | Valence | |
|---|---|---|---|---|---|---|
| | | | Accuracy | F1-score | Accuracy | F1-score |
| RF | Custom Wearable | Wearable | 49.73 | 0.63 | 57.89 | 0.73 |
| LDA | Custom Wearable | Wearable | 48.97 | 0.42 | 50.57 | 0.53 |
| HC + MLP | Custom Wearable | Wearable | 51.34 | 0.59 | 56.73 | 0.70 |
| CLSP MLP 5% | Custom Wearable | Wearable | 51.33 | 0.64 | 57.97 | 0.73 |
| CLSP MLP 25% | Custom Wearable | Wearable | 50.20 | 0.65 | 57.40 | 0.72 |
| CLSP MLP 50% | Custom Wearable | Wearable | 50.11 | 0.66 | 58.05 | 0.73 |
| CLSP CNN 5% | Custom Wearable | Wearable | 50.04 | 0.66 | 57.54 | 0.73 |
| CLSP CNN 25% | Custom Wearable | Wearable | 49.74 | 0.66 | 57.80 | 0.73 |
| CLSP CNN 50% | Custom Wearable | Wearable | 50.64 | 0.64 | 57.99 | 0.73 |
| RF | Custom Wearable | Lab Based Device | 50.87 | 0.63 | 40.00 | 0.63 |
| LDA | Custom Wearable | Lab Based Device | 52.24 | 0.45 | 49.41 | 0.56 |
| HC + MLP | Custom Wearable | Lab Based Device | 53.38 | 0.55 | 46.73 | 0.62 |
| CLSP MLP 5% | Custom Wearable | Lab Based Device | 48.97 | 0.64 | 45.88 | 0.62 |
| CLSP MLP 25% | Custom Wearable | Lab Based Device | 50.37 | 0.64 | 45.94 | 0.62 |
| CLSP MLP 50% | Custom Wearable | Lab Based Device | 51.01 | 0.63 | 46.38 | 0.63 |
| CLSP CNN 5% | Custom Wearable | Lab Based Device | 48.79 | 0.65 | 45.96 | 0.63 |
| CLSP CNN 25% | Custom Wearable | Lab Based Device | 49.52 | 0.65 | 46.08 | 0.63 |
| CLSP CNN 50% | Custom Wearable | Lab Based Device | 50.76 | 0.63 | 46.48 | 0.63 |

Table 53: Performance comparison of ML (RF, LDA), DL (HC + MLP), pretrained CLSP fine-tuned on **arousal and valence classification** using **EDA + PPG modality** with datasets of **Custom Wearable Group** as training cohort. Accuracy and F1 scores are reported.

| Model | Training Cohort | Testing Cohort | Arousal | | Valence | |
|---|---|---|---|---|---|---|
| | | | Accuracy | F1-score | Accuracy | F1-score |
| RF | Lab Based Device | Custom Wearable | 49.78 | 0.59 | 32.48 | 0.42 |
| LDA | Lab Based Device | Custom Wearable | 53.81 | 0.65 | 69.51 | 0.81 |
| HC + MLP | Lab Based Device | Custom Wearable | 54.24 | 0.65 | 56.58 | 0.70 |
| CLSP MLP 5% | Lab Based Device | Custom Wearable | 62.67 | 0.75 | 25.39 | 0.31 |
| CLSP MLP 25% | Lab Based Device | Custom Wearable | 62.27 | 0.74 | 38.68 | 0.50 |
| CLSP MLP 50% | Lab Based Device | Custom Wearable | 65.17 | 0.77 | 45.95 | 0.59 |
| CLSP CNN 5% | Lab Based Device | Custom Wearable | 58.74 | 0.70 | 47.51 | 0.61 |
| CLSP CNN 25% | Lab Based Device | Custom Wearable | 65.81 | 0.78 | 38.65 | 0.51 |
| CLSP CNN 50% | Lab Based Device | Custom Wearable | 64.97 | 0.77 | 47.01 | 0.61 |
| RF | Lab Based Device | Wearable | 50.73 | 0.55 | 45.73 | 0.31 |
| LDA | Lab Based Device | Wearable | 49.99 | 0.56 | 49.25 | 0.53 |
| HC + MLP | Lab Based Device | Wearable | 52.32 | 0.56 | 47.66 | 0.49 |
| CLSP MLP 5% | Lab Based Device | Wearable | 50.50 | 0.54 | 47.09 | 0.37 |
| CLSP MLP 25% | Lab Based Device | Wearable | 50.67 | 0.49 | 45.87 | 0.39 |
| CLSP MLP 50% | Lab Based Device | Wearable | 50.88 | 0.57 | 46.47 | 0.43 |
| CLSP CNN 5% | Lab Based Device | Wearable | 50.58 | 0.55 | 48.47 | 0.48 |
| CLSP CNN 25% | Lab Based Device | Wearable | 50.51 | 0.59 | 48.05 | 0.47 |
| CLSP CNN 50% | Lab Based Device | Wearable | 52.08 | 0.56 | 46.74 | 0.45 |

Table 54: Performance comparison of ML (RF, LDA), DL (HC + MLP), pretrained CLSP fine-tuned on **arousal and valence classification** using **EDA + PPG modality** with datasets of **Lab-based Devices Group** as training cohort. Accuracy and F1 scores are reported.

| Model | Training Cohort | Testing Cohort | Arousal | | Valence | |
|---|---|---|---|---|---|---|
| | | | Accuracy | F1-score | Accuracy | F1-score |
| RF | Lab Based Device | Lab Based Device | 55.65 | 0.56 | 52.47 | 0.52 |
| MLP | Lab Based Device | Lab Based Device | 52.14 | 0.51 | 51.79 | 0.52 |
| RF | Wearable | Wearable | 51.24 | 0.51 | 52.54 | 0.52 |
| MLP | Wearable | Wearable | 52.51 | 0.52 | 53.35 | 0.54 |
| RF | Custom Wearable | Custom Wearable | 45.12 | 0.45 | 45.23 | 0.47 |
| MLP | Custom Wearable | Custom Wearable | 43.51 | 0.50 | 43.22 | 0.45 |
| CLSP Zero Shot | — | Wearable | 47.02 | 0.52 | 49.31 | 0.53 |
| CLSP Zero Shot | — | Lab Based Device | 50.56 | 0.50 | 49.97 | 0.45 |
| CLSP Zero Shot | — | Custom Wearable | 46.61 | 0.58 | 41.21 | 0.54 |

Table 55: Performance comparison of **In-cohort** and **zero-shot** performance for different groups based upon device type on **arousal and valence classification** using **EDA + PPG modality**. Accuracy and F1 scores are reported.

| Model | Training Cohort | Testing Cohort | Arousal | | Valence | |
|---|---|---|---|---|---|---|
| | | | Accuracy | F1-score | Accuracy | F1-score |
| RF | Stimulus-Label | Self-report | 50.19 | 0.49 | 50.66 | 0.57 |
| LDA | Stimulus-Label | Self-report | 49.78 | 0.59 | 46.11 | 0.40 |
| HC + MLP | Stimulus-Label | Self-report | 48.85 | 0.57 | 50.29 | 0.56 |
| CLSP MLP 5% | Stimulus-Label | Self-report | 52.04 | 0.39 | 48.21 | 0.50 |
| CLSP MLP 25% | Stimulus-Label | Self-report | 49.47 | 0.53 | 47.21 | 0.48 |
| CLSP MLP 50% | Stimulus-Label | Self-report | 51.85 | 0.53 | 46.76 | 0.47 |
| CLSP CNN 5% | Stimulus-Label | Self-report | 51.14 | 0.43 | 46.54 | 0.40 |
| CLSP CNN 25% | Stimulus-Label | Self-report | 51.26 | 0.41 | 49.23 | 0.54 |
| CLSP CNN 50% | Stimulus-Label | Self-report | 52.69 | 0.57 | 50.35 | 0.54 |
| RF | Stimulus-Label | Expert | 24.19 | 0.24 | 79.89 | 0.87 |
| LDA | Stimulus-Label | Expert | 58.94 | 0.71 | 62.52 | 0.72 |
| HC + MLP | Stimulus-Label | Expert | 47.53 | 0.59 | 72.57 | 0.82 |
| CLSP MLP 5% | Stimulus-Label | Expert | 35.09 | 0.43 | 45.31 | 0.58 |
| CLSP MLP 25% | Stimulus-Label | Expert | 54.51 | 0.66 | 63.37 | 0.75 |
| CLSP MLP 50% | Stimulus-Label | Expert | 48.04 | 0.58 | 64.73 | 0.77 |
| CLSP CNN 5% | Stimulus-Label | Expert | 41.90 | 0.52 | 32.87 | 0.36 |
| CLSP CNN 25% | Stimulus-Label | Expert | 20.61 | 0.18 | 65.07 | 0.76 |
| CLSP CNN 50% | Stimulus-Label | Expert | 68.14 | 0.79 | 73.59 | 0.84 |

Table 56: Performance comparison of ML (RF, LDA), DL (HC + MLP), pretrained CLSP fine-tuned on **arousal and valence classification** using **EDA modality** with datasets of **Stimulus-based Labeling Group** as training cohort. Accuracy and F1 scores are reported.

| Model | Training Cohort | Testing Cohort | Arousal | | Valence | |
|---|---|---|---|---|---|---|
| | | | Accuracy | F1-score | Accuracy | F1-score |
| RF | Self-report | Stimulus-Label | 51.02 | 0.52 | 48.31 | 0.50 |
| LDA | Self-report | Stimulus-Label | 51.39 | 0.62 | 48.60 | 0.61 |
| HC + MLP | Self-report | Stimulus-Label | 50.63 | 0.52 | 48.53 | 0.46 |
| CLSP MLP 5% | Self-report | Stimulus-Label | 50.64 | 0.43 | 49.04 | 0.62 |
| CLSP MLP 25% | Self-report | Stimulus-Label | 47.90 | 0.36 | 49.72 | 0.63 |
| CLSP MLP 50% | Self-report | Stimulus-Label | 50.28 | 0.44 | 48.74 | 0.62 |
| CLSP CNN 5% | Self-report | Stimulus-Label | 51.17 | 0.42 | 50.47 | 0.61 |
| CLSP CNN 25% | Self-report | Stimulus-Label | 50.64 | 0.44 | 47.31 | 0.63 |
| CLSP CNN 50% | Self-report | Stimulus-Label | 50.52 | 0.53 | 45.91 | 0.57 |
| RF | Self-report | Expert | 45.65 | 0.59 | 37.99 | 0.41 |
| LDA | Self-report | Expert | 79.04 | 0.87 | 67.80 | 0.79 |
| HC + MLP | Self-report | Expert | 61.32 | 0.71 | 50.25 | 0.56 |
| CLSP MLP 5% | Self-report | Expert | 43.44 | 0.56 | 71.03 | 0.82 |
| CLSP MLP 25% | Self-report | Expert | 17.54 | 0.20 | 71.55 | 0.82 |
| CLSP MLP 50% | Self-report | Expert | 30.15 | 0.40 | 66.09 | 0.76 |
| CLSP CNN 5% | Self-report | Expert | 13.45 | 0.13 | 72.06 | 0.83 |
| CLSP CNN 25% | Self-report | Expert | 40.03 | 0.54 | 72.23 | 0.83 |
| CLSP CNN 50% | Self-report | Expert | 57.75 | 0.71 | 66.61 | 0.76 |

Table 57: Performance comparison of ML (RF, LDA), DL (HC + MLP), pretrained CLSP fine-tuned on **arousal and valence classification** using **EDA modality** with datasets of **Self-report-based Labeling Group** as training cohort. Accuracy and F1 scores are reported.

| Model | Training Cohort | Testing Cohort | Arousal | | Valence | |
|---|---|---|---|---|---|---|
| | | | Accuracy | F1-score | Accuracy | F1-score |
| RF | Expert | Self-report | 50.69 | 0.51 | 47.74 | 0.43 |
| LDA | Expert | Self-report | 51.22 | 0.65 | 57.32 | 0.69 |
| HC + MLP | Expert | Self-report | 49.90 | 0.53 | 50.45 | 0.55 |
| CLSP MLP 5% | Expert | Self-report | 50.80 | 0.65 | 52.07 | 0.61 |
| CLSP MLP 25% | Expert | Self-report | 50.41 | 0.64 | 56.38 | 0.69 |
| CLSP MLP 50% | Expert | Self-report | 50.72 | 0.63 | 55.86 | 0.69 |
| CLSP CNN 5% | Expert | Self-report | 49.92 | 0.62 | 51.29 | 0.59 |
| CLSP CNN 25% | Expert | Self-report | 50.61 | 0.61 | 55.05 | 0.67 |
| CLSP CNN 50% | Expert | Self-report | 50.43 | 0.61 | 55.75 | 0.67 |
| RF | Expert | Stimulus-Label | 50.08 | 0.45 | 55.58 | 0.42 |
| LDA | Expert | Stimulus-Label | 50.95 | 0.64 | 52.79 | 0.63 |
| HC + MLP | Expert | Stimulus-Label | 49.21 | 0.48 | 51.85 | 0.45 |
| CLSP MLP 5% | Expert | Stimulus-Label | 49.71 | 0.64 | 58.06 | 0.65 |
| CLSP MLP 25% | Expert | Stimulus-Label | 49.33 | 0.63 | 52.26 | 0.61 |
| CLSP MLP 50% | Expert | Stimulus-Label | 50.78 | 0.62 | 52.67 | 0.63 |
| CLSP CNN 5% | Expert | Stimulus-Label | 48.53 | 0.61 | 55.90 | 0.62 |
| CLSP CNN 25% | Expert | Stimulus-Label | 49.83 | 0.60 | 54.27 | 0.63 |
| CLSP CNN 50% | Expert | Stimulus-Label | 50.51 | 0.61 | 51.27 | 0.55 |

Table 58: Performance comparison of ML (RF, LDA), DL (HC + MLP), pretrained CLSP fine-tuned on **arousal and valence classification** using **EDA modality** with datasets of **Expert based Labeling Group** as training cohort. Accuracy and F1 scores are reported.

| Model | Training Cohort | Testing Cohort | Arousal | | Valence | |
|---|---|---|---|---|---|---|
| | | | Accuracy | F1-score | Accuracy | F1-score |
| RF | Stimulus-Label | Stimulus-Label | 54.38 | 0.54 | 61.45 | 0.61 |
| MLP | Stimulus-Label | Stimulus-Label | 52.89 | 0.52 | 56.40 | 0.56 |
| RF | Self-report | Self-report | 51.41 | 0.49 | 54.41 | 0.53 |
| MLP | Self-report | Self-report | 55.51 | 0.53 | 52.47 | 0.51 |
| RF | Expert | Expert | 70.32 | 0.52 | 70.24 | 0.55 |
| MLP | Expert | Expert | 75.40 | 0.48 | 75.38 | 0.56 |
| CLSP Zero Shot | – | Stimulus-Label | 50.57 | 0.60 | 48.85 | 0.63 |
| CLSP Zero Shot | – | Self-report | 50.25 | 0.60 | 56.46 | 0.70 |
| CLSP Zero Shot | – | Expert | 84.49 | 0.91 | 71.55 | 0.83 |

Table 59: Performance comparison of **In-cohort** and **zero-shot** performance for different groups based upon Labeling type on **arousal and valence classification** using **EDA modality**. Accuracy and F1 scores are reported.

| Model | Training Cohort | Testing Cohort | Arousal | | Valence | |
|---|---|---|---|---|---|---|
| | | | Accuracy | F1-score | Accuracy | F1-score |
| RF | Stimulus-Label | Self-report | 51.14 | 0.43 | 49.57 | 0.40 |
| LDA | Stimulus-Label | Self-report | 49.76 | 0.21 | 48.38 | 0.59 |
| HC + MLP | Stimulus-Label | Self-report | 48.84 | 0.39 | 51.80 | 0.51 |
| CLSP MLP 5% | Stimulus-Label | Self-report | 49.18 | 0.45 | 52.52 | 0.59 |
| CLSP MLP 25% | Stimulus-Label | Self-report | 51.23 | 0.50 | 50.77 | 0.57 |
| CLSP MLP 50% | Stimulus-Label | Self-report | 49.31 | 0.48 | 49.49 | 0.53 |
| CLSP CNN 5% | Stimulus-Label | Self-report | 49.90 | 0.40 | 49.37 | 0.52 |
| CLSP CNN 25% | Stimulus-Label | Self-report | 51.23 | 0.50 | 49.66 | 0.55 |
| CLSP CNN 50% | Stimulus-Label | Self-report | 52.17 | 0.51 | 49.66 | 0.55 |
| RF | Stimulus-Label | Expert | 53.91 | 0.69 | 66.66 | 0.76 |
| LDA | Stimulus-Label | Expert | 23.76 | 0.29 | 76.81 | 0.85 |
| HC + MLP | Stimulus-Label | Expert | 50.72 | 0.62 | 62.89 | 0.73 |
| CLSP MLP 5% | Stimulus-Label | Expert | 44.63 | 0.60 | 38.84 | 0.40 |
| CLSP MLP 25% | Stimulus-Label | Expert | 42.02 | 0.57 | 46.08 | 0.55 |
| CLSP MLP 50% | Stimulus-Label | Expert | 55.36 | 0.70 | 49.27 | 0.58 |
| CLSP CNN 5% | Stimulus-Label | Expert | 42.89 | 0.57 | 39.71 | 0.47 |
| CLSP CNN 25% | Stimulus-Label | Expert | 49.56 | 0.65 | 44.92 | 0.53 |
| CLSP CNN 50% | Stimulus-Label | Expert | 53.33 | 0.68 | 44.92 | 0.53 |

Table 60: Performance comparison of ML (RF, LDA), DL (HC + MLP), pretrained CLSP fine-tuned on **arousal and valence classification** using **PPG modality** with datasets of **Stimulus-based Labeling Group** as training cohort. Accuracy and F1 scores are reported.

| Model | Training Cohort | Testing Cohort | Arousal | | Valence | |
|---|---|---|---|---|---|---|
| | | | Accuracy | F1-score | Accuracy | F1-score |
| RF | Self-report | Stimulus-Label | 51.14 | 0.43 | 49.57 | 0.40 |
| LDA | Self-report | Stimulus-Label | 49.76 | 0.21 | 48.38 | 0.59 |
| HC + MLP | Self-report | Stimulus-Label | 48.84 | 0.39 | 51.80 | 0.51 |
| CLSP MLP 5% | Self-report | Stimulus-Label | 49.35 | 0.31 | 49.79 | 0.57 |
| CLSP MLP 25% | Self-report | Stimulus-Label | 50.85 | 0.36 | 48.45 | 0.56 |
| CLSP MLP 50% | Self-report | Stimulus-Label | 51.22 | 0.39 | 48.67 | 0.58 |
| CLSP CNN 5% | Self-report | Stimulus-Label | 51.49 | 0.44 | 51.15 | 0.61 |
| CLSP CNN 25% | Self-report | Stimulus-Label | 50.78 | 0.39 | 48.07 | 0.57 |
| CLSP CNN 50% | Self-report | Stimulus-Label | 49.31 | 0.35 | 47.75 | 0.60 |
| RF | Self-report | Expert | 53.91 | 0.69 | 66.66 | 0.76 |
| LDA | Self-report | Expert | 23.76 | 0.29 | 76.81 | 0.85 |
| HC + MLP | Self-report | Expert | 50.72 | 0.62 | 62.89 | 0.73 |
| CLSP MLP 5% | Self-report | Expert | 39.42 | 0.52 | 56.52 | 0.69 |
| CLSP MLP 25% | Self-report | Expert | 43.47 | 0.53 | 60.29 | 0.71 |
| CLSP MLP 50% | Self-report | Expert | 44.34 | 0.55 | 68.40 | 0.78 |
| CLSP CNN 5% | Self-report | Expert | 50.72 | 0.62 | 63.76 | 0.77 |
| CLSP CNN 25% | Self-report | Expert | 45.50 | 0.57 | 63.47 | 0.73 |
| CLSP CNN 50% | Self-report | Expert | 49.56 | 0.60 | 75.94 | 0.85 |

Table 61: Performance comparison of ML (RF, LDA), DL (HC + MLP), pretrained CLSP fine-tuned on **arousal and valence classification** using **PPG modality** with datasets of **Self-report based Labeling Group** as training cohort. Accuracy and F1 scores are reported.

| Model | Training Cohort | Testing Cohort | Arousal | | Valence | |
|---|---|---|---|---|---|---|
| | | | Accuracy | F1-score | Accuracy | F1-score |
| RF | Expert | Self-report | 49.42 | 0.59 | 52.36 | 0.56 |
| LDA | Expert | Self-report | 48.09 | 0.64 | 57.41 | 0.72 |
| HC + MLP | Expert | Self-report | 48.09 | 0.64 | 54.31 | 0.62 |
| CLSP MLP 5% | Expert | Self-report | 47.17 | 0.59 | 55.14 | 0.67 |
| CLSP MLP 25% | Expert | Self-report | 48.55 | 0.62 | 56.51 | 0.70 |
| CLSP MLP 50% | Expert | Self-report | 47.85 | 0.63 | 56.78 | 0.71 |
| CLSP CNN 5% | Expert | Self-report | 46.45 | 0.60 | 49.08 | 0.51 |
| CLSP CNN 25% | Expert | Self-report | 48.20 | 0.62 | 57.00 | 0.70 |
| CLSP CNN 50% | Expert | Self-report | 48.76 | 0.64 | 55.69 | 0.70 |
| RF | Expert | Stimulus-Label | 51.43 | 0.60 | 49.48 | 0.46 |
| LDA | Expert | Stimulus-Label | 57.41 | 0.72 | 48.74 | 0.65 |
| HC + MLP | Expert | Stimulus-Label | 51.20 | 0.65 | 49.06 | 0.52 |
| CLSP MLP 5% | Expert | Stimulus-Label | 48.58 | 0.60 | 48.24 | 0.58 |
| CLSP MLP 25% | Expert | Stimulus-Label | 50.01 | 0.64 | 49.74 | 0.62 |
| CLSP MLP 50% | Expert | Stimulus-Label | 49.74 | 0.65 | 50.47 | 0.63 |
| CLSP CNN 5% | Expert | Stimulus-Label | 49.64 | 0.62 | 49.62 | 0.50 |
| CLSP CNN 25% | Expert | Stimulus-Label | 49.98 | 0.64 | 51.12 | 0.64 |
| CLSP CNN 50% | Expert | Stimulus-Label | 50.03 | 0.65 | 50.37 | 0.65 |

Table 62: Performance comparison of ML (RF, LDA), DL (HC + MLP), pretrained CLSP fine-tuned on **arousal and valence classification** using **PPG modality** with datasets of **Expert based Labeling Group** as training cohort. Accuracy and F1 scores are reported.

| Model | Training Cohort | Testing Cohort | Arousal | | Valence | |
|---|---|---|---|---|---|---|
| | | | Accuracy | F1-score | Accuracy | F1-score |
| RF | Stimulus-Label | Stimulus-Label | 51.83 | 0.51 | 53.26 | 0.53 |
| MLP | Stimulus-Label | Stimulus-Label | 47.76 | 0.46 | 52.83 | 0.52 |
| RF | Self-report | Self-report | 51.35 | 0.50 | 43.45 | 0.41 |
| MLP | Self-report | Self-report | 54.72 | 0.52 | 46.87 | 0.48 |
| RF | Expert | Expert | 51.20 | 0.28 | 52.35 | 0.42 |
| MLP | Expert | Expert | 47.10 | 0.21 | 44.20 | 0.32 |
| CLSP Zero Shot | – | Stimulus-Label | 47.36 | 0.37 | 52.55 | 0.59 |
| CLSP Zero Shot | – | Self-report | 49.96 | 0.43 | 51.63 | 0.62 |
| CLSP Zero Shot | – | Expert | 27.82 | 0.39 | 48.98 | 0.60 |

Table 63: Performance comparison of **In-cohort** and **zero-shot** performance for different groups based upon Labeling type on **arousal and valence classification** using **PPG modality**. Accuracy and F1 scores are reported.

### 3.2.3 EDA + PPG

This section presents the labeling methods experiment results for the EDA+PPG modality across three tables: stimulus-based labels in Table 64, self-reports in Table 65, expert annotations in Table 66, in-cohort, and zero-short experiments in Table 67.

| Model | Training Cohort | Testing Cohort | Arousal | | Valence | |
|---|---|---|---|---|---|---|
| | | | Accuracy | F1-score | Accuracy | F1-score |
| RF | Stimulus-Label | Self-report | 51.99 | 0.52 | 49.29 | 0.56 |
| LDA | Stimulus-Label | Self-report | 52.23 | 0.63 | 41.25 | 0.31 |
| HC + MLP | Stimulus-Label | Self-report | 51.87 | 0.57 | 47.27 | 0.49 |
| CLSP MLP 5% | Stimulus-Label | Self-report | 50.75 | 0.49 | 44.47 | 0.44 |
| CLSP MLP 25% | Stimulus-Label | Self-report | 54.93 | 0.59 | 45.41 | 0.46 |
| CLSP MLP 50% | Stimulus-Label | Self-report | 54.75 | 0.54 | 49.00 | 0.55 |
| CLSP CNN 5% | Stimulus-Label | Self-report | 49.90 | 0.46 | 45.35 | 0.44 |
| CLSP CNN 25% | Stimulus-Label | Self-report | 52.75 | 0.54 | 46.25 | 0.46 |
| CLSP CNN 50% | Stimulus-Label | Self-report | 53.25 | 0.60 | 46.77 | 0.53 |
| RF | Stimulus-Label | Expert | 54.49 | 0.65 | 64.34 | 0.74 |
| LDA | Stimulus-Label | Expert | 72.46 | 0.82 | 43.76 | 0.49 |
| HC + MLP | Stimulus-Label | Expert | 60.58 | 0.71 | 51.30 | 0.59 |
| CLSP MLP 5% | Stimulus-Label | Expert | 43.18 | 0.58 | 52.46 | 0.60 |
| CLSP MLP 25% | Stimulus-Label | Expert | 66.08 | 0.76 | 55.65 | 0.67 |
| CLSP MLP 50% | Stimulus-Label | Expert | 45.50 | 0.55 | 64.34 | 0.73 |
| CLSP CNN 5% | Stimulus-Label | Expert | 57.68 | 0.69 | 48.40 | 0.55 |
| CLSP CNN 25% | Stimulus-Label | Expert | 55.65 | 0.67 | 54.49 | 0.62 |
| CLSP CNN 50% | Stimulus-Label | Expert | 58.55 | 0.69 | 64.92 | 0.74 |

Table 64: Performance comparison of ML (RF, LDA), DL (HC + MLP), pretrained CLSP fine-tuned on **arousal and valence classification** using **EDA + PPG modality** with datasets of **Stimulus based Labeling Group** as training cohort. Accuracy and F1 scores are reported.

## 3.3 Setting Group

The setting group results are presented across four tables, corresponding to different experimental contexts: lab-based, constrained, real-life settings, in-cohort, and zero-short experiments. These results illustrate the cross-group transferability of models across varying levels of experimental control and ecological validity. Specifically, they shed light on how well models trained in controlled environments (e.g., lab or constrained settings) generalize to more naturalistic, real-life scenarios, which are critical for real-world deployment. As with the other groups, the results are further subdivided by modality, EDA, PPG, and the combined EDA+PPG, to evaluate how each signal type supports generalization across experimental settings.

| Model | Training Cohort | Testing Cohort | Arousal | | Valence | |
|---|---|---|---|---|---|---|
| | | | Accuracy | F1-score | Accuracy | F1-score |
| RF | Self-report | Stimulus-Label | 51.61 | 0.43 | 50.10 | 0.40 |
| LDA | Self-report | Stimulus-Label | 50.76 | 0.52 | 48.31 | 0.59 |
| HC + MLP | Self-report | Stimulus-Label | 49.84 | 0.47 | 50.00 | 0.46 |
| CLSP MLP 5% | Self-report | Stimulus-Label | 49.81 | 0.48 | 47.44 | 0.60 |
| CLSP MLP 25% | Self-report | Stimulus-Label | 48.31 | 0.35 | 46.92 | 0.58 |
| CLSP MLP 50% | Self-report | Stimulus-Label | 51.61 | 0.51 | 50.68 | 0.51 |
| CLSP CNN 5% | Self-report | Stimulus-Label | 51.65 | 0.57 | 47.85 | 0.61 |
| CLSP CNN 25% | Self-report | Stimulus-Label | 49.89 | 0.42 | 48.00 | 0.56 |
| CLSP CNN 50% | Self-report | Stimulus-Label | 50.66 | 0.52 | 50.40 | 0.57 |
| RF | Self-report | Expert | 48.40 | 0.60 | 60.58 | 0.72 |
| LDA | Self-report | Expert | 73.33 | 0.84 | 78.84 | 0.87 |
| HC + MLP | Self-report | Expert | 54.78 | 0.67 | 57.97 | 0.67 |
| CLSP MLP 5% | Self-report | Expert | 42.60 | 0.59 | 75.07 | 0.85 |
| CLSP MLP 25% | Self-report | Expert | 42.60 | 0.56 | 73.62 | 0.84 |
| CLSP MLP 50% | Self-report | Expert | 62.89 | 0.75 | 68.40 | 0.78 |
| CLSP CNN 5% | Self-report | Expert | 53.04 | 0.66 | 47.85 | 0.87 |
| CLSP CNN 25% | Self-report | Expert | 51.59 | 0.65 | 75.07 | 0.84 |
| CLSP CNN 50% | Self-report | Expert | 54.20 | 0.68 | 71.01 | 0.80 |

Table 65: Performance comparison of ML (RF, LDA), DL (HC + MLP), pretrained CLSP fine-tuned on **arousal and valence classification** using **EDA + PPG modality** with datasets of **Self-report based Labeling Group** as training cohort. Accuracy and F1 scores are reported.

| Model | Training Cohort | Testing Cohort | Arousal | | Valence | |
|---|---|---|---|---|---|---|
| | | | Accuracy | F1-score | Accuracy | F1-score |
| RF | Expert | Self-report | 52.83 | 0.60 | 47.88 | 0.45 |
| LDA | Expert | Self-report | 53.93 | 0.69 | 62.09 | 0.75 |
| HC + MLP | Expert | Self-report | 53.03 | 0.66 | 51.79 | 0.57 |
| CLSP MLP 5% | Expert | Self-report | 53.74 | 0.69 | 43.61 | 0.44 |
| CLSP MLP 25% | Expert | Self-report | 53.43 | 0.69 | 63.25 | 0.76 |
| CLSP MLP 50% | Expert | Self-report | 53.19 | 0.69 | 62.46 | 0.76 |
| CLSP CNN 5% | Expert | Self-report | 53.36 | 0.67 | 59.23 | 0.72 |
| CLSP CNN 25% | Expert | Self-report | 52.54 | 0.67 | 63.55 | 0.77 |
| CLSP CNN 50% | Expert | Self-report | 52.87 | 0.68 | 62.49 | 0.76 |
| RF | Expert | Stimulus-Label | 50.01 | 0.55 | 54.42 | 0.40 |
| LDA | Expert | Stimulus-Label | 50.98 | 0.66 | 49.62 | 0.63 |
| HC + MLP | Expert | Stimulus-Label | 50.71 | 0.64 | 53.21 | 0.48 |
| CLSP MLP 5% | Expert | Stimulus-Label | 50.44 | 0.66 | 53.02 | 0.50 |
| CLSP MLP 25% | Expert | Stimulus-Label | 50.78 | 0.66 | 48.55 | 0.64 |
| CLSP MLP 50% | Expert | Stimulus-Label | 50.90 | 0.67 | 48.72 | 0.65 |
| CLSP CNN 5% | Expert | Stimulus-Label | 50.85 | 0.65 | 49.59 | 0.63 |
| CLSP CNN 25% | Expert | Stimulus-Label | 51.53 | 0.66 | 48.89 | 0.65 |
| CLSP CNN 50% | Expert | Stimulus-Label | 51.20 | 0.66 | 48.72 | 0.63 |

Table 66: Performance comparison of ML (RF, LDA), DL (HC + MLP), pretrained CLSP fine-tuned on **arousal and valence classification** using **EDA + PPG modality** with datasets of **Expert based Labeling Group** as training cohort. Accuracy and F1 scores are reported.

### 3.3.1 EDA

This section presents the experimental setting results for the EDA modality across three tables: lab-based settings in Table 68, constrained environments in Table 69, real-life scenarios in Table 70, in-cohort, and zero-short experiments in Table 71.

### 3.3.2 PPG

This section presents the experimental setting results for the PPG modality across three tables: lab-based settings in Table 72, constrained environments in Table 73, real-life scenarios in Table 74, in-cohort, and zero-short experiments in Table 75.

| Model | Training Cohort | Testing Cohort | Arousal | | Valence | |
|---|---|---|---|---|---|---|
| | | | Accuracy | F1-score | Accuracy | F1-score |
| RF | Stimulus-Label | Stimulus-Label | 53.72 | 0.53 | 52.39 | 0.52 |
| MLP | Stimulus-Label | Stimulus-Label | 55.88 | 0.55 | 51.61 | 0.51 |
| RF | Self-report | Self-report | 53.45 | 0.52 | 52.93 | 0.52 |
| MLP | Self-report | Self-report | 51.51 | 0.52 | 51.03 | 0.52 |
| RF | Expert | Expert | 52.37 | 0.41 | 52.28 | 0.41 |
| MLP | Expert | Expert | 44.82 | 0.48 | 45.65 | 0.49 |
| CLSP Zero Shot | – | Stimulus-Label | 48.33 | 0.50 | 50.39 | 0.48 |
| CLSP Zero Shot | – | Self-report | 47.58 | 0.54 | 47.52 | 0.53 |
| CLSP Zero Shot | – | Expert | 54.20 | 0.68 | 35.07 | 0.43 |

Table 67: Performance comparison of **In-cohort** and **zero-shot** performance for different groups based upon Labeling type on **arousal and valence classification** using **EDA + PPG modality**. Accuracy and F1 scores are reported.

| Model | Training Cohort | Testing Cohort | Arousal | | Valence | |
|---|---|---|---|---|---|---|
| | | | Accuracy | F1-score | Accuracy | F1-score |
| RF | Lab | Constraint | 39.26 | 0.19 | 64.21 | 0.76 |
| LDA | Lab | Constraint | 41.69 | 0.30 | 56.90 | 0.65 |
| HC + MLP | Lab | Constraint | 48.43 | 0.44 | 52.41 | 0.58 |
| CLSP MLP 5% | Lab | Constraint | 39.51 | 0.32 | 48.24 | 0.61 |
| CLSP MLP 25% | Lab | Constraint | 36.11 | 0.24 | 59.08 | 0.71 |
| CLSP MLP 50% | Lab | Constraint | 41.31 | 0.19 | 62.16 | 0.74 |
| CLSP CNN 5% | Lab | Constraint | 37.97 | 0.23 | 45.29 | 0.57 |
| CLSP CNN 25% | Lab | Constraint | 39.51 | 0.26 | 58.88 | 0.72 |
| CLSP CNN 50% | Lab | Constraint | 36.63 | 0.10 | 59.72 | 0.73 |
| RF | Lab | Real | 50.34 | 0.46 | 61.26 | 0.72 |
| LDA | Lab | Real | 50.03 | 0.55 | 42.68 | 0.36 |
| HC + MLP | Lab | Real | 54.75 | 0.59 | 55.12 | 0.61 |
| CLSP MLP 5% | Lab | Real | 45.77 | 0.46 | 49.13 | 0.58 |
| CLSP MLP 25% | Lab | Real | 50.55 | 0.58 | 50.39 | 0.57 |
| CLSP MLP 50% | Lab | Real | 44.04 | 0.40 | 55.33 | 0.65 |
| CLSP CNN 5% | Lab | Real | 44.30 | 0.46 | 47.72 | 0.53 |
| CLSP CNN 25% | Lab | Real | 48.24 | 0.52 | 53.28 | 0.64 |
| CLSP CNN 50% | Lab | Real | 46.51 | 0.47 | 56.01 | 0.67 |

Table 68: Performance comparison of ML (RF, LDA), DL (HC + MLP), pretrained CLSP fine-tuned on **arousal and valence classification** using **EDA modality** with datasets of **Lab-based settings** as training cohort. Accuracy and F1 scores are reported.

| Model | Training Cohort | Testing Cohort | Arousal | | Valence | |
|---|---|---|---|---|---|---|
| | | | Accuracy | F1-score | Accuracy | F1-score |
| RF | Constraint | Lab | 50.93 | 0.56 | 52.40 | 0.63 |
| LDA | Constraint | Lab | 50.02 | 0.47 | 49.87 | 0.36 |
| HC + MLP | Constraint | Lab | 50.67 | 0.50 | 49.95 | 0.42 |
| CLSP MLP 5% | Constraint | Lab | 49.48 | 0.56 | 54.08 | 0.64 |
| CLSP MLP 25% | Constraint | Lab | 47.54 | 0.52 | 53.67 | 0.66 |
| CLSP MLP 50% | Constraint | Lab | 47.07 | 0.56 | 53.41 | 0.63 |
| CLSP CNN 5% | Constraint | Lab | 46.02 | 0.56 | 51.00 | 0.59 |
| CLSP CNN 25% | Constraint | Lab | 47.50 | 0.54 | 52.48 | 0.63 |
| CLSP CNN 50% | Constraint | Lab | 47.11 | 0.54 | 53.73 | 0.64 |
| RF | Constraint | Real | 44.78 | 0.46 | 65.04 | 0.76 |
| LDA | Constraint | Real | 41.58 | 0.33 | 51.76 | 0.33 |
| HC + MLP | Constraint | Real | 44.62 | 0.42 | 52.60 | 0.60 |
| CLSP MLP 5% | Constraint | Real | 54.96 | 0.65 | 54.75 | 0.62 |
| CLSP MLP 25% | Constraint | Real | 45.88 | 0.42 | 62.31 | 0.66 |
| CLSP MLP 50% | Constraint | Real | 50.03 | 0.55 | 62.94 | 0.74 |
| CLSP CNN 5% | Constraint | Real | 52.49 | 0.62 | 55.22 | 0.65 |
| CLSP CNN 25% | Constraint | Real | 48.14 | 0.49 | 62.26 | 0.73 |
| CLSP CNN 50% | Constraint | Real | 47.03 | 0.49 | 59.58 | 0.70 |

Table 69: Performance comparison of ML (RF, LDA), DL (HC + MLP), pretrained CLSP fine-tuned on **arousal and valence classification** using **EDA modality** with datasets of **Constraint-based settings** as training cohort. Accuracy and F1 scores are reported.

| Model | Training Cohort | Testing Cohort | Arousal | | Valence | |
|---|---|---|---|---|---|---|
| | | | Accuracy | F1-score | Accuracy | F1-score |
| RF | Real | Constraint | 54.39 | 0.68 | 64.21 | 0.76 |
| LDA | Real | Constraint | 47.79 | 0.46 | 49.13 | 0.52 |
| HC + MLP | Real | Constraint | 51.25 | 0.55 | 52.66 | 0.60 |
| CLSP MLP 5% | Real | Constraint | 47.89 | 0.61 | 53.89 | 0.66 |
| CLSP MLP 25% | Real | Constraint | 50.73 | 0.56 | 53.42 | 0.68 |
| CLSP MLP 50% | Real | Constraint | 50.87 | 0.58 | 52.75 | 0.66 |
| CLSP CNN 5% | Real | Constraint | 49.74 | 0.60 | 50.70 | 0.62 |
| CLSP CNN 25% | Real | Constraint | 51.57 | 0.47 | 51.75 | 0.67 |
| CLSP CNN 50% | Real | Constraint | 50.06 | 0.58 | 52.71 | 0.65 |
| RF | Real | Lab | 48.66 | 0.62 | 53.04 | 0.63 |
| LDA | Real | Lab | 50.42 | 0.47 | 50.50 | 0.50 |
| HC + MLP | Real | Lab | 49.71 | 0.52 | 51.45 | 0.56 |
| CLSP MLP 5% | Real | Lab | 63.82 | 0.61 | 67.03 | 0.79 |
| CLSP MLP 25% | Real | Lab | 56.70 | 0.69 | 64.79 | 0.77 |
| CLSP MLP 50% | Real | Lab | 50.55 | 0.62 | 64.98 | 0.77 |
| CLSP CNN 5% | Real | Lab | 60.81 | 0.72 | 64.27 | 0.76 |
| CLSP CNN 25% | Real | Lab | 41.69 | 0.47 | 65.43 | 0.78 |
| CLSP CNN 50% | Real | Lab | 56.13 | 0.68 | 65.04 | 0.76 |

Table 70: Performance comparison of ML (RF, LDA), DL (HC + MLP), pretrained CLSP fine-tuned on **arousal and valence classification** using **EDA modality** with datasets of **Lab-based settings** as training cohort.

| Model | Training Cohort | Testing Cohort | Arousal | | Valence | |
|---|---|---|---|---|---|---|
| | | | Accuracy | F1-score | Accuracy | F1-score |
| RF | Lab | Lab | 50.63 | 0.50 | 53.86 | 0.54 |
| MLP | Lab | Lab | 50.31 | 0.50 | 53.09 | 0.54 |
| RF | Constraint | Constraint | 44.29 | 0.46 | 66.06 | 0.63 |
| MLP | Constraint | Constraint | 46.43 | 0.48 | 61.50 | 0.60 |
| RF | Real | Real | 49.53 | 0.45 | 44.11 | 0.37 |
| MLP | Real | Real | 54.10 | 0.49 | 46.13 | 0.41 |
| CLSP Zero Shot | – | Lab | 49.00 | 0.58 | 53.86 | 0.67 |
| CLSP Zero Shot | – | Constraint | 61.90 | 0.74 | 65.94 | 0.79 |
| CLSP Zero Shot | – | Real | 54.12 | 0.65 | 63.83 | 0.77 |

Table 71: Performance comparison of **In-cohort** and **zero-shot** performance for different groups based upon setting type on **arousal and valence classification** using **EDA modality**. Accuracy and F1 scores are reported.

| Model | Training Cohort | Testing Cohort | Arousal | | Valence | |
|---|---|---|---|---|---|---|
| | | | Accuracy | F1-score | Accuracy | F1-score |
| RF | Lab | Constraint | 43.36 | 0.42 | 60.06 | 0.72 |
| LDA | Lab | Constraint | 53.38 | 0.67 | 49.96 | 0.56 |
| HC + MLP | Lab | Constraint | 48.60 | 0.55 | 55.96 | 0.63 |
| CLSP MLP 5% | Lab | Constraint | 46.24 | 0.47 | 47.68 | 0.53 |
| CLSP MLP 25% | Lab | Constraint | 47.91 | 0.56 | 55.66 | 0.67 |
| CLSP MLP 50% | Lab | Constraint | 50.87 | 0.61 | 54.29 | 0.62 |
| CLSP CNN 5% | Lab | Constraint | 45.63 | 0.49 | 46.32 | 0.52 |
| CLSP CNN 25% | Lab | Constraint | 50.19 | 0.58 | 49.05 | 0.59 |
| CLSP CNN 50% | Lab | Constraint | 46.77 | 0.63 | 54.75 | 0.63 |
| RF | Lab | Real | 52.84 | 0.42 | 48.92 | 0.60 |
| LDA | Lab | Real | 56.16 | 0.69 | 47.85 | 0.54 |
| HC + MLP | Lab | Real | 49.41 | 0.55 | 52.64 | 0.54 |
| CLSP MLP 5% | Lab | Real | 46.77 | 0.46 | 53.33 | 0.59 |
| CLSP MLP 25% | Lab | Real | 52.64 | 0.61 | 53.62 | 0.64 |
| CLSP MLP 50% | Lab | Real | 53.72 | 0.63 | 50.29 | 0.57 |
| CLSP CNN 5% | Lab | Real | 49.22 | 0.55 | 52.25 | 0.56 |
| CLSP CNN 25% | Lab | Real | 52.94 | 0.63 | 49.02 | 0.54 |
| CLSP CNN 50% | Lab | Real | 52.94 | 0.63 | 47.44 | 0.54 |

Table 72: Performance comparison of ML (RF, LDA), DL (HC + MLP), pretrained CLSP fine-tuned on **arousal and valence classification** using **PPG modality** with datasets of **Lab-based settings** as training cohort. Accuracy and F1 scores are reported.

| Model | Training Cohort | Testing Cohort | Arousal | | Valence | |
|---|---|---|---|---|---|---|
| | | | Accuracy | F1-score | Accuracy | F1-score |
| RF | Constraint | Lab | 47.85 | 0.61 | 52.56 | 0.66 |
| LDA | Constraint | Lab | 51.68 | 0.41 | 49.87 | 0.45 |
| HC + MLP | Constraint | Lab | 49.79 | 0.48 | 50.28 | 0.56 |
| CLSP MLP 5% | Constraint | Lab | 50.74 | 0.53 | 51.40 | 0.60 |
| CLSP MLP 25% | Constraint | Lab | 49.20 | 0.52 | 53.50 | 0.64 |
| CLSP MLP 50% | Constraint | Lab | 48.35 | 0.54 | 52.66 | 0.63 |
| CLSP CNN 5% | Constraint | Lab | 48.40 | 0.53 | 53.03 | 0.67 |
| CLSP CNN 25% | Constraint | Lab | 47.27 | 0.56 | 53.02 | 0.66 |
| CLSP CNN 50% | Constraint | Lab | 50.40 | 0.53 | 53.11 | 0.66 |
| RF | Constraint | Real | 49.02 | 0.59 | 57.14 | 0.70 |
| LDA | Constraint | Real | 44.42 | 0.41 | 54.01 | 0.55 |
| HC + MLP | Constraint | Real | 46.67 | 0.50 | 58.32 | 0.63 |
| CLSP MLP 5% | Constraint | Real | 51.57 | 0.58 | 53.03 | 0.65 |
| CLSP MLP 25% | Constraint | Real | 49.71 | 0.55 | 52.84 | 0.66 |
| CLSP MLP 50% | Constraint | Real | 48.34 | 0.53 | 54.11 | 0.67 |
| CLSP CNN 5% | Constraint | Real | 48.34 | 0.48 | 54.80 | 0.65 |
| CLSP CNN 25% | Constraint | Real | 51.76 | 0.56 | 53.33 | 0.69 |
| CLSP CNN 50% | Constraint | Real | 52.74 | 0.58 | 53.43 | 0.68 |

Table 73: Performance comparison of ML (RF, LDA), DL (HC + MLP), pretrained CLSP fine-tuned on **arousal and valence classification** using **PPG modality** with datasets of **Constraint-based settings** as training cohort. Accuracy and F1 scores are reported.

| Model | Training Cohort | Testing Cohort | Arousal | | Valence | |
|---|---|---|---|---|---|---|
| | | | Accuracy | F1-score | Accuracy | F1-score |
| RF | Real | Constraint | 42.29 | 0.51 | 64.47 | 0.78 |
| LDA | Real | Constraint | 42.22 | 0.43 | 56.04 | 0.65 |
| HC + MLP | Real | Constraint | 44.57 | 0.47 | 61.81 | 0.73 |
| CLSP MLP 5% | Real | Constraint | 52.46 | 0.49 | 49.14 | 0.53 |
| CLSP MLP 25% | Real | Constraint | 53.41 | 0.48 | 51.69 | 0.63 |
| CLSP MLP 50% | Real | Constraint | 53.21 | 0.47 | 50.69 | 0.61 |
| CLSP CNN 5% | Real | Constraint | 53.63 | 0.48 | 50.55 | 0.58 |
| CLSP CNN 25% | Real | Constraint | 50.61 | 0.47 | 51.42 | 0.60 |
| CLSP CNN 50% | Real | Constraint | 53.50 | 0.43 | 48.67 | 0.48 |
| RF | Real | Lab | 53.51 | 0.46 | 53.58 | 0.69 |
| LDA | Real | Lab | 52.01 | 0.42 | 53.12 | 0.63 |
| HC + MLP | Real | Lab | 52.42 | 0.33 | 53.16 | 0.68 |
| CLSP MLP 5% | Real | Lab | 49.96 | 0.57 | 57.56 | 0.67 |
| CLSP MLP 25% | Real | Lab | 46.85 | 0.53 | 56.11 | 0.66 |
| CLSP MLP 50% | Real | Lab | 47.68 | 0.52 | 57.02 | 0.68 |
| CLSP CNN 5% | Real | Lab | 47.99 | 0.54 | 54.52 | 0.63 |
| CLSP CNN 25% | Real | Lab | 48.44 | 0.50 | 59.15 | 0.69 |
| CLSP CNN 50% | Real | Lab | 44.04 | 0.44 | 54.21 | 0.62 |

Table 74: Performance comparison of ML (RF, LDA), DL (HC + MLP), pretrained CLSP fine-tuned on **arousal and valence classification** using **PPG modality** with datasets of **Real-life settings** as training cohort. Accuracy and F1 scores are reported.

| Model | Training Cohort | Testing Cohort | Arousal | | Valence | |
|---|---|---|---|---|---|---|
| | | | Accuracy | F1-score | Accuracy | F1-score |
| RF | Lab | Lab | 50.85 | 0.51 | 47.16 | 0.48 |
| MLP | Lab | Lab | 49.62 | 0.50 | 49.35 | 0.50 |
| RF | Constraint | Constraint | 49.10 | 0.48 | 68.76 | 0.64 |
| MLP | Constraint | Constraint | 46.65 | 0.46 | 66.45 | 0.62 |
| RF | Real | Real | 49.14 | 0.48 | 37.22 | 0.35 |
| MLP | Real | Real | 43.53 | 0.41 | 39.81 | 0.41 |
| CLSP Zero Shot | | Lab | 52.08 | 0.15 | 46.87 | 0.28 |
| CLSP Zero Shot | | Constraint | 45.71 | 0.27 | 49.20 | 0.55 |
| CLSP Zero Shot | | Real | 46.58 | 0.31 | 50.10 | 0.50 |

Table 75: Performance comparison of **In-cohort** and **zero-shot** performance for different groups based upon setting type on **arousal and valence classification** using **PPG modality**. Accuracy and F1 scores are reported.

### 3.3.3 EDA + PPG

This section presents the experimental setting results for the EDA+PPG modality across three tables: lab-based settings in Table 76, constrained environments in Table 77, real-life scenarios in Table 78, in-cohort, and zero-short experiments in Table 79.

| Model | Training Cohort | Testing Cohort | Arousal | | Valence | |
|---|---|---|---|---|---|---|
| | | | Accuracy | F1-score | Accuracy | F1-score |
| RF | Lab | Constraint | 46.47 | 0.43 | 62.34 | 0.74 |
| LDA | Lab | Constraint | 52.62 | 0.64 | 51.56 | 0.58 |
| HC + MLP | Lab | Constraint | 51.56 | 0.57 | 49.81 | 0.52 |
| CLSP MLP 5% | Lab | Constraint | 46.32 | 0.51 | 51.63 | 0.61 |
| CLSP MLP 25% | Lab | Constraint | 53.15 | 0.60 | 56.34 | 0.66 |
| CLSP MLP 50% | Lab | Constraint | 46.24 | 0.49 | 56.49 | 0.67 |
| CLSP CNN 5% | Lab | Constraint | 51.86 | 0.57 | 54.14 | 0.64 |
| CLSP CNN 25% | Lab | Constraint | 42.60 | 0.50 | 53.38 | 0.65 |
| CLSP CNN 50% | Lab | Constraint | 48.37 | 0.52 | 52.01 | 0.63 |
| RF | Lab | Real | 39.09 | 0.32 | 63.11 | 0.76 |
| LDA | Lab | Real | 59.24 | 0.72 | 43.35 | 0.55 |
| HC + MLP | Lab | Real | 51.56 | 0.61 | 47.80 | 0.59 |
| CLSP MLP 5% | Lab | Real | 53.51 | 0.65 | 55.62 | 0.67 |
| CLSP MLP 25% | Lab | Real | 60.16 | 0.72 | 49.78 | 0.66 |
| CLSP MLP 50% | Lab | Real | 60.89 | 0.72 | 57.32 | 0.70 |
| CLSP CNN 5% | Lab | Real | 57.04 | 0.68 | 62.64 | 0.74 |
| CLSP CNN 25% | Lab | Real | 59.58 | 0.72 | 60.19 | 0.72 |
| CLSP CNN 50% | Lab | Real | 56.79 | 0.69 | 58.21 | 0.71 |

Table 76: Performance comparison of ML (RF, LDA), DL (HC + MLP), pretrained CLSP fine-tuned on **arousal and valence classification** using **EDA + PPG modality** with datasets of **Lab-based settings** as training cohort. Accuracy and F1 scores are reported.

## 3.4 Gender Group

The gender group results are presented across three tables, corresponding to Male and Female participants for three modality groups: EDA, PPG, and EDA+PPG.

### 3.4.1 EDA

This section presents the experimental setting results for the EDA modality across in Table 80.

### 3.4.2 PPG

This section presents the experimental setting results for the PPG modality across in Table 81.

| Model | Training Cohort | Testing Cohort | Arousal | | Valence | |
|---|---|---|---|---|---|---|
| | | | Accuracy | F1-score | Accuracy | F1-score |
| RF | Constraint | Lab | 48.76 | 0.60 | 53.54 | 0.67 |
| LDA | Constraint | Lab | 51.31 | 0.38 | 48.82 | 0.35 |
| HC + MLP | Constraint | Lab | 50.65 | 0.56 | 51.40 | 0.58 |
| CLSP MLP 5% | Constraint | Lab | 50.87 | 0.57 | 52.93 | 0.66 |
| CLSP MLP 25% | Constraint | Lab | 50.40 | 0.46 | 53.12 | 0.68 |
| CLSP MLP 50% | Constraint | Lab | 48.38 | 0.55 | 52.34 | 0.65 |
| CLSP CNN 5% | Constraint | Lab | 48.37 | 0.52 | 52.56 | 0.68 |
| CLSP CNN 25% | Constraint | Lab | 48.84 | 0.50 | 52.40 | 0.66 |
| CLSP CNN 50% | Constraint | Lab | 47.82 | 0.44 | 52.89 | 0.66 |
| RF | Constraint | Real | 53.34 | 0.64 | 79.54 | 0.88 |
| LDA | Constraint | Real | 45.38 | 0.50 | 52.65 | 0.64 |
| HC + MLP | Constraint | Real | 52.23 | 0.63 | 69.68 | 0.81 |
| CLSP MLP 5% | Constraint | Real | 59.63 | 0.73 | 60.80 | 0.72 |
| CLSP MLP 25% | Constraint | Real | 46.88 | 0.54 | 70.46 | 0.81 |
| CLSP MLP 50% | Constraint | Real | 58.16 | 0.70 | 71.88 | 0.83 |
| CLSP CNN 5% | Constraint | Real | 42.73 | 0.49 | 70.49 | 0.82 |
| CLSP CNN 25% | Constraint | Real | 51.67 | 0.60 | 69.32 | 0.81 |
| CLSP CNN 50% | Constraint | Real | 52.31 | 0.62 | 71.35 | 0.82 |

Table 77: Performance comparison of ML (RF, LDA), DL (HC + MLP), pretrained CLSP fine-tuned on **arousal and valence classification** using **EDA + PPG modality** with datasets of **Constraint-based settings** as training cohort. Accuracy and F1 scores are reported.

| Model | Training Cohort | Testing Cohort | Arousal | | Valence | |
|---|---|---|---|---|---|---|
| | | | Accuracy | F1-score | Accuracy | F1-score |
| RF | Real | Constraint | 44.27 | 0.54 | 64.54 | 0.77 |
| LDA | Real | Constraint | 47.84 | 0.40 | 49.13 | 0.51 |
| HC + MLP | Real | Constraint | 46.17 | 0.49 | 60.74 | 0.73 |
| CLSP MLP 5% | Real | Constraint | 49.00 | 0.64 | 53.46 | 0.69 |
| CLSP MLP 25% | Real | Constraint | 49.43 | 0.63 | 53.41 | 0.70 |
| CLSP MLP 50% | Real | Constraint | 50.10 | 0.63 | 54.27 | 0.69 |
| CLSP CNN 5% | Real | Constraint | 48.55 | 0.64 | 53.36 | 0.69 |
| CLSP CNN 25% | Real | Constraint | 50.31 | 0.61 | 53.77 | 0.69 |
| CLSP CNN 50% | Real | Constraint | 50.45 | 0.63 | 54.00 | 0.69 |
| RF | Real | Lab | 51.79 | 0.48 | 53.11 | 0.69 |
| LDA | Real | Lab | 52.53 | 0.37 | 49.95 | 0.53 |
| HC + MLP | Real | Lab | 52.58 | 0.38 | 52.34 | 0.66 |
| CLSP MLP 5% | Real | Lab | 56.64 | 0.71 | 64.69 | 0.78 |
| CLSP MLP 25% | Real | Lab | 54.06 | 0.68 | 65.60 | 0.79 |
| CLSP MLP 50% | Real | Lab | 51.33 | 0.65 | 65.60 | 0.78 |
| CLSP CNN 5% | Real | Lab | 55.89 | 0.70 | 64.84 | 0.78 |
| CLSP CNN 25% | Real | Lab | 52.23 | 0.64 | 65.76 | 0.78 |
| CLSP CNN 50% | Real | Lab | 54.06 | 0.65 | 64.47 | 0.77 |

Table 78: Performance comparison of ML (RF, LDA), DL (HC + MLP), pretrained CLSP fine-tuned on **arousal and valence classification** using **EDA + PPG modality** with datasets of **Real-life settings** as training cohort. Accuracy and F1 scores are reported.

### 3.4.3 EDA + PPG

This section presents the experimental setting results for the EDA + PPG modality across in Table 82.

## 3.5 Age Group

The age group results are presented in three tables, corresponding to each age group. In our case, we have two groups: young and old, and we have presented results for three modality groups: EDA, PPG, and EDA+PPG.

### 3.5.1 EDA

This section presents the experimental setting results for the EDA modality across in Table 83.

| Model | Training Cohort | Testing Cohort | Arousal | | Valence | |
|---|---|---|---|---|---|---|
| | | | Accuracy | F1-score | Accuracy | F1-score |
| RF | Lab | Lab | 52.13 | 0.52 | 48.78 | 0.50 |
| MLP | Lab | Lab | 51.28 | 0.52 | 50.44 | 0.51 |
| RF | Constraint | Constraint | 46.75 | 0.48 | 69.64 | 0.65 |
| MLP | Constraint | Constraint | 37.47 | 0.38 | 64.13 | 0.63 |
| RF | Real | Real | 50.45 | 0.46 | 46.04 | 0.42 |
| MLP | Real | Real | 45.59 | 0.44 | 42.85 | 0.42 |
| CLSP Zero Shot | – | Lab | 53.96 | 0.29 | 48.44 | 0.36 |
| CLSP Zero Shot | – | Constraint | 47.91 | 0.40 | 51.33 | 0.52 |
| CLSP Zero Shot | – | Real | 46.21 | 0.52 | 41.40 | 0.49 |

Table 79: Performance comparison of **In-cohort** and **zero-shot** performance for different groups based upon setting type on **arousal and valence classification** using **EDA + PPG modality**. Accuracy and F1 scores are reported.

| Model | Training Cohort | Testing Cohort | Arousal | | Valence | |
|---|---|---|---|---|---|---|
| | | | Accuracy | F1-score | Accuracy | F1-score |
| RF | Male | Female | 58.63 | 0.34 | 57.75 | 0.67 |
| LDA | Male | Female | 55.14 | 0.50 | 52.16 | 0.56 |
| HC + MLP | Male | Female | 54.92 | 0.48 | 53.30 | 0.55 |
| CLSP MLP 5% | Male | Female | 54.24 | 0.37 | 56.49 | 0.70 |
| CLSP MLP 25% | Male | Female | 56.81 | 0.40 | 56.97 | 0.67 |
| CLSP MLP 50% | Male | Female | 58.03 | 0.39 | 58.89 | 0.71 |
| CLSP CNN 5% | Male | Female | 54.56 | 0.36 | 53.24 | 0.61 |
| CLSP CNN 25% | Male | Female | 56.65 | 0.48 | 56.49 | 0.68 |
| CLSP CNN 50% | Male | Female | 58.33 | 0.38 | 58.11 | 0.66 |
| RF | Female | Male | 56.14 | 0.38 | 56.24 | 0.67 |
| LDA | Female | Male | 54.32 | 0.53 | 50.65 | 0.54 |
| HC + MLP | Female | Male | 55.48 | 0.56 | 52.03 | 0.55 |
| CLSP MLP 5% | Female | Male | 53.90 | 0.37 | 51.23 | 0.56 |
| CLSP MLP 25% | Female | Male | 54.96 | 0.32 | 56.86 | 0.69 |
| CLSP MLP 50% | Female | Male | 54.68 | 0.31 | 56.64 | 0.67 |
| CLSP CNN 5% | Female | Male | 55.22 | 0.29 | 51.21 | 0.58 |
| CLSP CNN 25% | Female | Male | 55.24 | 0.49 | 55.46 | 0.67 |
| CLSP CNN 50% | Female | Male | 55.26 | 0.25 | 54.06 | 0.63 |
| RF | Male | Male | 56.88 | 0.56 | 52.69 | 0.51 |
| HC + MLP | Male | Male | 56.94 | 0.56 | 53.14 | 0.53 |
| RF | Female | Female | 55.37 | 0.52 | 54.95 | 0.55 |
| HC + MLP | Female | Female | 53.49 | 0.51 | 55.50 | 0.55 |
| CLSP Zero Shot | – | Male | 48.25 | 0.56 | 55.80 | 0.69 |
| CLSP Zero Shot | – | Female | 48.45 | 0.54 | 55.99 | 0.69 |

Table 80: Performance comparison of traditional ML models (RF, LDA), deep learning models (HC + MLP), and the pretrained CLSP model (fine-tuned), evaluating both in-cohort and zero-shot performance across **gender groups** for **arousal** and **valence** classification using the **EDA modality**.

### 3.5.2 PPG

This section presents the experimental setting results for the PPG modality across in Table 84.

### 3.5.3 EDA + PPG

This section presents the experimental setting results for the EDA + PPG modality across in Table 85.

| Model | Training Cohort | Testing Cohort | Arousal | | Valence | |
|---|---|---|---|---|---|---|
| | | | Accuracy | F1-score | Accuracy | F1-score |
| RF | Male | Female | 59.53 | 0.29 | 55.76 | 0.69 |
| LDA | Male | Female | 53.52 | 0.51 | 54.22 | 0.53 |
| HC + MLP | Male | Female | 55.20 | 0.44 | 52.50 | 0.56 |
| CLSP MLP 5% | Male | Female | 54.62 | 0.39 | 53.68 | 0.59 |
| CLSP MLP 25% | Male | Female | 58.39 | 0.36 | 56.27 | 0.69 |
| CLSP MLP 50% | Male | Female | 57.83 | 0.38 | 55.44 | 0.68 |
| CLSP CNN 5% | Male | Female | 55.36 | 0.40 | 52.48 | 0.59 |
| CLSP CNN 25% | Male | Female | 55.16 | 0.38 | 55.64 | 0.64 |
| CLSP CNN 50% | Male | Female | 58.25 | 0.45 | 55.86 | 0.70 |
| RF | Female | Male | 55.24 | 0.27 | 58.07 | 0.71 |
| LDA | Female | Male | 54.30 | 0.51 | 54.64 | 0.60 |
| HC + MLP | Female | Male | 52.21 | 0.42 | 55.52 | 0.65 |
| CLSP MLP 5% | Female | Male | 54.80 | 0.32 | 52.17 | 0.60 |
| CLSP MLP 25% | Female | Male | 55.06 | 0.26 | 53.48 | 0.63 |
| CLSP MLP 50% | Female | Male | 55.02 | 0.35 | 54.20 | 0.62 |
| CLSP CNN 5% | Female | Male | 55.66 | 0.27 | 50.07 | 0.56 |
| CLSP CNN 25% | Female | Male | 55.68 | 0.43 | 56.64 | 0.67 |
| CLSP CNN 50% | Female | Male | 55.14 | 0.38 | 55.74 | 0.67 |
| RF | Male | Male | 51.02 | 0.50 | 50.02 | 0.48 |
| HC + MLP | Male | Male | 51.06 | 0.51 | 53.01 | 0.52 |
| RF | Female | Female | 57.12 | 0.54 | 49.23 | 0.49 |
| HC + MLP | Female | Female | 56.45 | 0.55 | 48.94 | 0.49 |
| CLSP Zero Shot | – | Male | 54.63 | 0.16 | 47.12 | 0.35 |
| CLSP Zero Shot | – | Female | 58.15 | 0.15 | 45.56 | 0.34 |

Table 81: Performance comparison of traditional ML models (RF, LDA), deep learning models (HC + MLP), and the pretrained CLSP model (fine-tuned), evaluating both in-cohort and zero-shot performance across **gender groups** for **arousal** and **valence** classification using the **PPG modality**.

| Model | Training Cohort | Testing Cohort | Arousal | | Valence | |
|---|---|---|---|---|---|---|
| | | | Accuracy | F1-score | Accuracy | F1-score |
| RF | Male | Female | 60.90 | 0.34 | 56.69 | 0.69 |
| LDA | Male | Female | 57.79 | 0.53 | 53.82 | 0.57 |
| HC + MLP | Male | Female | 56.37 | 0.49 | 55.10 | 0.60 |
| CLSP MLP 5% | Male | Female | 54.86 | 0.49 | 53.88 | 0.61 |
| CLSP MLP 25% | Male | Female | 53.84 | 0.48 | 58.07 | 0.70 |
| CLSP MLP 50% | Male | Female | 56.77 | 0.51 | 57.99 | 0.69 |
| CLSP CNN 5% | Male | Female | 55.88 | 0.45 | 53.96 | 0.62 |
| CLSP CNN 25% | Male | Female | 55.90 | 0.44 | 57.67 | 0.70 |
| CLSP CNN 50% | Male | Female | 54.66 | 0.48 | 57.69 | 0.69 |
| RF | Female | Male | 56.96 | 0.36 | 58.09 | 0.69 |
| LDA | Female | Male | 57.17 | 0.54 | 51.93 | 0.55 |
| HC + MLP | Female | Male | 56.28 | 0.48 | 55.00 | 0.61 |
| CLSP MLP 5% | Female | Male | 53.84 | 0.21 | 53.08 | 0.63 |
| CLSP MLP 25% | Female | Male | 55.86 | 0.34 | 56.62 | 0.68 |
| CLSP MLP 50% | Female | Male | 55.52 | 0.28 | 56.70 | 0.68 |
| CLSP CNN 5% | Female | Male | 54.98 | 0.41 | 54.20 | 0.62 |
| CLSP CNN 25% | Female | Male | 54.30 | 0.41 | 55.40 | 0.65 |
| CLSP CNN 50% | Female | Male | 54.88 | 0.13 | 58.07 | 0.70 |
| RF | Male | Male | 57.13 | 0.56 | 49.59 | 0.47 |
| HC + MLP | Male | Male | 56.30 | 0.56 | 47.48 | 0.46 |
| RF | Female | Female | 55.81 | 0.53 | 52.20 | 0.52 |
| HC + MLP | Female | Female | 55.21 | 0.56 | 53.15 | 0.54 |
| CLSP Zero Shot | – | Male | 53.42 | 0.24 | 48.91 | 0.42 |
| CLSP Zero Shot | – | Female | 55.47 | 0.25 | 50.15 | 0.42 |

Table 82: Performance comparison of traditional ML models (RF, LDA), deep learning models (HC + MLP), and the pretrained CLSP model (fine-tuned), evaluating both in-cohort and zero-shot performance across **gender groups** for **arousal** and **valence** classification using the **EDA+PPG modality**.

| Model | Training Cohort | Testing Cohort | Arousal | | Valence | |
|---|---|---|---|---|---|---|
| | | | Accuracy | F1-score | Accuracy | F1-score |
| RF | Old | Young | 58.51 | 0.28 | 58.71 | 0.71 |
| LDA | Old | Young | 52.04 | 0.50 | 53.77 | 0.58 |
| HC + MLP | Old | Young | 56.13 | 0.40 | 53.50 | 0.59 |
| CLSP MLP 5% | Old | Young | 48.28 | 0.47 | 59.01 | 0.72 |
| CLSP MLP 25% | Old | Young | 51.72 | 0.48 | 55.80 | 0.67 |
| CLSP MLP 50% | Old | Young | 55.93 | 0.40 | 54.70 | 0.65 |
| CLSP CNN 5% | Old | Young | 52.42 | 0.35 | 56.96 | 0.68 |
| CLSP CNN 25% | Old | Young | 57.21 | 0.26 | 54.75 | 0.63 |
| CLSP CNN 50% | Old | Young | 56.91 | 0.44 | 57.33 | 0.70 |
| RF | Young | Old | 55.96 | 0.32 | 59.06 | 0.71 |
| LDA | Young | Old | 53.95 | 0.51 | 50.60 | 0.51 |
| HC + MLP | Young | Old | 56.55 | 0.50 | 52.18 | 0.57 |
| CLSP MLP 5% | Young | Old | 52.42 | 0.31 | 56.53 | 0.67 |
| CLSP MLP 25% | Young | Old | 53.59 | 0.28 | 57.41 | 0.70 |
| CLSP MLP 50% | Young | Old | 53.93 | 0.19 | 59.85 | 0.73 |
| CLSP CNN 5% | Young | Old | 52.56 | 0.32 | 57.49 | 0.68 |
| CLSP CNN 25% | Young | Old | 54.51 | 0.32 | 58.62 | 0.72 |
| CLSP CNN 50% | Young | Old | 55.04 | 0.25 | 58.72 | 0.71 |
| RF | Old | Old | 56.93 | 0.55 | 54.87 | 0.53 |
| HC + MLP | Old | Old | 56.32 | 0.53 | 52.74 | 0.52 |
| RF | Young | Young | 56.24 | 0.54 | 56.62 | 0.54 |
| HC + MLP | Young | Young | 56.79 | 0.55 | 53.51 | 0.51 |
| CLSP Zero Shot | – | Old | 47.04 | 0.56 | 53.40 | 0.14 |
| CLSP Zero Shot | – | Young | 47.88 | 0.56 | 59.51 | 0.16 |

Table 83: Performance comparison of traditional ML models (RF, LDA), deep learning models (HC + MLP), and the pretrained CLSP model (fine-tuned), evaluating both in-cohort and zero-shot performance across **age groups** for **arousal** and **valence** classification using the **EDA modality**.

| Model | Training Cohort | Testing Cohort | Arousal | | Valence | |
|---|---|---|---|---|---|---|
| | | | Accuracy | F1-score | Accuracy | F1-score |
| RF | Old | Young | 59.44 | 0.18 | 56.88 | 0.67 |
| LDA | Old | Young | 53.97 | 0.43 | 51.37 | 0.50 |
| HC + MLP | Old | Young | 57.76 | 0.29 | 52.75 | 0.57 |
| CLSP MLP 5% | Old | Young | 54.53 | 0.31 | 55.25 | 0.64 |
| CLSP MLP 25% | Old | Young | 58.51 | 0.27 | 54.35 | 0.61 |
| CLSP MLP 50% | Old | Young | 58.19 | 0.33 | 55.58 | 0.63 |
| CLSP CNN 5% | Old | Young | 54.98 | 0.34 | 54.12 | 0.61 |
| CLSP CNN 25% | Old | Young | 56.48 | 0.40 | 53.15 | 0.58 |
| CLSP CNN 50% | Old | Young | 58.24 | 0.36 | 54.05 | 0.59 |
| RF | Young | Old | 54.19 | 0.43 | 59.77 | 0.72 |
| LDA | Young | Old | 51.69 | 0.53 | 57.23 | 0.63 |
| HC + MLP | Young | Old | 53.24 | 0.56 | 57.90 | 0.67 |
| CLSP MLP 5% | Young | Old | 55.02 | 0.48 | 54.03 | 0.62 |
| CLSP MLP 25% | Young | Old | 54.29 | 0.35 | 55.16 | 0.66 |
| CLSP MLP 50% | Young | Old | 54.81 | 0.42 | 59.10 | 0.72 |
| CLSP CNN 5% | Young | Old | 54.35 | 0.36 | 56.16 | 0.67 |
| CLSP CNN 25% | Young | Old | 55.16 | 0.42 | 57.96 | 0.69 |
| CLSP CNN 50% | Young | Old | 54.55 | 0.43 | 59.89 | 0.72 |
| RF | Old | Old | 57.83 | 0.55 | 57.43 | 0.57 |
| HC + MLP | Old | Old | 50.06 | 0.43 | 53.64 | 0.53 |
| RF | Young | Young | 55.23 | 0.53 | 54.59 | 0.51 |
| HC + MLP | Young | Young | 55.02 | 0.53 | 53.35 | 0.50 |
| CLSP Zero Shot | – | Old | 56.79 | 0.70 | 47.50 | 0.37 |
| CLSP Zero Shot | – | Young | 58.69 | 0.72 | 45.68 | 0.31 |

Table 84: Performance comparison of traditional ML models (RF, LDA), deep learning models (HC + MLP), and the pretrained CLSP model (fine-tuned), evaluating both in-cohort and zero-shot performance across **age groups** for **arousal** and **valence** classification using the **PPG modality**.

| Model | Training Cohort | Testing Cohort | Arousal | | Valence | |
|---|---|---|---|---|---|---|
| | | | Accuracy | F1-score | Accuracy | F1-score |
| RF | Old | Young | 61.12 | 0.21 | 57.96 | 0.69 |
| LDA | Old | Young | 57.41 | 0.45 | 52.52 | 0.52 |
| HC + MLP | Old | Young | 59.74 | 0.34 | 54.88 | 0.63 |
| CLSP MLP 5% | Old | Young | 51.09 | 0.47 | 53.25 | 0.61 |
| CLSP MLP 25% | Old | Young | 52.47 | 0.46 | 54.93 | 0.65 |
| CLSP MLP 50% | Old | Young | 55.52 | 0.47 | 52.65 | 0.56 |
| CLSP CNN 5% | Old | Young | 53.12 | 0.44 | 51.54 | 0.54 |
| CLSP CNN 25% | Old | Young | 55.65 | 0.43 | 55.63 | 0.65 |
| CLSP CNN 50% | Old | Young | 54.63 | 0.39 | 56.18 | 0.66 |
| RF | Young | Old | 55.62 | 0.43 | 60.80 | 0.73 |
| LDA | Young | Old | 53.40 | 0.55 | 55.74 | 0.60 |
| HC + MLP | Young | Old | 55.18 | 0.56 | 58.78 | 0.67 |
| CLSP MLP 5% | Young | Old | 53.38 | 0.21 | 57.71 | 0.71 |
| CLSP MLP 25% | Young | Old | 54.78 | 0.38 | 58.48 | 0.71 |
| CLSP MLP 50% | Young | Old | 53.95 | 0.29 | 58.48 | 0.70 |
| CLSP CNN 5% | Young | Old | 54.21 | 0.28 | 55.14 | 0.65 |
| CLSP CNN 25% | Young | Old | 54.33 | 0.44 | 56.28 | 0.65 |
| CLSP CNN 50% | Young | Old | 53.50 | 0.18 | 59.33 | 0.72 |
| RF | Old | Old | 53.87 | 0.53 | 53.29 | 0.53 |
| HC + MLP | Old | Old | 53.57 | 0.52 | 52.94 | 0.53 |
| RF | Young | Young | 57.86 | 0.57 | 52.13 | 0.48 |
| HC + MLP | Young | Young | 58.96 | 0.58 | 47.50 | 0.46 |
| CLSP Zero Shot | – | Old | 51.97 | 0.19 | 49.40 | 0.47 |
| CLSP Zero Shot | – | Young | 58.84 | 0.28 | 47.36 | 0.35 |

Table 85: Performance comparison of traditional ML models (RF, LDA), deep learning models (HC + MLP), and the pretrained CLSP model (fine-tuned), evaluating both in-cohort and zero-shot performance across **age groups** for **arousal** and **valence** classification using the **EDA+PPG modality**.