# OpenReview forum: "FEEL: Quantifying Heterogeneity in Physiological Signals for Generalizable Emotion Recognition"
_NeurIPS.cc/2025/Datasets_and_Benchmarks_Track — NeurIPS 2025 Datasets and Benchmarks Track poster_

### Official Review · Reviewer_SjV7 · 2025-06-30

**Rating:** 5
**Confidence:** 4

**Summary:**

This is a very solid and well-organized paper, presenting a thorough experimental analysis of the proposed evaluation framework. This paper introduces FEEL (Framework for Emotion Evaluation), a large-scale benchmarking framework for emotion recognition from physiological signals, specifically electrodermal activity (EDA) and photoplethysmography (PPG). The authors address a clear need in the community for standardized, reproducible evaluation across heterogeneous datasets. They curate 19 publicly available datasets spanning a wide variety of experimental settings, devices, and labeling strategies, and systematically benchmark 16 models across four paradigms: traditional machine learning, deep learning with handcrafted features, deep learning on raw signals, and contrastive language-signal pretraining (CLSP). Through extensive within-dataset and cross-dataset evaluation, the paper shows that fine-tuned CLSP models achieve the best results in most settings, but that simpler models with handcrafted features remain competitive, particularly in noisy or low-resource contexts. The cross-dataset analysis also provides important insights about how factors like device type, labeling method, and data collection setting affect model generalizability.

**Dataset Code Accessibility:**

NA; not applicable to this submission (e.g., no new dataset, benchmark, code, or data provided)

**Ethical Comments:**

No, there are no or only very minor ethics concerns

**Ethical Considerations:**

No, there are no or only very minor ethics concerns

**Final Justification:**

I keep my final rating.

**Limitations Weaknesses:**

(1) While the data curation and benchmarking are valuable, the methodological contributions in modeling are less pronounced, as most of the evaluated architectures are standard and previously established. The omission of more recent paradigms, such as large language models (LLMs) or reinforcement learning frameworks, represents a missed opportunity to fully explore the performance envelope. With only four representative modeling paradigms included, some readers may find it difficult to connect this benchmark with the newest trends in the field. LLMs would strengthen the work to incorporate analyses of how more advanced signal representations — beyond handcrafted features — could be integrated to improve EDA and PPG signal interpretation. On the other hand, reinforcement learning frameworks might provide clearer guidance or a roadmap on how to extend the dataset to better account for participants’ demographics, cultural background, and variations in physical and mental health. It would make the benchmark more equitable and broadly applicable with the specific design of the reward function.
(2) Show strong performance for CLSP models, but several datasets (e.g., CASE, ADARP, LAUREATE) have markedly low scores, needing deeper discussion.
(3) Several crucial results and implementation details (e.g., more precise feature lists, extended benchmarking results) are deferred to the appendix, which can reduce the self-contained clarity of the main text

**Strengths Contributions:**

The authors provide a unified benchmark spanning 19 publicly available emotion recognition datasets based on physiological signals, creating a valuable resource for the community. Their dataset curation involves detailed preprocessing and standardization, including a unified binning and labeling protocol for arousal and valence (Section 3.2, Table 1), which addresses a major challenge in this research area. The paper rigorously evaluates 16 models across four major paradigms — traditional machine learning, deep learning with handcrafted features, deep learning on raw signals, and self-supervised pretrained models (Section 3.3) — offering a representative and comprehensive overview of current methods. Extensive experiments using subject-independent cross-validation and multiple training/test regimes provide concrete benchmarking results, showing that CLSP-based models, particularly with fine-tuning, consistently outperform other approaches. Finally, the release of code and harmonized datasets supports reproducibility and encourages continued community engagement.

---

> ### Author Rebuttal · Authors · 2025-07-30
>
> We thank the reviewer for their encouraging and constructive comments on our work. Please find our detailed responses to Reviewer SjV7 comments below:
>
> **1) LLMs and RL-based methods in Benchmarking:** We thank the reviewer for their thoughtful feedback. We agree that recent developments, such as LLMs and RL frameworks, represent promising directions for advancing physiological emotion recognition. Recent efforts, including Time2Lang [1], PhysioLLM [2], and HealthLLM [3], further highlight the growing interest in leveraging these paradigms for sensor data interpretation. We acknowledge that incorporating such approaches could provide deeper insights, particularly in learning advanced signal representations or addressing personalization through adaptive mechanisms like reward design. **However, the primary goal of our current work was to establish a robust and reproducible benchmark rooted in widely adopted modeling paradigms.** These methods remain highly relevant in the field and provide a necessary foundation for evaluation across heterogeneous datasets. Furthermore, we emphasize that **our fine-tuning framework for CLSP models constitutes a key methodological contribution**, as it enables the adaptation of pretrained models to new datasets without the need for textual inputs, a crucial step toward scalable and practical deployment in real-world settings.
>
> Finally, we appreciate the reviewer’s suggestions and view on the integration of LLM and RL-based approaches as an exciting extension of our current work, which we aim to pursue in our future work.
>
> **2) CLSP Performance discussion:** We thank the reviewer for this observation. We agree that the markedly low scores for certain datasets, such as CASE, ADARP, and LAUREATE, require deeper discussion. **In the current version of the manuscript, we addressed these performance differences in Table 10, which presents a dataset-wise performance summary highlighting the strengths and limitations of each dataset in supporting effective arousal and valence classification.** The table explicitly separates high-performing datasets from those with poor or contrasting performance, and includes contextual information on the data collection methods and labeling strategies that may have impacted the overall model effectiveness, thus providing a data-centric view of classification performance.
> For instance, the **CASE** dataset exhibited close to random performance in both arousal and valence classification across all modeling paradigms, which we attribute to its use of continuous joystick-based labeling. This labeling method may have interfered with participants’ emotional immersion or may have reduced the accuracy of self-reports, leading to reduced signal-labels. Similarly, for the **LAUREATE** dataset CLSP model showed strong performance in arousal classification using PPG and EDA+PPG data. This likely reflects the nature of the self-reported engagement assessments used to derive arousal and valence labels; these assessments appear to capture the **intensity** of emotional experiences (arousal) more effectively than their **directionality** (valence, i.e., positive vs. negative emotion), and thus suggest better alignment between signals and arousal label. Similarly, for **ADARP** CLSP models demonstrated strong arousal classification across individual modalities (EDA and PPG), which we believe is due to the participant cohort, as all participants were diagnosed with Alcohol Use Disorder (AUD), who predominantly reported states such as anxiety, stress, and neutrality, that was more aligned with arousal than exhibiting clear shifts in emotional valence. Moreover, we observed a trend in LAUREATE and ADARP, both real-world datasets, suggesting that physiological signals are more aligned with arousal than with valence in naturalistic settings.
>
> Overall, these findings highlight the importance of strong alignment between the elicited physiological signals and the associated labels- particularly, labels should meaningfully reflect underlying physiological changes to support effective emotion classification.  Finally, we appreciate the reviewer’s suggestion and will further expand this discussion in Section 5.1 (Benchmarking Performance across Modeling Paradigms) and in Appendix A.6 (Results Discussion) to more clearly articulate the implications of dataset characteristics on model performance, and to aid readers in interpreting these benchmark results.
>
> **3) Implementation Details in Appendix:** We thank the reviewer for pointing this out. We acknowledge that deferring several important results and implementation details (such as detailed feature lists and extended benchmarking outcomes) to the appendix may impact the self-contained clarity of the main text. In the final version, we will incorporate the crucial details into the main text, as permitted by the page limit constraints.
>
> **References**
>
> 1] Pillai, A., Spathis, D., Nepal, S., Collins, A. C., Mackin, D. M., Heinz, M. V., ... & Campbell, A. (2025). Time2Lang: Bridging Time-Series Foundation Models and Large Language Models for Health Sensing Beyond Prompting. arXiv preprint arXiv:2502.07608.
>
> 2] Fang, C. M., Danry, V., Whitmore, N., Bao, A., Hutchison, A., Pierce, C., & Maes, P. (2024, November). Physiollm: Supporting personalized health insights with wearables and large language models. In 2024 IEEE EMBS International Conference on Biomedical and Health Informatics (BHI) (pp. 1-8). IEEE.
>
> 3] Kim, Y., Xu, X., McDuff, D., Breazeal, C., & Park, H. W. (2024). Health-llm: Large language models for health prediction via wearable sensor data. arXiv preprint arXiv:2401.06866.

---

> > ### Author Response · Authors · 2025-08-06
> >
> > Dear Reviewer SjV7,
> >
> > We hope our rebuttal has clarified your initial comments. We appreciate that you’ve acknowledged it, and would be grateful if you could share any feedback you may have on our response.

---

### Official Review · Reviewer_JFh6 · 2025-07-02

**Rating:** 4
**Confidence:** 3

**Summary:**

This paper introduces ​FEEL, the first large-scale benchmarking framework for physiological emotion recognition using electrodermal activity (EDA) and photoplethysmography (PPG) signals. Addressing the field's fragmentation, FEEL standardizes evaluation across ​19 diverse public datasets​ spanning lab, constrained, and real-world settings. The framework evaluates ​16 modeling approaches—including traditional ML, deep learning, and contrastive language-signal pretraining (CLSP)—for arousal/valence classification. Key findings reveal that ​fine-tuned CLSP models​ achieve the highest performance overall, while ​handcrafted features​ consistently outperform raw signal models. Cross-dataset analysis demonstrates strong generalization: models trained on ​real-world or expert-annotated data​ transfer effectively to controlled settings, and ​lab-device-trained models​ adapt well to wearables. FEEL provides critical insights into dataset heterogeneity (settings, devices, labeling strategies) and establishes a reproducible foundation for developing robust, deployable emotion recognition systems.

**Dataset Code Accessibility:**

Yes

**Dataset Code Comments:**

There is no new dataset proposed. Only code is available.

**Ethical Considerations:**

No, there are no or only very minor ethics concerns

**Final Justification:**

Thanks for the authors' response. Please add these additional results in the final version.

**Limitations Weaknesses:**

- The paper does not analyze how ​age, gender, cultural background, or health conditions​ affect model performance, despite known physiological differences across groups.

- Mapping diverse emotion labels (e.g., discrete emotions, stress) to binary arousal/valence ​loses nuanced emotional context. No sensitivity analysis on labeling strategies is provided.

- For datasets with self-reports, ​delays between physiological signals and retrospective labeling​ are ignored, potentially harming alignment.

- The CLSP fine-tuning relies on ​manually crafted textual prompts​ for classes. This introduces subjectivity and may not generalize optimally across cultures or languages.

- Raw-signal DL models are outperformed by simpler models. No efficiency comparison (e.g., FLOPs, latency) is provided.

- Only random oversampling is used during training. ​No testing of advanced techniques​ for highly skewed datasets (e.g., NURSE: 44 vs. 248 negative valence samples). During test, are the results reliable if the labels are imbalance in the test set?

**Strengths Contributions:**

- Benchmark:​​ Introduces the ​first large-scale, standardized benchmark​ for physiological emotion recognition, harmonizing ​19 diverse public datasets​ (spanning lab, constrained, and real-world settings) to address fragmentation in the field.


- Comprehensive Evaluation:​​ Rigorously evaluates ​16 diverse modeling approaches​ across four paradigms (traditional ML, feature-based DL, raw-signal DL, contrastive pretraining) for ​arousal/valence classification​ using EDA and PPG signals.
​

- Practical Impact & Reproducibility:​​ Provides a ​unified framework (FEEL)​​ for future benchmarking and robust model development. Shares ​open-source code, promoting transparency, reproducibility, and community adoption.

---

> ### Author Rebuttal · Authors · 2025-07-30
>
> We thank the reviewer for their constructive and thoughtful comments on our work. Please find our detailed responses to Reviewer JFh6 comments below:
>
> **1) Lack of Analysis on Age, Gender, Culture, and Health:** We thank the reviewer for raising this point. We agree that understanding how demographic factors such as age, gender, cultural background, and health conditions influence model performance is critical for building equitable and generalizable systems. However, **most of the datasets used in our benchmark do not provide complete or consistent metadata on these attributes, limiting the scope of such analyses**. That said, based on the available demographic information, we conducted additional transferability experiments focused on gender (using available binary male/female labels across 8 datasets) and age (grouped into 18–25 and 25+ age ranges across 6 datasets, depending on data availability). It is important to note that the age distribution is heavily skewed toward the 18–30 range, with limited representation of older individuals, which constrains conclusions about age-related generalization.
>
> The key takeaways from our gender-based transferability experiments are as follows: Gender-specific transferability is significantly higher for valence than arousal, likely due to lower inter-gender variability in the physiological patterns associated with valence. In our experiments, valence classification achieved an **F1 score of 0.71**, while arousal classification reached a maximum of only 0.563, with many results falling below random chance. These findings suggest that CLSP models are well-suited for cross-gender valence recognition, but struggle to generalize arousal-related patterns across genders. This highlights the stronger influence of gender on arousal-related physiological responses and underscores the need for future research focused on gender-aware modeling.
>
> These experiments offer initial insights into demographic transferability, and we will include the results in the revised version of the paper. We also acknowledge the broader importance of demographic-aware benchmarking and plan to further explore this in future work.
>
> **2) Binarization of Emotion Labels:** Thank you for raising this concern about the binarization of categorical emotion labels. **Our initial motivation for the binary label transformation was to unify label spaces across datasets with varying emotional category definitions, enabling consistent benchmarking and cross-dataset comparison.** However, we agree that binary classification may oversimplify the complexity of emotional states, and the benchmark should also consider comparison to more nuanced emotion classification.
>
> To address this, we will add new set of benchmarking experiments on a more nuanced **four-class classification based on the widely accepted circumplex model of affect** [1], which combines arousal and valence dimensions into following four classes: High Arousal Positive Valence (HAPV), High Arousal Negative Valence (HANV), Low Arousal Positive Valence (LAPV), and Low Arousal Negative Valence (LANV).
>
> This 4-class approach was chosen because of the following reasons:
> **i)** It is theoretically grounded, widely used in the emotion recognition and affective computing community, and **captures more fine-grained emotional distinctions than binary labels**.
> **ii) It is practically feasible and consistent across datasets in our benchmark**, as arousal and valence annotations are the most commonly available labels across our datasets.
> **iii)** It **maintains our cross-dataset experimental setting and label unification strategy**, while better reflecting the complexity of human emotions.
>
> We will add the results and a revised discussion in Section 5.1 of the main paper, and include the additional details about the four-class experiment in our appendix. We believe this will provide a more comprehensive analysis of labeling strategies on the model performance across datasets. Additionally, we would like to clarify that four-class classification was not feasible for several datasets (NURSE, UBFC_PHYS, MAUS, UNOBSTRUCTIVE, VERBIO, and ADARP) due to no samples in some class/classes or severe class imbalance, making a fair comparison on a more fine-grained classification task difficult across all datasets. This limitation guided our initial decision to adopt a two-class classification setup, which was consistently feasible across all datasets.
>
> **3) Label–Signal Delay in Self-Reported Datasets:** Thank you for highlighting this concern. We fully agree that the delay between physiological signals and retrospective emotion self-reports poses a challenge for precise alignment. This issue is indeed present in most of the datasets used in our benchmark, where emotional labels are collected after stimulus presentation, introducing a potential temporal misalignment. The one exception is the **CASE** dataset, in which participants provided real-time continuous annotations of their emotional experiences using a joystick-based interface, simultaneously reporting valence and arousal throughout the video. These annotations were later aggregated across participants for each segment, providing more temporally aligned labels. **Unfortunately, such continuous annotation is rare, and most available datasets only include post-hoc self-reports, which remain a recognized challenge in emotion recognition/affective computing.** Following common practice in prior literature, we treated these retrospective self-reports as ground truth in our benchmark. **While this may introduce some label noise due to delay, it reflects the current state of publicly available emotion datasets and was necessary to harmonize data across studies for large-scale cross-dataset evaluation.** We appreciate the reviewer’s point and will explicitly acknowledge this limitation in our revised manuscript (Section 6). Going forward, we aim to explore strategies to better handle signal-label alignment.
>
> **4) Subjectivity in CLSP Prompts:** We agree that textual prompts can introduce subjectivity, especially across cultures and languages. However, in our work, **the prompts used for CLSP fine-tuning were carefully designed to describe general physiological and emotional cues, such as heart rate changes, muscle tension, or energy levels, that are widely recognized across cultures**. These prompts reflect universal patterns of emotional experience, rather than culture-specific language or narratives. Moreover, **our method does not rely solely on fixed textual class descriptions**. Instead, the **text prompt is paired with context tokens that are learned for each input signal segment**. These tokens are trained to adapt the prompts dynamically for each sample, allowing the model to go beyond fixed definitions and reduce potential bias or rigidity. As a result, the model can generalize better across datasets within our experiments.
>
> **5) Efficiency Comparison:** Thank you for the comment. In Section 4 (line 270) of the paper, we provide a detailed comparison of training time and computational load across different modeling paradigms and datasets. Raw-signal-based deep learning models, as well as feature-based deep models, were the most computationally intensive, often requiring several GPU days as per the dataset size. In contrast, classical machine learning models and hybrid HC+MLP models typically completed training in under an hour. Fine-tuning the CLSP models was the most efficient, taking only 1 to 30 minutes of GPU time per dataset. We agree that including FLOPs and inference latency would offer a more hardware-independent and standardized perspective on the model efficiency. We will incorporate these metrics in the final version of the paper for a clearer view of performance across modeling paradigms.
>
> **6) Advanced Imbalance Handling Techniques:** We thank the reviewer for highlighting this point. We acknowledge that relying solely on random oversampling may not be sufficient for handling severe class imbalance, particularly in datasets like NURSE, which has a significant skew in valence labels (e.g., 44 vs. 248 samples). To address this, we conducted additional experiments using SMOTE (Synthetic Minority Over-sampling Technique) on highly imbalanced datasets to assess whether more advanced sampling methods improve performance. We will include the results in the revised manuscript. Regarding the test phase, we agree that label imbalance can affect the reliability of evaluation metrics. To account for this, in our results, we have reported the F1-score to provide a more comprehensive view of model performance under imbalance.

---

> > ### Comment · Reviewer_JFh6 · 2025-08-04
> >
> > Thanks for the authors' response. Please add these additional results in the final version.

---

### Official Review · Reviewer_CwHR · 2025-07-03

**Rating:** 5
**Confidence:** 4

**Summary:**

This paper offers a benchmarking framework for emotion recognition using two physiological signals: EDA and PPG. Study is done with 19 public datasets evaluating several ML models from traditional ones to self-supervised deep learning. Some studies are within datasets and some are cross-dataset settings. The authors aim to assess generalization across devices, labeling methods, and environments.Reported results show that the deep learning approach on ​​pretrained representation learning performs well overall, but simpler models with handcrafted features remain competitive, especially in noisy or low-resource settings. The study also finds strong cross-dataset transferability, such as from lab to real-world data and across different device types. Overall, this paper provides a useful and practical framework for studying robustness and generalization of wearable-based emotion recognition systems, and it is supported by open-source code.

**Dataset Code Accessibility:**

Partly

**Ethical Considerations:**

No, there are no or only very minor ethics concerns

**Final Justification:**

I am keeping my "accept" decision, as the authors’ response and proposed improvements align well with my comments.

**Limitations Weaknesses:**

I have some suggestions for improvement of the paper:

1. The benchmarking codebase would benefit greatly from a more structured and user-friendly tutorial. Currently, the codebase lacks proper guidance for researchers to adopt and extend it efficiently. Without clear documentation and usage instructions, the practical impact of the work may be diminished. I strongly encourage the authors to provide a detailed plan for improving the codebase, ideally including examples, setup guides, and usage scenarios to support broader adoption.


2. The decision to convert categorical emotion labels into binary ones is motivated for unifying the label space. However, this binning strategy made me concerned about potentially introducing bias into model evaluation. It may oversimplify the complexity of emotional states and meaningful distinctions between categories. I recommend that the authors either support this approach by some ablations or provide a more detailed justification and analysis of its impact on model performance and interpretability.


3. The inclusion of Contrastive Language-Signal Pretraining (CLSP) as a baseline method in the benchmark appears to me somewhat ad hoc. The paper does not sufficiently explain why this particular method was chosen over other self-supervised learning (SSL) approaches, nor does it detail the criteria or motivations guiding this selection. Moreover, CLSP relies on textual information, which is not typically available in many physiological datasets. This choice needs further justification to make the benchmarking’s method more solid. A clearer discussion on the compatibility of this approach with the benchmark's goals and datasets would enhance the work’s credibility.

**Strengths Contributions:**

I have enjoyed reading this paper. I believe this paper presents a timely and impactful contribution to the field of emotion recognition, particularly through its use of non-invasive, privacy-preserving wearable signals such as EDA and PPG. The direction of this paper offers some significant promise, as the paper is offering a more natural and scalable alternative to modalities like audio and video, and enabling a broader range of real-world applications.

I am aligned with the authors’ vision for the community on the call to treat data as a “shared resource” and to work toward synergizing across key dimensions such as signal representations and labeling strategies. These are valuable efforts to motivate more reproducible and collaborative research practices.

I also appreciate the authors' overall framing and motivation, which I believe many researchers in the community will find both engaging and insightful.

---

> ### Author Rebuttal · Authors · 2025-07-30
>
> We thank the reviewer for their encouraging and constructive comments on our work. Please find our detailed responses to Reviewer CwHR comments below:
>
> **1) Improving the Benchmarking Codebase Documentation and Usability:** Thank you for this valuable suggestion. We fully agree that improving the usability of the benchmarking codebase is essential to support broader adoption and collaboration in the research community. To address this, we will provide a structured, step-by-step tutorial in our GitHub repository.
>
> This tutorial will include:
> **i)** Environment setup and installation instructions in detail,
> **ii)** Dataset-wise standardization procedures,
> **iii)** Steps on how to run baseline models, CLSP models, fine-tuning procedures, and group-wise experiments,
>
> We will also include a **detailed walkthrough of the codebase, explaining the purpose and usage of each folder and script**. This will help researchers efficiently navigate and reproduce our results. Additionally, **we will add examples demonstrating how to extend the benchmark with new datasets or models using our harmonization strategy**. While all required folders and scripts are already present in the current version of the codebase, we acknowledge that clearer documentation is needed. We will add the above-mentioned tutorial and improved documentation to maximize the accessibility and impact of our work.
>
> **2) Emotion Label Binarization and Evaluation Bias:** Thank you for raising this concern about the binarization of categorical emotion labels. **Our initial motivation for the binary label transformation was to unify label spaces across datasets with varying emotional category definitions, enabling consistent benchmarking and cross-dataset comparison.** However, we agree that binary classification may oversimplify the complexity of emotional states, and the benchmark should also consider more nuanced classification.
>
> To address this, we will add new set of benchmarking experiments on a more nuanced **four-class classification based on the widely accepted circumplex model of affect** [1], which combines arousal and valence dimensions into following four classes: High Arousal Positive Valence (HAPV), High Arousal Negative Valence (HANV), Low Arousal Positive Valence (LAPV), and Low Arousal Negative Valence (LANV).
>
> This 4-class approach offers several advantages:
> **i)** It is theoretically grounded, widely used in the emotion recognition and affective computing community, and **captures more fine-grained emotional distinctions than binary labels**.
> **ii) It is practically feasible and consistent across datasets in our benchmark**, as arousal and valence annotations are the most commonly available labels.
> **iii)** It **maintains our cross-dataset experimental setting and label unification strategy**, while better reflecting the complexity of human emotions.
>
> We will add the results and a revised discussion in Section 5.1 of the main paper, and include the additional details about the four-class experiment in our appendix. We believe this will provide a more comprehensive analysis of labeling strategies on the model performance across datasets. Additionally, we would like to clarify that four-class classification was not feasible for several datasets (NURSE, UBFC_PHYS, MAUS, UNOBSTRUCTIVE, VERBIO, and ADARP) due to no samples in some class/classes or severe class imbalance, making a fair comparison on a more fine-grained classification task difficult across all datasets. This limitation guided our initial decision to adopt a two-class classification setup, which was consistently feasible across all datasets.
>
> **3) Justification for Including CLSP as a Baseline Method:** Thank you for this comment. We clarify that the inclusion of CLSP as a baseline was driven by its strong alignment with the goals of our benchmark and the unique suitability of its pretrained models for our experiments. Specifically, **CLSP offers publicly available pretrained models for EDA-only, PPG-only, and combined EDA+PPG inputs, directly matching the three input configurations we evaluate across our datasets.** Moreover, **it was trained on tasks that are fully aligned with our benchmark label space, namely, valence and arousal classification**, unlike other self-supervised models (e.g., LSM-2 [2], PaPaGei [3], pulsePPG [4]), which are either modality-specific (PaPaGei and pulse-PPG are PPG-only) or do not have an open-source pre-trained model (LSM-2 is not open-source) or are trained on unrelated tasks like anxiety or hypertension detection [LSM-2], thus limiting our baseline choices for benchmarking on self-supervised models.
>
> We acknowledge the reviewer’s point regarding CLSP’s reliance on textual supervision during pretraining. However, we clarify that in our experiments, **textual data is not required at inference or fine-tuning time**. CLSP was initially trained using paired signal–text data, but our experiments focused on evaluating its transferability to datasets without textual annotations, a common real-world constraint in physiological emotion recognition. **The strong zero-shot performance observed in our initial zero-shot evaluations across our 19 datasets indicated that the representations learned by CLSP generalize well even without further textual input.** This motivated our adoption of a COCO-Op–based fine-tuning strategy, which allows efficient adaptation of CLSP models to new datasets using only physiological signals and binary (valence/arousal) labels. This makes CLSP practically usable across all datasets in our benchmark, regardless of whether textual descriptions are available.
>
> Furthermore, this setting mirrors practices common in the vision community with CLIP, where models pretrained on image–text pairs are successfully transferred to purely image-labeled tasks. We believe CLSP offers a similar modality-bridging foundation for physiological data. Overall, the availability of pretrained models across EDA, PPG, and combined modalities and robust transfer to unseen datasets without text makes it a strong candidate for our experiments. We will expand this in Section 3.3 of the main paper to make these motivations more explicit.
>
> **Reference:**
>
> 1] Russell, J. A. (1980). A circumplex model of affect. Journal of personality and social psychology, 39(6), 1161.
>
> 2] Xu, M. A., Narayanswamy, G., Ayush, K., Spathis, D., Liao, S., Tailor, S. A., ... & McDuff, D. (2025). LSM-2: Learning from Incomplete Wearable Sensor Data. arXiv preprint arXiv:2506.05321.
>
> 3] Pillai, A., Spathis, D., Kawsar, F., & Malekzadeh, M. (2024). Papagei: Open foundation models for optical physiological signals. arXiv preprint arXiv:2410.20542.
>
> 4] Saha, M., Xu, M. A., Mao, W., Neupane, S., Rehg, J. M., & Kumar, S. (2025). Pulse-ppg: An open-source field-trained ppg foundation model for wearable applications across lab and field settings. arXiv preprint arXiv:2502.01108.

---

> > ### Comment · Reviewer_CwHR · 2025-08-03
> >
> > Thanks for your thoughtful responses. I believe your revision plan adequately addresses my comments.The proposed improvements to the codebase documentation, the addition of supplementary four-class emotion classification, and the clear justification for including CLSP as a baseline (compared to other alternatives) will strengthen the quality of this paper. I encourage you to incorporate these discussions and clarifications into the main paper.

---

### Official Review · Reviewer_69Nf · 2025-07-03

**Rating:** 5
**Confidence:** 3

**Summary:**

In this paper, the authors have introduced FEEL (Framework for Emotion Evauation), that is the first large-scale benchmarking study of emotion recognition using electrodermal activity (EDA) and photoplethysmography (PPG) signals across 19 publicly available datasets. They evaluated 16 architectures spanning traditional machine learning, deep learning, and self-supervised pretraining approaches, structured into four representative modeling paradigms.

**Additional Feedback:**

None

**Dataset Code Accessibility:**

Yes

**Dataset Code Comments:**

None

**Ethical Considerations:**

No, there are no or only very minor ethics concerns

**Final Justification:**

This paper introduced FEEL (Framework for Emotion Evauation), a large-scale benchmarking study of emotion recognition using electrodermal activity (EDA) and photoplethysmography (PPG) signals across 19 publicly available datasets. The authors comprehensively evaluated 16 architectures spanning traditional machine learning, deep learning, and self-supervised pretraining approaches, structured into four representative modeling paradigms.  I can recommend the acceptance of this paper.

**Limitations Weaknesses:**

Weaknesses:
1.	While the authors stated that models leveraging handcrafted features (109/114) consistently outperform those trained on raw signal segments, underscoring the value of domain knowledge in low-resource, noisy settings. Does it mean that manual features outperform automatic features extracted by deep learning models? More discussion should be provided.
2.	In fact, the manual features often highly rely on specific expert experience in a given domain, which lacks generalization while being extended to other tasks. The authors should further explain why manual feature extraction perform well in generalizable emotion recognition.
3.	During fine-tuning stage, the modulation networks were designed in two architectural variants: one using two linear layers, and another employing two stacked 1D convolutional layers applied to the signal embedding sequence. More structure details should be presented.

**Strengths Contributions:**

Strengths:
1.	This paper is well organized and written.
2.	The authors curated a diverse collection of 19 publicly available datasets covering a wide range of experimental conditions and labeling strategies.
3.	They also performed a meta-analysis of data quality and benchmarked this dataset suite using four representative modeling approaches commonly employed in prior studies.

---

> ### Author Rebuttal · Authors · 2025-07-30
>
> We thank the reviewer for their encouraging and constructive comments on our work. Please find our detailed responses to the  Reviewer 69Nf comments below:
>
> **1) Clarification on Manual vs. Automatic Feature Performance:** Our results demonstrate that handcrafted features, when used with classical machine learning models or hybrid deep learning models (i.e., those that incorporate learnable layers after feature extraction), consistently outperform deep learning models that have to learn discriminative features directly from raw signals. **However, this observation should not be viewed as a tradeoff between “manual” and “automatic” approaches.** Instead, it reflects the fact that handcrafted features can encode key physiological invariances that end-to-end models must learn from scratch. **Extracting such patterns directly from raw and often noisy signals is inherently more difficult and typically requires large datasets, making deep models less effective in low-data scenarios.** In contrast, statistical features provide structured, informative inputs that enable more efficient learning even with limited data. That said, while end-to-end deep models may eventually outperform feature-based approaches when trained on sufficiently large and diverse datasets, we did not observe such a trend across our dataset collection. This reinforces the importance of dataset harmonization for enabling effective deep learning. We will incorporate this discussion into Section 5.1, highlighting the practical advantages of feature-based models in low-data settings, as suggested by the reviewer.
>
> **2) Generalizability of Handcrafted Features:** While the reviewer raises a valid concern that handcrafted features can often depend heavily on domain expertise and lack generalization, we want to clarify that this does not apply to our approach. In our work, **"handcrafted/manual features"** refer to standard statistical features derived from both time and frequency domains of raw signal segments, such as mean, slope, variance, entropy, etc., computed automatically using well-established signal-processing libraries like *NeuroKit2*. **These features are based on general physiological signal-specific properties and do not require manual intervention, expert tuning, or dataset-specific adjustments.** Once signal segments are defined, the extraction process is fully automated and reproducible across datasets. To further support generalization, we also experimented with a hybrid approach where these statistical features are passed through trainable linear layers within the CLSP architectures. This allows the model to adapt and refine the fixed features based on task-specific data, combining the robustness of structured physiological descriptors with the flexibility of end-to-end learning. This combination has proven effective across our diverse set of emotion recognition datasets in our study, suggesting that the feature-enabled deep learning approach we used, while rooted in prior domain knowledge, is scalable and well-suited for generalizable emotion recognition.
>
> **3) Details on Modulation Network Architectures:** As noted by the reviewer, we explored two architectural variants for the MetaNet (modulation network):
> **i) MLP-based MetaNet:** This variant consists of two linear layers. The first maps the 100-dimensional input embedding to a 32-dimensional hidden representation, followed by a ReLU activation. The second linear layer projects this 32-dimensional vector to 768 dimensions to align with the CLSP text embedding space.
> **ii) 1D-CNN-based MetaNet:** This version of the model takes a 1D input signal of length 100 with one channel. The first convolutional layer uses a kernel size of 3, stride of 1, and padding of 1, and outputs 24 channels, followed by a ReLU activation. The second convolutional layer, using the same kernel and padding, reduces the output back to a single channel. The result is then flattened and passed through a linear layer to generate a 768-dimensional vector.
> We have included these details in Appendix A.3.5 of the submitted version (referenced in line 213), and we will further revise this section to make the structure of both variants more explicit and easier to interpret in the final version of the paper as suggested.

---

> > ### Author Response · Authors · 2025-08-06
> >
> > Dear Reviewer 69Nf,
> >
> > We hope our rebuttal has addressed your initial comments clearly. Please let us know if you require any further clarification or additional details.

---

> > > ### Comment · Reviewer_69Nf · 2025-08-07
> > >
> > > Thanks for the authors' response, which have well addressed my concerns. I can increase my rating at this time.

---

### Decision · Program_Chairs · 2025-09-18

**Decision:**

Accept (poster)

**Comment:**

The paper received recommendations of three accept and one borderline accept. The reviewers initially had some concerns with the work; however, the authors gave a detailed rebuttal for each concern. The reviewers acknowledged this in their responses. Emotion recognition with physiological signals is an important and challenging problem. The proposed framework includes a large number of datasets and evaluated models. The framework offers valuable insight into creating generalizable models for emotion recognition. The evaluation is robust with within and cross-dataset evaluations. The work is a nice contribution to the field and can fuel future research in this area.

===== FINAL UPDATE FROM DB Track PCs ====

The final decision for this paper has been taken by the program chairs after consultation with the SACs. All Senior Area Chairs have ranked papers according to the feedback from the AC during the review process. We decided to leave the original meta-review to reflect the opinion of the AC in light of the initial discussions with reviewers and SAC.